# Atmospheric Mercury: Recent advances in theoretical, computational, experimental, observational and isotopic understanding to decipher its redox transformations in the upper and lower atmosphere and interactions with Earth surface reservoirs

Jonas O. Sommar[1], Xinyu Shi[1,2], Xueling Tang[1,2], Guangyi Sun[1], Che-Jen Lin[3] and Xinbin Feng[1,2].

[1] State Key Laboratory of Environmental Geochemistry, Institute of Geochemistry, Chinese Academy of Sciences, Guiyang 550081, China.
[2] University of Chinese Academy of Sciences, Beijing 100045, China.
[3] Department of Mechanical Engineering, University of West Florida, Pensacola, Florida, United States.

*Correspondence to*: Jonas O. Sommar (jonassommar@icloud.com)

**List of content**

## Abstract

Mercury is a volatile heavy element with no known biological function. It is present in trace amounts (on average, ~80 ppb) but is not geochemically well blended in the Earth's crust. As a result, it occurs in extremely high concentrations (up to a few percent) in certain locations. It is found along tectonic plate faults in deposits of sulfide ores (cinnabar), and it has been extensively mobilized during the Anthropocene. Mercury is currently one of the most targeted global pollutants, with methylmercury compounds being particularly neurotoxic. Over 5,000 tons of mercury are released into the atmosphere annually through primary emissions and secondary re-emissions. Much of the re-emitted mercury resulting from exchanges with surface reservoirs is related to (legacy) human activities, such as direct releases. Understanding the dynamics of the global Hg cycle is critical for assessing the impact of emission reductions under the UN Minamata Convention, which became legally binding in 2017. This review of atmospheric mercury focuses on fundamental advances in field, laboratory, and theoretical studies, including six stable Hg isotope analytical methods, which have contributed recently to a more mature understanding of the complexity of the atmospheric Hg cycle and its interactions with the Earth's surface ecosystem.

## 1 Introduction

Mercury (Hg) is a potent neurotoxin that, via methylmercury (MMHg$^+$) food exposure, poses a global health threat (e.g., IQ decrement and

heart attack) (Zhang et al., 2021b). The atmosphere plays a pivotal role in the Hg biogeochemical cycle, functioning as the most important transient reservoir, a conduit for transport and transformation, and a site rich in redox chemistry. In part due to concerns about global Hg transport, the multilateral UN Environment Convention on Hg was negotiated and entered into force in 2017 with a mandate to reduce the intentional use and emissions of Hg (UNEP, 2018). Research on Hg biogeochemical cycling gained momentum after an outbreak of mass $MMHg^+$ poisoning severely affected the population of Minamata Bay, Japan, in the 1950s and 1960s through the consumption of contaminated seafood, and it became clear that $MMHg^+$ was present at chronically high levels in predatory fish in many lakes, particularly those in the boreal forest belt, through long-range transport and biomagnification (Lindqvist et al., 1991). The earliest known series of measurements of airborne elemental Hg, possibly the first systematic study of its kind, was conducted in Pacific North America during the second half of the 1960s (Williston, 1968). It was recognized as early as the 1970s that Hg circulated globally through the atmosphere (Nriagu, 1979). Somewhat later, Slemr et al. (1985) published an influential paper whose results on the distribution, speciation and budget of atmospheric Hg reproduce fairly well the qualitative features of the atmospheric Hg cycle, such as atomic vapor ($Hg^0$) dominating the atmospheric pool and showing an interhemispheric difference with higher concentrations in the northern hemisphere, and being relatively well mixed vertically through the troposphere with an extensive residence time (concept as a "global pollutant").

Knowledge of the physical and chemical processes that govern the dynamics of Hg in the atmosphere has developed gradually. Over time, through technological leaps (stable isotope sampling in natural probes, refined methods in the theoretical and experimental field, etc.), its full complexity began to be appreciated. In earlier research, the prevailing view was that water-phase oxidation by ozone could be the primary mechanism initiating the removal of tropospheric $Hg^0$ (Pleijel and Munthe, 1995; Seigneur et al., 1994). However, newer data have indicated that gaseous oxidized mercury (GOM) could also be present in the atmosphere (Xiao et al., 1997; Lindberg and Stratton, 1998), in addition to the particulate form (PBM). Specifically, the observation that $Hg^0$ was periodically depleted in the planetary mixing layer during the polar spring (Schroeder et al., 1998) prompted a reassessment of Hg chemistry in favor of homogeneous gas-phase chemistry (Hynes et al., 2009). The two-step gas-phase oxidation of $Hg^0$ initiated by Br atoms has emerged as the most important global channel for tropospheric conversion to $Hg^{II}$ (Donohoue et al., 2006; Holmes et al., 2010). Gas-phase $O_3$ was previously considered an oxidizing agent for $Hg^0$ to $Hg^{II}$. Although this route was discarded, $O_3$ has been found to effectively oxidize intermediate $Hg^I$ species (Gómez Martín et al., 2022). This suggests that OH- and less certain I-initiated oxidation of $Hg^0$, which produces less stable intermediates than Br and Cl do, may also be important for Hg turnover in parts of the troposphere and beyond (Dibble et al., 2020; Lee et al., 2024). A novel finding is that major $Hg^{I,II}$ species, which are expected to be formed in the atmosphere upon oxidation of $Hg^0$, are themselves photolabile and undergo gas-phase reduction (Francés-Monerris et al., 2020; Saiz-Lopez et al., 2019). The complexity of rapid redox Hg chemistry involving multiple gas-phase oxidation states (0, +1 and +2) is further compounded by the impact of multiphase interactions, including reactive uptake and homogeneous and heterogeneous processes in condensed-phase media, on the dynamics of atmospheric Hg. An indicator of the maturation of our understanding of atmospheric Hg chemistry is the inclusion of bromine chemistry in critically evaluated datasets for use in atmospheric studies (Burkholder et al., 2019). Over the past two decades, measurements of Hg stable isotope ratios in natural samples have emerged as valuable tools for gaining insights into the atmospheric Hg cycle. One notable outcome of isotope analysis is the recognition that dry $Hg^0$ deposition exerts a more pronounced influence on a global scale than was previously understood, with wet and dry deposition of the atmospheric $Hg^{II}$ fraction being of lesser importance (Jiskra et al., 2018).

Hg in the atmosphere has been the subject of reviews over the past 45 years: *topics* including *biogeochemical cycling* (Lindqvist and Rodhe, 1985; Lindqvist et al., 1991; Schroeder and Munthe, 1998; Selin, 2009; Lyman et al., 2020), *observations* (Slemr et al., 2003; Sprovieri et al., 2010; Dommergue et al., 2010; Fu et al., 2015; Steffen et al., 2015; Mao et al., 2016; Zhang et al., 2019c; Custódio et al., 2022; Bencardino et al., 2024), *isotopic observational data* (Kwon et al., 2020; Liu et al., 2024), *atmospheric measurement techniques* (Pandey et al., 2011; Huang et al., 2014; Gustin et al., 2015; Davis and Lu, 2024; Gustin et al., 2024), *anthropogenic emissions* (Carpi, 1997; Zhang et al., 2016; Cheng et al., 2023), *natural volcanism* (Edwards et al., 2021), *physical removal and air-surface exchange* (Zhang et al., 2009; Sommar et al., 2013; Zhu et al., 2016; Agnan et al., 2016; Cooke et al., 2020; Sommar et al., 2020; Zhou et al., 2021; Liu et al., 2024) with emphasis on *global change* (Obrist et al., 2018; Sonke et al., 2023), *polar atmospheric surface layer mercury depletion events* (Steffen et al., 2008), *chemical conversion in the atmosphere* (Schroeder et al., 1991; Lin and Pehkonen, 1999; Lin et al., 2011; Si and Ariya, 2018), *aqueous*

*homogeneous and heterogeneous photoredox chemistry* (Zhang, 2006; Si et al., 2022), *multi-phase atmospheric chemistry* (Ariya et al., 2015), *assessment of critical atmospheric chemical processes using state-of-the-art experimental and computational chemistry methods* (Ariya and Peterson, 2005; Ariya et al., 2008; Hynes et al., 2009), *receptor-* (Cheng et al., 2015) *and global models* (Lin et al., 2006; Lin et al., 2007; Subir et al., 2011, 2012; Amos et al., 2015; Travnikov et al., 2017). This review is based on the perspective of atmospheric scientists, with synthesis and a comprehensive account of the results of fundamental research, including field, laboratory, and theoretical studies, that have contributed to an understanding of the complexity of the atmospheric Hg cycle and its interactions with the Earth's surface ecosystem at the molecular level. This work does not address several topics related to Hg in the atmosphere. These include anthropogenic and natural emission inventories, corresponding top-down constraints and inverse modeling from atmospheric observations, accounting for long-term air data series and their temporal and spatial trends, observations of the PBM and its particle size distributions, wet deposition, future scenarios for the effects of regulatory measures (Minamata Convention), ongoing climate change and many more topics. Our goal is to provide a comprehensive review of the atmospheric chemistry of both inorganic and organic Hg in the lower and upper atmosphere, coupled with a compilation of updated, critically evaluated kinetic, thermochemical, photochemical, and isotopic fractionation data. Where appropriate, we introduce the basic concepts and fundamental aspects of Hg chemistry, including those of condensed phases. In atmospheric Hg isotope chemistry, our approach is comprehensive, encompassing a range of activities from field observations of air and $Hg^0$ gas exchange with natural surfaces to laboratory studies of processes that may be relevant to the atmosphere. We also highlight areas of persistent uncertainty or lack of consensus, such as measurement methods for atmospheric Hg speciation and the partitioning of $Hg^{II}$ in atmospheric water between inorganic and organic ligands.

## 2 Physical chemistry of elemental mercury

Hg is the only metal that is a liquid at standard temperature and pressure (freezing point of -38.8 °C and boiling point of 356.7 °C), and its vapor is monatomic. Under these conditions, the mixing ratio of neurotoxic Hg vapor in equilibrium with metallic liquid is already at the hazardous level of approximately 1.7 ppm (Huber et al., 2006). Liquid Hg possesses properties that have given it a wide range of applications in the past despite its known toxicity, including exceptional surface tension (nearly seven times greater than that of water at 25 °C), high specific gravity, high electrical conductivity (a reference substance for measuring the SI unit $\Omega$), low compressibility, and a constant volume of expansion coefficient in the liquid state. Hg forms solid alloys (amalgams) with most metals except iron. This property enables its application in gold panning (HgAu), dental fillings (HgAg), or as an electrode material in the chloralkali industry (NaHg). The electronic configuration of the mercury atom has filled f- and d-orbitals with a high density of 6s-valence electrons near the nucleus ($[Xe]4f^{14}5d^{10}6s^2$), which is related to a relativistic radial contraction of s- and p-orbitals as the inner electrons approach a significant fraction of the speed of light (which for a Hg 1s electron is 58%, implying a radial shrinkage of 23%; Pyykkö, 1988). It also follows that oxidation states 0 and +2 (mercuric, $d^{10}$ metal ion) are the most stable for Hg. Nevertheless, Hg differs from other metals in its propensity to readily form a polycation in the aqueous phase, the mercurous ion, $Hg_2^{2+}$, which is, however, only metastable in the gaseous phase (Strömberg and Wahlgren, 1990). The solubility of $Hg^0$ in water is limited to 0.3 μM (Sanemasa, 1975), and the gas–water equilibrium is governed by Henry's law. The Henry's law coefficient ($k_H^{cp}$) for $Hg^0$ is 0.11 M atm$^{-1}$ at 25 °C (Andersson et al., 2008), whereas the value is more than seven orders of magnitude greater for the $HgCl_2$ molecule at the same temperature (Sommar et al., 2000). Since the mid-19th century, light production, including sharp lines at 184.9 and 257.3 nm, has been achieved by passing an electric arc through $Hg^0$ vapor in a glass bulb. Conversely, absorption spectroscopy uses these atomic deep-UV lines to analytically detect Hg in samples, as discussed in detail in **Section 3.1**.

## 3 Atmospheric environment

### 3.1 Atmospheric measurements of mercury species

Hg is the only trace gas, other than the noble gases (Burnard, 2013), that exists as free atoms ($Hg^0$) in the atmosphere, making this pollutant exceptional in terms of low detection limits by optical measurement techniques. This makes it possible to measure $Hg^0$ vapor emissions in real time, for example, from mining, chloralkali production and geothermal activities, as has been done in Europe for decades via light detection and ranging (LIDAR) in differential absorption mode by mobile laser systems (Svanberg, 2002). If the optical path length in the measuring cell of an instrument is sufficiently long (i.e., using multipath techniques such as cavity ring-down), then the conditions exist for continuous measurement of $Hg^0$ in ambient air (at the sub-ppt level, $\sim 5 \times 10^6$ atoms cm$^{-3}$ in the northern

hemisphere) via atomic absorption spectroscopy (AAS) with Zeeman background correction (Osterwalder et al., 2020). The application of Zeeman AAS in Hg stable isotope analysis has also been described (Lu et al., 2019). As an alternative to Zeeman splitting of the Hg($6^3$P) level for sensitive, selective detection of Hg$^0$ (Sholupov et al., 2004), sequential two-photon laser-induced fluorescence schemes have been used (Bauer et al., 2002; Bauer et al., 2014; Hynes et al., 2017). For initial excitation of the Hg($6^1$S$_0$) $\rightarrow$ Hg($6^3$P$_1$) transition at 253.7 nm, a light beam from a Hg discharge lamp or the frequency-doubled output of a dye laser pumped by the third harmonic of a Nd:YAG laser is used. As shown in **Fig. 1a**, further excitation involves the sequential excitation of different atomic transitions by two laser systems, both starting from the Hg($6^3$P$_1$) state, followed by the detection of blue- (Hg($6^1$P$_1$) $\rightarrow$ Hg($6^1$S$_0$) at 184.9 nm) or redshifted (e.g., at 578.9 nm) fluorescence. The detection of Hg$^0$ with such a sophisticated apparatus is an exception to typical measurements, which are made via cold vapor atomic fluorescence spectroscopy (CV-AFS) after preconcentration sampling on gold (Ambrose, 2017). Smaller non-Hg$^0$ portions of atmospheric Hg are challenging to speciate because of their low concentrations. Instead, they are fractionated operationally based on their oxidation state (Hg$^0$ versus GOM) or phase state (GOM versus PBM). Since gold does not selectively trap Hg$^0$ but also captures other Hg species (Dumarey et al., 1985; Gačnik et al., 2024), the GOM and PBM must be individually collected upstream of the sample air to accurately measure the triad Hg$^0$–GOM–PBM. KCl-coated annular denuders have been utilized for fractionating ambient GOM by gas–phase diffusion for over two decades. Nonetheless, upon the development of techniques to assess its accuracy in measuring ambient air regularly, the method was found to be biased in a nonsystematic manner toward lower values (Jaffe et al., 2014; Lyman et al., 2010; McClure et al., 2014). The automated KCl denuder method, with its variable efficiency, can thus

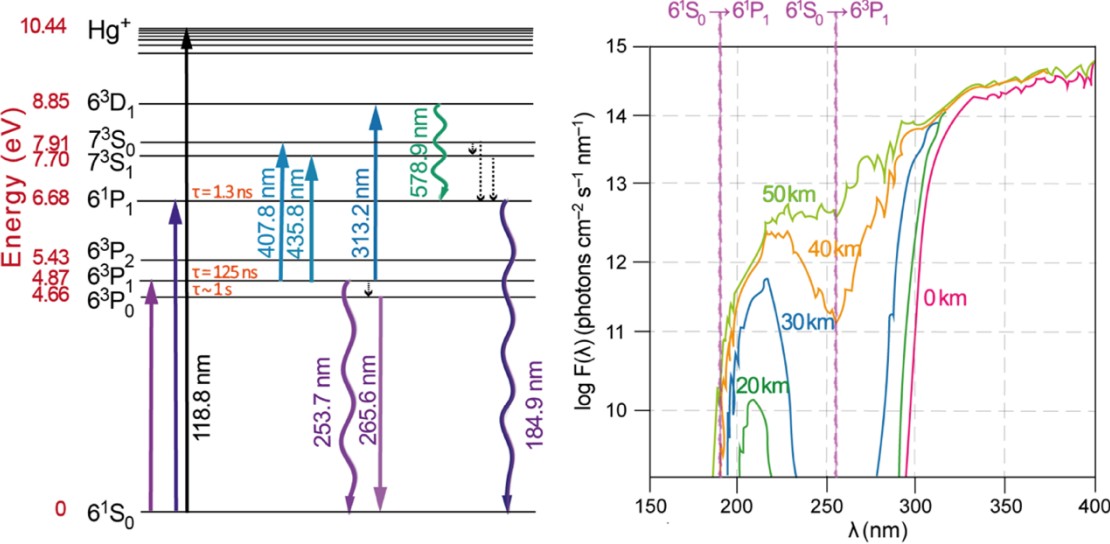

**Figure 1:** Left (a). Energy level diagram of the Hg atom. The wave-shaped arrows represent resonant radiation. Right (b): Actinic fluxes as a function of altitude. The wavelengths of the Hg($^1$S$_0$) $\rightarrow$ Hg($^3$P$_1$) and $\rightarrow$ Hg($^1$P$_1$) transitions at 253.7 and 184.9 nm, respectively, are given.

lead to serious underestimation of the GOM, whereas the refluxing mist chamber method, which is an alternative, carries the risk of cosampling the GOM with the PBM (Gustin et al., 2021). However, the KCl-covered denuder does not have full penetration of PBMs < 2.5 µm, but aerosols of a hundred nm or less are increasingly trapped by the salt surface (Ghoshdastidar et al., 2019). When compared, refluxing mist chambers yielded ambient GOM concentrations that were 3 to 4 times higher on average than those obtained with KCl-coated annular denuders (Landis et al., 2002). A decade later, the capture and retention efficiency of the KCl denuder method for GOM was evaluated, which was close to 95% in synthetic Hg$^0$-free air but decreased drastically to between 20% and 54% when exposed to ambient air, where ozone and humidity were found to cause severe reductive losses such as Hg$^0$ (McClure et al., 2014). In fact, ozone gas can heterogeneously reduce particle-bound Hg$^{II}$ halides, as recent experiments have shown (Ai et al., 2023). In high-humidity marine applications, KCl denuder technology operates at very low efficiency; for example, He and Mason (2021) reported average losses of 80% during oceanographic expeditions in the Pacific. By determining total airborne mercury (TAM; Steffen et al., 2002; Slemr et al., 2018) and Hg$^0$ in air, a measure of reactive mercury (RM) is obtained as the sum of GOM + PBM by subtracting Hg$^0$ from TAM. In turn, Hg$^0$ is obtained by passing an air stream through a filter and a cation exchange membrane (CEM) in series, whereas TAM is measured as Hg$^0$ after a pyrolysis unit held at 800 °C converts all Hg in the sample air to elemental vapor (Lyman et al., 2020b). CEM is capable of capturing and retaining Hg$^{II}$ quantitatively over long storage periods but has no affinity for Hg$^0$ (Miller et al., 2019). However, when two quantities that are usually close to each other are subtracted, the precision of the RM determination is low. Hynes et al. (2017) used two-photon laser-induced fluorescence as an online detection method for

RM (by switching between ambient and pyrolyzed air as the source for the $Hg^0$ analyte) and concluded that the variability in ambient $Hg^0$ severely limits the sensitivity of dual-channel difference RM measurements. For the separation of the semivolatile GOM fraction from the PBM in ambient air, various membranes have been examined, but with recognized limitations (Dunham-Cheatham et al., 2023; Gustin et al., 2023). The realization of NIST-traceable GOM calibration systems has recently progressed (Gacnik et al., 2022). Several studies have been carried out with the aim of experimentally deciphering the molecular identities (speciation) of the GOM pool in ambient air. Most methods are based on a preconcentration process of GOM on a substrate, which is then thermodesorbed in a gas stream following a programmed temperature ramp and detected as $Hg^0$ after pyrolysis (Gustin et al., 2015), alternatively focused on a capillary column and analyzed by different types (chemical ionization CI; electron impact ionization) of mass spectrometry (MS) (Deeds et al., 2015; Jones et al., 2016). In the former case, standards are used in the form of a number of commercially available Hg chemicals (such as $HgBr_2$, $HgCl_2$, HgO, $Hg(NO_3)_2$, and $HgSO_4$) that are assumed to be representative surrogates for GOM (Huang et al., 2017; Sexauer Gustin et al., 2016). As inferred by Khalizov et al. (2020), this speciation is indirect, as it has not been confirmed that the GOM molecules adsorbed on the substrate can be desorbed in the same chemical form as they are in air.

In contrast, studies have shown that aerosol reactions lead to the re-speciation of mercuric halides on surfaces (Mao et al., 2021; Mao and Khalizov, 2021). The authors reported that their ion-drift (ID) CI-MS system, which is sensitive enough for detection in laboratory studies, can achieve an LOD at a 1 amu resolution of $(0.8–2.0) \times 10^5$ molecules $cm^{-3}$ toward ambient GOM by switching to multistage atmospheric pressure ID-CI-MS. The feasibility of using proton transfer reaction mass spectrometry (PTR-MS) to study the reaction products (GOM) of Br-initiated $Hg^0$ oxidation has been evaluated by Dibble et al. (2014) but is not recommended because it cannot be applied in multi-stage atmospheric pressure systems (Khalizov et al., 2020). In summary, direct measurements of ambient GOM species have not yet been achieved. No method exists for chemically characterizing the GOM fraction, which is semivolatile and may contain species that are photolytically unstable. Since previous GOM measurements are considered unreliable (Lyman et al., 2020a; Slemr et al., 2016) and emerging RM data (Lyman et al., 2020b; Slemr et al., 2018; Swartzendruber et al., 2009; Gratz et al., 2015; Lyman and Jaffe, 2012) are still too sparse and spatially limited, it is not possible to draw deterministic conclusions on atmospheric $Hg^{II}$. Sampling methods for organic Hg (dimethylmercury; He et al., 2022 and monomethylated $Hg^{II}$ species; Lee et al., 2003) in ambient air, as opposed to inorganic Hg species, are more unambiguous. The speciation of Hg in atmospheric waters is discussed in **Section 4.6**. Hg measurement data from air and precipitation, ground-based or aircraft (Slemr et al., 2018; Slemr et al., 2016) observations that fall outside the scope of this review, including those reported from continental (Cole et al., 2014; Cole et al., 2013; Schmolke et al., 1999; Wängberg et al., 2001; Gay et al., 2013; Fu et al., 2015) to hemispherical (Bencardino et al., 2024; Szponar et al., 2020; Slemr et al., 2020; Sprovieri et al., 2017; Sprovieri et al., 2016) monitoring networks, some of which have been in operation since before the turn of 2000 (Custódio et al., 2020), have been reviewed elsewhere (Mao et al., 2016; Lyman et al., 2020a; Howard et al., 2017; Angot et al., 2016; Kim et al., 2012; Zhang et al., 2017). In the case of the isotopic characterization of atmospheric Hg, however, we feel justified in compiling, analyzing, and discussing the considerable body of recent observations (**Section 8.2**).

### 3.2 Stability of atmospheric $Hg^0$

$Hg^0$ represents the primary form of atmospheric mercury in both the troposphere and stratosphere. Considering the spatial variability of $Hg^0$ concentrations [1], which depart from a uniform vertical distribution throughout the atmosphere (Slemr et al., 2018), a singular global atmospheric lifetime is not appropriate. A more pertinent measure is the effective lifetime of $Hg^0$, expressed on an annual basis and as a function of its horizontal and vertical location within the atmosphere. The observed disparity in tropospheric $Hg^0$ concentrations between the Northern and Southern Hemispheres of a factor of ~1.41 (Tang et al., 2025), despite anthropogenic emissions in the Northern Hemisphere being approximately 2.5 times greater than those in the Southern Hemisphere (Streets et al., 2019; Sonke et al., 2023), implies that $Hg^0$ has a relatively short effective lifetime in comparison to the interhemispheric air mass

---

[1] There is a widespread practice in the Hg research community to report $Hg^0$ air concentrations in ng $m^{-3}$ referenced to one atmosphere (101.325 kPa) and 0 °C (STP). By that, the unit represents a mixing ratio not an absolute (mass) concentration.

exchange time of approximately 1.3 years (Geller et al., 1997). As $Hg^0$ crosses the intertropical convergence zone, it undergoes convective uplift, enabling its transport into the stratosphere. Troposphere-to-stratosphere $Hg^0$ transport has been regarded as limited (100–176 Mg yr$^{-1}$; Lyman and Jaffe, 2012; Horowitz et al., 2017). Nevertheless, recent modeling suggests that the stratosphere is crucial for biogeochemical Hg cycling, acting as the primary pathway for $Hg^0$ exchange between hemispheres and explaining the minor interhemispheric gradient (Saiz-Lopez et al., 2025). As posited by models developed by Shah et al. (2021) and Saiz-Lopez et al. (2025), approximately 17% of the aggregate atmospheric Hg load is situated within the stratosphere, whereas a previous study reported 12% (Horowitz et al., 2017). Given that the stratosphere's mass ($9.06 \times 10^{17}$ kg) constitutes approximately 18% of the total atmospheric air mass ($5.13 \times 10^{18}$ kg; Warneck and Williams, 2012), one might infer that the fraction of mercury present in the stratosphere is comparable to the proportion of stratospheric air relative to the entire atmosphere. However, this scaling is not supported by empirical data. Aerial measurements of Hg in the troposphere and lower stratosphere reveal a steep Hg gradient around and above the tropopause with lower Hg mixing ratios in the upper atmospheric layers (Radke et al., 2007; Talbot et al., 2007; Slemr et al., 2018) linked to a larger contribution of oxidized Hg species partitioned to aerosols (Murphy et al., 1998) from the gas phase.

With respect to the tropospheric Hg budget, there is a relative consensus that the Hg load is close to 4 Gg (Saiz-Lopez et al., 2020; $3.9 \pm 1.0$, Saiz-Lopez et al., 2025; 3.8, Zhang et al., 2023b; 4.0, Shah et al., 2021; 3.9, Horowitz et al. 2017), with exceptions suggesting that it is closer to 5 - 6 Gg (Holmes et al., 2010; Zhang et al., 2025) and that anthropogenic emissions, excluding biomass burning, are approximately $2.2 - 2.6$ Gg yr$^{-1}$ (Horowitz et al., 2017; Shah et al., 2021; Zhang et al., 2023b; Geyman et al., 2024; Saiz-Lopez et al., 2025), with significant reductions across developed countries in the Northern Hemisphere observed in the near term (Custódio et al., 2022; Feinberg et al., 2024). Aircraft-based observations reveal a relatively consistent mixing ratio of $Hg^0$ within the troposphere below the tropopause, encompassing the planetary boundary layer in regions characterized by low primary emissions (Banic et al., 2003; Talbot et al., 2007; Swartzendruber et al., 2008; Weigelt et al., 2016b; Bieser et al., 2017). This uniformity supports the adoption of a steady-state procedure (Seinfeld and Pandis, 2006), where the inverse of the $Hg^0$ lifetime ($\tau_{troposphere}$) is approximated by the sum of its loss rates:

$$1/\tau_{troposphere} = 1/\tau_{rxn} + 1/\tau_{ocean} + 1/\tau_{land} + 1/\tau_{wash} + 1/\tau_{stratosphere} \qquad (1)$$

where the indices rxn, ocean, land, wash, and stratosphere are used to represent net oxidation, oceanic uptake, assimilation in land ecosystems, processes that lead to wet deposition and net transfer to the tropopause/stratosphere, respectively. As discussed subsequently, all the terms in equation 1 are subject to significant uncertainties. However, as is the case with many other trace gases, the chemical lifetime ($\tau_{rxn}$) undoubtedly plays a controlling role in determining the effective lifetime of $Hg^0$. Representing net oxidation, $\tau_{rxn}$ encompasses the duration of the initial two-step oxidation to molecular forms and the subsequent redox cycling of the photolabile fraction of these molecules in the gas phase and aerosols prior to deposition. According to the latest redox schemes (Shah et al., 2021; Castro Pelaez et al., 2022; Saiz-Lopez et al., 2025), the extent of bidirectional Hg mass flux by atmospheric chemical conversion (oxidation and reduction, 10.4 - 13.0 vs. 6.0 - 6.9 Gg yr$^{-1}$, respectively) appears to be much greater than previously assumed (e.g., 8.0 vs. 3.7 Gg yr$^{-1}$, Holmes et al. 2010), which also holds for the bidirectional fluxes (emission and depositional uptake) that occur in the gas exchange of $Hg^0$ between the atmosphere and the land and ocean. Aggregate atmospheric emissions and dry deposition have been approximated at 7.4–11.2 and 2.9–6.8 Gg $Hg^0$ yr$^{-1}$, respectively (Horowitz et al., 2017; Shah et al., 2021; Sonke et al., 2023; Zhang et al., 2023b), following a tendency of researchers toward augmenting the role of re-emission of legacy Hg from the oceans (3.7–7.2 Gg $Hg^0$ yr$^{-1}$) and gross biospheric assimilation from the atmosphere (1.2–3.2 Gg $Hg^0$ yr$^{-1}$; Horowitz et al., 2017; Yuan et al., 2019; Obrist et al., 2021; Zhou and Obrist, 2021; Feinberg et al., 2022; Wang et al., 2022; Szponar et al., 2025), respectively. To transfer $Hg^0$ from the ocean into the atmosphere, the mass transfer rate is usually parameterized via wind speed dependencies that have been tested for $CO_2$ emissions. However, recent evidence (Osterwalder et al., 2021) suggests that $Hg^0$, which is less soluble than $CO_2$, behaves similarly to $O_2$ and $N_2$, where the impact of bubble-mediated transfer is greater. As a result, ocean emissions play an increased role in the global Hg budget, accounting for approximately 60% of total Hg emissions to the atmosphere due to a wind speed dependence with a cubic power exponent instead of a quadratic power exponent in model simulations (Zhang et al., 2023b). The greater gross emissions from seawater must be balanced by gross deposition of $Hg^0$, which is, within uncertainties,

of comparable magnitude to that of $Hg^{II}$ deposition over oceans (Jiskra et al., 2021) and much higher than previously assumed (Soerensen et al., 2010). The global net exchange of $Hg^0$ from the oceans has been estimated at 0.8 - 4.0 Gg $Hg^0$ $yr^{-1}$ (Lamborg et al., 2002; Strode et al., 2007; Selin et al., 2008; Holmes et al., 2010; Chen et al., 2014; Horowitz et al., 2017; Shah et al., 2021), and the fraction of $Hg^0$ emissions resulting from $Hg^{II}$ reduction in surface waters is at an upper limit of 2.25 ± 0.89 $Hg^0$ Gg $yr^{-1}$ (Tang et al., 2025). In summary, the latter terms in Equation (1) correspond to lifetimes, the spans of which are conservatively estimated to exceed one year. However, their inverses, referring to Eq. 1, when summed, can shorten $\tau_{troposphere}$ by tens of percent beyond what the tropospheric chemical lifetime of $Hg^0$ ($\tau_{rxn}$) dictates, taking into account the inherent uncertainties. Currently, $Hg^0$ is estimated to have a $\tau_{troposphere}$ of between 3.8 and 7 months (Shah et al., 2016; Horowitz et al., 2017; Saiz-Lopez et al., 2020; Shah et al., 2021; Saiz-Lopez et al., 2025) and an average atmospheric lifetime (troposphere + stratosphere) of 8.2 months (Saiz-Lopez et al., 2025).

The sources of atmospheric $Hg^{II}$ are twofold: primary $Hg^{II}$ emissions from anthropogenic sources and atmospheric $Hg^0$ oxidation. Compared with that of $Hg^0$, the proportion of $Hg^{II}$ in anthropogenic emissions in the troposphere is not well defined. One estimate suggests that 74% of cumulative anthropogenic Hg emissions into the air are $Hg^0$ (Streets et al., 2017). Currently, East Asia has the most emissions worldwide (Streets et al., 2019); however, compelling evidence indicates that the magnitude of total Hg air emissions in this region has already peaked (Zhang et al., 2023a) and has declined in recent years (Wu et al., 2023; Feinberg et al., 2024). Nevertheless, a shift in the contributions of distinct source categories, with cement production emerging as the predominant source since 2009 in China (Wu et al., 2016), suggests an increase in the proportion of $Hg^{II}$ within Hg emissions (Zhang et al., 2016; Wang et al., 2024). Hg speciation profiles from anthropogenic sources may vary significantly across regions; for example, in continental Europe, the $Hg^{II}$ contribution from coal-fired power plants may represent less than 25% (Weigelt et al., 2016a), whereas in the tropics, artisanal and small-scale gold mining represent a substantial yet largely unconstrained source of atmospheric $Hg^0$ (Obrist et al., 2018). On average, contemporary global models employ 60 to 65% $Hg^0$ speciation in current anthropogenic emissions to the atmosphere (Horowitz et al., 2017; Shah et al., 2021; Zhang et al., 2023b). There are significant differences in the estimates of the tropospheric pool of $Hg^0$ (~3.3–4.8 Gg), separated from $Hg^{II}$ (0.1–1.0 Gg), within the above-mentioned constrained budgets for the total tropospheric Hg load in contemporary models. Having estimated the atmospheric load of $Hg^{II}$ up to 20 km at ~0.36 Gg on the basis of a synthesis of RM measurements at different heights in the atmosphere (Saiz-Lopez et al., 2020), a later contribution (Saiz-Lopez et al., 2025) involving stratospheric transport and chemistry deployed a much larger tropospheric $Hg^{II}$ pool (0.51 Gg) associated with downward transport (0.35 Gg $yr^{-1}$) of mostly photostable $Hg^{II}$ from the stratosphere ($Hg^{II}$ pool of ~0.2 Gg). The corresponding amount of (wet and dry) $Hg^{II}$ deposited on Earth's surface is 6.92 ± 1.70 Gg $yr^{-1}$, which is outside the previously estimated range of 4.8--6.8 Gg $yr^{-1}$ $Hg^{II}$ (Strode et al., 2007; Zhang et al., 2019b; Feinberg et al., 2022; Sonke et al., 2023). The effective $\tau_{troposphere}$ of $Hg^{II}$ is a few weeks (Horowitz et al., 2017), whereas $Hg^I$ species are intermediates (lifetime << 1 s) in the $Hg^0$/$Hg^{II}$ redox cycle, and their tropospheric mass is negligible (Shah et al., 2021).

$Hg^0$ in the planetary boundary layer can be consumed at a surprisingly high rate, leading to low concentration levels that approach complete depletion. Thus, chemical oxidation by reactive bromine species in a catalytic cycle ("bromine explosion", Toyota et al., 2014; Gao et al., 2022) can explain atomic Hg depletion events (AMDEs) during the polar spring after sunrise (Schroeder et al., 1998; Sommar et al., 2007; Nerentorp Mastromonaco et al., 2016) and those observed over the Dead Sea (Obrist et al., 2011) (**Fig. 2**). Br-controlled oxidation via the intermediate $^\bullet Hg^I Br$ is critical for the tropospheric oxidation of $Hg^0$, as described later in the section on gas-phase oxidation. Upon entry into the stratosphere, thermal oxidation with $Br^\bullet$ remains important for conversion to $Hg^{II}$, but with increasing altitude in the lower stratosphere, Cl chemistry plays the most important role, with OH-directed chemistry in second place at a slow net oxidation rate. With the maximum concentration of the $O_3$ layer (~25 km) as the dividing line, there is a strong dichotomy between the Hg chemistry in the upper and lower stratosphere. The former is UVC driven (Sun et al., 2022, the UV window > 30 km provides a substantial photon flux at $\lambda = 253.7$ nm, **Fig. 1b**), involving optically excited $Hg^0$ states with a strong electrophilic character. The electronic excitation of $Hg^0$ from the ground state (singlet, $^1S_0$) at 253.7 nm is spin-forbidden (leading to a triplet state, $^3P_1$ with a radiative lifetime of ~125 ns; **Fig. 1a**). The metastable dark $Hg(^3P_0)$ state cannot be produced directly from

Hg($^1$S$_0$) by light absorption but can be produced by spin-orbit relaxation of Hg($^3$P$_1$) atoms involving energy transfer to surrounding (air) molecules. In N$_2$, the equilibrium constant between the $^3$P$_0$ and $^3$P$_1$ states at room temperature (297 K) is $1.87 \times 10^3$ (Callear and Shiundu, 1987), but in the presence of O$_2$, their distribution changes profoundly. Although O$_2$ is a slightly less effective quencher for Hg($^3$P$_0$) than for Hg($^3$P$_1$) (Callear, 1987), their effective lifetimes in air at atmospheric pressure differ by only one order of magnitude (~1.1 ns and ~0.2 ns, respectively; Saiz-Lopez et al., 2022). In addition to physical quenching to the ground state, both Hg($^3$P$_0$) and

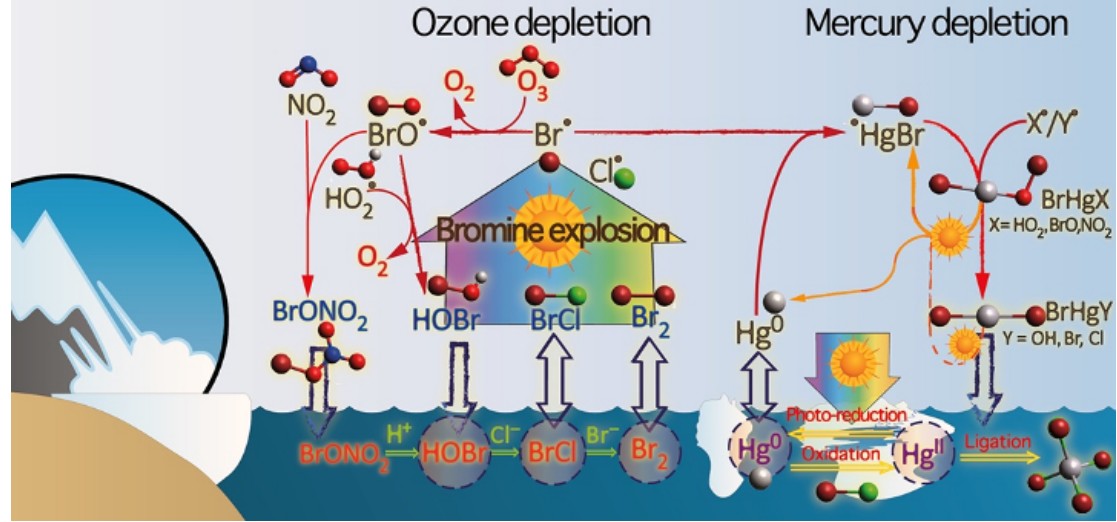

**Figure 2.** The chemistry behind bromine explosion events and related surface layer ozone and mercury depletion events.

Hg($^3$P$_1$) may undergo chemical oxidation to mercury oxide(s) (Callear et al., 1959), although metastable atoms are expected to be less reactive. The chemical conversion of excited Hg atoms by O$_2$ releases Hg$^{II}$, which can further react with more stable species, giving Hg$^0$ in the upper stratosphere a tiny lifetime against oxidation compared with that of transfer to the lower atmosphere (Saiz-Lopez et al., 2022). In the uppermost stratosphere, there appears to be access to deeper UVC (**Fig. 1b**) such that at 184.9 nm, a spin-allowed electronic transition from Hg($^1$S$_0$) to Hg($^1$P$_1$) occurs, with a light absorption cross-section nearly two orders of magnitude greater than that for the Hg($^1$S$_0$) $\rightarrow$ Hg($^3$P$_1$) transition (Morton, 2000). Like Hg($^3$P$_1$), the more energetic Hg($^1$P$_1$) reacts with O$_2$ at a rate approaching the collision frequency, but the HgO product formed in the latter case is so vibrationally hot that it promptly decays into Hg and O atoms. As a result, the chemistry of Hg($^1$P$_1$) is expected to play a minor role in the turnover of Hg in the upper stratosphere. The calculated lifetime of Hg$^0$ in the middle to upper stratosphere is altitude-dependent, ranging from a fraction to a few hundred hours (Saiz-Lopez et al., 2022), and is most comparable to that of Hg$^0$ during AMDEs. However, the underlying governing physicochemical processes are completely different.

## 4 Kinetics, thermodynamics and general chemistry

### 4.1 Fundamental kinetics and thermodynamic principles

A chemical process can be decomposed into a sequence of one or more single-step processes as elemental reactions. Elementary processes involve a transition between two atomic or molecular states, separated by a potential energy barrier that represents the activation energy. The rate of a gas–phase reaction depends on the number of collisions between the reactants and the thermodynamics of their interactions (i.e., the change in entropy, $\Delta S$, and enthalpy, $\Delta H$, upon passing through the transition state), whereas for the rate of a reaction in aqueous solution, there are a number of additional factors that can influence the rate, such as solvation, ionic strength, pH, and diffusion rates. Processes that release heat as products and increase the entropy of the system favor the reaction. The balance between $\Delta H$ and $\Delta S$ is given by the Gibbs free energy equation, where T is the absolute temperature: $\Delta G = \Delta H - T\Delta S$. If the Gibbs free energy is negative, the reaction is spontaneous from a thermodynamic perspective. The index is used to distinguish the enthalpy of reaction ($\Delta H_R$) from, e.g., the enthalpy of formation of a substance ($\Delta H_f$). We can calculate the equilibrium constant, K, using $\ln K = -\Delta G_R/RT$ and determine the ratio of the forward and reverse rate coefficients from $K = k_f/k_r$. Examples of important types of gas-phase reactions are as follows:

| Reaction order | Type | Unit |
|---|---|---|
| Unimolecular step | Thermal dissociation | $s^{-1}$ |
| Bimolecular step | Recombination | $cm^3 molecule^{-1} s^{-1}$, $L\ mol^{-1} s^{-1}$ |
| Termolecular step | Recombination assisted by a third body (M = $N_2/O_2$) | $cm^6 molecule^{-2} s^{-1}$, $L^2 mol^{-2} s^{-1}$ |

Termolecular reactions are pressure (M)-dependent at low pressures with an effective rate coefficient (k) of third order but become pressure independent at high pressures. The transition from third- to second-order behavior is known as the fall-off region. For most atmospheric reactions, we can expect that the rate coefficient is at the low-pressure limit. However, there are exceptions, which are listed in **Table 3**. While *two-body* collisions are common in the gas phase, *three-body* collisions are much less probable, and *four-body* collisions can be ignored because of their low probability. An *overall* reaction includes two or more *elementary* reactions. The temperature dependence of the rate coefficients can be fit over a relatively narrow temperature range via the empirical *Arrhenius equation*: $k(T) = A \exp(-E_a/RT)$, where $E_a$ is the activation energy and R is the gas constant. The pre-exponential factor A, a constant in the original Arrhenius equation, is weakly temperature dependent for most reactions (varying as the square root of T according to collision theory). For a wider temperature range, the modified expression $k(T) = (T/300)^n \cdot \exp(-E_a/RT)$ provides a better fit to the experimental data. If the activation energy is high enough, there is a large endothermic barrier that prevents even a reaction with a strongly negative $\Delta G_R$ from occurring at measurable rates. In select cases, the experimental data show a negative activation energy, suggesting that the reaction proceeds by the addition of reactants to form an intermediate species with excess energy that must be dissipated before decomposing into the final products. The rate constant for termolecular reactions between small molecules in the atmosphere can usually be well approximated by a combination of three parameters $k_0$ ($cm^6\ molecule^{-2}\ s^{-1}$), $k_\infty$ ($cm^3\ molecule^{-1}\ s^{-1}$) and $F_C$. The first two correspond to the low- and high-pressure limits, and $F_C$ is a form factor that describes the transition region.

$$k = \frac{k_0 \cdot k_\infty \cdot [M]}{k_\infty + k_0 \cdot [M]} F_C^{\left(1 + [\log(k_0 \cdot [M]/k_\infty)]^2\right)^{-1}} \tag{2}$$

The temperature dependence of k is expressed by parameterizing $k_0$ and $k_\infty$ as a function of temperature with the following expression:

$$k_0^T = k_0^{300}(T/300)^{-n} \quad \text{and} \quad k_\infty^T = k_\infty^{300}(T/300)^{-m} \tag{3}$$

## 4.2 Surface kinetics

Atmospheric aerosols have a high surface-to-volume ratio that concentrates most of their constituents at the surface. Furthermore, the influence of surface chemistry increases with decreasing particle size. Gas–to-particle reactions, among other heterogeneous reactions, begin with adsorption, which links molecules from the gas phase to the surface of a solid or liquid. This process can be physical, with low adsorption energy (*physisorption*, van der Waals forces) or chemical (*chemisorption*) when chemical bonding occurs as molecules approach the surface, overcome the activation energy barrier, and become reactive when the adsorbent reacts with sites on the surface. Importantly, gases and solutes adsorbed at an interface frequently exhibit physicochemical properties that diverge from their bulk properties, including reactivity and spectral shifts. Surface reaction kinetics are often expressed by the uptake probability ($\gamma$), which represents the fraction of gas collisions with a substrate surface that yield uptake or reactions. The net uptake of gas $\gamma_{net}$ is quantified in terms of conductances ($\Gamma$), which are normalized to the rate of gas surface collisions:

$$\gamma_{net}^{-1} = \Gamma_g^{-1} + \alpha^{-1} + (\Gamma_{rxn} + \Gamma_{sol})^{-1} \tag{4}$$

where $\Gamma_g$, $\Gamma_{rxn}$, and $\Gamma_{sol}$ represent the processes of gas-phase diffusion to the surface, solubility, and reaction in the bulk liquid phase, respectively, and $\alpha$ represents the (reversible) mass accommodation ("sticking") across the gas–particle interface. In addition to $\alpha$, these processes are related to the diffusion constants in gas ($D_g$) and liquid ($D_l$) phases, Henry's law coefficient ($k_H^{cp}$), and the rate constant of the first-order reaction in the condensed phase bulk (Finlayson-Pitts and Pitts, 2000). For solids, bulk diffusion is generally too slow to allow bulk solubilities or bulk kinetics to control uptake. To justify the use of the formulation of additive kinetic conductances (**Eq. 4**) to solve the continuity equation and thus to be sufficient in laboratory studies to measure the net loss of a gas over a condensed phase of known volume and surface area, it is preferable to conduct experiments at low pressure. These experiments are typically performed in a tube reactor (radius r) with fast laminar flow (FF) conditions. To vary the reaction time, a moving injector is employed to change the exposed surface length in this technique. The net flux of the gas X into the condensed phase ($J_x$) can in

this case be expressed as **Eq. 5**:

$$J_X = \frac{2 \cdot k_{obs}}{r} \left( [X]_{g,\infty} - \frac{[X]_{surf}}{k_H^{cp}} \right) \tag{5}$$

where $k_{obs}$ is the experimentally observed first-order rate coefficient and where the indices g, $\infty$ and surf represent the gas bulk and surface, respectively. In turn, $k_{obs}$ is approximately related to $\gamma_{net}$ as shown in **Eq. 6**:

$$k_{obs} = r^{-1} \left( \frac{r}{3.66 \cdot D_g} + \frac{2\gamma_{net}}{\bar{v}_X} \right)^{-1} \tag{6}$$

where $D_g$ is the diffusivity of the gas and where $\bar{v}_X$ is its mean thermal velocity. The value of $\gamma_{net}$ changes as the surface is covered by molecules and depends on the concentrations of the reactants and the reaction time. The initial phase is denoted by $\gamma_{net}^0$, whereas the steady-state phase is denoted by $\gamma_{net}^\infty$. The calculated $\gamma_{net}$ can be employed to estimate the lifetime of gas X ($\tau_X$) with respect to the reactive uptake on particles. The following formula has been applied to the uptake of aerosols with a polydisperse distribution (Mao et al., 2021; Sander, 1999; Schwartz, 1986):

$$\tau_X = \left[ \sum 4\pi r^2 \left( \frac{\Delta N}{\Delta \log r} \right) \Delta \log r \left( \frac{r}{D_g} + \frac{4}{\gamma_{net} \cdot \bar{v}_X} \right) \right]^{-1} \tag{7}$$

The uptake of the only $Hg^{II}$ species studied thus far, $HgCl_2$, follows a Hinshelwood–Langmuir mechanism (Mao et al., 2021), where $HgCl_2(g)$ must first be adsorbed to a site ($\parallel$) on the surface and then react as a surface complex with a reactive center (e.g., anions) R on the surface, forming a product released from $\parallel$, which becomes vacant again:

$$HgCl_2(g) \underset{K = k_{ads}/k_{des}}{\overset{}{\rightleftharpoons}} \parallel\text{--}HgCl_2 \overset{R}{\rightarrow} product(s) + \parallel \tag{8}$$

where K in the above equation is referred to as the Langmuir constant. Deposition velocities and partitioning coefficients constitute an empirical framework for parameterizing heterogeneous atmospheric processes. A coefficient for absorptive partitioning of compound X onto existing aerosols, $K_{gp}$, was proposed as Pankow (2007):

$$K_{gp} = \frac{[X]_p/PM}{[X]_g} \tag{9}$$

where the index gp represents gas–particle partitioning; $[X]_p$ and $[X]_g$ represent the mass concentrations of compound X in the gas phase and particle phase, respectively, in a unit volume of air; and PM represents the total mass concentration of the particles.

## 4.3 Aqueous redox equilibria

The Gibbs free energy change ($\Delta G$) presented previously is related to the electrode potential (E) as the equation:

$$\Delta G = -nFE \tag{10}$$

where n is the number of moles of electrons transferred in the reaction and F is the Faraday constant (96485 C mol$^{-1}$). The standard potentials for the mercury-mercurous-mercuric free cation couples are as follows:

$$Hg_2^{2+}(aq) + 2\,e^- \rightleftharpoons 2\,Hg^0(aq) \quad E^0 = 0.789\,V \tag{Rxn 1}$$

$$2\,Hg^{2+}(aq) + 2\,e^- \rightleftharpoons Hg_2^{2+}(aq) \quad E^0 = 0.908\,V \tag{Rxn 2}$$

$$Hg^{2+}(aq) + 2\,e^- \rightleftharpoons Hg^0(aq) \quad E^0 = 0.854\,V \tag{Rxn 3}$$

These positive potentials indicate that the reduction of $Hg^{2+}/Hg_2^{2+}$ to Hg is favored under standard conditions. It is also evident that $Hg^0$ can be oxidized to $Hg_2^{2+}(aq)$ rather than to $Hg^{2+}(aq)$ only by agents with potentials ranging from –0.79 to –0.85 V. None of the common oxidizing agents meet this narrow potential range. Therefore, in excess of the oxidizing agent, $Hg^0$ is completely oxidized to $Hg^{2+}(aq)$. Only when the excess $Hg^0$ exceeds 50% does oxidation lead to $Hg_2^{2+}(aq)$. Ligation and hydrolysis have a major impact on standard potentials, including those listed in **Rxn 1–3**. For example, $Hg(OH)_2 + 2\,e^- \rightleftharpoons Hg^0 + 2\,HO^-$, analogous to **Rxn 3**, has an $E^0$ value of 0.206 V.

## 4.4 Chemical properties of aqueous Hg[I,II]

The $Hg^{2+}$ aqua ion, $[Hg(H_2O)_6]^{2+}$, exists only in distinctly acidic aqueous solutions containing a weakly coordinating anion (e.g., $ClO_4^-$). It readily undergoes hydrolysis at pH > 1 (log $\beta_{10}$ $[Hg(OH)]^+$ = 10.3, Powell et al., 2005). Owing to its size and stable electron configuration, $Hg^{2+}(aq)$ can be easily polarized by ligands and, therefore, has the potential to form strong covalent bonds.

This property allows $Hg^{2+}(aq)$ to interact with organic C to readily form Hg–C bonds through mercury–hydrogen substitution (*mercuration*), addition (*oxy-* and *amino-mercuration, etc.),* and decarboxylation reactions. An example is aniline, which forms a covalent complex with $Hg^{2+}$ readily in aqueous solution at room temperature:

(Rxn 4)

The formation of organomercurials by mercuration in aqueous solution is generally slow because of the reduced electrophilicity of $Hg^{2+}$ caused by hydrolysis of the metal center. However, the presence of a polar solvent has little influence on other processes

of organomercurial formation, such as decarboxylation. Therefore, abiotic Hg methylation can occur in aqueous solutions with the assistance of, e.g., light carboxylic acids (Deacon et al., 1986). In the case of keto-enolic organic compounds such as acetylacetone (R = H) and malonate (R = OH), the mercuric ion can, in principle, adopt a C-bond, an O-bond or a chelate structure:

Highly toxic $CH_3Hg^+$ ($MMHg^+$) species are by far the most abundant organic Hg in the environment and are formed from inorganic $Hg^{II}$, mainly by the action of $Fe^{III}$ and $SO_4^{2-}$ reducing bacteria. In addition to monomethylation, permethylation can also occur

anaerobically (Sommar et al., 1999). $(CH_3)_2Hg$ (DMHg) is detected mainly in deep-sea waters, but by upwelling waters (Conaway et al., 2009), it may reach the mixed layer, where gas exchange with the atmosphere can occur. DMHg has also been detected in landfills (Lindberg et al., 2005; Feldmann et al., 1994), sewage gas (Sommar et al., 1999), flood plains (Wallschläger et al., 1995) and rice paddies (Wang et al., 2019c). The binding affinity of $Hg^{2+}$ to ligands is often qualitatively rationalized by Lewis acid–base theory, with the message that mercurials (type B metals) prefer soft ligands such as heavier halides and hydrochalcogenides (e.g., $I^-$

and $SH^-$, respectively) to hard ones (e.g., $OH^-$ and $F^-$). In fact, $Hg^{2+}$ is the softest metal ion that acts as a Lewis acid. The preference for low coordination numbers ($\leq 4$, typically linear two-coordinate) in $Hg^{II}$ complexes is related to the fact that relativistic effects come into play for the heaviest elements (Tossell and Vaughan, 1981). The interactions between $Hg^{2+}(aq)$ and inorganic ligands (**Table 1**) and low-molecular-weight organics (**Table 2**) are given as stability constants. The tables show that $Hg^{2+}$ also binds strongly to nitrogenous bases. Interactions with inorganic compounds, such as ammonia, are extensive and complex (Breitinger and

Brodersen, 1970). For organic nitrogen ligands, there is a parallel between the basicity of the ligand and the stability of the Hg-ligand complex (e.g., guanidine). Heterocyclic nitrogen compounds, such as histidine, also form strong complexes with mercuric ions. The hard–soft acid–base principle applies only to highly polar solvents, such as aqueous solutions, as a result of solvation (hydrolysis) effects (Riccardi et al., 2013). In the gaseous phase, an inverse relationship prevails (Riccardi et al., 2013) and can be illustrated by the fact that gaseous $Hg(OH)_2$ is a stable molecule, whereas in aqueous solution, $Hg^{2+}$ and 2 $OH^-$ can form the intermediate molecule

$Hg(OH)_2$ (Yang et al., 2020b), which eliminates $H_2O$ and precipitates solid HgO. Therefore, solid $Hg(OH)_2$ is not known (Wang and Andrews, 2005). Furthermore, in the aqueous phase, the univalent state (mercurous species) is represented by the metal-metal bound ion $Hg_2^{2+}(aq)$, which is ordinarily stable. Like $Hg^{2+}(aq)$, $Hg_2^{2+}(aq)$ is a soft Lewis acid.

Hg-ligand complexation is ubiquitous in the environment. This process involves a significant energy shift due to solvation effects, which results in a reduction in the number of solvating water molecules and an increase in the interaction between ligands/anions in

the complexes and water. Unlike the dimer cation, the discrete $Hg^{\bullet+}$ cation is paramagnetic and was detected for the first time via

electron spin resonance (Symons and Yandell, 1971). Free $Hg^{\bullet+}$ is a highly potent reducing agent with a one-electron reduction potential, $E^0(Hg^{2+}/Hg^{\bullet+})$, estimated to be well below -2.0 V (Gårdfeldt and Jonsson, 2003). However, hydrolyzed or ligated forms are less reactive (Gårdfeldt and Jonsson, 2003; Kozin and Hansen, 2013). The dissociation $Hg_2^{2+}(aq) \rightleftarrows 2\,Hg^{\bullet+}(aq)$ is considerably less significant than the disproportionation $Hg_2^{2+}(aq) \rightleftarrows Hg^0(aq) + Hg^{2+}(aq)$, with a conservative upper bound for the ratio $[Hg^{\bullet+}]/[Hg_2^{2+}]$ of $10^{-7}$ (Moser and Voigt, 1957). Free cation acidity decreases in the order of $Hg^{2+}$ (pK 3.4), $Hg_2^{2+}$ (pK 4.9) and $Hg^{\bullet+}$ (pK 5.1). $Hg_2^{2+}(aq) \rightleftarrows Hg^0(aq) + Hg^{2+}(aq)$ has an equilibrium constant of $5.5 \times 10^{-9}$ M (Moser and Voigt, 1957), which indicates that a solution of initially only $Hg_2^{2+}$ in pure water will contain only a single percent $Hg^{2+}$ in the absence of ligands that form complexes with $Hg^{2+}$. However, in the presence of ligands that form complexes with $Hg^{2+}$, disproportionation is rapid, and $Hg_2^{2+}$ is consumed. The same applies when $Hg^0(aq)$ is removed from the solution, e.g., by a gas stream. $Hg_2^{2+}$ can be a major speciation component in heavily polluted waters (Fang et al., 2024) but is insignificant in the atmosphere.

### 4.5. Chemical equilibria data

For a general complex equilibrium with $Hg^{2+}$ and the ligand L, $Hg^{2+} + q\,L + r\,H_2O \rightleftarrows \left[HgL_q(OH)_r\right]^{(2-r)+} + r\,H^+$, a stability constant $\beta_{qr}$ is defined as $\left[\{HgL_q(OH)_r\}^{(2-r)+}\right]\left[H^+\right]^r / ([Hg^{2+}][L]^q)$. When the complex is not hydrolyzed, $\beta_{qr}$ is reduced to $\beta_q = \left[\{HgL_q\}^{2+}\right] / ([Hg^{2+}][L]^q)$. For the equilibrium obtained by adding a ligand (L) to a metal complex in a stepwise manner, $K_q$ is used, which is related to $\beta_q$ by $\prod_{i=1}^q K_q$. **Tables 1** and **2** present the equilibrium constants for $Hg^{2+}$ associated with a range of inorganic and organic natural ligands, respectively, without being comprehensive. Quantitative details are available through the open-access AQUAMER database and web server dedicated to Hg, which provides direct speciation results by combining web-based interfaces with a speciation calculator, thermodynamic constant databases, and a computational chemistry toolbox for input to other software to estimate missing constants. (Lian et al., 2020).

### 4.6 Speciation of $Hg^{II}$ in atmospheric waters

$Hg^{II}$ speciation in atmospheric waters such as clouds and fog is governed by interactions with inorganic nucleophiles, low-molecular-weight organics (LMWO), and high-molecular-weight dissolved organic matter (DOM). The identified LMWOs typically make up a smaller mass fraction of the DOM in ambient cloud and fog droplets. Despite its limited abundance (0.5–3% in freshwater), sulfurized DOM exerts control over Hg cycling in terrestrial aquatic systems by forming predominantly strong HgL (logK ~21.9–23.6) and $HgL_2$ (logK ~30.1–31.6) complexes (Dong et al., 2011), where L represents functional groups with reduced sulfur. Although sulfur-containing DOM (with the elemental compositions of CHSO and CHNSO) is also relatively ubiquitous in atmospheric organic matter (AOM), sulfur is present mainly in hexavalent form, with reduced sulfur being rare (Zhao et al., 2013; Bianco et al., 2018; Jiang et al., 2022). In contrast to sub-zero valence S, which is not relevant in this context, conjugate bases of strong oxo acids that are common in AOM, such as organic nitrates and sulfates, form only weak complexes with $Hg^{II}$. Therefore, the application of speciation by equilibrium modeling on a geospherical basis to assess the atmospheric interaction between atmospheric DOM and $Hg^{II}$, as in some studies (Li et al., 2018; Zhen et al., 2023), is questionable. Bittrich et al. (2011) used pH, a confined set of inorganic ions ($NH_4^+$, $NO_3^-$, $SO_4^{2-}$ and $Cl^-$), and LMWO acids to observe dissolved $Hg^{II}$ in a study of cloud and fog water. Strongly dependent on pH, at < 5, even moderate $Cl^-$ levels can control speciation ($HgCl_2$), whereas in more alkaline waters (e.g., influenced by $NH_3$), speciation is represented by $Hg(OH)_2$, Hg(OH)Cl, and to some extent $[Hg(NH_3)_2]^{2+}$. A more realistic approach is to include DOM in speciation. In this regard, Yang et al. employed $Hg^{II}$ complexation with fulvic acids under conditions of binding to mainly O-donors (1:2 complexes with $\log\beta_{20} = 5.6$, Haitzer et al., 2002) as surrogates for AOM interaction, which, when applied, was found to dominate in the $Hg^{II}$ speciation of rainwater samples in rural and urban France (Yang et al., 2019). Studies of cloud water in eastern China revealed a marked change in acidity and other chemical compositions in the post-2008 period, where $Hg^{II}$, although the concentration was unchanged over time, in the former acidic environment was mainly bound by DOM (~79%) (Li et al., 2018), and in the latter more neutral environment, was more homogeneously distributed in addition to DOM among hydrolyzed and halide (X = Cl, Br)-bound species ($Hg(OH)_2$, HOHgX, and $HgX_2$; Zhen et al., 2023). In conclusion, until the complexation of $Hg^{II}$ with AOM is well understood, there is considerable uncertainty regarding the partitioning of aquatic $Hg^{II}$ between stable and reduction-labile complexes in the photic atmosphere.

**Table 1.** $Hg^{2+}$ – inorganic ligand complexes. Omitted in the table are, e.g., interactions with reduced sulfur (HS–, R–S–), which can be found in, e.g., Skyllberg (2011).

| Ligand/ion | | $\log \beta_{10}$ / $pKa_1$ | $\log \beta_{20}$ / $pKa_2$ | $\log \beta_{30}$ | $\log \beta_{40}$ | $\log \beta_{11}$ | Reference |
|---|---|---|---|---|---|---|---|
| Elemental mercury | $Hg^0$ | 8.46 | | | | | Hietanen and Sillén, 1956 |
| Hydroxide | $HO^-$ | 10.3 | 21.4 | | | | Powell et al., 2005 |
| Fluoride | $F^-$ | 1.6 / 3.17 | | | | | Martell and Smith, 1976 |
| Chloride | $Cl^-$ | 7.3 / < 0 | 14.0 | 14.9 | 15.5 | 18.0 | Powell et al., 2005 |
| Bromide | $Br^-$ | 9.0 / < 0 | 17.1 | 19.4 | 21.0 | | Martell and Smith, 1976 |
| Iodide | $I^-$ | 12.87 / <0 | 23.82 | 27.6 | 29.8 | | Martell and Smith, 1976 |
| Ammonia/amide | $NH_3$/$-NH_2$ | 8.8 / 9.25 | 17.4 | 18.4 | 19.1 | | Martell and Smith, 1976 |
| Carbonate | $CO_3^{2-}$ | 10.7 / 6.35 | 14.5/15.7 / 10.33 | | | 5.47[2] | Puigdomenech, 2013 |
| Cyanide | $C\equiv N^-$ | 17.0 / 9.21 | 32.8 | 36.3 | 39.0 | | Martell and Smith, 1976 |
| Thiocyanate | $N\equiv CS^-$ | 9.08 / 0.9 | 17.3 | 20.0 | 21.8 | | Martell and Smith, 1976 |
| Selenocyanate | $N\equiv CSe^-$ | — / — | — / — | 26.4 | 28.9 | | Martell and Smith, 1976 |
| Sulfite | $SO_3^{2-}$ | 13.3 / 1.81 | 24.1 / 6.97 | 26.0 | | | Martell and Smith, 1976; van Loon et al., 2001 |
| Selenite | $SeO_3^{2-}$ | / 2.35 | 12.5 / 7.94 | | | | Martell and Smith, 1976 |
| Sulfate | $SO_4^{2-}$ | 1.34 / < 0 | 2.4 / 1.99 | | | | Martell and Smith, 1976 |
| Thiosulfate | $S_2O_3^{2-}$ | / 1.6 | 29.23 | 30.6 | | | Martell and Smith, 1976 |
| Selenosulfate | $SeSO_3^{2-}$ | | 36.8 | | | | Martell and Smith, 1976 |
| Selenide | $Se^{2-}$ | 51.2[3] / 3.89 | 61.0[4] / 15.0 | | | 52.8[5] | Foti et al., 2009 |
| Nitrate | $NO_3^-$ | 0.11 / < 0 | | | | | Martell and Smith, 1976 |

**Table 2.** $Hg^{2+}$ –organic ligand complexes.

| Ligand/ion | Structure formula | $\log \beta_{10}$ / $pKa_1$ | $\log \beta_{20}$ / $pKa_2$ | $\log \beta_{30}$ | $\log \beta_{40}$ | $\log \beta_{11}$ | Reference |
|---|---|---|---|---|---|---|---|
| Oxalate |  | 9.66/10.5 / 1.25 | / 4.27 | | | | Bartels-Rausch et al., 2011; Martell and Smith, 1982 |
| Formate |  | 3.66/3.55 / 3.55 | 7.10/7.35 / — | | | | Martell and Smith, 1982 |
| Acetate |  | 3.74/4.3 / 4.5 | 7.01/8.7 / — | | | | Martell and Smith, 1982 |
| Pivalate |  | 5.92 / 5.03 | / — | | | | Martell and Smith, 1977 |
| Monochloroacetate |  | 2.95. / 2.87 | 5.61 / — | | | | Martell and Smith, 1977 |
| Trichloroacetate |  | 3.08 / 0.66 | / — | | | | Martell and Smith, 1977 |
| Glycolate |  | 3.6 / 3.83 | 7.05 | | | | Martell and Smith, 1982 |
| Mercaptoacetate |  | 34.2 / 3.43 | 42.6 / 10.1 | | | 36.3 | Cardiano et al., 2011 |

[2] $Hg^{2+}+HCO_3^- \rightleftarrows (HgHCO_3)^+$

[3] $Hg^{2+} + HO^- + HSe^- \rightleftarrows HgSe$

[4] $Hg^{2+} + 2\,HO^- + 2\,HSe^- \rightleftarrows HgSe_2^{2-}$

[5] $Hg^{2+} + HO^- + 2\,HSe^- \rightleftarrows HgHSe_2^{2-}$

| Compound | Structure | | | | | | Reference |
|---|---|---|---|---|---|---|---|
| **Metoxyacetate** | | 3.54 | 6.91 | | | | Martell and Smith, 1982 |
| | | 3.57 | | | | | |
| **Acetylacetonate** | | 12.9 | 20.1 | | | | van der Linden and Beers, 1975 |
| | | 9.00 | | | | | |
| **Phtalate** | | 4.9 | | | | | Martell and Smith, 1977 |
| | | 2.7 | 4.9 | | | | |
| **D-tartarate** | | 5.4 | | | | 15.5 | Kornev and Kardapol'tsev, 2008 |
| | | 2.8 | 3.9 | | | | |
| **Thiomalate** | | 9.94 | 18.07 | | | | Martell and Smith, 1977 |
| | | 3.3 | 4.6 | | | | |
| **Iminodiacetate** | | 13.1 | 20.2 | | | | van der Linden and Beers, 1975 |
| | | 2.65 | | | | | |
| **Dimercaprol (BAL)** | | 25.7 | 34.3 | | | | Martell and Smith, 1982 |
| | | 8.76 | 10.78 | | | | |
| **Citrate** | | 4.1 | 6.1 | 11.1 | 15.0 | 17.8 | van der Linden and Beers, 1975; Kornev and Kardapol'tsev, 2008 |
| | | 3.0 | 4.1 | | | | |
| **Ascorbate** | | 4.2 | 8.7 | | | | Kleszczewska, 1999 |
| | | 4.1 | | | | | |
| **Urea** | | 2.1 | | | | | Martell and Smith, 1977 |
| | | | | | | | |
| **Thiourea** | | 11.4 | 21.7 | 24.6 | 26.4 | | Martell and Smith, 1982 |
| | | | | | | | |
| **Selenourea** | | | 24.0 | 30.2 | 32.9 | | Martell and Smith, 1977 |
| | | | | | | | |
| **Semicarbazide** | | | 11.6 | 15.2 | | | Martell and Smith, 1977 |
| | | 3.53 | | | | | |
| **Thio-semicarbazide** | | | 22.4 | 24.8 | 25.8 | | Martell and Smith, 1977 |
| | | 1.6 | | | | | |
| **Seleno-semicarbazide** | | | 26.9 | 30.4 | 32.4 | | Martell and Smith, 1977 |
| | | 0.8 | | | | | |
| **Guanidine** | | | 24.5 | | | | Martell and Smith, 1982 |
| | | 13.5 | | | | | |
| **Ethylenediamine (en)** | | 13.85 | 23.3 | | | 10.2 | Martell and Smith, 1982 |
| | | 9.79 | 16.82 | | | | |
| **Alanine** | | 12.4 | 19.6 | | | | van der Linden and Beers, 1974 |
| | | 2.50 | 9.80 | | | | |
| **Arginine** | | 11.5 | 18.8 | | | | van der Linden and Beers, 1974 |
| | | 2.19 | 9.21 | | | | |
| **Asparagine** | | 11.4 | 18.6 | | | | van der Linden and Beers, 1974 |
| | | 2.14 | 8.85 | | | | |
| **Glycine** | | 12.2 | 19.2 | 18.82 | 31.42 | 6.98 | van der Linden and Beers, 1974 |
| | | 2.44 | 9.68 | | | | |
| **Glutamine** | | 11.5 | 18.7 | | | | van der Linden and Beers, 1974 |
| | | 2.27 | 9.16 | | | | |
| **Leucine** | | 11.9 | 19.5 | | | | van der Linden and Beers, 1974 |
| | | 2.37 | 9.62 | | | | |
| **Iso-leucine** | | 12.4 | 19.6 | | | | van der Linden and Beers, 1974 |
| | | 2.40 | 9.66 | | | | |
| **Phenylalanine** | | 12.4 | 19.6 | | | | van der Linden and Beers, 1974 |
| | | 2.21 | 9.18 | | | | |
| **Proline** | | 12.2 | 20.1 | | | | van der Linden and Beers, 1974 |
| | | 2.04 | 10.52 | | | | |

| Compound | | | | | | Reference |
|---|---|---|---|---|---|---|
| Serine | 11.7 | 19.1 | | | | van der Linden and Beers, 1974 |
| | 2.21 | 9.13 | | | | |
| Threonine | 11.7 | 18.7 | | | | van der Linden and Beers, 1974 |
| | 2.24 | 8.86 | | | | |
| Tryptophan | 13.9 | 21.4 | | | | van der Linden and Beers, 1974 |
| | 2.39 | 9.43 | | | | |
| Valine | 11.7 | 18.7 | | | | van der Linden and Beers, 1974 |
| | 2.38 | 9.59 | | | | |
| Lysine | 11.3 | 18.7 | | | | van der Linden and Beers, 1974 |
| | 2.18 | 9.18 | 10.72 | | | |
| Tyrosine | 12.3 | 19.5 | | | | van der Linden and Beers, 1974 |
| | 2.34 | 9.11 | 10.16 | | | |
| Cysteine | | 39.4 | | | | van der Linden and Beers, 1974 |
| | 1.96 | 8.48 | 10.55 | | | |
| Aspartic acid | 14.86 | 19.15 | 33.1 | | 7.4 | van der Linden and Beers, 1975; Kornev and Kardapol'tsev, 2008 |
| | 1.94 | 3.70 | 9.62 | | | |
| Glutamic acid | 12.8 | 19.2 | | | | van der Linden and Beers, 1974 |
| | 2.39 | 4.21 | 9.54 | | | |
| Histidine | 15.75 | 20.48 | 34.4 | | 7.4 | Martell and Smith, 1982 |
| | 1.79 | 6.00 | 9.16 | | | |
| Methionine | 12.8 | 19.5 | | | | van der Linden and Beers, 1974 |
| | 2.26 | 9.13 | | | | |
| Succinic acid Succinate (suc$^{2-}$) | 9.46 | 14.22 | | | 3.3 | Martell and Smith, 1982 |
| | 5.20 | 9.17 | | | | |
| 1,2,3,4-butane tetracarboxylic acid (btc$^{4-}$) | 11.61 | 17.14 | 21.5 | | 4.8 | Martell and Smith, 1982 |
| | 6.42 | 11.67 | | | | |
| 1,2,3,4,5,6-benzene hexacarboxylic acid (mlt$^{6-}$) | 18.4 | 22.6 | 25.6 | | 14.3 | Martell and Smith, 1982 |
| | 6.55 | 12.11 | | | | |
| Glutathione (H$_2$GsH) | 26.0 | 33.4 | | | 32.4 | Smith et al., 2004 |
| | 2.12 | 3.53 | 8.66 | 9.12 | | |
| Penicillamine | 18.9 | 25.0 | | | | Strand et al., 1983 |
| | 1.8 | 7.83 | | | | |

## 4.7 Chemical reaction data

The subsequent two principal sections address the chemical redox reactions in the gaseous phase (**Section 5**) and in the aqueous phase (**Section 6**). **Table 3** summarizes the gas-phase reactions, along with the rate coefficients considered most accurate and the corresponding reaction enthalpies. The reaction numbers are designated with the prefix G (G1, G2, etc.). The aqueous-phase reaction numbers are designated with prefix W and are listed in **Table 4** with the corresponding rate coefficients. Notably, several chemical reactions that are not labeled with G or W and are not assigned to **Tables 3** and **4 appear in the text**. This is particularly the case for heterogeneous (multiphase) processes (**Section 7**), such as reactive uptake and reduction on surfaces, which consequently have no prefix and follow sequential numbering throughout the document.

## 5 Gas-phase atmospheric Hg chemistry

### 5.1 Inorganic species

#### 5.1.1 Initial reactions of ground-state Hg$^0$

The homogeneous gas-phase oxidation of Hg$^0$ in the electronic ground state is limited to a few reactive species produced

photolytically. In the atmosphere, multi-step reactions involving both $Hg^I$ and $Hg^{II}$ species are crucial for Hg transformation. Atmospheric oxidation of $Hg^0$ occurs largely in the gas phase, whereas the rates of aqueous phase reactions in deliquescent aerosols are relatively slower on a unit air volume basis and are inherently limited by the low water solubility of $Hg^0$. The oxidation of $Hg^0$ vapor by closed-shell molecules, such as halogenation chemistry with reference to the gas phase, has been studied in the laboratory at various temperatures (Hall, 1992; Qu et al., 2009; Chi et al., 2009; Ariya et al., 2002; Sumner et al., 2005; Raofie and Ariya, 2004; Raofie et al., 2008; Wilcox, 2009) since Ogg et al. (1936). Direct oxidation by free halogens $(X_2)$ via the insertion reaction $Hg + X_2 \rightarrow XHg^{II}X$ is highly exothermic but very slow under atmospheric conditions due to large energy barriers (Auzmendi-Murua et al., 2014), whereas the abstraction $Hg + X_2 \rightarrow {}^\bullet Hg^I X + X^\bullet$ proceeds at significant rates only at high temperatures (Niksa et al., 2001). Thus, free halogen chemistry is important for the conversion of Hg in flue gas from power generation systems (Wilcox, 2009), such as coal-fired (CFPP) systems, but not in the atmosphere. The same applies to the $Hg + NO_2$ reaction, which is barrierless and whose pathway to $Hg^{II}(ONO)_2$ shows a negative temperature dependence (Li et al., 2022c). However, reactions that are important only in combustion and flue gas cleaning systems are outside the scope of this review.

**Hg + XO (X = O₂, NO₂ and Br)**

Although oxidation of $Hg^0$ vapor by the common atmospheric oxidants $O_3$ (Sumner et al., 2005; Hall, 1995; Pal and Ariya, 2004b; Snider et al., 2008), $BrO^\bullet$ (Raofie and Ariya, 2004; Spicer et al., 2002) and $NO_3^\bullet$ (Sommar et al., 1997; Sumner et al., 2005) has been observed in the laboratory, the identity and phase of the product(s) are in doubt. Laboratory studies of gas-phase oxidation of ppb levels of $Hg^0$ (the atmospheric level is sub-ppt) have revealed product particles in the accumulation mode, suggesting that gas-to-particle conversion takes place (Raofie and Ariya, 2004; Sun et al., 2016). These data attributed to the gas phase are almost certainly compromised by complex kinetics, including reactions at the reactor wall (Hynes et al., 2009). In all cases, gas-phase oxidation pathways leading to HgO by O atom transfer are endothermic (**Rxn G5–G7**, **Table 3**). Furthermore, the measured pre-exponential factors for the Hg–$O_3$ reaction, $\sim 10^{-16}$–$10^{-18}$ $cm^3$ $molecule^{-1}$ $s^{-1}$ (Hall, 1995; Pal and Ariya, 2004b), are much smaller than expected for simple O atom transfer (Calvert and Lindberg, 2005). Alternative $O_3$ oxidation via a weakly bound ($\sim$16 kJ $mol^{-1}$) adduct, $HgO_3$, lacks exothermic dissociation pathways (i.e., $HgO + O_2$, **Rxn G5a**) and is therefore unlikely to occur in the atmosphere. However, in laboratory experiments, $HgO_x$ can conceivably diffuse to surfaces and be deposited as solid HgO possibly via oligomerization (Tossell, 2006). Recombination of $Hg^0$ with $NO_3^\bullet$ results in weakly bound ${}^\bullet Hg^I NO_3$ ($\sim$27 kJ $mol^{-1}$), which dissociates in the lower troposphere before oxidation to $Hg^{II}$ species of the type $O_2NOHgO^\bullet$ or $O_2NOHgY$ can occur (Edirappulige et al., 2024). Abstractions (e.g., $Hg + BrO^\bullet \rightarrow HgO + Br^\bullet$ or $Hg + BrO^\bullet \rightarrow {}^\bullet Hg^I Br + O$, **Rxn G7a & b**) are endothermic, whereas direct insertion reactions (e.g., $Hg + BrO^\bullet \rightarrow BrHg^{II}O^\bullet$, **Rxn G7c**) are exothermic (–84 kJ $mol^{-1}$, Shepler 2006) but affected by large barriers (170 kJ $mol^{-1}$) and are therefore unlikely to proceed (Balabanov and Peterson, 2003). The remaining exit channels, namely, the recombination of Hg and $BrO^\bullet$ (**Rxn G7d**), leading to the formation of the geometric isomers of $BrHg^{II}O^\bullet$ (${}^\bullet Hg^I BrO$ and ${}^\bullet Hg^I OBr$), are also inconceivable, as these adducts are thought to be very weakly bound (Shepler, 2006). Stable $Hg^I$ species of this type have been reported, suggesting that $BrO^\bullet$ is important during AMDEs (Raofie and Ariya, 2004). However, other field (Wang et al., 2019a) and model (Xie et al., 2008; Ahmed et al., 2023) studies have shown that the synchronous disappearance of $Hg^0$ and $O_3$ during AMDEs can best be described solely as the action of Br atoms, with an upper limit for $k_{Hg + BrO}$ of $1 \times 10^{-15}$ $cm^3$ $molecule^{-1}$ $s^{-1}$, but that the reaction product ${}^\bullet Hg^I Br$ (**Fig. 2**) rapidly adds $BrO^\bullet$, presumably mainly to $BrHg^{II}OBr$, which is 117 kJ $mol^{-1}$ more stable than the isomer $BrHg^{II}BrO$ (Jiao and Dibble, 2017a). Despite its thermal stability, $BrHg^{II}OBr$ is rapidly photolyzed (**Figs. 2** and **4**) and therefore does not constitute a significant component of the $Hg^{II}$ pool following an AMDE.

**Table 3.** Atmospheric gas-phase reactions. Except where otherwise noted in the reference column, the thermodynamic data have been compiled from the following sources of information: CRC Handbook of Chemistry and Physics (Lide, 2008), Hepler and Olofsson (1975), Chemical Kinetics and Photochemical Data for Use in Atmospheric Studies (Burkholder et al., 2019), Guzman and Bozzelli (2019), Saiz-Lopez et al. (2020; 2022), Balabanov and Peterson (2003; 2004), and Shepler (2006). The photolysis frequencies are calculated via the global annual average photon flux in the troposphere.

| ID | Elementary reaction | Rate coefficient[6] | $\Delta H_R$ (kJ mol⁻¹)[7] | Reference | Remarks |
|---|---|---|---|---|---|
| colspan Initial reactions of ground state Hg⁰ | | | | | |
| G1 | $Hg + Br^\bullet \xrightarrow{M} BrHg^\bullet$ | $1.46 \times 10^{-32} \times (T/298)^{-1.86} \times [M]$ | −69 | Donohoue et al., 2006 | |
| G2 | $Hg + Cl^\bullet \xrightarrow{M} ClHg^\bullet$ | $2.2 \times 10^{-33} \times \exp(680/T) \times [M]$ | −104 | Donohoue et al., 2005 | |
| G3 | $Hg + HO^\bullet \xrightarrow{M} HOHg^\bullet$ | $3.34 \times 10^{-33} \times \exp(43/T) \times [M]$ | −60 to −30 | Sommar et al., 2001; Pal and Ariya, 2004a; Dibble et al., 2020; Bauer et al., 2003 | |
| G4 | $Hg \xrightarrow{h\nu} Hg(^3P_1)$ | | 471 | Saiz-Lopez et al., 2022 | Only significant in the stratosphere |
| G5 | $Hg + O_3 \xrightarrow{M} \begin{array}{l}\rightarrow HgO + O_2 \\ HgO_3\end{array}$ | | 93 | Hall, 1995; Pal and Ariya, 2004b; Hynes et al., 2009 | Thermodynamically unfeasible or adducts weakly bound |
| G6 | $Hg + NO_3^\bullet \rightarrow HgO + NO_2$ | | 195 | Sommar et al., 1997; Spicer et al., 2002; Edirappulige et al., 2024 | |
| G7 | $Hg + BrO^\bullet \xrightarrow{M} \begin{array}{l}\rightarrow HgO + Br^\bullet \\ \rightarrow BrHg^\bullet + O \\ BrHgO^\bullet \\ \rightarrow HgBrO\end{array}$ | | 219 166 114 −85 | Shepler, 2006; Raofie and Ariya, 2004 | |
| G8 | $Hg + ClOO^\bullet \rightarrow ClHg^\bullet + O_2$ | | −80 | Hynes et al., 2009 | |
| colspan Reactions of excited state Hg⁰ | | | | | |
| G9 | $Hg(^3P_1) \rightarrow Hg + h\nu$ | $8.4 \times 10^6$ | | Kramida et al., 2023 | Only significant in the stratosphere |
| G10 | $Hg(^3P_1) + N_2 \rightarrow Hg(^3P_0) + N_2$ | $5.1 \times 10^{-11} \times \exp(-701/T)$ | −21 | Callear and Shiundu, 1987 | |
| G11 | $Hg(^3P_0) + O_2 \rightarrow Hg + O_2(^3\Sigma_u^+)$ | $1.8 \times 10^{-10} \times (T/300)^{0.167}$ | −27 | Callear, 1987 | |
| G12a | $Hg(^3P_1) + O_2 \rightarrow Hg + O_2(^3\Sigma_u^+)$ | $1.3 \times 10^{-10} \times (T/300)^{-0.29}$ | −6 | Saiz-Lopez et al., 2022 | |
| G12b | $Hg(^3P_1) + O_2 \rightarrow HgO(^3\Pi) + O$ | $1.7 \times 10^{-10} \times (T/300)^{0.53}$ | −48 | | |
| G13 | $Hg(^3P_1) + H_2O \rightarrow HOHg^\bullet + H^\bullet$ | | −37 | | |
| colspan Hg^I & Hg^II bromine chemistry | | | | | |
| G14a | $BrHg^\bullet \xrightarrow{M} Hg + Br^\bullet$ | $1.6 \times 10^{-9} \times \exp(-7801/T) \times [M]$ | 69 | Saiz-Lopez et al., 2019; Dibble et al., 2012 | |
| G14b | $BrHg^\bullet \xrightarrow{h\nu} Hg + Br^\bullet$ | $4.3 \times 10^{-2}$ | | Shah et al., 2021 | |
| G15a | $BrHg^\bullet + Br^\bullet \rightarrow Hg + Br_2$ | $3.90 \times 10^{-11}$ | −124 | Balabanov et al., 2005 | |
| G15b | $BrHg^\bullet + Br^\bullet \xrightarrow{M} HgBr_2$ | $2.5 \times 10^{-10} \times (T/298)^{0.57}$ | −301 | Goodsite et al., 2004 | |
| G16 | $BrHg^\bullet + HO^\bullet \xrightarrow{M} HOHgBr$ | $2.5 \times 10^{-10} \times (T/298)^{0.57}$ | −314 | Goodsite et al., 2004 | |
| G17 | $BrHg^\bullet + Cl^\bullet \xrightarrow{M} ClHgBr$ | $3.00 \times 10^{-11}$ | −338 | Shah et al., 2021 | |
| G18 | $BrHg^\bullet + NO \rightarrow Hg + BrNO$ | $7.0^{+1.2}_{-0.9} \times 10^{-12}$ | −56 | Wu et al., 2022 | |
| G19 | $BrHg^\bullet + O_2 \rightleftarrows BrHgOO^\bullet$ | $1.4 \times 10^{-26} \times \exp(3650/T)$[8] | −30 | Wu et al., 2022 | |
| G20 | $BrHg^\bullet + NO_2 \xrightarrow{M} \begin{array}{l} BrHgONO \\ \rightarrow Hg + BrNO_2 \end{array}$ | $k_0 = (4.3 \pm 0.5) \times 10^{-30} \times (T/298)^{-(5.9\pm0.8)}$ $k_\infty = 1.2 \times 10^{-10} \times (T/298)^{-1.9}$ $F_C = 0.6$ $3.0 \times 10^{-12}$ | −176 −45 | Jiao and Dibble, 2017b; Wu et al., 2020 | |
| G21 | $BrHg^\bullet + HO_2^\bullet \xrightarrow{M} BrHgOOH$ | $k_0 = 4.3 \times 10^{-30} \times (T/298)^{-5.9}$ $k_\infty = 6.9 \times 10^{-11} \times (T/298)^{-24}$ $F_C = 0.6$ | −167 | Jiao and Dibble, 2017b | |
| G22 | $BrHg^\bullet + O_3 \rightarrow BrHgO^\bullet + O_2$ | $(7.5 \pm 0.6) \times 10^{-11}$ | −140 | Gómez Martín et al., 2022 | |
| G23 | $BrHg^\bullet + O \rightarrow Hg + BrO^\bullet$ | $(5.3 \pm 0.4) \times 10^{-11}$ | −168 | Gómez Martín et al., 2022 | |
| G24 | $BrHg^{II}O^\bullet + O \rightarrow BrHg^\bullet + O_2$ | $(9.1 \pm 0.6) \times 10^{-11}$ | −252 | Gómez Martín et al., 2022 | |
| G25 | $BrHg^{II}O^\bullet + O_3 \begin{array}{l}\rightarrow {}^\bullet Hg^I Br + O_2 + O_2 \\ \rightarrow BrHg^{II}OO^\bullet + O_2 \end{array}$ | $< 5 \times 10^{-12}$ | −143 −171 | Gómez Martín et al., 2022 | |

---

[6] The basics of gas phase kinetics have been introduced in Section 4.1. Unimolecular rate coefficients are in s⁻¹ (photolysis frequencies refer to excitation energies at lambda >290 nm calculated according to $\int \phi(\lambda, T) \cdot \sigma(\lambda, T) \cdot F(\lambda) d\lambda$, where $\phi$ is the quantum yield (≤ 1), $\sigma$ is the absorption cross section (cm² molecule⁻¹), F is the photon flux (photons cm² s⁻¹), $\lambda$ is the wavelength and T is absolute temperature), bimolecular reaction rate coefficients are in cm³ molecule⁻¹ s⁻¹ (expressed as a rate constant or as a coefficient with an Arrhenius or other type of temperature dependence) and three-body reactions according to **Eq. 2**, i.e., $k = \frac{k_0[M]}{1 + k_0[M]/k_\infty} \cdot F_C^{\left\{1 + \left[\log_{10} k_0[M]/k_\infty\right]^2\right\}^{-1}}$ are in cm⁶ molecule⁻² s⁻¹ (where [M] is the number density of air molecules, $k_0$ (cm³ molecule⁻¹ s⁻¹) is the low-pressure limiting rate coefficient, $k_\infty$ (cm⁶ molecule⁻² s⁻¹) is the high-pressure limiting rate coefficient. The temperature dependence of $k_0$ and $k_\infty$ is expressed with **Eq. 3**.

[7] Refers to the calculated enthalpy (0 K) or to the experimental ditto (298 K).

[8] Equilibrium coefficient (unit: cm³ molecule⁻¹)

| | | | | | |
|---|---|---|---|---|---|
| G26 | $BrHgO^\bullet + CH_4 \rightarrow BrHgOH + CH_3^\bullet$ | $4.1 \times 10^{-12} \times exp(-856/T)$ | −10 | Lam et al., 2019a | |
| G27 | $BrHgO^\bullet + CO \rightarrow HgBr^\bullet + CO_2$ | $6.0 \times 10^{-10} \times exp(-550/T)$ | −282 | Khiri et al., 2020 | |
| G28 | $BrHgO^\bullet + HCHO \rightarrow BrHgOH + CO + H^\bullet$ | $(4.7 - 5.5) \times 10^{-11}$[9] | −109 | Lam et al., 2019a | |
| G29 | $BrHgO^\bullet + NO \xrightarrow{M} BrHgONO$ | $2.9 \times 10^{-11}$[10] | −226 | Lam et al., 2019b | |
| G30 | $BrHgO^\bullet + NO_2 \xrightarrow{M} BrHgONO_2$ | $1.7 \times 10^{-11}$[11] | −242 | Lam et al., 2019a | |
| G31 | $BrHgO^\bullet \xrightarrow{hv} 0.56\ HgO + 0.44\ Hg + Br^\bullet + 0.44\ O$ | $2.9 \times 10^{-2}$ | | Francés-Monerris et al., 2020 | |
| G32 | $HgBr_2 \xrightarrow{hv} 0.6\ BrHg^\bullet + 1.4\ Br^\bullet + 0.4\ Hg$ | $1.5 \times 10^{-6}$ | | Shah et al., 2021 | |
| G33 | $BrHgOH \xrightarrow{hv} 0.35\ HOHg^\bullet + 0.85\ Br^\bullet + 0.5\ Hg + 0.65\ HO^\bullet + 0.15\ BrHg^\bullet$ | $1.3 \times 10^{-5}$ | | Shah et al., 2021 | |
| G34 | $BrHgCl \xrightarrow{hv} 0.6\ BrHg^\bullet + Cl^\bullet + 0.4\ Br^\bullet + 0.4\ Hg$ | | | Sitkiewicz et al., 2019 | Only significant in the stratosphere |
| G35 | $BrHgONO \xrightarrow{hv} 0.9\ BrHgO^\bullet + 0.1\ NO_2 + 0.9\ NO + 0.1\ BrHg^\bullet$ | $1.1 \times 10^{-3}$ | | Shah et al., 2021 | |
| G36 | $BrHgOOH \xrightarrow{hv} 0.31\ BrHgOH + 0.66\ Br^\bullet + 0.66\ Hg + 0.69\ HO_2^\bullet + 0.03\ BrHg^\bullet$ | $1.5 \times 10^{-2}$ | | Shah et al., 2021 | |
| **Hg$^I$ & Hg$^{II}$ chlorine chemistry** | | | | | |
| G37a | $ClHg^\bullet \xrightarrow{M} Hg + Cl^\bullet$ | $9.0 \times 10^{-11} \times exp(-8980/T) \times [M]$ | 104 | Khalizov et al., 2003; Donohoue et al., 2005 | |
| G37b | $ClHg^\bullet \xrightarrow{hv} Hg + Cl^\bullet$ | $2.5 \times 10^{-2}$ | | Shah et al., 2021 | |
| G38 | $ClHg^\bullet + Br^\bullet \rightarrow ClHgBr$ | $3.0 \times 10^{-11}$ | −307 | Shah et al., 2021 | |
| G39 | $ClHg^\bullet + Cl^\bullet \rightarrow HgCl_2$ | $3.0 \times 10^{-11}$, $(4\pm1) \times 10^{-12}$[12] | −346 | Shah et al., 2021; Taylor et al., 2005 | |
| G40 | $ClHg^\bullet + HO^\bullet \rightarrow ClHgOH$ | $3.0 \times 10^{-11}$ | −315 | Shah et al., 2021 | |
| G41 | $ClHg^\bullet + NO_2 \xrightarrow{M} ClHgONO$ | $k_0 = 4.3 \times 10^{-30} \times (T/298)^{-5.9}$<br>$k_\infty = 1.2 \times 10^{-10} \times (T/298)^{-1.9}$<br>$F_C = 0.6$ | −165 | Shah et al., 2021 | |
| G42 | $ClHg^\bullet + HO_2^\bullet \xrightarrow{M} ClHgOOH$ | $k_0 = 4.3 \times 10^{-30} \times (T/298)^{-5.9}$<br>$k_\infty = 6.9 \times 10^{-11} \times (T/298)^{-2.4}$<br>$F_C = 0.6$ | −183 | Shah et al., 2021 | |
| G43 | $ClHg^\bullet + O_3 \rightarrow ClHgO^\bullet + O_2$ | $1.0 \times 10^{-10} \times (T/300)^{0.5}$ | −151 | Saiz-Lopez et al., 2022 | |
| G44 | $ClHgO^\bullet + CH_4 \rightarrow ClHgOH + CH_3^\bullet$ | $1.5 \times 10^{-11} \times exp(-1290/T)$ | −23 | Shah et al., 2021 | |
| G45 | $ClHgO^\bullet + CO \rightarrow ClHg^\bullet + CO_2$ | $6.0 \times 10^{-11} \times exp(-550/T)$ | −275 | (Shah et al., 2021) | |
| G46 | $ClHgO^\bullet + HCl \rightarrow HgCl_2 + HO^\bullet$ | $7.9 \times 10^{-11} \times (T/300)^{-0.916}$ | −84 | Saiz-Lopez et al., 2022 | |
| G47 | $ClHgOH + HCl \rightarrow HgCl_2 + H_2O$ | $1.3 \times 10^{-12} \times (T/300)^{-1.6}$ | −122 | Saiz-Lopez et al., 2022 | |
| G48 | $ClHgO^\bullet \xrightarrow{hv} 0.673\ HgO + 0.327\ Hg + Cl^\bullet + 0.327\ O$ | | | Saiz-Lopez et al., 2022 | Only significant in the stratosphere |
| G49 | $HgCl_2 \xrightarrow{hv} 0.6\ ClHg^\bullet + 1.4\ Cl^\bullet + 0.4\ Hg$ | | | Saiz-Lopez et al., 2022 | |
| G50 | $ClHgOH \xrightarrow{hv} 0.063\ HgOH + 0.969\ Cl^\bullet + 0.906\ Hg + 0.937\ HO^\bullet + 0.031\ ClHg^\bullet$ | $1.3 \times 10^{-5}$ | | Shah et al., 2021 | |
| G51 | $ClHgONO \xrightarrow{hv} 0.9\ ClHgO^\bullet + 0.1\ NO_2 + 0.9\ NO + 0.1\ ClHg^\bullet$ | $1.1 \times 10^{-3}$ | | Shah et al., 2021 | |
| G52 | $ClHgOOH \xrightarrow{hv} 0.31\ ClHgOH + 0.66\ Cl^\bullet + 0.66\ Hg + 0.69\ HO_2^\bullet + 0.03\ ClHg^\bullet$ | $1.5 \times 10^{-2}$ | | Shah et al., 2021 | |
| **Hg$^I$ & Hg$^{II}$ HO$_x$ chemistry** | | | | | |
| G53a | $HOHg^\bullet \xrightarrow{M} Hg + HO^\bullet$ | $3.5 \times 10^{-9} \times exp(-5269/T) \times [M]$ | 30 to 60 | Saiz-Lopez et al., 2022 | |
| G53b | $HOHg^\bullet \xrightarrow{hv} Hg + HO^\bullet$ | $1.6 \times 10^{-2}$ | | Saiz-Lopez et al., 2019 | |
| G54 | $HOHg^\bullet + Br \xrightarrow{M} BrHgOH$ | $3.0 \times 10^{-11}$ | −306 | Shah et al., 2021 | |
| G55 | $HOHg^\bullet + Cl \xrightarrow{M} ClHgOH$ | $3.0 \times 10^{-11}$ | −273 | Shah et al., 2021 | |
| G56 | $HOHg^\bullet + HO^\bullet \xrightarrow{M} Hg(OH)_2$ | $3.0 \times 10^{-11}$ | −321 | Shah et al., 2021 | |
| G57 | $HOHg^\bullet + NO_2 \xrightarrow{M} HOHgONO$ | $k_0 = 3.69 \times 10^{-17} \times T^{-4.75}$<br>$k_\infty = 1.26 \times 10^{-5} \times T^{-2.04}$<br>$F_C = 0.6$ | −189 | Jiao and Dibble, 2017b | |
| G58 | $HOHg^\bullet + HO_2 \xrightarrow{M} HOHgOOH$ | $k_0 = 7.68 \times 10^{-19} \times T^{-4.25}$<br>$k_\infty = 1.24 \times 10^{-4} \times T^{-2.53}$<br>$F_C = 0.6$ | −184 | Jiao and Dibble, 2017b | |
| G59 | $HOHg^\bullet + O_3 \rightarrow HOHgO^\bullet + O_2$ | $10^{-10} \times (T/300)^{0.17}$ | −162[13] | Saiz-Lopez et al., 2022; Castro Pelaez et al., 2022 | |
| G60 | $HOHgO^\bullet + H_2O \rightarrow Hg(OH)_2 + HO^\bullet$ | $5.3 \times 10^{-12} \times exp(-2894/T)$ | −26[14] | Saiz-Lopez et al., 2022 | |
| G61 | $HOHgO^\bullet + HO_2 \rightarrow Hg(OH)_2 + O_2$ | $7.2 \times 10^{-11} \times (T/300)^{-0.436}$ | −282 | Saiz-Lopez et al., 2022 | |
| G62 | $HOHgO^\bullet + CH_4 \rightarrow Hg(OH)_2 + CH_3^\bullet$ | $4.4 \times 10^{-12} \times exp(-1650/T)$ | −40 | Saiz-Lopez et al., 2022 | |
| G63 | $HOHgO^\bullet + CO \rightarrow HOHg^\bullet + CO_2$ | $6.0 \times 10^{-11} \times exp(-550/T)$ | −252 | Edirappulige et al., 2023 | |
| G64 | $HOHgO^\bullet + HCHO \rightarrow Hg(OH)_2 + CO + H^\bullet$ | $\leq 4.7 \times 10^{-11}$ | −109 | Edirappulige et al., 2023 | |
| G65 | $HOHgO^\bullet + NO \xrightarrow{M} HOHgONO$ | $2.9 \times 10^{-11}$[15] | −226 | Edirappulige et al., 2023 | |

[9] Over the interval 333 K to 200 K.

[10] Estimated value from CH$_3$O + NO

[11] Estimated value from CH$_3$O + NO$_2$

[12] Valid for 395-573 K

[13] Based on calculation on exit-channel complexes at SC-NEVPT2 level of theory

[14] Based on $\Delta_f H^0(HOHgO^\bullet) = 63.2$ kJ mol$^{-1}$

[15] Estimated value from CH$_3$O + NO

| | | | | | |
|---|---|---|---|---|---|
| G66 | HOHgO• + NO$_2$ $\xrightarrow{M}$ HOHgONO$_2$ | $1.7 \times 10^{-11}$[16] | −242 | Edirappulige et al., 2023 | |
| G67 | Hg(OH)$_2$ + HCl → HOHgCl + H$_2$O | $1.5 \times 10^{-12} \times (T/300)^{-2.14}$ | −125 | Saiz-Lopez et al., 2022 | |
| G68 | HOHgCl + HCl → HgCl$_2$ + H$_2$O | $1.3 \times 10^{-12} \times (T/300)^{-2.14}$ | −122 | Saiz-Lopez et al., 2022 | |
| G69 | HOHgO• $\xrightarrow{h\nu}$ 0.5 HgO + 0.5 Hg + HO• + 0.5 O | | | | Tentative |
| G70 | Hg(OH)$_2$ $\xrightarrow{h\nu}$ 0.5 HOHg• + 1.5 HO• + 0.5 Hg | | | Saiz-Lopez et al., 2022 | Only significant in the stratosphere |
| G71 | HOHgONO $\xrightarrow{h\nu}$ HOHg• + NO$_2$ | $1.1 \times 10^{-3}$ | | Shah et al., 2021 | |
| G72a | HgO $\xrightarrow{M}$ Hg + O | $8.4 \times 10^{-11} \times \exp(-3150/T) \times [M]$ | 27.6 | Saiz-Lopez et al., 2022 | |
| G72b | HgO $\xrightarrow{h\nu}$ Hg + O | 0.54 | | Francés-Monerris et al., 2020 | |
| G73 | HgO + H$_2$O $\xrightarrow{M}$ Hg(OH)$_2$ <br> → HOHg• + HO• | | −240[17] <br> 40 | Saiz-Lopez et al., 2022 | |
| G74 | HgO + HCl → ClHg• + HO• | $7.1 \times 10^{-11} \times (T/300)^{-1.6}$ | −61 | Saiz-Lopez et al., 2022 | |
| G75 | HgO + O$_2$ → Hg + O$_3$ | $3.4 \times 10^{-13} \times \exp(-1993/T)$ | −300 | Saiz-Lopez et al., 2022 | |
| **Dimethylmercury chemistry** | | | | | |
| G76 | CH$_3$HgCH$_3$ + Cl• → CH$_3$HgCl + CH$_3$• <br> → CH$_3$HgCH$_2$• + HCl <br> → CH$_3$Hg• + CH$_3$Cl → CH$_3$• + Hg + CH$_3$Cl | $(2.8 \pm 0.3) \times 10^{-10}$ | −121 <br> −21 <br> −9 | Niki et al., 1983b | |
| G77 | CH$_3$HgCH$_3$ + HO• → CH$_3$HgOH + CH$_3$• <br> → CH$_3$HgCH$_2$• + H$_2$O <br> → CH$_3$Hg• + CH$_3$OH → CH$_3$• + Hg + CH$_3$OH | $(2.0 \pm 0.2) \times 10^{-11}$ | −39[18] <br> −88 <br> −177 | Niki et al., 1983a | |
| G78 | CH$_3$HgCH$_3$ + NO$_3$• → CH$_3$HgONO$_2$ + CH$_3$• → HgO +2 CH$_3$• + NO$_2$ <br> → CH$_3$HgCH$_2$• + HNO$_3$ <br> → CH$_3$Hg• + CH$_3$OH → CH$_3$• + Hg + CH$_3$OH | $3.2 \times 10^{-11} \times$ <br> $\exp((-1760 \pm 400)/T)$ | −93 <br> −98 <br> −100 | Sommar et al., 1996; Sommar et al., 1997 | |

**Hg + X• (X = Br, Cl, OH and I)**

In addition to bromine atoms (Br•), hydroxyl radicals (HO•) and, to a lesser extent, chlorine (Cl•) and possibly iodine (I•) atoms have been proposed to initiate the global gas-phase oxidation of Hg$^0$ in the ground state in the atmosphere:

$$Hg + X• \xrightarrow{M} •Hg^IX \qquad \text{(Rxn G1 – G3)}$$

The reaction rates for X = Cl (**Rxn G2**, Donohoue et al., 2005; Taylor et al., 2005) and Br (**Rxn G1**, Donohoue et al., 2006) have been determined via pulsed laser photolysis-laser-induced fluorescence (PLP-LIF) for a range of pressures and temperatures. The reaction is apparently termolecular, i.e., it shows a linear dependence on pressure (M), a slightly negative temperature dependence, and a significant difference in deactivation efficiency, with N$_2$ and He as third bodies (Donohoue et al., 2005). There are also several

experimental static studies of halogen atom reactions carried out at 1-atm pressure, which, with the exception of the studies by Horne et al. (1968) and Greig et al. (1970), have used the relative rate (RR) technique at room temperature (Ariya et al., 2002; Spicer et al., 2002; Sun et al., 2016; Guérette, 2011). The Hg + X• rate expression determined by Donohoue et al. over 0.26–0.79 atm and 243–293 K by the preferred PLP-LIF technique gives rate coefficients of $5.4 \times 10^{-13}$ (Donohoue et al., 2005) and $3.6 \times 10^{-13}$ (Donohoue et al., 2006) cm$^3$ molecule$^{-1}$ s$^{-1}$ at 298 K and 1 atm pressure in air for the Cl•- and Br•- reactions, respectively. Although the rate

constant of the chlorine atom reaction is 50% greater than that of the bromine atom reaction, the significance of the former is small in the remote troposphere, considering the low concentration of chlorine atoms. Notably, a significant increase in the apparent recombination rate coefficient of Hg + Cl• was observed in the presence of air. This result has been rationalized on the basis that secondarily formed ClO$_x$ species may also react rapidly with Hg$^0$ (Donohoue, 2008). A plausible candidate is Hg + ClO$_2$ → •Hg$^I$Cl + O$_2$ (**Rxn G8**), which is exothermic ($\Delta H_R = -80$ kJ mol$^{-1}$), but the channel has not been investigated further. Computational studies

(Shepler et al., 2007; Goodsite et al., 2004; Goodsite et al., 2012) reported a slightly larger rate constant ($\sim 10^{-12}$ cm$^3$ molecule$^{-1}$ s$^{-1}$) for the Hg + Br• reaction than the absolute PLP-LIF determination at STP. On the other hand, experimental RR studies generally yield rate constants that exceed the limit obtained from theoretical calculations, suggesting complex kinetics, including reactions at the reactor wall.

The reaction with X = OH (**Rxn G3**) was studied with PLP-LIF using an excess of Hg$^0$ over •OH (generated from the photolysis of

590 HNO$_3$ at 266 nm) without evidence of a reaction, resulting in an upper rate limit of (<) $1.2 \times 10^{-13}$ cm$^3$ molecule$^{-1}$ s$^{-1}$ (Bauer et al.,

---

[16] Estimated value from CH$_3$O + NO$_2$

[17] refers to *singlet* Hg(OH)$_2$, but is 10 kJ mol$^{-1}$ endothermic for formation of spin-conserving *triplet* Hg(OH)$_2$

[18] Assuming $\Delta_f H^0$(CH$_3$HgCl) = $\Delta_f H^0$(CH$_3$HgOH)

2003). The rate constant of Hg + $^\bullet$OH → products determined by Sommar et al. (2001) relative to cyclohexane + $^\bullet$OH → products of $8.7 \times 10^{-14}$ cm$^3$ molecule$^{-1}$ s$^{-1}$ falls below this limit at 295 K and 1 atm air, as does the temperature-resolved kinetic RR study of Pal and Ariya (2004a) extrapolated to 295 K ($\sim 1 \times 10^{-13}$ cm$^3$ molecule$^{-1}$ s$^{-1}$). External re-analysis of Pal and Ariya (Calvert and Lindberg, 2005) and Sommar et al. (Dibble et al., 2020) data via kinetic modeling revealed that $^\bullet$Hg$^I$OH under experimental conditions exclusively reacts with NO$_2$ ($^\bullet$Hg$^I$OH + NO$_2$ → HOHg$^{II}$ONO, **Rxn G57**) rather than dissociating. The temporal resolution in the PLP-LIF study also allowed a lower-bound estimate of the equilibrium constant $K_{\bullet HgOH} = [^\bullet HgOH]/([Hg][HO^\bullet])$ of $5 \times 10^{-16}$ cm$^3$ molecule$^{-1}$ (Bauer et al., 2003). This equilibrium constant has been estimated via computational studies. Recently, high-level quantum chemical calculations (Dibble et al., 2020) performed at 200–320 K yielded a $K_{\bullet HgOH}$ of $\sim 7 \times 10^{-16}$ cm$^3$ molecule$^{-1}$ at 298 K, corresponding to a $k_{13}$ of $9.5 \times 10^{-14}$ cm$^3$ molecule$^{-1}$ s$^{-1}$ at 1 atm. In contrast, Saiz-Lopez et al. (2022) reported that $K_{\bullet HgOH}$ was more than an order of magnitude smaller ($\sim 5 \times 10^{-17}$ cm$^3$ molecule$^{-1}$) at the corresponding temperature.

The kinetics of the reaction between Hg$^0$ and iodine atoms (by photolysis of CH$_2$I$_2$/CF$_3$I) were studied in an early work by monitoring $^\bullet$Hg$^I$I by absorption spectroscopy at 403–438 K (Greig et al., 1971) and in a later study by following the Hg$^0$ loss by MS at 296 K (Raofie et al., 2008). In the first study, sufficiently high $^\bullet$Hg$^I$I densities could not be generated to gauge a reaction, for which the rate constant was lower than that of the competing I$^\bullet$ + I$^\bullet$ $\xrightarrow{M}$ I$_2$ reaction of $\sim 1 \times 10^{-13}$ cm$^3$ molecule$^{-1}$ s$^{-1}$. The latter study lacks conclusive results on the Hg + I$^\bullet$ $\xrightarrow{M}$ $^\bullet$Hg$^I$I reaction but provides a limit on the rate constant for the reaction of Hg with molecular iodine vapor ($\leq 1.3 \times 10^{-19}$ cm$^3$ molecule$^{-1}$ s$^{-1}$), a reaction that lacks any atmospheric significance. A rate coefficient of $4.0 \times 10^{-13}$ (T/298)$^{-2.38}$ cm$^3$ molecule$^{-1}$ s$^{-1}$ was calculated for the Hg + I$^\bullet$ $\xrightarrow{M}$ $^\bullet$Hg$^I$I reaction at 1 atm N$_2$ and T between 180 and 400 K via Rice-Ramsperger-Kassel-Markus (RRKM) theory based on the calculated binding energy (46 kJ mol$^{-1}$) and molecular properties of $^\bullet$HgI($^2\Sigma$) (Goodsite et al., 2004).

## 5.1.2 Stability of $^\bullet$Hg$^I$X

The first step (termolecular reactions **G1–G3**), which is exothermic, produces Hg$^I$ radical intermediates ($^\bullet$Hg$^I$X), which can revert to Hg$^0$ both thermally and photolytically:

$$^\bullet Hg^I X \underset{\xrightarrow{hv}}{\xrightarrow{\Delta}} Hg + X^\bullet \qquad\qquad\qquad (Rxn\ G14a,b/G37a,b/G53a,b)$$

**Photolytic and thermal dissociation**

The first excited electronic state of $^\bullet$Hg$^I$X (designated A$^2\Pi$ for halogenated radicals) is exclusively repulsive, resulting in dissociation with visible light for wavelengths exceeding $\sim 460$ nm, where the absorption maxima are predicted at $\sim 480$, $\sim 575$, $\sim 650$, and $\sim 690$ nm for $^\bullet$Hg$^I$OH, $^\bullet$Hg$^I$Cl, $^\bullet$Hg$^I$Br, and $^\bullet$Hg$^I$I, respectively (Saiz-Lopez et al., 2019; **Fig. 3**). While the bond strengths of Hg–Cl and Hg–Br are well defined in relative terms (89.5–98.0 kJ mol$^{-1}$, Tellinghuisen et al., 1982; Shepler et al., 2005; Saiz-Lopez et al., 2022; Cremer et al., 2008 and 60.2–68.1 kJ mol$^{-1}$, Goodsite et al., 2004; Shepler et al., 2005; Cremer et al., 2008; Tellinghuisen and Ashmore, 1983, respectively), there is significant variation in the estimates of the bond strengths of $^\bullet$Hg$^I$I and $^\bullet$Hg$^I$OH, ranging from $\sim 33$ to 46 kJ mol$^{-1}$ (Goodsite et al., 2004; Shepler et al., 2005; Cremer et al., 2008; Jordan et al., 1993; Salter et al., 1986) and $\sim 23$ to 55 kJ mol$^{-1}$ (Dibble et al., 2020; Tossell, 2003; Goodsite et al., 2012; Guzman and Bozzelli, 2019; Cremer et al., 2008), respectively. Therefore, the stability of $^\bullet$Hg$^I$OH and $^\bullet$Hg$^I$I is uncertain, and it is debatable whether their thermal lifetimes in the atmosphere are long enough for these radicals to be further oxidized to mercuric species to any significant degree. The question has been raised recently since it was experimentally established that $^\bullet$Hg$^I$Br is kinetically oxidized by O$_3$ without a reaction barrier (**Rxn G22**), which was also theoretically established to be true at least for $^\bullet$Hg$^I$Cl and $^\bullet$Hg$^I$OH (**Rxn G43 & G59**, respectively, **Section 5.1.3**). A study using RRKM theory suggested that the recombination rate coefficients of Hg with I$^\bullet$ and HO$^\bullet$ are similar in the free troposphere, while the thermal dissociation of $^\bullet$Hg$^I$I gradually exceeds that of $^\bullet$Hg$^I$OH at lower temperatures (Goodsite et al., 2004). $^\bullet$Hg$^I$I is the $^\bullet$Hg$^I$X species with the shortest photolytic lifetime in the troposphere globally ($\sim 17$ s), according to computational chemistry theory (Saiz-Lopez et al., 2019). Recently, Dibble et al. estimated the HO–Hg binding energy to be 46 kJ mol$^{-1}$ using high-level quantum chemical calculations (Dibble et al., 2020). Compared with a global photolytic lifetime of just over one minute (Shah et al., 2021), the thermal lifetime of $^\bullet$Hg$^I$OH in the lower troposphere is significantly shorter (according to data from Dibble et al., 2020 $\sim 10$ ms

at the surface up to approximately ten seconds at the tropopause). For the lighter mercurous halides (excluding $^\bullet Hg^I I$), the relationship is reversed with respect to the importance of photolytic versus thermal dissociation. The lifetimes of the former channel are ~20 and ~40 s for $^\bullet Hg^I Br$ and $^\bullet Hg^I Cl$, respectively, while the thermal decay is slower for $^\bullet Hg^I Br$ above the planetary boundary layer and $^\bullet Hg^I Cl$ is much less thermally unstable.

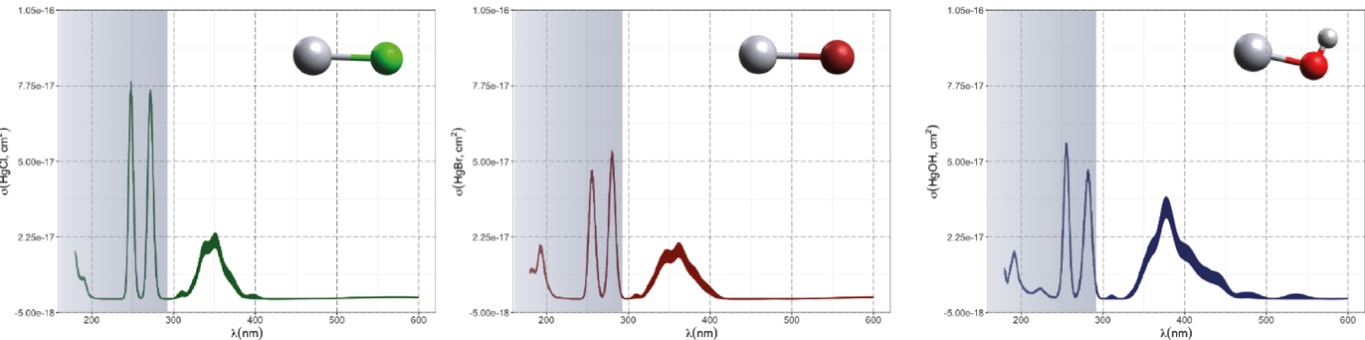

**Figure 3.** Computed absorption spectra of the atmospherically important mercurous chloride, bromide, and hydroxyl radicals. Wavelengths accessible in the troposphere are to the right of the colored area. Data from Saiz-Lopez et al. (2019).

### 5.1.3 Bimolecular reactions of $^\bullet Hg^I X$

In addition to thermal and photolytic decomposition, the fate of $^\bullet Hg^I X$ in the atmosphere is controlled by further oxidation to thermally stable mercuric species molecules. Experimental studies of the specific bimolecular reaction kinetics of $^\bullet Hg^I X$ are limited to X = Br and Cl (**Rxn G15 & G39**). Taylor et al. (2005) studied the reactions of $^\bullet Hg^I Cl$ with $Cl_2$, HCl and $Cl^\bullet$ at temperatures characteristic of post-combustion conditions. The observed reaction with free chlorine to form $HgCl_2$ was rapid (1.2 × 10$^{-11}$ cm$^3$ molecule $^{-1}$ s$^{-1}$) and temperature independent. Jiao and Dibble (2017b) used computational chemistry to determine the rate constant and product yield for reactions of $^\bullet Hg^I Br$ with abundant atmospheric $NO_2$ (**Rxn G20**) and HOO (**Rxn G21**) radicals. Analogous to the experimental $^\bullet Hg^I Cl$ study, these reactions were calculated to be rapid, with the rate constant for oxidation by $NO_2$ being approximately twice that for oxidation by HOO$^\bullet$. This theoretical study indicated that the $^\bullet Hg^I Br + NO_2$ reaction occurs along two competing channels (**Rxn G20a, b**), one proceeding via oxidative addition, resulting in BrHg$^{II}$ONO, and the other operating via reductive displacement, resulting in $Hg^0 + BrNO_2$. The dichotomy occurs because $^\bullet Hg^I Br$ ($^2\Sigma^+$) possesses a delocalized electron that spreads more equivalent spin density over the molecule ($^\bullet Hg^I Br \leftrightarrow Hg^I Br^\bullet$), whereas the spin density of the HgOH ($^1 A'$) radical is most localized on the Hg atom. A reaction with another radical center occurs for $^\bullet Hg^I OH$ when the reactant is oriented toward Hg, leading to addition, while for HgBr, reductive displacement is also possible when the collision involves the Br atom (Castro Pelaez et al., 2022). The existence of a branching ratio was also confirmed by an experimental study of the $^\bullet Hg^I Br + NO_2$ reaction by Wu et al. (2020) using PLP-LIF, who reported that the computed rate coefficients for both reduction and oxidation were greatly overestimated. This study deduced that the importance of the reductive channel increases slowly with increasing altitude from the ground level to the tropopause but is only ~10% as fast as the oxidation reaction. Wu et al. (2022) also experimentally studied the interaction between NO and $^\bullet Hg^I Br$, leading to $Hg^0 + BrNO$. $^\bullet Hg^I Br + O_2 \rightarrow BrHg^{II} OO^\bullet$ (**Rxn G19**) is slightly exothermic, while that leading to $Hg^0 + BrOO^\bullet$ is less feasible due to endothermicity. The $^\bullet Hg^I Br + O_2$ reaction is thus described by $^\bullet Hg^I Br + O_2 \rightleftarrows BrHg^{II} OO^\bullet$, with an equilibrium constant that decreases with increasing temperature (Wu et al., 2022). To the extent that BrHg$^{II}$OO$^\bullet$ can be attributed significance, it is a reservoir for $^\bullet Hg^I Br$ at low temperatures, with an upper limit of ~50% stored at 220 K. Wu et al. (2022) argued that BrHg$^{II}$OO$^\bullet$ behaves like a peroxyl radical (HOO$^\bullet$/ROO$^\bullet$) in reactions with atmospheric radicals. Recently, Saiz-Lopez et al. (2020) implied missing oxidation pathways to better reconcile their GEOS-Chem global atmospheric chemistry model simulations with field observations. Suggested by Shepler (2006) and later Lam (2019) as a potential pathway of Hg$^I$ oxidation, the Saiz–Lopez group has carried out theoretical (Saiz–Lopez et al., 2020) and experimental (Gómez Martín et al., 2022) investigations of the system $^\bullet Hg^I Br + O_3$. In addition, Castro Palaez et al. (2022) carried out theoretical calculations for rate constants and product yields, including $^\bullet Hg^I OH + O_3$. $^\bullet Hg^I X + O_3 \rightarrow XHg^{II}O^\bullet + O_2$ (**Rxn G22**, **G43** and **G59**) is highly exothermic (172 kJ mol$^{-1}$ for X = Br), proceeds without a

substantial activation barrier and is currently considered to be important for the atmospheric oxidation of $^\bullet Hg^I X$, with $XHg^{II}O^\bullet$ as a key intermediate. As a radical, $XHg^{II}O^\bullet$ is relatively thermally stable with strong Hg–O bond (333 and 294 kJ mol$^{-1}$ for X = Cl & Br, respectively; Balabanov and Peterson, 2003). Gómez Martin et al. (2022) determined the rate coefficient of the $^\bullet Hg^I Br$ + $O_3$ reaction at 295 K via a PLP-LIF system. To generate $^\bullet Hg^I Br$ (photolysis of $HgBr_2$ at 248 nm by a KrF excimer laser), the introduced $O_3$ would inevitably be photolyzed to some extent before it could react with $^\bullet Hg^I Br$. This led to complications due to the following potential chemistry:

$$^\bullet Hg^I Br + O_3 \rightarrow BrHg^{II}O^\bullet + O_2 \tag{Rxn G22}$$

$$^\bullet Hg^I Br + O(^3P) \quad \begin{array}{l} \rightarrow BrHg^{II}O^\bullet \\ \rightarrow Hg^0 + BrO^\bullet \end{array} \tag{Rxn G23}$$

$$BrHg^{II}O^\bullet + O(^3P) \rightarrow {}^\bullet Hg^I Br + O_2 \tag{Rxn G24}$$

$$BrHg^{II}O^\bullet + O_3 \quad \begin{array}{l} \rightarrow {}^\bullet Hg^I Br + O_2 + O_2 \\ \rightarrow BrHg^{II}OO^\bullet + O_2 \end{array} \tag{Rxn G25}$$

By performing experiments at different KrF laser energies and ozone concentrations and by numerical modeling of the data, Gómez Martin et al. isolated k($^\bullet HgBr + O_3$), k($^\bullet HgBr + O$) and k($BrHgO^\bullet + O$) as 7.5, 5.3 and 9.1 (all $\times 10^{-11}$ cm$^3$ molecule$^{-1}$ s$^{-1}$), respectively. They presented an upper limit for $BrHg^{II}O^\bullet + O_3$ (k $< 5 \times 10^{-12}$ cm$^3$ molecule$^{-1}$ s$^{-1}$), which was considered infeasible by theoretical calculations due to steric hindrance. Instead of leading primarily to $BrHg^{II}O^\bullet$, as is the case for the $^\bullet Hg^I Br + O_3$ reaction, $^\bullet Hg^I Br + O$ results in reductive elimination ($Hg^0 + BrO^\bullet$) for all collision geometries. $Hg^0$ is also produced in the rapid reaction between $BrHg^{II}O^\bullet + O$. In the lower atmosphere ($\leq 25$ km), the content of free O atoms is low, and therefore, its role as an oxidant is minor (Calvert et al., 2015). The energetic O($^1D$), formed primarily by photolysis of $O_3$ by UV light ($< 340$ nm), is rapidly consumed through two competitive channels: deactivation to O($^3P$) by collision with air molecules or reaction with the ubiquitous water vapor to form OH radicals. O($^3P$), also formed by the photolysis of $NO_2$ ($< 430$ nm), reacts rapidly and thermally with $O_2$ in the atmosphere to form ozone (Calvert et al., 2015). Importantly, k($^\bullet HgBr + O_3$) is more than twice as fast as k($^\bullet HgBr + NO_2$) when the experimental results are extrapolated to the atmospheric surface layer (1 atm, 295 K). The combination of a high k($^\bullet HgBr + O_3$) and the abundance of ozone relative to other radicals, such as $NO_2$ and HOO, suggests that $^\bullet Hg^I Br + O_3$ is predominant in the conversion of Hg$^I$ to Hg$^{II}$ in the atmosphere. The experimentally determined k($^\bullet HgBr + O_3$) is close to the upper limit of $1 \times 10^{-10}$ cm$^3$ molecule$^{-1}$ s$^{-1}$ estimated by Saiz-Lopez et al. (2020), which excludes steric effects. For an updated chemical mechanism in the global atmospheric model GEOS-Chem, Shah et al. (2021) used a conservative rate constant of $3 \times 10^{-11}$ cm$^3$ molecule$^{-1}$ s$^{-1}$ for the oxidation of $^\bullet Hg^I X$ with $O_3$ (X = Cl, Br and OH). By postulating k($^\bullet HgOH + O_3$) = k($^\bullet HgBr + O_3$), simulations by Shah et al. (2021) revealed that the OH-initiated pathway accounts for one-third of global Hg$^{II}$ production. In contrast, by not including $^\bullet Hg^I OH + O_3$ in their model, Dibble et al. (2020) reported that the OH-initiated channel is largely irrelevant, with only some regional significance in areas with high levels of photochemical smog. More recently, Castro Pelaez et al. (2022) compared $^\bullet Hg^I Br + O_3$ and $^\bullet Hg^I OH + O_3$ systems via computational chemistry and reported that the former has a slight tendency ($\leq 0.1\%$) to undergo reductive elimination ($Hg + BrO^\bullet + O_2$) rather than oxidation ($BrHg^{II}O^\bullet + O_2$) when the orientation of the terminal oxygen in ozone is toward the Br atom. There was no such tendency for $^\bullet Hg^I OH + O_3$. It was also found that k($^\bullet HgBr + O_3$) and k($^\bullet HgOH + O_3$) are likely similar at 298 K in the range of (6.6 - 8.5) $\times 10^{-11}$ cm$^3$ molecule$^{-1}$ s$^{-1}$. The positive covariation of $O_3$ and $^\bullet OH$, as opposed to $^\bullet Br$ and $O_3$ ($O_3$ titrates $^\bullet Br$, **Fig. 2**), suggests precedence for OH-initiated Hg oxidation in air with secondary pollutants (Rutter et al., 2012). Field observations of GOM in urban air may suggest radical-initiated Hg$^0 \rightarrow$ Hg$^{II}$ gas-phase transformation, which is claimed to be completed by certain radicals (Peleg et al., 2015; Hong et al., 2016; Edirappulige et al., 2024). An interesting case is urban Jerusalem, where episodes of elevated daytime and nighttime gaseous Hg$^{II}$ levels covary with $O_3$ (max 250 μg m$^{-3}$) and $NO_3$ (430 ng m$^{-3}$), respectively (Peleg et al., 2015). To the east of the city lies the Dead Sea basin, where effective bromine-controlled oxidation of Hg$^0$ has been observed (Tas et al., 2012). Finally, the reactivity of $^\bullet Hg^I X$ toward volatile hydrocarbons is low, as $^\bullet Hg^I X$ does not abstract a hydrogen atom from an alkane (e.g., from $CH_4$), nor does it significantly add to a double bond of an alkene (e.g., to $CH_2=CH_2$) (Dibble and Schwid, 2016).

**5.1.4 Stability of Hg$^{II}$XY**

**Photoreduction and stoichiometric yields**

Although atmospheric Hg$^{II}$ species are generally more stable than Hg$^{I}$ species are, many Hg$^{II}$ molecules are still labile, and the atmospheric pool contains mercuric species with different thermal and photolytic stabilities. Most of the atmospherically relevant gas-phase species have well-defined absorption bands in deep UV, in some cases extending into the UV-B and UV-A regions. Early theoretical studies (Strömberg et al., 1989; Strömberg et al., 1991), when knowledge of the atmospheric chemistry of Hg was rudimentary, indicated that the photoreduction of HgCl$_2$ and Hg(CN)$_2$ in actinic light at the Earth's surface was negligible, while that of Hg(OH)$_2$ and Hg(SH)$_2$ was extremely slow. The UV absorption spectra of mercuric halides are increasingly red-shifted as the halogen becomes heavier. HgCl$_2$ vapor absorbs only radiation below 240 nm (**Fig. 8a**), HgBr$_2$ absorbs mainly deep-UV light with a tiny tail (< 10$^{-19}$ cm$^2$ molecule$^{-1}$, **Fig. 8c**) into UV-B, while HgI$_2$ has significant absorption in the entire UV region (Maya, 1977; Sitkiewicz et al., 2019). However, binary compounds such as HgBr$_2$ or HgCl$_2$ do not completely dominate the atmospheric Hg$^{II}$(g) speciation. Mixed compounds such as BrHg$^{II}$Y molecules (Y= ONO, OOH, OH, OCl, OBr, etc.) and XHg$^{II}$O radicals (X = Br, OH) are also predicted to be important. Saiz-Lopez et al. (2018) computed the absorption spectra of mixed compounds and found that abundant BrHg$^{II}$Y molecules absorb in UV-B. The rapidly photolyzed Hg$^{II}$ species identified include BrHgONO (**Rxn G35**), BrHgOOH (**Rxn G36**) and BrHgOBr (with lifetimes of a few min to less than a second, **Fig. 4a-c**), with BrHgOH being comparatively long-lived (> 1 day, **Fig. 4d**) in terms of photodissociation. In their modeling study, HgCl$_2$ and Hg(OH)$_2$ were estimated to be photolytically stable in the troposphere by Shah et al. (2021), while the photolysis frequency of HgBr$_2$ was calculated to be just over an order of magnitude lower than that of BrHgOH (1.2 × 10$^{-6}$ and 1.3 × 10$^{-5}$ s$^{-1}$, respectively).

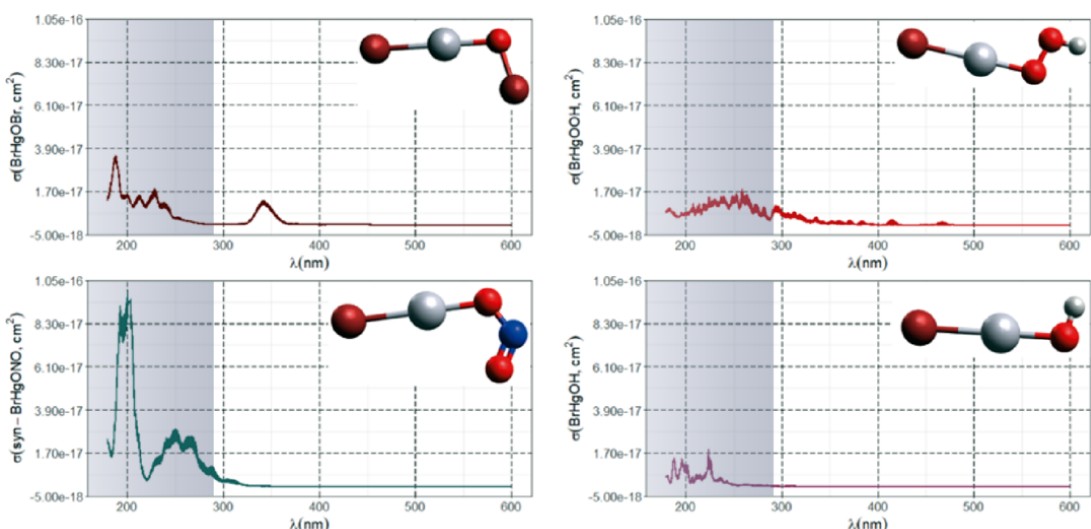

**Figure 4.** Computed absorption spectra of the atmospherically important (a) BrHgOBr, (b) BrHgOOH, (c) syn-BrHgONO and (d) BrHgOH. Wavelengths accessible in the troposphere are to the right of the colored area (Francés-Monerris et al., 2020).

The photodissociation mechanism (quantum and product yield) of BrHg$^{II}$Y has been studied using computer-aided calculations based on 2D potential energy surfaces, with the result that photodynamics lead to different channels in which the Hg-containing products can exhibit +II, +I and 0 oxidation states (Francés-Monerris et al., 2020; Lam et al., 2019b). Photolysis of BrHg$^{II}$ONO results in the formation of NO and BrHg$^{II}$O$^\bullet$ in 90% of cases, while the remainder reverts to $^\bullet$Hg$^{I}$Br and NO$_2$ (Francés-Monerris et al., 2020). Consistently, a large dominance of the photoproducts BrHg$^{II}$O$^\bullet$ + NO was predicted by the calculations of Lam et al. (2019b), in contrast to an early work by Saiz-Lopéz et al. (2018) that favored $^\bullet$Hg$^{I}$Br and Hg$^0$ formation. During the photolysis of BrHg$^{II}$OOH, the Hg–Br, Hg–O and O–O bonds can be broken, resulting in three main exit channels:

$$BrHg^{II}OOH \xrightarrow{h\nu} \begin{array}{l} Hg + Br + {}^\bullet OOH \ (66\%) \\ BrHg^{II}O^\bullet + {}^\bullet OH \ (31\%) \\ {}^\bullet Hg^I Br + O{\sim}O{-}H \ (\leq 3\%) \end{array} \qquad \text{(Rxn G36)}$$

Thus, the photodissociation of BrHg$^{II}$OOH produces Hg$^0$, $^\bullet$Hg$^{I}$Br and BrHg$^{II}$O$^\bullet$ to varying degrees (Francés-Monerris et al., 2020). In the case of BrHg$^{II}$OH, the photolytic formation of BrHg$^{II}$O$^\bullet$ is negligible, while in half of the cases (49%), reduction to elemental Hg occurs, and in the other half, $^\bullet$Hg$^{I}$Br or $^\bullet$Hg$^{I}$OH is formed, with the former being predominant (~70%) (Francés-Monerris et al.,

2020). The photolysis of $BrHg^{II}ONO$ and $BrHg^{II}OOH$ thus results in significant yields of $BrHg^{II}O^\bullet$, the radical form of $Hg^{II}$ described above as the major product of the rapid reaction between $^\bullet Hg^{I}Br$ and $O_3$. In this series of reported compositional chemical results, the only $YHg^{II}O^\bullet$ species that has been experimentally characterized is the fluorine analog that is formed along with $FOHg^{II}F$ when excited Hg atoms react with $OF_2$ (Andrews et al., 2012). Although $FHg^{II}O^\bullet$ has no atmospheric significance, its experimentally determined properties are important benchmarks for other homologs in the series. $YHg^{II}O^\bullet$ has two strong bonds (the dissociation energy for YHg-O is ~250 kJ mol$^{-1}$) and is thermally stable in the gas phase. However, $YHg^{II}O^\bullet$ is photolabile under UV–VIS light (cf. **Fig. 5b**) and decomposes photolytically along two channels. The calculated branching ratios for both Y = Cl and Br favor the formation of HgO (67% and 56%, respectively, Saiz-Lopez et al., 2022) over splitting into atoms, as shown below:

$$YHg^{II}O^\bullet \overset{h\nu}{\rightarrow} \begin{array}{l} HgO + Y^\bullet \ (56\%) \\ Hg + O + Y^\bullet \ (44\%) \end{array} \qquad \text{(Rxn G31 \& G48)}$$

For $HOHg^{II}O^\bullet$, there are no stoichiometric calculations for the photoproducts. The main product generated, HgO with a $^3\Pi$ ground state, as a monomer in the gas phase (Sun et al., 2022), possesses a weak Hg–O bond of disputed magnitude (15–30 kJ mol$^{-1}$, Tossell, 2006; Balabanov and Peterson, 2003; Cremer et al., 2008; Filatov and Cremer, 2004; Shepler and Peterson, 2003; Peterson et al., 2007), which is only ≤ 10% as strong as in $YHg^{II}O^\bullet$. HgO can be reduced to $Hg^0$ by reaction with $O_2$ and by thermal and photo-dissociation:

$$HgO \overset{M}{\rightarrow} Hg + O \qquad \text{(Rxn G71a)}$$
$$HgO \overset{h\nu}{\rightarrow} Hg + O \qquad \text{(Rxn G71b)}$$
$$HgO + O_2 \rightarrow Hg + O_3 \qquad \text{(Rxn G74)}$$

The $HgO + O_2$ reaction is exothermic but is subject to a barrier, which, using transition state theory, results in a rate coefficient of $3.4 \times 10^{-13} \exp(-1993/T)$ cm$^3$ molecule$^{-1}$ s$^{-1}$ (Saiz-Lopez et al., 2022). The enthalpy of thermal decay of HgO is only weakly endothermic and therefore favored by high temperature, with a dependence of $8.4 \times 10^{-11} \exp(-3150/T)$ cm$^3$ molecule$^{-1}$ s$^{-1}$ as calculated by RRKM theory (Saiz-Lopez et al., 2022). In addition, HgO is more photolabile than $^\bullet Hg^{I}OH$ is, with a calculated global annual mean J(HgO) of 0.54 s$^{-1}$ for the troposphere (Saiz-Lopez et al., 2018, absorption spectrum in **Fig. 5a**). These suggest that gas-phase HgO in the troposphere is highly unstable. Although the decay slows at lower temperatures and pressures as the reaction is collisionally activated, the thermal lifetime is still only about 1 ms at 250 K and 0.1 atm. Analogous to the photolysis of $Hg^{I}$ compounds, the quantum yield for the photo-dissociation of $Hg^{II}$ compounds is assumed to be unity.

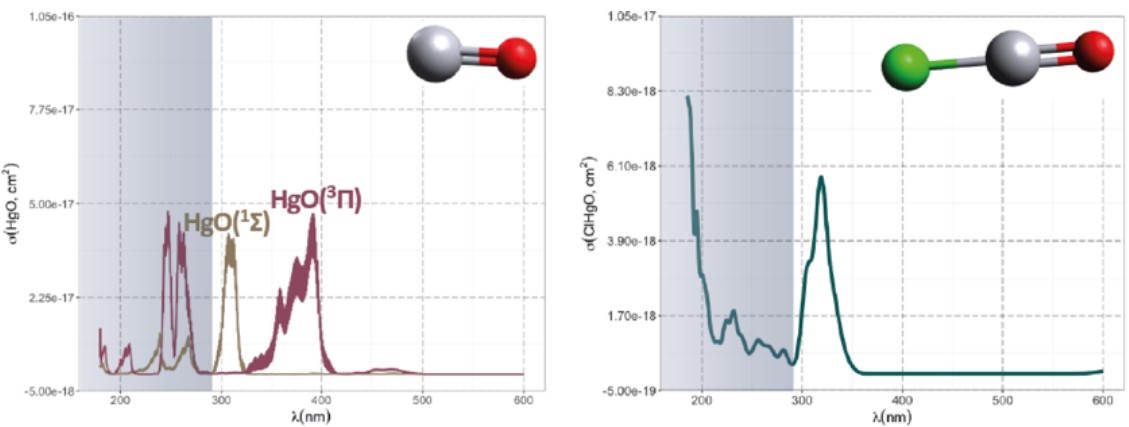

**Figure 5.** Computed absorption spectra of (a) HgO (lowlying $^3\Pi$ and $^1\Sigma$ states) and (b) ClHgO radical (Saiz-Lopez et al., 2018; Saiz-Lopez et al., 2022). Wavelengths accessible in the troposphere are to the right of the colored area.

**Thermochemistry of $YHg^{II}O^\bullet$**

Experimental data on the kinetics and mechanisms of the atmospheric chemistry of $YHg^{II}O^\bullet$ are scarce (the reaction $BrHg^{II}O^\bullet + O_3$ to $^\bullet Hg^{I}Br$ and $O_2$ has been described above, Gómez Martín et al., 2022). Initially, the focus of computer simulations was on the Br analog and its reactions. Later, the scope was expanded to include the thermochemistry of the OH and Cl analogs, which will be recapitulated below. Dibble and colleagues (Lam et al., 2019b; Khiri et al., 2020; Lam et al., 2019a) concluded that the bimolecular reaction with $CH_4$ is of primary importance for the disappearance of $BrHg^{II}O^\bullet$: $BrHg^{II}O^\bullet + CH_4 \rightarrow BrHg^{II}OH + ^\bullet CH_3$ (**Rxn G26**). Unlike $^\bullet Hg^{I}X$, $BrHg^{II}O^\bullet$ readily abstracts hydrogen atoms from saturated hydrocarbons and overcomes a modest energy barrier with a rate expression of $4.1 \times 10^{-12}$

$\times$ exp(-856/T) cm$^3$ molecule$^{-1}$ s$^{-1}$ for BrHg$^{II}$O$^\bullet$ + CH$_4$. Computational modeling suggested that BrHg$^{II}$O$^\bullet$ mimics the OH radical in terms of reaction selectivity. In addition to BrHg$^{II}$O$^\bullet$ abstract H from aliphatic hydrocarbons, it adds to the unsaturated bonds of olefins (such as the biogenic isoprene), NO (**Rxn G29**) and NO$_2$ (**Rxn G30**) and interacts with CO (**Rxn G27**). The addition of NO produces BrHg$^{II}$ONO, which is susceptible to photolytic decomposition to predominantly BrHg$^{II}$O$^\bullet$, whereas the addition of NO$_2$ promptly produces peroxynitrites of the BrHg$^{II}$OONO type, which are likely isomerized to BrHg$^{II}$ONO$_2$. Whether bromomercuric nitrate is photolabile in the troposphere is not yet known. Another source of BrHg$^{II}$OH is the reaction of BrHg$^{II}$O$^\bullet$ with aldehydes (e.g HCHO, **Rxn G28**). The pathway for the BrHg$^{II}$O$^\bullet$ + HCHO reaction bifurcates into two processes leading to different products (Khiri et al., 2020). The dominant reaction is H-abstraction, leading to BrHg$^{II}$OH and a formyl radical. The alternative route involves the addition of the oxygen atom in BrHg$^{II}$O$^\bullet$ to the carbon center in HCHO to form a methoxy radical, which eliminates a hydrogen atom unimolecularly or in the presence of O$_2$ to form a formate salt (BrHg$^{II}$OCHO). Secondary chemistry initiated by O$_2$ after the addition of BrHg$^{II}$O$^\bullet$ to a carbon double bond (such as in ethene) also involves alkoxy radicals formed after titration of the primary peroxyl radical formed by NO. The atmospheric fate of these mercuric alkoxy and alkyl peroxyl radicals (with one Hg–O bond) is similar to the general characteristics of organic oxidation in the atmosphere described in detail elsewhere (Finlayson-Pitts and Pitts, 2000). However, apart from the CH$_4$ reaction, the interaction between BrHg$^{II}$O$^\bullet$ and VOCs is considered limited in the atmosphere.

Analogous to $^\bullet$OH + CO, the reaction between BrHg$^{II}$O$^\bullet$ and CO is not a simple bimolecular reaction. However, the intermediate BrHgOCO is much less stable than HOCO with respect to the release of CO$_2$. The very weakly bound BrHgOCO promptly dissociates in $^\bullet$Hg$^I$Br + CO$_2$ (Khiri et al., 2020). The above reaction is highly exothermic (> 280 kJ mol$^{-1}$); therefore, the product $^\bullet$Hg$^I$Br can be chemically activated to the extent that it increasingly decomposes into atoms. The importance of this Hg reduction channel has been identified as difficult to constrain theoretically, as the shape of the potential energy surface is unfavorable for the application of standard kinetic simulation methods. Nevertheless, by using an inverse Laplace transformation method, Khiri et al. (2020) calculated the range for the rate coefficient at two temperatures: $(9.4 - 52) \times 10^{-12}$ cm$^3$ molecule $^{-1}$ s$^{-1}$ at 298 K and $(3.8-29) \times 10^{-12}$ cm$^3$ molecule $^{-1}$ s$^{-1}$ at 220 K. These data are the basis for the current inclusion of the YBr$^{II}$O$^\bullet$ + CO → $^\bullet$Hg$^I$Y + CO$_2$ reaction in chemical models (Shah et al., 2021; Saiz-Lopez et al., 2022), with an average expression of $6.0 \times 10^{-11} \times$ exp(-550/T) cm$^3$ molecule $^{-1}$ s$^{-1}$. With this numerical characterization, the YHg$^{II}$O$^\bullet$ + CO reaction becomes profoundly important when implemented in simulations, as it largely counteracts the effect of the $^\bullet$Hg$^I$X + O$_3$ reaction, thereby extending the predicted lifetime of Hg$^0$ in the troposphere. However, other candidates have emerged that, like CH$_4$, may react with HOHg$^{II}$O$^\bullet$ to form the stable Hg(OH)$_2$ molecule, namely, water vapor. The reaction HOHg$^{II}$O$^\bullet$ + H$_2$O → Hg(OH)$_2$ + $^\bullet$OH (**Rxn G60**) is nearly thermoneutral due to the stability of Hg(OH)$_2$ ($\Delta H_f = -226$ kJ mol$^{-1}$; Wang and Andrews, 2005) and Saiz-Lopes et al. (2022) give a temperature-dependent rate constant expression of $5.3 \times 10^{-12} \times$ exp($-2894$/T) cm$^3$ molecule$^{-1}$ s$^{-1}$ without further details. Since both the calculated HOHg$^{II}$O$^\bullet$ + H$_2$O rate coefficient and the H$_2$O(g) mixing ratio vary considerably across the troposphere, the HOHg$^{II}$O$^\bullet$ loss due to this channel may largely exceed or fall below the more monotonic rate of hydrogen abstraction by HOHg$^{II}$O$^\bullet$ from CH$_4$, depending on the circumstances. The fate of HOHg$^{II}$O$^\bullet$ is thus influenced by several exit channels (Edirappulige et al., 2023), none of which have been investigated experimentally. Particularly, the uncertainty of the CO and H$_2$O reactions makes it difficult to determine the importance of OH-initiated oxidation to the atmospheric Hg$^{II}$ pool.

**Can mercury species nucleate in the atmosphere?**

While Hg$^0$ vapor has been observed to nucleate homogeneously in laboratory experiments conducted under high pressures (Martens et al., 1987), neither Hg$^0$ atoms nor GOM species, which are molecular rather than ionic entities, have a vapor pressure that is sufficiently low and a concentration that is sufficiently high in the atmosphere to nucleate new particles by simple condensation (Murphy et al., 1998). However, the concerted action of a foreign gas-phase precursor (e.g., amines, highly oxygenated organics, sulfuric, nitric, and iodic acids, etc.; Lehtipalo et al., 2025; He et al., 2021) or heterogeneous condensation on pre-existing nuclei of subcritical or critical size may result in the transfer of GOM species to aerosols (Ariya et al., 2015). Measurements of individual aerosol particles have shown that a significant portion of the aerosols present in the lowest kilometers of the stratosphere contain small yet measurable amounts of Hg$^{II}$. Interestingly, Hg$^{II}$ is empirically correlated with bromine and iodine in these organic-sulfate-type particles and has the highest relative concentrations in the

stratosphere near the tropopause. However, Hg$^{II}$ is rarely observed in the relatively pure sulfuric acid particles characteristic of the main stratospheric aerosol (Junge) layer (Murphy et al., 2006). While bromine and iodine aerosols are also observed throughout the troposphere, no Hg can be detected in these aerosols (Murphy et al., 2006). Both Br and I, with oceans as the primary sources, are injected into the stratosphere, where they account for most of the ozone depletion caused by halogens (Koenig et al., 2020). It is challenging to determine

whether there is a causal mechanistic relationship and, if so, what can explain the observed correlation between aerosol Hg, Br, and I. Nevertheless, a plethora of clues can be utilized to assemble a coherent narrative. First, the combination of Br• (**Rxn G14a**) and O$_3$ (**Rxn G22**) constitutes a significant oxidation pathway for Hg$^0$ to Hg$^{II}$. However, as mentioned above, there is no firm evidence that this reaction pathway is relevant when I• is a substitute for Br•. Second, the gas phase system I• + O$_3$ + H$_2$O has been identified as a substantial precursor of particle nucleation (as iodine oxoacids) and growth that is highly important within marine (Sipilä et al., 2016) and stratospheric

(Koenig et al., 2020) environments. Third, the condensed phases Br$^-$ and I$^-$ act as robust complexing ligands (**Table 1**) for the GOM to partition into the aerosol, thereby impeding its recycling back to the gas phase. Presumably, the fundamentals are similar for a particle formation event observed in the context of the polar spring partial AMDE in East Antarctic pack ice by Humphries et al. (2015), where the formation of 3 nm particles lags the phase of gaseous Hg$^0$ loss in the air mass. Observations over a decade in the Canadian high Arctic region clearly show that PBM transmitted by KCl-coated denuders dominated Hg$^{II}$ fractionation over the GOM during the early

period of AMDEs, where the highest frequencies of depleted Hg$^0$ occurred between –45 and –40 °C, whereas during the late period of higher temperatures and lower particulate concentrations (AMDEs then occurred most frequently between –25 and –20 °C), Hg$^{II}$ fractionation shifted to a clear dominance of the GOM (Steffen et al., 2014). The KCl-denuder technique used cannot selectively separate nano- to submicron-sized mercuric halide clusters completely from GOM (Ghoshdastidar and Ariya, 2019), which, along with the other nonsystematic bias of the method previously mentioned, makes separation into Hg$^{II}$ fractions tentative. The measurement methodology

deficiencies make establishing empirical gas–particle partitioning schemes highly uncertain (Amos et al., 2012; Rutter and Schauer, 2007b, a). This complicates the assumption and verification of model parameterization, which relies on accurate atmospheric concentration measurements. For example, the release of Hg$^{II}$ from aerosols into the gas phase is assumed to be entirely in the form of the tropospherically stable HgCl$_2$ molecule (Shah et al., 2021). Coupling an oxidized Hg vapor source or a reactor where oxidized Hg is formed by gas-phase oxidation of Hg$^0$ to particle characterization instruments (such as scanning mobility or optical particle sizers)

provides conclusive evidence that mercuric halide molecules readily form clusters that undergo particle growth (Ghoshdastidar and Ariya, 2019). In experimental studies of the vapor-phase oxidation of volatile Hg forms Hg$^0$ and (CH$_3$)$_2$Hg, aerosol-phase products have been detected (Raofie et al., 2008; Raofie and Ariya, 2004; Sun et al., 2016; Niki et al., 1983a; Niki et al., 1983b). For example, using a scanning mobility particle sizer, Sun et al. (2016) found that, well below the saturation pressure of HgX$_2$, reaction products from X (Cl and Br)-initiated Hg$^0$ vapor oxidation began to generate particles that grew from the Aitken nuclei range (few tens of nm) into the

accumulation range (> 100 nm) over the course of a few hours (**Fig. 6**). **Fig. 7** summarizes the main elements of the Hg gas phase chemistry in the troposphere.

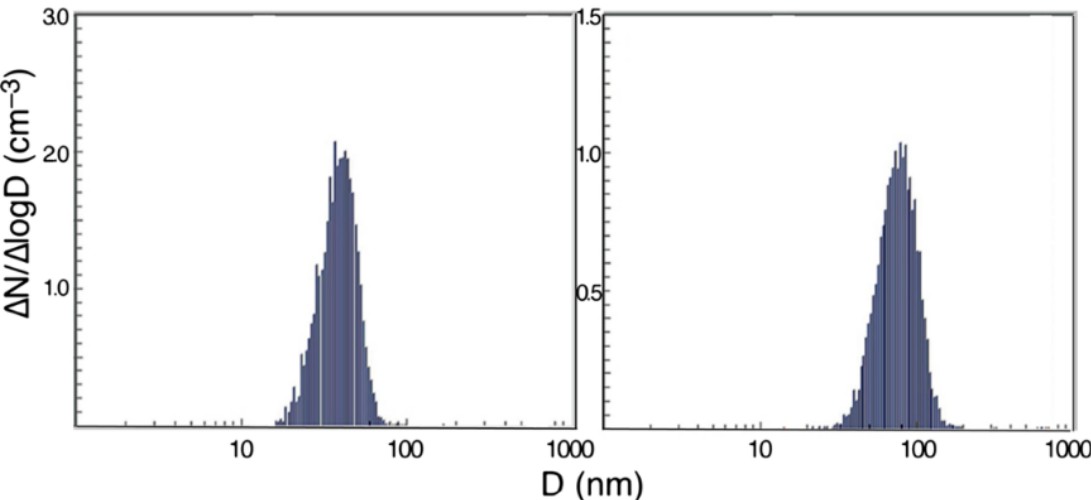

**Figure 6.** Particle growth of the reaction products from the halogen atom (X = Cl, Br) induced oxidation of Hg$^0$ vapor studied after the same degree of Hg$^0$ conversion (~75%, 5–8 ppb) but at different reaction times a and b (~45 min, Hg$^0$ + Br and ~4 h, Hg$^0$ + Cl, respectively). Adopted from Sun et al. (2016).

## 5.1.5 Chemical transformation of Hg in the lower stratosphere

In the lower stratosphere, chlorine atoms and hydroxyl radicals initiate most of the oxidation of $Hg^0$. This is because the concentrations of these species increase with altitude, and the channels in which they are contained produce more photostable products, such as $Hg(OH)_2$ and $HgCl_2$ (**Fig. 8a, b**). The prediction of these model calculations that $Hg^0$ converts to long-lived (photostable) oxidized forms and thus leads to a higher RM/TAM ratio is supported by hundreds of profile measurements made with an Airbus 340-600 passenger aircraft in intercontinental traffic as an upper troposphere-lowermost stratosphere observatory (Slemr et al., 2018). In addition to the frequently observed higher $RM/Hg^0$ ratios, a steep decrease in the $Hg^0$ mixing ratio occurs when crossing the tropopause. In the stratosphere, the latter ratio decreases to 0.25–0.7 ng m$^{-3}$ (STP), measured up to an altitude of 4 km (Slemr et al., 2018). The results of both studies above show a more than tenfold increase in the lifetime of $Hg^0$ in the lower stratosphere compared with that in the troposphere. The chemical lifetime of $Hg^0$ increases and approaches 10 years as the concentration in the lower stratosphere bound by the ozone layer increases. (Saiz-Lopes et al. 2025).

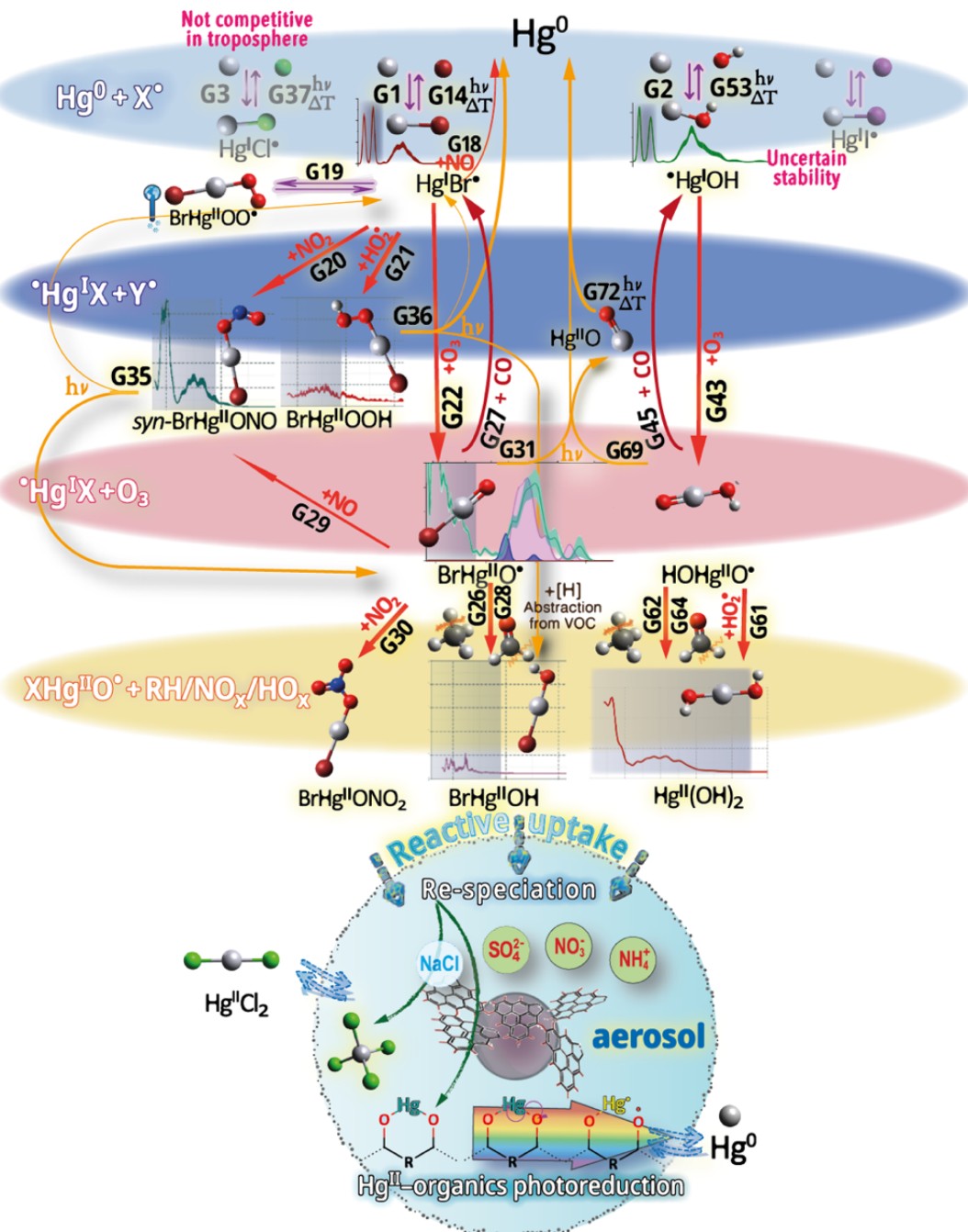

**Figure 7.** Outline of tropospheric gas-phase Hg chemistry. The reactions of the type $Hg + XO^•$ ($X = O_2$, Br and $NO_2$) directly leading to mercuric species are, for the (varying) reasons given in the text, impossible processes in the homogeneous gas phase. In the troposphere, gas-phase oxidation of $Hg^0$ in termolecular reactions is initiated by a few radicals in which Br atoms and, with some uncertainty, OH radicals have the

greatest importance, leading to the formation of thermally and photolytically labile mercurous radical species (i.e., $\bullet Hg^I Br$ and $\bullet Hg^I OH$). These
$Hg^I$ species are further oxidized to $Hg^{II}$ not only by ozone but also by radicals (in descending order of abundance), such as $NO_2 > HO_2 > BrO \approx$
$OH$. NO cannot efficiently oxidize $Hg^I$ to $Hg^{II}$ but instead induces thermal reduction, e.g., $\bullet Hg^I Br + NO \rightarrow Hg + BrNO$. As $O_3$ is a closed shell
species, it directly oxidizes $\bullet Hg^I Br/\bullet Hg^I OH$ to mercuric radical species $YHg^{II}O^\bullet$; for example, $HOO^\bullet$ and $BrO^\bullet$ are added to linear mercuric
molecules (e.g., $BrHg^{II}OOH$) that are photolytically labile while those resulting from, e.g., $NO_2$, $\bullet OH$ and $Br^\bullet$ are more photostable. The
photolysis of many of the major thermally more stable $Hg^{II}$ species, such as syn-$BrHg^{II}ONO$, $BrHg^{II}OOH$ and $BrHg^{II}OH$, leads to several
species-specific photoproducts (potentially $Hg^0$, $Hg^I$ species or $YHg^{II}O^\bullet$) with various yields (**Table 3**). The remarkably thermally stable YHgO
radical exhibits versatile thermochemistry, such as abstracting hydrogen from VOCs, adding to double bonds and being reduced by CO. Some
of its bimolecular reactions, such as with $CH_4$, directly form fairly stable $Hg^{II}$ compounds such as $Hg(OH)_2$, and $BrHg^{II}OH$. When these
encounter hydrometeors, they dissolve and are re-speciated by rapid equilibrium with major aqueous ligands. This leads to the formation of
strong complexes, e.g., by $Cl^-$ to chloromercurates $HgCl_2$, $HgCl_3^-$ and $HgCl_4^{2-}$. Thus, molecular $HgCl_2$ is released into the gas phase when the
particle dries. $HgCl_2$ is completely photostable and is enriched in the troposphere (a major $Hg^{II}$ species), with dry and wet deposition as the only
sink processes.

### 5.1.6 Chemical transformation of Hg in the upper stratosphere

The presence of UVC radiation above the ozone layer maximum opens completely new reaction pathways for redox cycling of
stratospheric Hg. New insights into its conceptual stratospheric chemistry (Saiz-Lopez et al., 2022; 2025) and associated anomalous
isotope fractionation (Sun et al., 2022; Fu et al., 2021) have been presented. The gas-phase oxidation of $Hg^0$ is rapid ($10^3$–$10^4$ times
faster than in the troposphere, Saiz-Lopez et al., 2022) and is driven entirely by the oxidation of electronically excited Hg atoms by
one of the major constituents of air, $O_2$.

### $Hg^0(^3P)$ reaction with elemental oxygen

Already involved in the discovery of the element oxygen toward the end of the 18$^{th}$ century, the chemistry of the system of $Hg + O_2$
has exhibited intricate complexity. These early observations, made independently in northern and western Europe, address an
important aspect of the thermochemistry of the system. A direct combination of liquid Hg and $O_2$ occurs just below the boiling point
of Hg to form HgO, but the reaction is reversed above 400 °C. While the reaction of ground-state Hg vapor ($Hg(^1S)$) with $O_2$ is
negligibly slow (Hall et al., 1995), deep UV light excitation of singlet to triplet Hg atoms ($Hg(^3P)$) leads to significant homogeneous
reactions with $O_2$. In contrast, further excitation of $Hg(^1P_1)$ in blue and subsequent reaction with $O_2$, as discussed above, is unlikely
to result in the net formation of mercury oxides. The gas-phase reactions of $Hg(^3P)$ have been studied in the laboratory since 1922
(Cario and Franck, 1922). In particular, $O_2$/air has been used as a route for ozone synthesis since the mid-1920s (Dickinson and
Sherrill, 1926). While larger quantities of ozone are produced by Hg photochemistry (photo-sensitization), the elemental vapor is
oxidized more slowly, resulting in the deposition of a yellow–brown film of solid HgO on the reactor walls downstream of the
irradiation zone (Volman, 1953). However, the $Hg(^3P) + O_2$ mechanism is unclear because of the controversy regarding the molecular
intermediates, and whether there is a direct route from $Hg(^3P)$ to gaseous HgO or oxidation starting from the $Hg(^1S)$ state remains
undetermined (Callear et al., 1959; Volman, 1953; Hippler et al., 1978; Morand and Nief, 1968). The dark homogeneous reaction
$Hg(^1S) + O_3 \rightarrow HgO + O$, supported by early researchers (Callear et al., 1959; Volman, 1953; Pertel and Gunning, 1959) as driving
the oxidation in the photochemical experiments, can now be rejected for the reasons discussed above in **Section 5.1.1**. Considering
more recent results (Wang and Andrews, 2005; Hall, 1995), e.g., those obtained by refined computational chemistry, the following
mechanism seems to be the most plausible:

$$Hg(^1S_0) + h\nu\ (\lambda=253.7\ nm) \rightarrow Hg(^3P_1) \hspace{4cm} \text{(Rxn G4)}$$
$$Hg(^3P_1) \rightarrow Hg(^1S_0) + h\nu\ (\lambda=253.7\ nm) \hspace{3.5cm} \text{(Rxn G9)}$$
$$Hg(^3P_1) + N_2 \rightarrow Hg(^3P_0) + N_2 \hspace{4.5cm} \text{(Rxn G10)}$$
$$Hg(^3P_1) + O_2 \xrightarrow{M} HgO_2^* \xrightarrow{M} OHgO \begin{array}{l} \rightarrow Hg(^1S_0) + O_2(^3\Sigma_u^+) \\ \rightarrow HgO(^3\Pi) + O(^3P) \end{array} \hspace{1.5cm} \text{(Rxn G12a,b)}$$
$$Hg(^3P_0) + O_2 \rightarrow Hg(^1S_0) + O_2(^3\Sigma_u^+) \hspace{3.5cm} \text{(Rxn G11)}$$
$$O_2(^3\Sigma_u^+) + O_2 \rightarrow O_3 + O(^3P) \hspace{4.2cm} \text{(Rxn 5)}$$
$$O_2 + O(^3P) \rightarrow O_3 \hspace{5.5cm} \text{(Rxn 6)}$$

Photoexcitation (**Rxn G4**) has been discussed, but its reverse (**Rxn G9**), i.e., the spontaneous emission of a photon that brings $Hg(^3P_1)$
to the ground state, is spin-forbidden, and the radiative lifetime is relatively long (0.12 μs corresponding to $k_9 = 8.4 \times 10^6$ s$^{-1}$). The
quenching of $Hg(^3P)$ states (i.e **Rxn G9 & G10 – G12a**) for several gases has been studied, with $Hg(^3P_1)$ atoms being 21.3 kJ mol$^{-1}$
more energetic than $Hg(^3P_0)$ atoms. The two main constituents of air play different roles in the quenching process, with $N_2$ almost
exclusively deactivating $Hg(^3P_1)$ to $Hg(^3P_0)$ with $k_{G10} = 5.1 \times 10^{-11} \exp(-701/T)$ cm$^3$ molecule $^{-1}$ s$^{-1}$, while $O_2$ quenches both $Hg(^3P_1)$

and Hg($^3P_0$) directly to Hg($^1S_0$) with $k_{G12a}$ and $k_{G11}$ values of $1.3 \times 10^{-10}$ (T/300)$^{-0.29}$ and $1.8 \times 10^{-10}$ (T/300)$^{0.167}$ cm$^3$ molecule$^{-1}$ s$^{-1}$, respectively. In the stratosphere (T = 240 K), the $k_{G12a}/k_{G10}$ ratio is ~50, suggesting that O$_2$ is a much better physical quencher than N$_2$, which is true throughout the atmosphere. Of primary interest here, however, is the spin-conserving **Rxn G12b**, which allows the oxidation of Hg and is overall nearly thermoneutral (exothermic by ~6 kJ mol$^{-1}$), yielding HgO ($^3\Pi$) with low vibrational energy, as noted by Saiz-Lopez et al. (2022), which is important for increasing the lifetime of this weakly bound molecule. First tentatively identified as an intermediate in a low-temperature UVC-irradiated matrix consisting of Hg, O$_2$ and H$_2$ yielding discrete Hg(OH)$_2$ molecules (Wang and Andrews, 2005), linear OHg$^{II}$O as the initial product is calculated to be 275 kJ mol$^{-1}$ lower in energy than the reactants Hg($^3P$) + O$_2$ and therefore sufficiently stable over time to participate in barrier rearrangement to Hg($^1S$) + O$_2$* alongside dissociation to HgO and O. Experimental data suggest that the branching ratio between **Rxn G12b** and **G12a** is low, making oxidation the minor process. Sun et al. reported a quantum yield of up to a few percent for the oxidation step in experiments using synthetic air at 46–88 kPa and 233–298 K (Sun et al., 2022). Callear et al. (1959) observed a faster reaction in air than in O$_2$, suggesting that Hg($^3P_0$) may also react with O$_2$ to form HgO, analogous to Hg($^3P_1$).

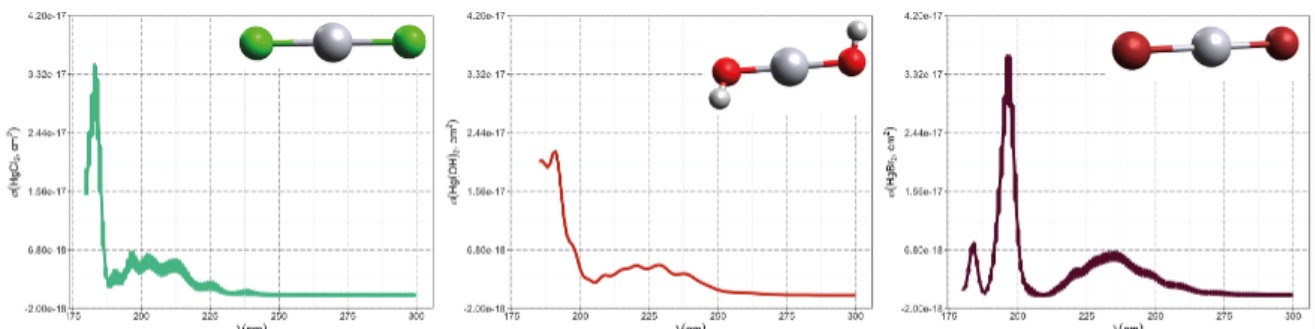

**Figure 8.** Computational absorption spectra of HgCl$_2$, Hg(OH)$_2$ and HgBr$_2$. Data from Saiz-Lopez et al. (2022) and Sitkiewicz et al. (2019).

**Chemical turnover of HgO in the stratosphere. Formation of HgCl$_2$**

The instability of the HgO molecule and its unimolecular decay to elemental Hg was discussed earlier. Produced in greater quantities by rapid photosensitized but nearly thermoneutral oxidation, the initially vibrational cold stratospheric HgO is more likely to survive in the colder part of the upper stratosphere until it can react further into less unstable oxidized forms. The most abundant trace gases in this part of the stratosphere are water vapor, hydrochloric acid and ozone (Calvert et al., 2015). H$_2$O can oxidize Hg($^3P$) (Gunning and Strausz, 1963; Gruss et al., 2017) and may react with HgO:

$$\text{Hg}(^3P) + \text{H}_2\text{O} \rightarrow {}^\bullet\text{Hg}^\text{I}\text{OH} + \text{H}^\bullet \qquad \text{(Rxn G13)}$$

$$\text{HgO}(^3\Pi) + \text{H}_2\text{O} \begin{array}{l} \xrightarrow{\text{M}} \text{Hg(OH)}_2 \\ \rightarrow {}^\bullet\text{Hg}^\text{I}\text{OH} + \text{HO}^\bullet \end{array} \qquad \text{(Rxn G72)}$$

However, the reaction of Hg($^3P$) + H$_2$O is so exothermic (~200 kJ mol$^{-1}$) that the product $^\bullet$Hg$^\text{I}$OH can be expected to be vibrational hot and dissociate rapidly with less time for further bimolecular oxidation. A possible reaction between water vapor and HgO is strongly exothermic if the final product is singlet Hg(OH)$_2$ but weakly endothermic if the triplet form is formed instead. Nevertheless, there is currently no evidence to suggest that HgO can be converted to Hg(OH)$_2$ in a direct reaction with moisture. According to Saiz-Lopez et al. (2022), the reaction between stratospheric HgO and HCl is fast enough (close to the collision limit) to allow some Hg$^{II}$ to be converted to $^\bullet$Hg$^\text{I}$Cl rather than being reduced to elemental vapor:

$$\text{HgO} + \text{HCl} \rightarrow {}^\bullet\text{Hg}^\text{I}\text{Cl} + {}^\bullet\text{OH} \qquad \text{(Rxn G73, } \Delta H_R = -61 \text{ kJ mol}^{-1})$$

As with $^\bullet$Hg$^\text{I}$OH and $^\bullet$Hg$^\text{I}$Br, the reaction between $^\bullet$Hg$^\text{I}$Cl and O$_3$ is barrierless and rapid; in this case, ClHg$^{II}$O$^\bullet$ is produced:

$$^\bullet\text{Hg}^\text{I}\text{Cl} + \text{O}_3 \rightarrow \text{ClHg}^{II}\text{O}^\bullet + \text{O}_2 \qquad \text{(Rxn G43)}$$

Of the versatile tropospheric chemistry presented for YHg$^{II}$O$^\bullet$, hydrogen abstraction (**Rxn G44**) is still important in the stratosphere and is again dominated by CH$_4$ (which is not photolyzed and reacts with the OH radical as the main sink). The product ClHg$^{II}$OH, like ClHg$^{II}$O$^\bullet$, is further converted by reaction with HCl to HgCl$_2$ (**Rxn G46 & G47**), which is the most thermally and photolytically stable of the Hg$^{II}$ molecules present. The photolytic lifetime of HgCl$_2$ in the upper stratosphere is close to one hour and about twice that of Hg$^0$, so the oxidized Hg species dominate (of which ≥ 90% is HgCl$_2$). The Hg$^\text{I}$ concentration increases rapidly above 50 km with increasing UVC photon flux, so the ratio $^\bullet$Hg$^\text{I}$Cl/Hg$^{II}$Cl$_2$ approaches unity at 60 km. An overview of the gas-phase Hg chemistry in the upper stratosphere is given in **Fig. 9** below.

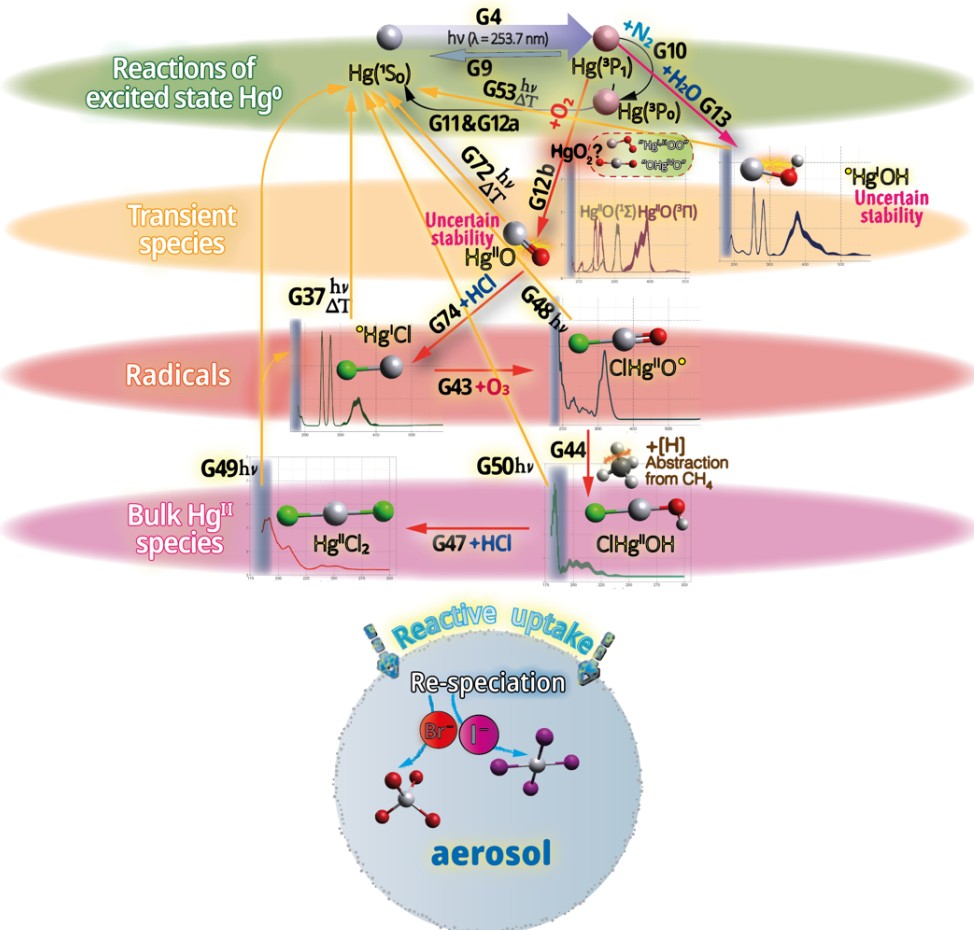

**Figure 9**. At approximately 35 km, the stratosphere begins to contain actinic radiation, which can electronically excite Hg (at 253.7 nm), but below it is
absorbed by $O_3$ in the Hartley bands (with a maximum at 254 nm). Electronically excited $Hg^0$ reacts primarily with $O_2$, with one of the exit channels
leading to the formation of HgO via the intermediate $OHg^{IV}O$. Before HgO can fully decompose into elements, it reacts further via secondary HCl-
driven fast chemistry to $HgCl_2$, the major constituent of Hg in the mid-upper stratosphere. A 2:1 steady state between $HgCl_2$ and $Hg^0$ occurs because
the former photo dissociates more slowly than photosensitized Hg oxidizes, both at significant rates (Saiz-Lopez et al. 2022). The reactive uptake
in aerosols, followed by $Hg^{II}$ complexation with the heavier halides, reflects the observed situation in the lower stratosphere (Murphy et al. 1998).

## 5.2 Organic species

### 5.2.1 Dimethylmercury

In addition to $Hg^0$, "supertoxic" DMHg is another volatile Hg species that exists in nature (Siegler et al., 1999). DMHg has a boiling
point below 100 °C, a high vapor pressure and a Henry's law coefficient equivalent to that of $Hg^0$ (Schroeder and Munthe, 1998).
Both DMHg and $MMHg^+$ species have been detected in ambient air (Lee et al., 2003; Bloom et al., 2005; Weiss-Penzias et al., 2018;
Baya et al., 2015; Zhang et al., 2019a). DMHg has no known sources in the atmosphere. Its occurrence is due mainly to volatilization
from surface waters, where it is transported by upwelling conditions from the deep sea, where it is formed under anoxic conditions
(Conaway et al., 2009; Pongratz and Heumann, 1999). Polar sea ice harbors Hg-methylating microbes and is thought to be a source
of DMHg that can be degassed as ice melts (Schartup et al., 2020). Recently, DMHg has been measured in marine air and
corresponding surface water and has an air–sea gas flux that is 1/30 of the magnitude of the simultaneously measured $Hg^0$ flux (He
et al., 2022). The atmospheric transformation of DMHg is the main source of atmospheric $MMHg^+$ species (Sommar et al., 1997).
DMHg vapor does not absorb actinic light (Terenin, 1934; Terenin and Prileshajewa, 1935) and is therefore not photolyzed in the
planetary boundary layer, where it is only expected to be found (Sommar et al., 1996). DMHg appears to be prone to rapid gas phase
transformation and, depending on the products formed, could be an important source of atmospheric $MMHg^+$ on a regional scale.
However, in addition to $MMHg^+$ species (Niki et al., 1983a; Niki et al., 1983b), inorganic Hg compounds (Thomsen and Egsgaard,
1986; Sommar et al., 1997) have also been reported as products of radical reactions with DMHg. Aware of its acute toxicity (Siegler
et al., 1999), it has been more than a quarter of a century since any laboratory kinetic and reaction mechanistic studies of the
atmospheric gas-phase chemistry of DMHg have been reported and, in retrospect, some comments are worth making. There are three

thermodynamically accessible bimolecular pathways that can potentially initiate the gas phase transformation of DMHg, where $X^\bullet$ below denotes a radical oxidant:

$$\begin{aligned} &\rightarrow CH_3HgX + CH_3^\bullet \\ CH_3HgCH_3 + X^\bullet &\rightarrow CH_3HgCH_2^\bullet + HX \\ &\rightarrow CH_3Hg^\bullet + CH_3X \end{aligned} \qquad (Rxn\ G76 - G78)$$

The existence of the $CH_3Hg$ radical formed in the latter reaction was tentatively demonstrated in a matrix isolation study (Snelson, 1970). The small dissociation energy of the methylmercury bond of the radical (Kominar and Price, 1969) together with a predicted barrierless $CH_3Hg^\bullet \rightarrow {}^\bullet CH_3 + Hg$ reaction (Kallend and Purnell, 1964) suggest rapid decomposition to metallic Hg without time, e.g., a reaction with $O_2$ to form a methylmercury peroxyl radical ($CH_3HgOO^\bullet$). In contrast, a composite reaction

such as $CH_3HgCH_3 + X^\bullet \rightarrow {}^\bullet Hg^I X + 2\ CH_3^\bullet$, which directly produces inorganic Hg, is endothermic and, therefore, less plausible. In a high-pressure study of the gas phase reaction between atomic F and DMHg of low atmospheric relevance, ~10% of the reacted DMHg was reported to be converted to $CH_3F$ via the above substitution reaction (McKeown et al., 1983). However, a static FT–IR study of the Cl-initiated gas–phase reaction in the presence and absence of $O_2$ at atmospheric pressure revealed the importance of the displacement reaction that generates $CH_3HgCl$. The remaining $CH_3$ group is converted to $CH_3Cl$ in $N_2$ as a bath

gas in a chain reaction that regenerates Cl atoms, whereas the end products of the group in air can be attributed to the self-reaction of the $CH_3OO$ radical. The reaction $CH_3HgCH_3 + {}^\bullet OH$ was studied with the same static method by photolysis of a mixture of $CH_3HgCH_3$, ethyl nitrite, and NO in air, which primarily followed the displacement reaction. The rate constant of ~$2 \times 10^{-11}$ $cm^3$ $molecule^{-1}$ $s^{-1}$ indicates that the lifetime of DMHg in the planetary boundary layer with respect to the OH channel active during the day is a few to tens of hours. In the nocturnally active DMHg + $NO_3^\bullet$ reaction, studied by fast flow discharge technique with

Hg/CI–MS detection (Sommar et al., 1996) and a static long path FT–IR system (Sommar et al., 1997), both $CH_3$ groups contained in DMHg react. A small but significant yield of $Hg^0$ was detected along with a product with m/z = 78 ($CH_3ONO_2$) after the reaction of DMHg and $NO_3^\bullet$ under fast flow discharge conditions, indicating that substitution had occurred. In the DMHg

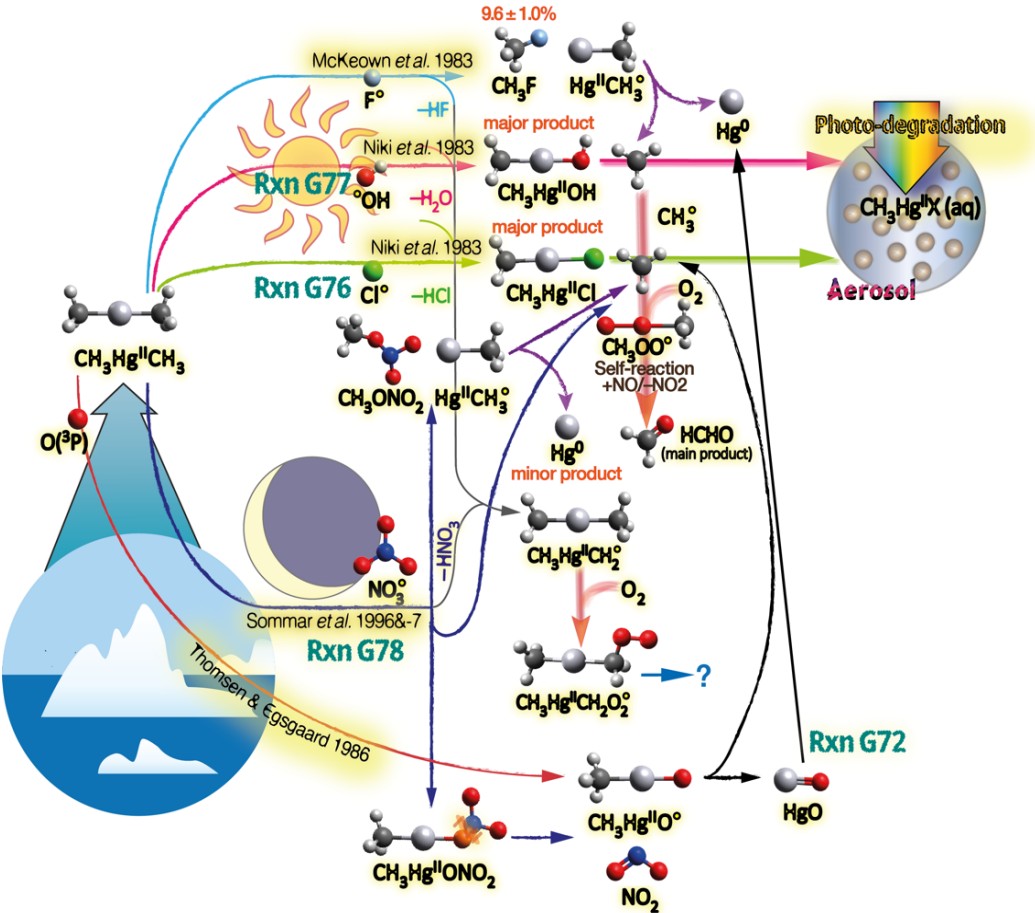

**Figure 10.** Schematics of the atmospheric fate of DMHg.

+ O study, ~95% of the Hg in the converted DMHg was recovered as HgO downstream of the injector in the fast-flow experiments. In the DMHg + $NO_3^\bullet$ batch reactor study, carbon and nitrogen mass balances ruled out the formation of $MMHg^+$ entities, and no Hg compound other than HgO could be considered an end product. Experimental evidence for the tentative intermediate $CH_3HgO^\bullet$ (indicated in **Fig. 10**) is lacking. The rate constant of the nitrate radical reaction (**Rxn G78**) evaluated in the temperature range 258–358 K can be described by the Arrhenius expression $3.2 \times 10^{-11} \times \exp[-(1760 \pm 400)/T]$ $cm^3$ $molecule^{-1}$ $s^{-1}$, and the reaction is fast enough to put the lifetime of DMHg during the night in the same time range as that for $^\bullet$OH-initiated degradation during the day (Sommar et al., 1997). To summarize this section, the degradation of DMHg in the atmosphere is illustrated in **Fig. 10**.

### 5.2.2 Monomethylmercury species

Although experimental data are lacking, gaseous $MMHg^+$ species (e.g., MMHgCl) are expected to react with atmospheric radicals, which leads to demethylation similar to the process that occurs in the reactions of $CH_3HgCH_3$ and OH/Cl radicals. However, the rate constant is likely lower, and the uptake of particles is more important for the atmospheric fate of $MMHg^+$ species.

## 6. Red-ox transformations in the aqueous phase

### 6.1 Inorganic Hg species

Aqueous redox reactions of $Hg^{II}$ complexes can include primary (intramolecular) processes involving direct electron transfer and secondary (intermolecular, usually bimolecular) reactions caused by reactive intermediates. Atmospheric aerosols serve as microreactors for redox Hg reactions (Lin and Pehkonen, 1999). Both oxidation and reduction occur in the aqueous phase. Since $Hg^{2+}$(aq) has a rapid ligand exchange rate, the formation of $Hg^{II}$ hydrated complexes does not limit the redox reaction rates and can therefore be treated separately as chemical equilibria. The aqueous speciation of $Hg^{II}$, where pH is often a critical parameter, is important for the reaction kinetics, not least for the reduction pathways. Thus, aerobic reduction pathways in principle require the formation of specific complexes, since $Hg^0$ cannot be formed from $Hg^{2+}$ by successive bimolecular (single-electron) reduction steps, since dissolved $O_2$ instantaneously re-oxidizes $Hg^{\bullet+}$:

$$XHg^\bullet + O_2 \rightleftarrows \ ^\bullet OOHgX \rightleftarrows HgX^+ + O_2^{\bullet-} \qquad \text{(Rxn W11)}$$

The overall forward rate constant for **Rxn W11** is at the diffusion limit (k~$10^9$ $M^{-1}$ $s^{-1}$, Nazhat and Asmus, 1973). In contrast, $Hg^0$ can be formed by fragmentation of a ligand bound to $Hg^{II}$ (reductive elimination, van Loon et al., 2000). Such photo- or thermolabile $Hg^{II}$ complexes are characterized by low-energy ligand-to-metal charge transfer (LMCT) excited states, which tend to induce internal redox processes leading to oxidation of a ligand and reduction of the mercuric ion. There is evidence that $Hg^{2+}$ complexes can undergo both one-electron and two-electron LMCT. An example is mercuric oxalate, where 2e-LMCT is photoinduced and occurs as part of a concerted series of electron rearrangements (heterolytic cleavage of σ-bonds in the complex), resulting in the oxalate ligand being eliminated as two molecules of $CO_2$ and the oxidation state of the metal ion decreasing by two units. This mechanism occurs without any detectable intermediates such as free radicals:

(Rxn W16a)

As described in **Sections 8.1 & 8.4**, 1e- and 2e-LMCT reactions produce isotopic effects, the specific fractionation of which can be used to identify the reaction mechanism. In addition to the quenching of triplet complex states, the presence of dissolved $O_2$ leads to the scavenging of radicals such as $Hg^{\bullet+}$ produced by the 1e-LMCT mechanism, resulting in reoxidation to $Hg^{2+}$ (**Rxn W11**, Zhao et al., 2021). As previously noted (Pehkonen and Lin, 1998), in certain laboratory experiments, such as reduction experiments, sufficiently elevated Hg concentrations are employed such that the $Hg^0$ formed exceeds its solubility, thereby existing predominantly in a colloidal form. **Table 4** below outlines the potentially significant redox reactions occurring in the aqueous phase, which are then elaborated upon in the subsequent text.

**Table 4.** Aqueous-phase redox chemistry

| ID | Reaction type | Reactant | Co-reactant | Reaction mechanism | Conditions | Technique/ Comments | Rate coefficient $(M^{-1}\,s^{-1})$[19] | References |
|---|---|---|---|---|---|---|---|---|
| **W1** | Oxidation, bimolecular | $Hg^0$ (aq) | Ozone ($O_3$) | $Hg^0 + O_3 \xrightarrow{H^+} Hg^{2+} + OH^- + O_2$ | pH 5.2–6.2 | Relative rate[20] | $(4.7 \pm 2.2) \times 10^7$ | Munthe, 1992 |
| **W2** | | | Hydroxyl radical ($^\bullet OH$) | $Hg^0 + HO^\bullet \rightarrow HOHg^\bullet \xrightarrow{H^+,\, O_2} Hg^{2+} + O_2^{\bullet-}$ | pH 5.6 – 5.9 | Steady-state[21] | $2.0 \times 10^9$ | Lin & Pehkonen, 1997 |
| | | | | | pH 7.9 | Relative rate[22] | $(2.4 \pm 0.3) \times 10^9$ | Gårdfeldt et al., 2001 |
| **W3** | | | Carbonate radical ($CO_3^{\bullet-}$) | $Hg^0 + CO_3^{\bullet-} \rightarrow products$ | pH 8 | Relative rate[23] | | He et al., 2014 |
| **W4** | | | Hypochloric acid (HOCl) | $Hg^0 + HOCl \rightarrow Hg^{2+} + Cl^- + HO^-$ | pH 6.5 – 8.4 | Steady-state concentration of reactant by hydrolysis of the precursor $NH_2Cl$ | $(2.1 \pm 0.1) \times 10^6$ | Lin & Pehkonen, 1998b |
| | | | Hypochlorite ($ClO^-$) | $Hg^0 + ClO^- \xrightarrow{H^+} Hg^{2+} + Cl^- + HO^-$ | pH 6.5 – 8.4 | | $(2.0 \pm 0.1) \times 10^6$ | |
| **W5** | | | Hypobromic acid (HOBr) | $Hg^0 + HOBr \rightarrow Hg^{2+} + Br^- + HO^-$ | pH 6.7 – 6.8 | Steady state disproportionation of $Br_2$ | $0.28 \pm 0.02$ | Wang & Pehkonen, 2004 |
| | | | Hypobromite ($BrO^-$) | $Hg^0 + BrO^- \xrightarrow{H^+} Hg^{2+} + Br^- + HO^-$ | pH 11.7 – 11.8 | | $0.27 \pm 0.04$ | |
| | | | Bromine ($Br_2$) | $Hg^0 + Br_2 \rightarrow Hg^{2+} + 2\,Br^-$ | pH 2.0 – 2.1 | | $0.20 \pm 0.03$ | |
| **W6** | | | Peracids (peracetic and perbenzoic acid) | [reaction scheme] | | Screening study | | Wigfield & Perkins, 1985a |
| **W7** | Oxidation, complexation | | 2–mercapto propionic acid | [reaction scheme] | pH 7, anoxic | Absolute | 0.61 | Zheng et al., 2013 |
| **W8** | Comprop., bimolecular | | Mercuric ion ($Hg^{2+}$) | $Hg^0 + Hg^{2+} \rightarrow Hg_2^{2+}$ | pH 3 – 4 | | $5.9 \times 10^8$ | Buxton et al., 1995 |
| **W9** | Dimerization, bimolecular | $Hg^{\bullet+}$ (aq) | Mercurous ion radical ($Hg^{\bullet+}$) | $2\,Hg^{\bullet+} \rightarrow Hg_2^{2+}$ | | $e_{aq}^-$ (pulse radiolysis) | $\geq 10^9$ | |
| **W10** | Disprop., bimolecular | | | $2\,Hg^{\bullet+} \rightarrow Hg^{2+} + Hg^0$ | pH 3.15 | | $2.6 \times 10^9$ | |
| **W11** | Oxidation, bimolecular | | Oxygen ($O_2$) | $XHg^\bullet + O_2 \rightleftharpoons\, ^\bullet OOHgX \rightleftharpoons HgX^+ + O_2^{\bullet-}$ | | | $(1-4) \times 10^{9}$[24] | Jungbluth et al., 1976 |
| **W12** | | | p–benzoquinone | [reaction scheme] | pH 5.0 – 5.5 | | $(1-4) \times 10^9$ | |
| **W13** | Reduction, 2e–LMCT, thermal | $Hg^{II}$ (aq) | Sulfite ($SO_3^{2-}$) | $Hg^{2+} + SO_3^{2-} \rightleftharpoons HgSO_3 \rightarrow Hg^0 S^{VI} O_3 \rightarrow Hg^0 + S(VI)$ | pH 3.0 – 4.8 | Absolute[25] | $0.6\ s^{-1}$ | Munthe et al., 1991 |
| | | | | | 280 – 307 K | Absolute[26] | $T \cdot \exp\left[\frac{(31.971 \cdot T - 12595)}{T}\right]$ | van Loon et al., 2000 |
| | | | | | pH 3, 298 K | | $0.0106 \pm 0.0009\ s^{-1}$ | Feinberg et al., 2015 |
| **W14** | Reduction, bimolecular | | Carbon dioxide anion radical ($CO_2^{\bullet-}$) | $HgCl_2 + CO_2^{\bullet-} \rightarrow ClHg^{\bullet+} + CO_2 + Cl^-$ | pH 1 – 2, anoxic | Relative rate[27] | $1.8 \times 10^8$ | Berkovic et al., 2010 |
| **W15** | | | Superoxide anion radical ($O_2^{\bullet-}$) | $HgCl_2 + O_2^{\bullet-} \rightarrow ClHg^{\bullet+} + O_2 + Cl^-$ | pH 6 | Relative rate[28] | $5 \times 10^3$ | Gårdfeldt & Jonsson, 2003 |
| | | | Hydroperoxy radical ($HO_2$) | $C_2O_4^{\bullet-} \rightarrow CO_2 + CO_2^{\bullet-},\ CO_2^{\bullet-} + O_2 \rightarrow CO_2 + O_2^{\bullet-},$ $H^+ + O_2^{\bullet-} \rightleftharpoons HO_2^\bullet,\ Hg^{2+} + HO_2^\bullet \rightarrow products$ | pH 3.9 | Absolute[29] | $1.7 \times 10^4$ | Pehkonen and Lin, 1998 |
| **W16** | Reduction, 2e$^-$ and 1e$^-$ LMCT, photolytic | | Oxalate | $Hg^{2+} + C_2O_4^{2-} \rightleftharpoons HgC_2O_4 \xrightarrow{hv} Hg^0 + 2\,CO_2$ $HgC_2O_4 \xrightarrow{hv} Hg^{\bullet+} + C_2O_4^{\bullet-}$ $Hg^{2+} + C_2O_4^{\bullet-} \rightarrow Hg^{\bullet+} + 2\,CO_2,$ $Hg^{\bullet+} + C_2O_4^{2-} \rightarrow Hg^0 + C_2O_4^{\bullet-}$ | pH 3 – 6, anoxic | | $15.7 \pm 2.8$[31] | Zhao et al., 2021 |
| | | | | | pH 3, anoxic | Absolute[30] | $(1.2 \pm 0.2) \times 10^{4}$[32] | Si & Ariya, 2008 |
| | | | Malonate (R = $CH_2$), Succinate (R = $C_2H_4$) | $Hg^{2+} + R(COO)_2^{2-} \rightleftharpoons Hg(OOC)_2R$ $\xrightarrow{hv} Hg^0 + CO_2 + HORCOOH - H_2O$ $Hg(OOC)_2R \xrightarrow{hv} Hg^{\bullet+} +\, ^\bullet OOCRCOO^-,$ $Hg^{2+} +\, ^\bullet OOCRCOO^- \xrightarrow{H_2O} Hg^{\bullet+} + CO_2 + HORCOOH$ $Hg^{\bullet+} + R(COO)_2^{2-} \rightarrow Hg^0 +\, ^\bullet OOCRCOO^-$ | pH 3, anoxic | | $(4.9 \pm 0.8) \times 10^3$ $(2.8 \pm 0.5) \times 10^3$ | Si & Ariya, 2008 |

---

[19] Unless otherwise stated.

[20] with $SO_3^{2-}$ as reference.

[21] using $C_6H_6$ as $^\bullet OH$ scavenger. $NO_3^-$ photolysis as $^\bullet OH$ source.

[22] with $CH_3Hg^+$ as reference. $NO_3^-$ photolysis as $^\bullet OH$ source.

[23] The loss of $Hg^0$(aq) was followed, but neither $^\bullet OH$ nor $CO_3^{\bullet-}$, which co-occur in the solution, were quantified. $NO_3^-$ photolysis as $^\bullet OH$ source, $^\bullet OH$ reaction with carbonate anion as a source of $CO_3^{\bullet-}$

[24] Concerns various mercurous halide and pseudohalide radicals $^\bullet HgX$ (X = Cl, Br, I, SCN and CN)

[25] Followed by decay of $[Hg(SO_3)_2]^{2-}$ absorption ($\lambda = 230$ nm)

[26] Followed by the formation of $Hg_2^{2+}$ ($\lambda = 236$ nm)

[27] Methyl viologen as reference. $S_2O_8^{2-}$ photolysis in the presence of HCOOH.

[28] Methyl viologen as reference. Reduction of $O_2$ by $e_{aq}^-$ (pulse radiolysis). Determination of K = 2.5 for **W11/W16** and using **W11** for calculating **W16**.

[29] Dithizone colorimetric quantification of $Hg^{II}$ decay. $HO_2^\bullet$ produced by $C_2O_4^{2-}$ photolysis with air bubbling. $[HO_2^\bullet]$ estimated from production of $H_2O_2$.

[30] After analytically determining the initial $Hg^{II}$ concentration, the reduction process was studied by measuring the production of $Hg^0$ by CV-AFS.

[31] The oxalate ion ($C_2O_4^{2-}$) was identified as the sole reducing agent (complexing ligand), hence $k = k_{obs}/[C_2O_4^{2-}]$.

[32] Based on the total concentration, the second-order rate coefficient is expressed as $k = k_{obs}/([H_2C_2O_4] + [HC_2O_4^-] + [C_2O_4^{2-}])$.

| | Reaction type | Hg species | Chemical | Reaction / mechanism | Conditions | Type | Rate constant | Citation |
|---|---|---|---|---|---|---|---|---|
| | Reduction, thermal | Hg$^{II}$(aq) | Ascorbate (H$_2$asc, Hasc$^-$) (enolic acids) | HOHg(Hasc) $\longrightarrow$ Hg$^0$ + H$_2$O + dehydroascorbate | pH 4 – 5.5 | | $2.8 \times 10^{-3}$ s$^{-1}$ | This work |
| | Reduction, photolytic | | Salicylic acid | (reaction scheme: Solvo-mercuration $-$H$^\oplus$; $h\nu$; $+$H$^\oplus$ $\rightarrow$ O=C=O, Hg, phenol) | pH 4.3 | | $1.0 \times 10^{-4}$ s$^{-1}$ | This work |
| | | | p-aminobenzoic acid p-hydroxybenzoic acid | See Rxn 7 | pH 4.9 pH 5.6 | | $3.1 \times 10^{-4}$ s$^{-1}$ $1.1 \times 10^{-5}$ s$^{-1}$ | |
| W17 | 1e$^-$LMCT, followed by bimolecular reduction, photolytic | | Anthraquinone–2,6–disulfonate (AQDS) | (reaction scheme: $h\nu$; Red $-$Ox; Hg + Products) | UVB, pH 3.4 | Absolute[33] | $(9.9 \pm 2.7) \times 10^{-4}$ s$^{-1}$ | Zhao et al., 2021 |
| W18 | Reduction, photolytic | | 1–alkanethiols | Hg(RS)$_2$ $\xrightarrow{h\nu}$ Hg$^0$ + RS–SR | UV–VIS, pH 7 | Absolute | $(2.0 \pm 0.2) \times 10^{-7}$ s$^{-1}$ (R = C$_3$H$_7$) $(1.4 \pm 0.1) \times 10^{-7}$ s$^{-1}$ (R = C$_4$H$_9$) $(8.3 \pm 0.5) \times 10^{-8}$ s$^{-1}$ (R = C$_5$H$_{11}$) | Si & Ariya, 2011 |
| | Reduction etc. Photolytic | | Thioglycolic acid | Hg(O(=O)CCH$_2$S) $\xrightarrow{h\nu}$ Hg$^0$ + products $\xrightarrow{h\nu}$ HgS + products | UV–VIS, pH 4 | | $(2.3 \pm 0.4) \times 10^{-5}$ s$^{-1}$ | Si & Ariya, 2015 |

### 6.1.1 Oxidation channels

The mass transfer (diffusion) of gas-phase Hg$^0$ into typical size regimes of aerosols (radius of 0.1–10 μm) does not limit the rate of aqueous Hg$^0$ oxidation. The concentration of dissolved Hg$^0$ in a droplet is at a steady state governed by Henry's law (Lin and Pehkonen, 1998a).

**Elemental mercury**

**Inorganic oxidants**

**Rxn W1. Ozone (O$_3$)**

The presence of O$_3$ in atmospheric water is due mainly to the scavenging of gaseous O$_3$ ($k_H^{cp} = 0.013$ M atm$^{-1}$ at 298 K). An early study of the oxidation of Hg$^0$ by O$_3$ in the aqueous phase was carried out by Iverfeldt and Lindqvist (1986) using a flow system in which 70–200 ppb O$_3$ was introduced. Their results suggested a conversion rate of 1–4% h$^{-1}$ when applied to atmospheric conditions. Munthe and coworkers (McElroy and Munthe, 1991; Munthe, 1992) studied the ozone reaction with the mercurous cation in an acidic solution (pH = 1 - 3) in a stopped-flow system and with elemental Hg using the relative rate technique (sulfite as a reference compound, pH 5.2 - 6.2) and obtained pH-independent rate constants of $(9.2 \pm 0.9) \times 10^6$ and $(4.7 \pm 2.2) \times 10^7$ M$^{-1}$ s$^{-1}$, respectively.

**Rxn W2. Hydroxyl radical ($^\bullet$OH)**

The OH radical in atmospheric water can come from the air ($k_H^{cp} = 30$ M atm$^{-1}$) or from aqueous phase production via pathways including photolysis of H$_2$O$_2$, HONO, O$_3$ and NO$_3^-$ (Finlayson-Pitts and Pitts, 2000). The reaction rate of Hg$^0$ + $^\bullet$OH in the aqueous phase was determined by Lin and Pehkonen (1997) using a steady-state technique with the photolysis of NO$_3^-$ as the $^\bullet$OH source and C$_6$H$_6$ as the $^\bullet$OH scavenger to $2.0 \times 10^9$ M$^{-1}$ s$^{-1}$ at pH 5.6–5.9. Like the first step (Hg$^0$ + $^\bullet$OH $\rightarrow$ $^\bullet$Hg$^I$OH), the second step, which is mediated by dissolved O$_2$ (**Rxn W11**), is near the diffusion limit. Gårdfeldt et al. (2001) subsequently studied the same reaction at pH 7.9 but with the reaction between CH$_3$Hg$^+$ and $^\bullet$OH as a reference but with similar results ($2.4 \times 10^9$ M$^{-1}$ s$^{-1}$).

**Rxn W3. Carbonate radical (CO$_3^{\bullet-}$)**

In water, the carbonate system (HCO$_3^-$ and CO$_3^{2-}$) can react with OH radicals to form strongly oxidizing carbonate radicals (CO$_3^{\bullet-}$) in fast reactions ($8.5 \times 10^6$ and $3.9 \times 10^8$ M$^{-1}$ s$^{-1}$, respectively). In a comparative study, He et al. (2014) studied the disappearance of Hg$^0$ in aqueous solutions where NO$_3^-$ was photolyzed via UV–VIS in the absence and presence of CO$_3^{2-}$ at pH = 8. When both NO$_3^-$ (0.23 mM) and CO$_3^{2-}$ (2.75 mM) were present in the irradiated solutions (electron paramagnetic resonance spin trapping analysis detected the presence of $^\bullet$OH and CO$_3^{\bullet-}$), the rate of oxidation of Hg$^0$ (aq) (1.44 h$^{-1}$) was 8 times faster than that observed when only NO$_3^-$ (which produces $^\bullet$OH) was irradiated. The carbonate radical is a single-electron oxidant and reacts according to Hg$^0$ + CO$_3^{\bullet-} \rightarrow$ Hg$^{\bullet+}$ + CO$_3^{2-}$. In addition to identifying the carbonate radical as an effective oxidant of Hg$^0$ dissolved in water alongside the hydroxyl radical, this study

---

[33] AQDS is not the reductant rather photohydroxylated reduced AQDS forms.

investigated the role of $^1\Delta_g$ O$_2$ (singlet oxygen) as an oxidant for Hg$^0$(aq). However, the latter species, an excited state of O$_2$, does not initiate any measurable oxidation. Notably, the absolute rate constant for Hg$^0$ + CO$_3^{\bullet-}$ remains to be determined.

**Rxn W4. Aqueous chlorine (HOCl/ClO$^-$)**

Aqueous chlorine is formed mainly by the scavenging of gaseous Cl$_2$ ($k_H^{cp}$ = 7.61 × 10$^{-2}$ M atm$^{-1}$ at 298 K) into the aqueous phase and the oxidation of chloride ions by $^\bullet$OH. Once incorporated into the aqueous phase, it dissociates to form HOCl/OCl$^-$ (pK$_a$ = 7.5) and Cl$^-$

, the former being the primary oxidant and increasing the solubility of total chlorine. It is a nocturnal oxidant, as both Cl$_2$ and HOCl are readily photolyzed by solar radiation. The prospects for Hg$^0$ oxidation by aqueous chlorine were investigated by Kobayashi (1987) and Munthe and McElroy (1992). In the former, rapid dissolution of Hg was reported when a gas stream containing Hg$^0$ was passed through a solution containing dissolved chlorine (HClO), while in the latter, Hg$_2^{2+}$(aq) was used as a proxy for Hg$^0$, whose oxidation was observed to be "relatively fast" in a solution containing HClO. A detailed kinetic study (Lin and Pehkonen, 1998b) of the reaction between Hg$^0$

and HClO/ClO$^-$ was carried out using a steady-state method with chloramine as a reservoir of free hypochlorous acid formed by hydrolysis: NH$_2$Cl + H$_2$O → NH$_3$ + HClO. The turnover of Hg$^0$ was studied in the pH range of 6.5 - 8.5 around the pK$_a$ (HClO) to investigate the influence of HClO (aq) and ClO$^-$ (aq), which were found to be closely equivalent according to the rate constants for Hg$^0$ + HClO and Hg$^0$ + ClO$^-$ of (2.09 ± 0.06) × 10$^6$ and (1.99 ± 0.06) × 10$^6$ M$^{-1}$ s$^{-1}$, respectively. The products of both reactions (2 electrons are transferred) are chloride and hydroxide anions with a stoichiometry of 1:1 together with a mercuric cation, which rapidly forms a

strong complex (logβ$_{11}$ = 18.0).

**Rxn W5. Aqueous bromine (HOBr/BrO$^-$/Br$_2$)**

Bromine has a higher $k_H^{cp}$ (0.725 atm M$^{-1}$) than chlorine does, but the disproportionation of Br$_2$ to HBrO/BrO$^-$ (pK$_a$ = 8.7) and Br$^-$ is slow, and the equilibrium is shifted in favor of Br$_2$. In contrast, Br$^{+1}$ is formed by the action of O$_3$ on bromide ions and exists in the presence of Cl$^-$ largely as BrCl (Liu and Margerum, 2001). Aqueous bromine (Br$_2$, HOBr) oxidizes Hg$^0$ only slowly (0.2–0.3 M$^{-1}$ s$^{-1}$,

Wang and Pehkonen, 2004). However, BrCl is likely important, as it is used as an oxidant for Hg in current analytical methods, although the kinetics have not been investigated.

**Organic oxidants**

**Rxn W6. Peroxides**

H$_2$O$_2$ cannot oxidize Hg$^0$ (aq) (Kobayashi, 1987) but participates in the metal-catalyzed oxidation of Hg$^{0,}$ as in Fenton's system.

Fenton's reagent itself, Fe$^{2+}$ + H$_2$O$_2$, produces OH radicals, for which Hg$^0$, Fe$^{2+}$ and H$_2$O$_2$ compete for oxidation. The latter reaction, H$_2$O$_2$ + $^\bullet$OH, produces the HO$_2$ radical, which propagates a chain reaction (Fenton's reaction) supported by Fe$^{3+}$ acting as a catalyst to decompose H$_2$O$_2$ to O$_2$ and H$_2$O, during which a stable concentration of Fe$^{2+}$ is produced as a source of $^\bullet$OH. Hg$^0$ oxidation is most pronounced when the ferrous part of the Fenton reaction dominates over the ferric part, corresponding to a higher concentration of OH radicals (Liu, 2011). The -OOH functional group in organic hydroperoxides, like that in hydrogen peroxide, lacks the ability

to oxidize Hg$^0$, whereas that in peroxocarboxylic acids (peracetic and perbenzoic acid) seems to possess it, tentatively forming a mercuric carboxylate by a cyclic mechanism (Wigfield and Perkins, 1985b; Wigfield and Perkins, 1985a).

**Rxn W7. Thiocarboxylic acids**

Thiol compounds, as substituted carboxylic acids, including cysteine and glutathione, can oxidize Hg$^0$(aq) both thermally under anoxic conditions (Gu et al., 2011). For example, Zheng et al. (2013) reported that 2-sulphanylpropanoic acid in greater excess (1000:1) oxidized

Hg$^0$ at a rate of 2.18 ± 0.13 h$^{-1}$. The presence of an electron acceptor (such as a quinone) further increased the reaction rate. The reaction mechanism has been described as oxidative complexation. Hg$^0$, which is polarizable, interacts with a thiol group, leading to ligand-induced oxidative complexation in which hydrogen participates in charge transfer (Cohen-Atiya and Mandler, 2003).

**Mercurous radical species ($^\bullet$Hg$^I$X)**

**Inorganic oxidants**

**Rxn W11. Oxygen (O$_2$)**

The reaction between Hg$^{\bullet+}$ and O$_2$ has been studied for a variety of ligands and over a range of pH values well into the alkaline range

using pulse radiolysis, ~~with a homogeneous kinetic result~~ (Nazhat and Asmus, 1973; Jungbluth et al., 1976; Fujita et al., 1975; Fujita et al., 1973; Liu et al., 1983; Pikaev et al., 1975). Mercurous species are formed by the reduction of corresponding mercuric species by the action of solvated electrons and H atoms derived from $H_2O$ radiolysis: $HgX_2 + e_{aq}^- \rightarrow {}^\bullet Hg^I X + X^-$ and $HgX_2 + H^\bullet \rightarrow {}^\bullet Hg^I X + H^+ + X^-$.

All types of ${}^\bullet Hg^I X$ species react rapidly ($\geq 1 \times 10^9$ M$^{-1}$ s$^{-1}$) with $O_2$ (aq): $Hg^{\bullet +} + O_2 \rightleftarrows {}^\bullet OOHg^+ \rightleftarrows Hg^{2+} + O_2^{\bullet -}$, where the equilibrium is very strongly shifted to the right. In one case (X = CN, Jungbluth et al., 1976 the reaction takes place without the clear formation of a peroxyl radical intermediate. In an air-saturated solution (~0.2 mM $O_2$), the lifetime of ${}^\bullet Hg^I X$ is about 1 µs (Jungbluth et al., 1976).

### Organic oxidants

### Rxn W12. Quinones

Both $Hg^{\bullet +}$ and ${}^\bullet OOHg^+$ are rapidly oxidized by benzoquinone ($\gtrsim 10^9$ and $\lesssim 10^9$ M$^{-1}$ s$^{-1}$, Jungbluth et al., 1976), which accepts an electron to form a semiquinone anion. Lalonde et al. (2001) observed that $Hg^0$ is oxidized (~0.6 h$^{-1}$) in UVB-irradiated aqueous solutions containing both benzoquinone (32 nM) and chloride ions (0.5 M) without being able to fix the mechanism.

### 6.1.2 Reduction channels

### Mercuric compounds ($Hg^{II}$)

### Inorganic reductants

### Rxn W13. Sulfite ($SO_3^{2-}$)

$SO_2$ dissolves in water ($k_H^{cp}$ = 1.36 M atm$^{-1}$) to form the weak acid $H_2SO_3$ (aq), which can be deprotonated to $HSO_3^-$ and $SO_3^{2-}$. The oxidation of sulfite to sulfate is rapid in the atmosphere and takes a few hours under typical oxygenated conditions in atmospheric droplets. $SO_3^{2-}$ is a soft ligand that forms strong complexes with $Hg^{2+}$ (**Table 1**), such as $HgSO_3$ and $[Hg(SO_3)_2]^{2-}$, the latter completely

dominating under natural conditions where the sulfite content greatly exceeds that of $Hg^{2+}$. The reduction of aqueous $Hg^{II}$ by the sulfite system was first investigated by Munthe et al. (1991). $[Hg(SO_3)_2]^{2-}$ is stable, whereas $HgSO_3$ decomposes readily to $Hg^0$ and sulfate with firstorder rate constants of $<10^{-4}$ s$^{-1}$ and 0.6 s$^{-1}$, respectively. Scott and co-workers (van Loon et al., 2001, 2000) carried out a thorough re-examination and confirmed that the bis-sulfite complex is thermally stable but that the reduction of $HgSO_3$, which is strongly temperature dependent (k approximately quadruples with each 10 °C increase in temperature) and weakly pH dependent, is more than

50 times slower than that reported by Munthe et al. (0.011 vs. 0.6 s$^{-1}$ at 25 °C). The reaction mechanism is intramolecular with 2e-LMCT and heterolytic cleavage of the Hg–S bond: $Hg^{2+} + SO_3^{2-} \rightarrow Hg^{II}S^{IV}O_3 \rightarrow Hg^0 S^{VI}O_3 \xrightarrow{H_2O} Hg^0 + SO_4^{2-}$.

### Rxn W14. Carbon dioxide anion radical ($CO_2^{\bullet -}$)

The carbon dioxide radical ($CO_2^{\bullet -}$) can be formed in nature by the oxidation of carboxylic acids (see above under oxalic acid). It is strongly reducing and occurs in anaerobic environments. Berkovic et al. (2010) studied the $CO_2^{\bullet -}$–mediated reduction of $Hg^{2+}$ at low

pH by laser flash photolysis of a dilute mixture of $HgCl_2$, formic acid and sodium peroxydisulfate at 266 nm. The one-electron reaction $Hg^{2+} + CO_2^{\bullet -} \rightarrow Hg^{\bullet +} + CO_2$ is exothermic, with a rate constant of $1.8 \times 10^8$ M$^{-1}$ s$^{-1}$. The $Hg^{\bullet +}$ formed can only be further reduced to $Hg^0$ in the absence of $O_2$.

### Rxn W15. Superoxide anion/hydroperoxy radical ($O_2^{\bullet -}/HO_2^\bullet$)

$HO_2^\bullet/O_2^{\bullet -}$ (pK$_a$ 5.5) is a one-electron reductant of $Hg^{2+}$ to $Hg^{\bullet +}$. Gårdfeldt and Jonsson (2003) determined the one-electron

reduction potential for the pair $HgCl_2/{}^\bullet HgCl$ vs. NHE at [Cl$^-$] = 0.05 M to be –0.47 V, which, together with that for $O_2/O_2^{\bullet -}$ vs. NHE of –0.155 V, gives an equilibrium constant for $HgCl_2 + O_2^{\bullet -} \rightleftarrows {}^\bullet HgCl + O_2 + Cl^-$ of $5 \times 10^{-6}$ at the aforementioned [Cl$^-$]. Given that the rate constant for the reaction ${}^\bullet Hg^I Cl + O_2$ is ~$10^9$ M$^{-1}$ s$^{-1}$ (**Rxn W11**), the bimolecular rate constant between $HgCl_2$ and $O_2^{\bullet -}$ can be estimated to be $5 \times 10^3$ M$^{-1}$ s$^{-1}$. Pehkonen and Lin (1998) studied the photoreduction of mercuric ions to $Hg^0$ with nitrate or chloride as counterions in the presence of formic, acetic or oxalic acid at neutral (7.0) and acidic (3.9) pH values

in aerated solutions. Only in the presence of oxalic acid does significant photoreduction occur, and as in the later studies by Zhao et al. (2021) and Si and Ariya (2008), an increase in the reaction rate is observed with increasing ratios of oxalic acid to $Hg^{II}$. The reduction is also suppressed in the presence of Cl$^-$. Photoreduction results in an exponential increase in $H_2O_2$ formation, which is

due to the presence of hydroperoxyl radicals in solution ($2\,HO_2^\bullet \rightarrow H_2O_2 + O_2$). In retrospect (see above), this follows from the homolytic decomposition of $Hg(\eta^2\text{-}C_2O_4)$ into radicals in an aerated solution ($CO_2^{\bullet-} + O_2 \rightarrow O_2^{\bullet-} + CO_2$) and does not necessarily mean that $HO_2^\bullet/O_2^{\bullet-}$ can reduce $Hg^{II}$ to $Hg^0$.

**Organic reductants**

In the atmospheric environment, $Hg^{II}$ complexation by DOM plays a pivotal role in the redox chemistry of Hg (Åkerblom et al., 2015). The chemical-reducing effect of DOM (humic substances) on $Hg^{II}$ has been recognized for nearly 50 years (Alberts et al., 1974). These heterogeneous macromolecular ligands contain not only building blocks that can form complexes with $Hg^{II}$ but also redox-active aromatic chromophores that can photolytically convert Hg. The fractions of DOM contributing to $Hg^{II}$ photoreduction include fulvic- and flavin-like fractions that contain more quinone and flavin moieties than usual (Yang et al., 2020a). Furthermore, DOM contains several functional groups that can reduce complex-bound mercuric ions to $Hg^0$ by a 2e-LMCT reaction (**Table 5**).

**Table 5.** Main functional groups of DOM that can (photo)reduce ligated $Hg^{2+}$.

| Binding atom | Ligand oxidation process |
|---|---|
| Oxygen (O) |  |
| Nitrogen (N) |  |
| Sulfur (S) |  |

**Rxn W16. Organic acids**

The low-molecular-weight organic acids present in the atmosphere can reduce $Hg^{II}$ to $Hg^0$ in the presence of $O_2$. These include dicarboxylic acids, ortho-substituted aromatic carboxylic acids, and enolic acids. Since 1880 (Eder, 1880), the salt of the lightest dicarboxylic acid, oxalate, has been known to reduce $Hg^{II}$ in daylight. Oxalic acid is formed from, e.g., ethylene or acetylene, by atmospheric oxidation over several reaction cycles (chemical aging, Warneck, 2003). Mercuric ions form a complex with oxalate in a 1:1 ratio ($Hg(\eta^2\text{-}C_2O_4)$), characterized by $\log \beta_{10} = 9.66$, which is most photolabile under UVB irradiation. Si and Ariya (2008) studied the kinetics and products of the photoreduction of $Hg^{II}$ in a series of experiments with different concentrations of the lightest dicarboxylic acids, $C_2$–$C_4$, at an initial pH of 3.0 and a temperature of $296 \pm 2$ K, while the kinetic, product and isotopic study of Zhao et al. (2021) involved the system $Hg^{II}$ + oxalic acid with $ClO_4^-$ as a counterion in the pH range of 2.7–6.3 and a small temperature range of 295–303 K. The pH-resolved experiments show that in the $C_2O_4^{2-}$, $HC_2O_4^-$, $H_2C_2O_4$ – system, only the oxalate ion reduces $Hg^{2+}$ with a $k_{Hg^{2+}+C_2O_4^{2-}}$ of $15.7 \pm 2.8$ M$^{-1}$ s$^{-1}$ at $295 \pm 1$ K. Si and Ariya reported a much larger bimolecular rate constant between $Hg^{2+}$ and the total oxalic acid concentration of $1.2 \times 10^4$ M$^{-1}$ s$^{-1}$ at pH 3.0. The magnitude is surprisingly large and is comparable to the rate constant between $Hg^{2+}$ and $HO_2/O_2^-$ radicals (see below). When this higher rate constant, which is based on the total oxalic acid concentration, is implemented in regional air quality models, the impact is significant (Bash et al., 2014). However, with respect to the reaction mechanism, there is more consensus that it follows a branched route. $Hg(\eta^2\text{-}C_2O_4)$ undergoes photolysis followed by partial reductive elimination in one step (insensitive to the presence of $O_2$): $Hg(\eta^2\text{-}C_2O_4) \xrightarrow{h\nu} Hg^0 + 2\,CO_2$ and, in part, homolysis of a Hg–O bond, which initiates a chain reaction: $Hg(\eta^2\text{-}C_2O_4) \xrightarrow{h\nu} Hg^{\bullet+} + C_2O_4^{\bullet-}$. $Hg^0$ should form from the reaction of $Hg^{\bullet+}$ with the bulk ligand $C_2O_4^{2-}$, where $Hg^{\bullet+}$ is reformed from the reaction between bulk $Hg^{2+}$ and the oxalyl ($C_2O_4^{\bullet-}$) or carbon dioxide anion ($CO_2^{\bullet-}$) radical. The reduction to $Hg^0$ in the chain reaction is inhibited by $O_2$, which reacts rapidly with both $C_2O_4^{\bullet-}$ and $CO_2^{\bullet-}$ and re-oxidizes $Hg^{\bullet+}$ to $Hg^{2+}$. Like dicarboxylic acids, aqueous solutions of aromatic ortho- and para-substituted carboxylic acids exposed to UVB can oxidize $Hg^{2+}$ to $Hg^0$ via elimination of $CO_2$, and $Hg^{2+} \rightarrow Hg^0$ photoreduction is attenuated but not completely inhibited by the presence of dissolved $O_2$ and competing counterions. Previously, He et al. (2012) studied the aqueous photoreduction of $Hg^{2+}$ coupled with a series of aromatic carboxylic acid derivatives in the absence of $O_2$ at pH 4.3 and suggested that the reaction

proceeded via a radical mechanism. However, studies of the same reactants in our laboratory (unpublished results, shown in **Table 4**) have shown that $Hg^0$ is formed even in the presence of dissolved $O_2$, suggesting the existence of an additional non-radical reduction pathway. We propose that this channel requires solvo-mercuration to an arylmercurial intermediate followed by photolytically induced 2e-LMCT as part of a concerted series of electron rearrangements, including cleavage of a Hg-C bond yielding $Hg^0$, $CO_2$ and a decarboxylated aromatic as end products. Taking p-aminobenzoic acid as an example:

(Rxn 7)

Photo-reduction has also been observed in the presence of dissolved $O_2$ when $Hg^{2+}$ is bound mainly to the amino acid serine (HSer), similar to $HgSer_2$ (Motta et al., 2020b), which can be explained as the result of reductive elimination with $CO_2$ and 2-aminoethanol as byproducts in addition to $Hg^0$ and involves an intermediate with a photolabile Hg–C bond (Zhao et al., 2021). In cysteine-mediated photoreduction (Motta et al., 2020b; Zheng and Hintelmann, 2010b), the ligand is converted from a thiol to a disulfide (**Table 5**). Ascorbic acid, as a representative enolic acid, can readily reduce inorganic divalent Hg in aqueous solutions to $Hg^0$. Studies in our laboratory have shown that the reaction is thermal and not affected by actinic light. When ascorbic acid is in excess (>10:1) relative to $Hg^{2+}$, the reaction rate is not significantly affected by increasing ascorbic acid concentration. The reaction rate is highest in the pH range where the hydrogen ascorbate ion ($HAsc^-$) is dominant and the hydrolysis of $Hg^{2+}$ is not complete, i.e., typical pH values for atmospheric hydrometeors ($\leq 5.5$). Presumably, $HgOH^+$ (aq) forms a reactive complex with $HAsc^-$, $Hg(HAsc)^+$, which is labile to the elimination of water in a heterolytic process, forming $Hg^0$ and dehydroascorbate as the final products. Enols act as atmospheric intermediates, and it is unclear whether they are present in high concentrations, which makes them interesting reducing agents for atmospheric $Hg^{2+}$. In any case, $k_{Hg^{2+} + HAsc^-}$ is relatively high (~0.17 min$^{-1}$, **Rxn W16c**, **Table 4**).

**Rxn W17. Hydroquinones and polyphenols**

The quinonic (Zheng et al., 2012) and fulvic (Yang et al., 2020a) units in DOM act as key red-ox centers. How this happens at the molecular level is being investigated by studying model compounds that contain redox-active groups but lack other functional groups (Zhao et al., 2021). The simplest quinone forms a red-ox pair with the corresponding hydroquinone in the half-reaction:

$E^0 = -0.699$ V

(Rxn 8)

Combined with the half-reaction in **Rxn 3**, this gives a $\Delta E^0 > 0$ for $Hg^{2+} + C_6H_4(OH)_2 \rightarrow Hg^0 + C_6H_4(=O)_2 + 2\ H^+$, i.e., thermodynamically feasible. Relatively slow reduction of $Hg^{2+}$ to $Hg^0$ by hydroquinone occurs in the dark in a dilute aqueous solution $(8.2 \pm 2.4) \times 10^{-5}$ s$^{-1}$). These results are consistent with a reaction mechanism involving a hydroxyphenoxymercuric complex or via *ipso*-mercuration followed by electron shuttling and elimination of $Hg^0$ and $H_2O$:

(Rxn 9)

The aqueous photochemistry of quinones is complicated and can involve both ground and excited state reactions as well as free radicals (Görner, 2019). With respect to the interaction of benzoquinone with Hg under actinic light, one study revealed significant oxidation (~0.6 h$^{-1}$) of $Hg^0$ in Cl$^-$-enriched water (see above, Lalonde et al., 2001), whereas another study reported photoreduction of $Hg^{II} \rightarrow Hg^0$ of about the same magnitude (~0.8 h$^{-1}$) under anaerobic conditions and in the absence of strongly complexing inorganic ligands (Zhao et al., 2021). An anthraquinone (AQ) derivative (AQ-2,6-disulfonate) is an effective electron shuttle that facilitates electron transfer from metal-reducing bacteria (MRB) to $Hg^{II}$ (Lee et al., 2018), as well as from $Hg^0$ (aq) to organic thiols (R-SH) during oxidative complexation

to form Hg(SR)$_2$ (Zheng et al., 2013). Zheng et al. (2013) reported that AQDS(aq) alone is unable to oxidize Hg$^0$ or reduce Hg$^{II}$ under dark and anaerobic conditions. AQDS-assisted biotic Hg$^{II}$ reduction by the MRB *Shewanella oneidensis* MR-1 is associated with negative charge scavenging, which temporarily increases the content of reduced AQDS species, such as AQH$_2$DS and semiquinone radicals (Lee et al., 2018). The reduced species AQH$_2$DS alone is a potent reductant of Hg$^{II}$ in the dark. On the other hand, Hg$^{II}$ is efficiently reduced to Hg$^0$ in a UVB-irradiated aqueous solution containing dissolved AQDS (~10$^{-3}$ M$^{-1}$ s$^{-1}$). The reactive species is tentatively photohydrated AQDS (AQH$_2$(OH)DS), which interacts with Hg$^{II}$ by forming a photolabile bidentate O–coordinated mercuric complex. In conjunction with a strong isotope effect (**Section 8.4**), the photoreaction is likely to occur via a paramagnetic intermediate (a mercurous semiquinone biradical complex). The reaction rate decreases to ~0.2 h$^{-1}$ in the presence of dissolved O$_2$ (Zhao et al., 2021). Hg$^{II}$ interacts with ortho-QH$_2$ moieties such as those in the natural polyphenols of humic substances and tannins (Jerzykiewicz, 2013). A direct reaction yields redox-active Hg$^{I}$ complexes with ligands of semiquinone radical character that may eventually decompose into Hg$^0$ (Jerzykiewicz et al., 2015). Reaction kinetic and mechanistic studies that are more applicable to the environment are not available.

**Rxn W18. Thiols**

Hg$^{2+}$ and CH$_3$Hg$^+$ bind extremely strongly to heavier hydrochalcogenide groups (such as RSH and RSeH) and other corresponding groups of reduced chalcogenides, such as sulfides and disulfides (Skyllberg, 2011). Most relevant, both inorganic (e.g., H$_2$S, CS$_2$) and organic (CH$_3$SH, CH$_3$SCH$_3$) low-molecular-weight reduced sulfur compounds have short lifetimes (Warneck, 1988) and therefore have no effect on aqueous Hg speciation. It is questionable whether reduced sulfur/thiol groups associated with macromolecular organic compounds in aerosols influence internal Hg speciation. The photoreduction of divalent Hg by lighter aliphatic thiols is slow (< 10$^{-7}$ s$^{-1}$, Si and Ariya, 2011), whereas that by thioglycolic thiols is slightly faster (2.3 × 10$^{-5}$ s$^{-1}$, Si and Ariya, 2015) but hardly significant in the atmosphere.

## 6.2 Organic mercury

### 6.2.1 Demethylation channels

Biogenically produced organo-Hg in the environment is almost exclusively methylated Hg, although there are few reports of the presence of ethyl Hg (Wu et al., 2023b), which must be derived from a natural source. However, only methylated Hg has been detected in air. As mentioned above, DMHg is a major source of MMHg$^+$ compounds in the atmosphere through gas-phase degradation. Gaseous MMHg$^+$ species (Lee et al., 2003) can potentially react homogeneously to inorganic Hg, but as MMHg$^+$ species are only semi-volatile and have a high $k_H^{cp}$, they are more likely to be rapidly absorbed on aerosols. MMHg$^+$ species have been detected in cloud water (Li et al., 2018; Weiss-Penzias et al., 2018), fog water (Weiss-Penzias et al., 2012), rainwater (Conaway et al., 2010; Won et al., 2019) and snow (St Louis et al., 2007). Photolytic demethylation of dissolved DMHg occurs in pure water (Chen et al., 2024) incubated with sunlight (CH$_3$HgCH$_3$ $\xrightarrow{h\nu, +H^+}$ CH$_3$Hg$^+$ + CH$_4$, ~0.32 ± 0.07 d$^{-1}$, West et al., 2022). Acidolytic demethylation of DMHg to MMHg$^+$ species is of very minor importance and occurs only at low pH (Maguire and Anand, 1976; Wolfe et al., 1973). A theoretical study of CH$_3$HgOH$_2^+$ and CH$_3$HgOH, which dominate the speciation of MMHg$^+$ in natural waters without significant levels of Cl$^-$ and reduced sulfur ligands, including DOM, indicated that CH$_3$HgOH$_2^+$ can be excited to the triplet state by sunlight and that this state dissociates into CH$_3$ and Hg$^{I}$ radicals (Tossell, 1998). An room-temperature study of the photo-degradation of CH$_3$HgOH (aq) when irradiated by a Xe lamp with filter blocking wavelengths < 290 nm reported a rate constant of (2.2 ± 0.2) × 10$^{-4}$ s$^{-1}$ (Gårdfeldt et al., 2003). Rapid indirect demethylation of MMHg$^+$ species by a bimolecular process with the OH radical occurs at the limit of what diffusion allows (9.83 ± 0.66) × 10$^9$ M$^{-1}$ s$^{-1}$, Chen et al., 2003). In natural water, select reactive oxygen species, such as singlet oxygen (see above, Suda et al., 1993; Zhang and Hsu-Kim, 2010), have been suggested to cause Hg$^{II}$ demethylation, but their reactivity has not been directly quantified. Instead, its presence has been suggested based on the results of added scavenger/promoter tests, some of which may yield misleading results for some water compositions (Han et al., 2017). Chen et al. (2003) concluded that OH-initiated demethylation is comparable to the rates of MMHg$^+$ photodegradation reported in situ in natural waters. These researchers reported that, in addition to inorganic Hg$^{II}$, Hg$^0$ was a by-product of OH-initiated degradation in an O$_2$-saturated system, presumably by homolytic substitution.

### 6.2.2 Methylation channels

The paucity of empirical data renders the budgets of tropospheric $MMHg^+$ species highly uncertain. A recent estimate of the $MMHg^+$ pool size is 5.5 Mg, associated with a lifetime of 1.9 d, of which one of the major sources is inferred to be in-cloud methylation (Wu et al., 2024b). The potential for atmospheric biotic methylation is considered limited, despite the presence of pathogens and bacteria in aerosols and hydrometeors, because Hg-methylating microbes (possessing two important methylation genes, hgcA and hgcB, Parks et al., 2013) usually thrive in anaerobic environments, in contrast to the distinctly oxic environment of atmospheric waters. However, many unknowns about the potential for $Hg^{II}$ methylation under oxic conditions need to be resolved (Sonke et al., 2023). There have been extensive studies on the abiotic methylation of $Hg^{2+}$ (Ullrich et al., 2001). Methylating agents that are important for $MMHg^+$ formation in the atmosphere are oxygenated hydrocarbons containing a methyl group (Yin et al., 2012; Hammerschmidt et al., 2007). Some of them have properties that allow competitive photochemical reduction and methylation of $Hg^{2+}$ (Yin et al., 2012). Earlier studies have investigated photochemical $Hg^{2+}$ methylation by deep UV irradiation (Yin et al., 2012; Akagi et al., 1974; Hayashi et al., 1977), making it impossible to generalize these results to the lower atmosphere. The formation of $MMHg^+$ species was observed in the dark in dilute $Hg^{II}$ solutions (1 nM) containing an excess of acetic acid (100:1 M/M), with an apparent first-order rate constant of $5.4 \times 10^{-6}\,s^{-1}$ in artificial rainwater (pH 4.9 Gårdfeldt et al., 2003). When the system is exposed to sunlight, photo-demethylation occurs, which counteracts $MMHg^+$ formation mediated by acetic acid/acetate, and within hours, the $MMHg^+$ concentration reaches a steady state (~2.5% of inorganic $Hg^{II}$). Hammerschmidt et al. (2007) noted that the average ratio of $MMHg^+$ to reactive $Hg^{II}$ measured in North American continental precipitation ($2.5 \pm 0.6\%$) agrees with the findings of the above laboratory study. Methylation takes place intramolecularly in the acetato-mercuric complexes present in solution concerted with decarboxylation (Gårdfeldt et al., 2003; Yin et al., 2012; Akagi et al., 1974): $\left[Hg(CH_3COO)_n\right]^{2-n} \rightarrow CH_3Hg^+ + CO_2 + (n-1)\,CH_3COO^-$.

## 7 Multi-phase transformations

Multiphase transformations address dynamics and chemistry at interfaces and media, such as aerosol particles and cloud droplets, which interact heterogeneously with gases and solute species. Despite a wealth of studies addressing the multiphase chemical or physical transformation of Hg under processes such as those under simulated post-combustion conditions, which undoubtedly pertain to interactions with certain environmental surfaces, the findings offer limited insight into the surface and heterogeneous atmospheric Hg chemistry. The subsequent chapter addresses the studies that have been identified as contributing meaningfully to the advancement of understanding in this domain.

### 7.1 Gas–particle partitioning and reactive gas uptake

The behaviors of gaseous $Hg^0$ atoms and $Hg^{II}$ molecules in interacting with the atmospheric condensed phase differ. The dominant $Hg^0$ pool has limited water solubility, and the uptake of $Hg^0$ vapor to aerosol surfaces is low, to the limited extent that it has been investigated. Gas phase $Hg^{II}$ molecules, "GOM", have $k_H^{cp}$ several orders of magnitude greater than that of $Hg^0$, favoring the liquid phase. The heterogeneous processes that allow GOM to be adsorbed reversibly or irreversibly, modified by ligand exchange, or dissociated to $Hg^0$ by reduction on surfaces are key parameters that need to be characterized to appropriately parameterize chemical transport models.

#### 7.1.1 HgCl₂

Understanding the transformation from GOM to PBM through gas phase processes (condensation, **Section 5.1.4**) and aerosol surface interaction (**Section 4.2**) is crucial for parameterizing deposition. Since the separation of GOM from PBM with current methods is tentative, the accuracy of studies of $Hg^{II}$ distribution between the gas and condensed phases, performed by preconcentration in laboratory experiments with nebulized aerosols (Rutter and Schauer, 2007b, a) and in the field (Amos et al., 2012), is retrospectively ambiguous. Fitting observational data to an equilibrium $GOM + PM_{2.5} \overset{1/T}{\Leftrightarrow} PBM$ according to a van't Hoff-type relationship $\log_{10}\left(K_{gp}^{-1}\right) = a + b/T$ is used in models to calculate the volatilization of GOM from atmospheric aerosols, where $K_{gp}$ (**Eq. 9**) is weighted by the inverse of the mass concentration of fine particulate matter ($PM_{2.5}$; Shah et al., 2021). The partitioning expression does not consider that the interaction between GOM and a surface is significantly influenced by the composition of the surface layer. $HgCl_2(g)$ partitions among particles consisting of typical alkali metal salts such as chlorides, nitrates, and sulfates (Mao et al., 2021; Malcolm et al., 2009). To compensate, global Hg models

treat the uptake of GOM onto sea salt particles separately as an irreversible first-order process parameterized by wind speed and humidity. The equilibrium studies conducted at atmospheric pressure do not provide insights into the dynamics of the system, as the experiments are limited by mass transport, which negates the possibility of obtaining quantitative information on reactive uptake. As an alternative (Liu et al., 2022), partition coefficients have been calculated for individual GOM species based on theoretical predictions of both adsorption and absorption (Wu et al., 2024a). The reactive uptake of $HgCl_2(g)$ on surfaces representative of inorganic and organic primary and secondary atmospheric aerosols has recently been studied via the fast flow technique coupled with an ion drift chemical ionization mass spectrometer (ID-CI-MS). The reported data (Mao et al., 2021; Khalizov and Mao, 2023) are summarized in **Table 6**.

**Table 6.** Reactive uptake of $HgCl_2(g)$ on surfaces

| Chemical | Structural formula[34] | $\gamma_{net}$[35] | | Surface coverage ($\theta$, %) | Lifetime ($\tau$, days) |
|---|---|---|---|---|---|
| | | $\gamma_{net}^0$, | $\gamma_{net}^\infty$ | | |
| **Inorganic aerosol surrogates** | | | | | |
| $Na_2SO_4$ | | $3.1 \times 10^{-2}$ | $1.7 \times 10^{-3}$ | 98 | 0.1 |
| NaCl | | $2.2 \times 10^{-2}$ | $1.9 \times 10^{-3}$ | 65 | 0.1 |
| $(NH_4)_2SO_4$ | | $1.4 \times 10^{-2}$ | $7.0 \times 10^{-4}$ | 5.6 | 0.2 |
| $NH_4NO_3$ | | $3.6 \times 10^{-3}$ | $3.3 \times 10^{-4}$ | 0.3 | 0.7 |
| **Primary organic aerosol (POA) surrogates** | | | | | |
| Levoglucosan | | $1.1 \times 10^{-2}$ | $2.9 \times 10^{-4}$ | 9.6 | 0.2 |
| Pyrene | | $2.1 \times 10^{-3}$ | $5.0 \times 10^{-4}$ | 1.3 | 1.2 |
| Perylene | | $3.0 \times 10^{-3}$ | $5.2 \times 10^{-4}$ | 3.8 | 0.8 |
| Soot | | $8.9 \times 10^{-5}$ | | 0.1 | 20.2 |
| **Secondary organic aerosol (SOA) surrogates** | | | | | |
| Citric acid ($H_3cit$) | | $< 1 \times 10^{-5}$ | $< 1 \times 10^{-5}$ | $< 0.02$ | $> 242$ |
| $NaH_2cit$ | | $6.9 \times 10^{-5}$ | $5.0 \times 10^{-5}$ | $< 0.02$ | 35 |
| $Na_2Hcit$ | | $2.4 \times 10^{-3}$ | $2.3 \times 10^{-4}$ | 1.2 | 1.0 |
| $Na_3cit$ | | $8.4 \times 10^{-3}$ | $6.6 \times 10^{-4}$ | 7.5 | 0.3 |
| Pimelic acid ($H_2pim$) | | $1.1 \times 10^{-3}$ | $1.8 \times 10^{-4}$ | 1.0 | 2.2 |
| NaHpim | | $2.2 \times 10^{-3}$ | $3.1 \times 10^{-4}$ | 1.4 | 1.1 |
| $Na_2pim$ | | $8.2 \times 10^{-3}$ | $8.0 \times 10^{-4}$ | 11.6 | 0.3 |
| Succinic acid ($H_2suc$) | | $9.3 \times 10^{-4}$ | $1.0 \times 10^{-4}$ | 0.02 | 2.6 |
| NaHsuc | | $2.0 \times 10^{-3}$ | $3.6 \times 10^{-4}$ | 0.7 | 1.2 |
| $Na_2suc$ | | $8.3 \times 10^{-3}$ | $6.6 \times 10^{-4}$ | 6.2 | 0.3 |
| Dioctyl sebacate | | $2.6 \times 10^{-2}$ | $7.1 \times 10^{-3}$ | 153 | 0.1 |

The data in **Table 6** are for dry surfaces, where $\gamma_{net}^0$ is the initial uptake coefficient, which is relevant throughout the lifetime of the aerosol, as the surface coverage by atmospheric $HgCl_2$ remains unchanged and low. In the presence of sea salt aerosols (>0.6 µm, initially at pH 8) that dominate in marine air, where NaCl represents >95% of its mass, the lifetime of $HgCl_2$ (g) is expected to be between 4 and 20 h depending on aerosol loading (Mao et al., 2021). When the relative humidity exceeds ~75%, a hygroscopic sea salt droplet is formed as the salt deliquesces, and a highly mobile surface phase in which $Hg^{II}$ is equilibrated in ionic form as $HgCl_4^{2-}$ may contribute to more rapid GOM loss in marine air (Holmes et al., 2009). Ammonium salts such as nitrates and sulfates are primarily found in secondary particles, typically in urban and agricultural-rural air. Although $HgCl_2$ uptake is lower here, its lifetime is comparable because of the higher particle number and the large surface area they generally represent. These semi-volatile

---

[34] For soot, a clichéd structure is used that does not claim to be accurate.
[35] Calculated by Eq. 5

ammonium salts do not occur in isolation but coexist with oxygenated organics formed through photochemical activity, resulting in the formation of secondary aerosols, which constitute the primary fraction of the atmospheric burden of organic aerosols (OA, Jimenez et al., 2009). The acidity of secondary organic aerosols (SOAs), a dominant component of PM2.5, affects HgCl2 uptake by controlling the acid–base equilibria of characteristic chemical species such as aliphatic dicarboxylic acids, aromatic polycarboxylic acids, and other oxygenated multi-functional organics in aerosols. For the diprotic acids in **Table 6**, the reactivity becomes noticeable only after the first deprotonation step at pH 4.5 - 5.5. For the triprotic citric acid, activation occurs after the second step at pH 6.5. The adsorption of HgCl2 on primary organic aerosol (POA) surfaces is significant in the presence of levoglucosan, an anhydrosugar, which is a fingerprint of fire activity. Nevertheless, the interaction between HgCl2 and polyaromatic hydrocarbons (PAHs) derived from carbonaceous fuel combustion is more constrained, occurring between the electrophile HgCl2 and the $\pi$ electrons delocalized over the aromatic fused ring skeleton. The observed adsorption on fresh soot, which is porous and graphitic with a high specific surface area, is more than one order of magnitude lower than that for the minor type of PAH studied (pyrene, perylene). If morphology affects uptake, so does the state of the surface phase, as a diester of sebacic acid (a close homolog of pimelic acid), octyl sebacate, a lubricant, is more reactive to HgCl2 than the microcrystalline pimelic acid film is. The adsorption of HgCl2 on mineral surfaces (dust aerosols) represented by iron (hydr)oxides has not been studied experimentally, but calculations indicate a partition coefficient ($K_{gp}$) for $\alpha$-Fe2O3 that exceeds that for NaCl by three orders of magnitude (Tacey et al., 2018b). The studies listed in **Table 6** were performed without observing redox chemistry (i.e., no $Hg^0$ was detected to be emitted from the HgCl2-exposed surfaces when heated to 120 °C), but a combined study using FF-ID-CIMS and Raman spectroscopy revealed that exchange reactions between gaseous mercuric compounds are catalyzed by surfaces such that HgCl2 and HgBr2 molecules in the presence of a deactivated surface produce mixed BrHgCl molecules (Mao and Khalizov, 2021), which are also volatile. Owing to rapid exchange reactions, the prospect of accurately speciating GOMs by pre-concentration on filters and cation exchange membranes, as discussed previously (**Section 3.1**), is unlikely.

**7.1.2 Hg$^0$**

A challenge in studying gas-phase- or liquid-phase-initiated reactions is the potential for side reactions and phase changes to occur during experiments. Thus, a portion of the loss of gas-phase $Hg^0$ in laboratory experiments designed to study homogeneous oxidation (e.g., by O3, Snider et al., 2008; $NO_3^{\bullet}$, Sommar et al.,1997, etc.) has been linked to a heterogeneous rate component ($k_{surf}$) occurring on new surfaces that form during experiments (product clusters undergoing particle growth in free suspension, **Section 5.1.4**) and/or on initially deactivated existing surfaces (reactor walls) that begin to catalyze $Hg^0$ surface oxidation as deposits form (Sommar et al., 1997; Medhekar et al., 1979). For example, in a series of spherical reactors with varying surface-to-volume ratios (S/V), Pal and Ariya (2004b) reported the loss of $Hg^0$ by reacting with excess O3 in N2 as follows:

$$-d[Hg^0]/dt= \left\{k_{gas} + \frac{S}{V}\cdot k_{surf}\right\}\cdot[Hg^0]\cdot[O_3] = k_{net}\cdot[Hg^0]\cdot[O_3] \qquad (11)$$

where $k_{gas}$ (cm$^3$ molecule$^{-1}$ s$^{-1}$) is the gas-phase reaction rate, S/V (cm$^{-1}$), $k_{surf}$ (cm$^4$ molecule$^{-1}$ s$^{-1}$) is the surface rate loss, and [O3] (molecules cm$^{-3}$) is the gas-phase O3 concentration. In the S/V range of 0.28–0.93 cm$^{-1}$, $k_{net}$ increased by 30% simultaneously with the formation of particles (Snider et al., 2008) during the experiments, which started homogeneously. Using a fluorocarbon film smog chamber (9 m$^3$, S/V = 0.03 cm$^{-1}$), Rutter et al. (2012) studied the influence of SOA (yielding an ~100-fold increase in the surface area of the system) and secondarily formed $^{\bullet}$OH (at ambient level due to added scavenger) generated from an irradiated mixture of O3 and various biogenic and anthropogenic VOCs (at a level ~one order of magnitude greater than ambient) on the oxidation of Hg atoms (at a level ~two orders of magnitude greater than ambient). Neither Rutter et al. nor subsequent researchers (Lyman et al., 2022) have been able to identify evidence that interactions with photochemical smog particles significantly contribute to the oxidation of $Hg^0$. Nevertheless, few studies concerning $Hg^0$ uptake have been conducted with a sufficiently rigorous standard, employing techniques used in specific studies of heterogeneous processes to produce a kinetic formalism that can be related to atmospheric models. These studies, which were conducted with a coated-wall laminar flow tube reactor, focused on the light- and moisture-dependent uptake of $Hg^0$ (detected by CV-AFS), which may be photocatalytic, on the major metal oxides (TiO2, Fe2O3,

FeOOH, and $Al_2O_3$) present in mineral dust aerosols (Kurien et al., 2017; Lee et al., 2022). The first three metal oxides have semiconductor properties with band gaps that allow photoexcitation in the UVA ($\leq$ 395 nm) and visible ($\leq$590 nm) regions, while $Al_2O_3$, the second most abundant mineral oxide in the Earth's crust after $SiO_2$, is an insulator but has some thermal conductivity. It has been established for over half a century that $Hg^0$ vapor in the presence of $O_2$ over an irradiated $TiO_2$ surface is consumed by reactive uptake (Kaluza and Boehm, 1971) via the following tentative mechanism:

$$TiO_2 \overset{h\nu}{\rightarrow} e^-_{CB} + h^+_{VB}$$
$$e^-_{CB} + O_2 \text{ (ads)} \rightarrow O_2^{\bullet-} \text{ (ads)}$$
$$h^+_{VB} + H_2O|OH^- \text{ (ads)} \rightarrow HO^\bullet \text{ (ads)} + (H^+\text{(ads)})$$
$$Hg^0 \text{ (ads)} + HO^\bullet \text{ (ads)} \rightarrow {}^\bullet Hg^I OH \text{ (ads)} \qquad \text{(Rxn 10a-g)}$$
$${}^\bullet Hg^I OH \text{ (ads)} + O_2 \text{ (ads)} \rightarrow HgO \text{ (ads)} + HO_2^\bullet \text{ (ads)}$$
$$HO^\bullet \text{ (ads)} + HO_2^\bullet \text{ (ads)} \rightarrow H_2O \text{ (ads)} + O_2 \text{ (ads)}$$
$$HO_2^\bullet \text{ (ads)} + O_2^{\bullet-} \text{ (ads)} \overset{H^+}{\rightarrow} H_2O_2 \text{ (ads)} + O_2 \text{ (ads)}$$

When excited by light of a wavelength shorter than the band gap energy, the generation of electron–hole pairs ($e^-_{CB}$, $h^+_{VB}$) occurs in the conduction and valence bands (**Rxn 10a**). The electrons and holes transported to the particle surface initiate redox chemistry by reacting with $H_2O$ and $O_2$ molecules to form reactive oxygen species (ROS, **Rxn 10b, c**). The oxidation potential of $h^+_{VB}$ exceeds +2.27 eV in the $TiO_2$, $Fe_2O_3$ and FeOOH cases, which is sufficient to generate hydroxyl radicals from surface water (**Rxn 11c**) that can oxidize adsorbed $Hg^0$ (**Rxn 10d**). The reported uptake coefficients are in the range of $<10^{-10}$ to $>10^{-4}$ (based on the Brunauer–Emmett–Teller surface area), with relative reactivities of $Fe_2O_3 \lesssim$ FeOOH $< Al_2O_3 < TiO_2$, where $\gamma^\infty_{net}$ without irradiation is below the detection limit. The uptake of $Hg^0$ on iron (hydr)oxides is less than $10^{-8}$ under both UV and visible light and is inhibited by humidity, as is the case for $Al_2O_3$, which shows measurable uptake under UV irradiation ($\gamma^\infty_{net} = 1.2 \times 10^{-8}$). The photo-initiated uptake of $Hg^0$ on $TiO_2$ is significant, especially under UV light at low humidity ($\gamma^\infty_{net} > 3 \times 10^{-5}$, diffusion-controlled limit). However, as with $Al_2O_3$, it shows reversibility (desorption of $Hg^0$) in the presence of water vapor during darkness (Lee et al., 2022), whereas $Hg^0$ exhibits almost irreversible binding to iron (hydr)oxides at the temperatures studied (< 150 °C, Kurien et al., 2017). Based on limited published data, only under conditions of low humidity and very high mineral dust aerosol loading can the uptake of $Hg^0$ be considered to have any effect on the atmospheric cycling of $Hg^0$. Notably, there are no corresponding experimental data for $HgCl_2$ uptake on mineral dust surrogates.

The uptake of $Hg^0$ on ice, which involves the migration of radioactive Hg isotopes into ice spheres in a packed bed flow tube exposed to a strong temperature gradient, can be described as reversible adsorption without significant solvation. The observations were in accordance with a Langmuir isotherm, where the adsorption equilibrium can be described thermodynamically by Bartels-Rausch et al. (2008):

$$-RT\ln K = \Delta H^0_{ads} - T\Delta S^0_{ads} = -28000 + 38 \cdot T \qquad \text{(12)}$$

where K is the Langmuir absorption constant (**Eq. 8**), R is the gas constant, T is the absolute temperature, and $\Delta H^0_{ads}$ and $\Delta S^0_{ads}$ are the enthalpy and entropy of adsorption, respectively. Compared with $k^{cc}_H$ for $Hg^0$ (0.18 at 5 °C), the Langmuir adsorption coefficient on ice, which is expressed in a dimensionless way, is much smaller even at temperatures lower than the freezing point of the metal ($2.2 \times 10^{-5}$ at 220 K), which is most relevant for polar regions and the upper troposphere. Therefore, in both atmospheric and polar environments, the uptake of $Hg^0(g)$ on ice surfaces is negligible.

## 7.2 Reduction of mercurial species on surfaces

Computational chemistry studies report that the adsorption of mercuric halides on dry salt- or mineral-like surfaces reduces the energy required for reduction to $Hg^0$ (Tacey et al., 2016) and that the reduction of $HgCl_2$ and $HgBr_2$ to $Hg^0$ on iron oxide aerosols requires the presence of actinic light (Tacey et al., 2018a). Breaking the first Hg–X bond is possible either thermally or photolytically, while the second requires photons with $\lambda \leq 461$ nm. To release $Hg^0$ from the surface, an excitation energy of 2.59 eV ($\lambda \leq 479$ nm) is required in a photoinduced charge transfer process between the surface and the adsorbate.

The photoreduction of particle-bound $Hg^{II}$ has been the subject of experimental investigations (Tong et al., 2013; Tong et al., 2014). In these experiments, aerosol surrogates doped with $HgCl_2$ were generated and dried in laboratory air and subsequently captured on filters, which were then exposed to light with three spectral options in a flow-through reactor. Photoreduction of NaCl aerosols occurs under actinic light (both UV and visible light, with approximately 2.5% and 2.0% of $Hg^{II}$ reduced, respectively, during a 30-minute exposure, normalized per 100 W $m^{-2}$ irradiation). However, the presence of iron species (mainly $Fe^{III}$ rather than $Fe^{II}$) has been observed to exert some inhibitory effects (Tong et al., 2013). In contrast, photoreduction on carbon-based synthetic aerosols has been demonstrated to be more significant but also more variable. For example, $Hg^{II}$ on adipic acid aerosols is reduced by 8% (per same time unit and normalization as above), while on levoglucosan, it is less than 2% (Tong et al., 2014). Notably, however, these experiments were carried out without $O_2$ in the carrier gas stream.

The reduction of $Hg^{II}$ in ice in the presence of organics has been studied in an ice-coated flow tube at atmospheric pressure under irradiation with light between 300 nm and 420 nm (Bartels-Rausch et al., 2011). $O_2$-free ice matrices containing 60 nM Hg were doped with a stoichiometric excess (up to 50:1 M/M) of either benzophenone (a strong photosensitizer), oxalic acid-oxalate (forming photolabile $Hg^{II}$ complexes), or humic acid (ditto photolabile complexes), which, upon irradiation, accelerated the release of $Hg^0$, which was most rapid in the presence of benzophenone at high pH. The presence of $O_2$ (20% in the gas stream), the introduction of sea ice-like conditions, or a large drop in temperature (from 270 to 250 K) or pH (to 4) resulted in diminished photoreduction. The mechanism by which $Hg^{II}$ reduction is sensitized by benzophenone is challenging to ascertain. One potential mechanism involves the dissociation of an excited state of the major species, $Hg(OH)_2$, which has been reported to be photolabile as a solute in water (Xiao et al., 1994). A controlled laboratory study of light-irradiated natural snow samples at a temperature of -10 °C revealed that the release of $Hg^0$ follows first-order kinetics with a coefficient between 0.18 and 0.25 $h^{-1}$, corresponding to a natural lifetime of 4–5.6 h (Dommergue et al., 2007). However, no monitoring of $Hg^{II}$ in the condensed phase has been conducted. Given that light does not penetrate the entire snowpack, it can be assumed that a $Hg^{II}$ gradient toward depletion at the top is established.

Brominated mercurials that are present in the Arctic environment during AMDE may play a role in light-induced Hg re-emission from the cryosphere to the atmosphere (cf. **Fig. 2**). A computational study (Carmona-García et al., 2025) suggested that, compared with $HgBr_2$ in the gas phase, $HgBr_2$ in solution has an increased absorption cross section for wavelengths longer than 290 nm, whereas bromomercurate anions ($Hg^{II}Br_3^-$ and $Hg^{II}Br_4^{2-}$) have a comparatively greater absorption in actinic light. The low-energy excited states of $HgBr_2$, $Hg^{II}Br_3^-$, and $Hg^{II}Br_4^{2-}$ in solution are characterized by electronic transitions in which the electron density is mainly transferred from the Br atoms to the Hg atom, indicating a significant photoreductive character upon light absorption, leading to the generation of $Hg^I$ species ($^\bullet Hg^I Br$, $^\bullet Hg^I Br_2^-$ and $^\bullet Hg^I Br_3^{2-}$) and a bromine atom. The photoreductive character is also recognized for the aforementioned $Hg^I$ species in their electonically excited states, which plausibly dissociate via an LMCT mechanism with $Hg^0$ as the product. The predicted peak photolysis constants for the polar spring (March, ~80°N) are $3.9 \times 10^{-6}$, $3.8 \times 10^{-4}$ and $7.9 \times 10^{-5}$ $s^{-1}$ for $HgBr_2$, $HgBr_3^-$ and $HgBr_4^{2-}$, respectively.

For pure heterogeneous reduction, there is experimental evidence that $SO_2(g)$ can reduce HgO(s) at room temperature via $Hg^I_2SO_4$ (Zacharewski and Cherniak, 1987) to $Hg^0$, HgS and $HgSO_4$ as stable products (Scott et al., 2003) and that $O_3(g)$ in the presence of actinic light can reduce $HgCl_2/HgBr_2(s)$ to mercurous species (Ai et al., 2023, which may tentatively undergo $Hg^0/Hg^{II}$ disproportionation). In the latter exploratory study, single-particle reactors, 10–50 μm in size, synthesized from mercuric halides in single-walled carbon nanotubes were prepared to levitate during the experiments via optical tweezers. The turnover of $HgX_2$ by breaking a Hg–X bond was measured by time- and position-resolved Raman spectroscopy, which also showed that the decomposed X atom was bound to the carbon material (X = Cl, Br). Heterogeneous reactions of this type, i.e.,

$$O_3 \xrightarrow{h\nu} O_2 + O(^3P)$$
$$HgX_2\ (s) + O(^3P) \rightarrow\ {}^\bullet Hg^I X\ (s) + XO^\bullet\ (g)$$
$$XO^\bullet + O(^3P) \rightarrow X^\bullet + O_2 \qquad\qquad \text{(Rxn 11)}$$
$$2\ {}^\bullet Hg^I X\ (s) \rightarrow Hg_2X_2\ (s) \rightleftarrows Hg^0\ (ads) + HgX_2\ (s)$$

may explain why KCl-coated denuders do not work as a robust quantitative method for measuring GOM in ambient air (Lyman et al.,

2010). Since the gas-phase reaction $HgX_2 + O/O_3 \rightarrow {}^\bullet Hg^I X + XO^\bullet (+ O_2)$ is endothermic ($\geq 66$ kJ mol$^{-1}$) and therefore unlikely, the results of a steady-state study (Tong et al., 2021) claiming gas-phase photoreduction of $HgX_2$ in the presence of $O_3$ and light can instead be attributed to the above-mentioned heterogeneous reactions. Additionally, voltammetry can provide valuable insights into the redox chemistry of mercury. $Hg^0$ is frequently employed as the working electrode and has a high overpotential for the reduction of $H_3O^+$ to $H_2$.

This enables the utilization of standard potentials as negative as -1 V in acidic solutions and -2 V in basic solutions. The surface of the hanging mercury drop electrode (HMDE) can be readily renewed by extruding a new drop. In a study by Giannakopoulos et al. (2012), the interfacial adsorption mechanism of gallic acid onto HMDE was investigated, and a series of easily reducible $Hg^{II}$ complexes with mono-, di-, or tridentate gallic acid ligation were identified.

### 7.3 Dark oxidation of $Hg^0$ accelerated by freeze-concentration effects

Slow oxidation of dissolved $Hg^0$ by $O_2$ occurs in aquatic systems in the presence of $Cl^-$ ions (Amyot et al., 2005; Wang et al., 2023). However, upon freezing, most of the solutes are separated from the forming ice phase and concentrated in the remaining liquid at a significantly reduced pH (Bartels-Rausch et al., 2011). In experimental mimics of the micro-pockets of solutions that occur in ice, experiments in the presence of $O_2$, $H_2O_2$, and HONO each result in significant $Hg^0$ oxidation. It has been postulated that protonated forms, $HO\text{-}OH_2^+$ and $ONOH_2^+$, are responsible for oxidation processes, which can be classified as strongly exothermic on the basis of

the provided thermodynamic data (O'Concubhair et al., 2012). Moreover, neither dilute $H_2O_2$ (aq) nor HONO (aq) will oxidize $Hg^0$ (aq) to any significant extent at room temperature (Kobayashi, 1987).

### 7.4 Surface-catalyzed reduction of $Hg^{II}$ in aqueous solution

In the presence of a solid phase of ferric (hydr)oxide and dissolved di- or monocarboxylic acids under oxic conditions, the reduction of $Hg^{II}$ in aqueous solution to $Hg^0$ occurs upon UV irradiation (Lin and Pehkonen, 1997). The systems studied for the photoreduction of

1440 $Hg^{II}$ are goethite ($\alpha$-FeOOH) + oxalate/formate, hematite ($\alpha$-Fe$_2$O$_3$) + oxalate, and maghemite ($\gamma$-Fe$_2$O$_3$) + oxalate. The experiments with filtered Xe light were conducted with 10 μM HgCl$_2$, 1 mM organic acid, and 0.1 g L$^{-1}$ ferric hydr(oxide) suspension, with a starting pH of 3.9. During some of the experiments, the pH increased substantially, resulting in the dominance of oxalate over hydrogen oxalate. Unlike oxalate, formate alone is not capable of reducing $Hg^{II}$ to $Hg^0$ under actinic light. It requires irradiation in the deep UV by processes such as (Leonori and Sturgeon, 2019):

$$Hg^{2+} + 2\,HCOO^- \rightleftarrows Hg(OOCH)_2 \xrightarrow{h\nu} Hg^0 + 2\,CO_2 + H_2$$

$$Hg^{2+} + HCOO^- \rightleftarrows Hg^{II}(OOCH)^+ \xrightarrow{h\nu} Hg^{\bullet+} + HCO_2^\bullet \ \text{and}\ Hg^{\bullet+} + HCOO^- \rightarrow Hg^0 + HCO_2^\bullet \quad \text{(Rxn 12)}$$

One study described $Hg^{II}$ reduction mediated by the carbon dioxide radical anion ($CO_2^{\bullet-}$) generated from formic acid via photosensitization by visible light-excited naphthoquinone (Berkovic et al., 2012). Iron(III) complexes with formate and oxalate are photolabile under UVA and visible, where a fast 1e–LMCT step generates $Fe^{2+}$ and eventually $CO_2^{\bullet-}$, which initiates a chain process (Mangiante et al., 2017; Baxendale and Bridge, 1955):

$$\equiv Fe^{III}\text{--}OH + HC_2O_4^- | HCOO^- \rightleftarrows \equiv Fe^{III}\text{--}C_2O_4^- | OOCH + H_2O$$

$$\equiv Fe^{III}\text{--}C_2O_4^- | OOCH \xrightarrow{h\nu} \equiv Fe^{II} + C_2O_4^{\bullet-} | HCO_2^\bullet$$

$$\equiv Fe^{II} \rightleftarrows \equiv + Fe^{2+} \quad \text{and} \quad HCO_2^\bullet \rightleftarrows H^+ + CO_2^{\bullet-}$$

$$C_2O_4^{\bullet-} \rightarrow CO_2 + CO_2^{\bullet-},\ CO_2^{\bullet-} + O_2 \rightarrow CO_2 + O_2^{\bullet-}\ \text{and}\ H^+ + O_2^{\bullet-} \rightleftarrows HO_2^\bullet \rightarrow \tfrac{1}{2}\,O_2 + \tfrac{1}{2}\,H_2O_2$$

$$Fe^{2+} + H_2O_2 \rightarrow Fe^{III} + HO^- + HO^\bullet \quad \text{(Rxn 13)}$$

$$Hg^{II} + CO_2^{\bullet-} | Fe^{2+} \equiv Fe^{II} \rightarrow Hg^I + CO_2 | Fe^{III} \equiv Fe^{III}\ ,\ Hg^I + CO_2^{\bullet-} | Fe^{2+} \equiv Fe^{II} \rightarrow Hg^0 + CO_2 | Fe^{III} \equiv Fe^{III}\ \text{(red.)}$$

$$2\,Hg^I \rightleftarrows Hg^0 + Hg^{II}\ \text{(disprop.)}$$

$$Hg^{0/I} + HO^\bullet \rightarrow Hg^{I/II} + HO^-,\ Fe^{2+} + HO^\bullet \rightarrow Fe^{III} + HO^-\ \text{and}\ H_2O_2 + HO^\bullet \rightarrow HO_2^\bullet + H_2O\ \text{(ox.)}$$

The oxic reaction system described by **Rxn 13** reaction formulas contains a number of ROS with different designations, such as strongly

reducing $CO_2^{\bullet-}$ and strongly oxidizing $HO^\bullet$, as extreme cases. One subsystem is Fenton's reagent (**Section 6.1.1**, **Rxn W6**), which produces $HO^\bullet$, for which each of the $Hg^{0/I}$, $Fe^{2+}$, and $H_2O_2$ competes to be oxidized. Except for the heterolytic photolysis of $Hg(\eta^2\text{-}C_2O_4)$, which produces $Hg^0$ from $Hg^{II}$ in a single step (**Section 6.1.2**, **Rxn W16**), the remaining redox steps involving metals are of the

single-electron type. The reduction of $Hg^{II}$ occurs via reactions with $HCO_2^{\bullet}|CO_2^{\bullet-}$ nucleophiles (**Section 6.1.2**, **Rxn W14**), which are both homogeneous and heterogeneous with dissolved and adsorbed ferrous species, respectively. A second-order homogeneous reaction coefficient of ~120-313 $M^{-1}$ $s^{-1}$ has been determined in the near-neutral pH range, with $Hg(OH)_2$ and $FeOH^+$ identified as the reactive species in solution (Amirbahman et al., 2013; Schwab et al., 2023). Under anoxic conditions, the rate of $Hg^0$ production derived from surface-catalyzed reduction on hematite and goethite has been described by the expression $k_{het}[\equiv Fe^{II}][Hg(OH)_2]$, with $k_{het}$ values of ~89 and ~78 $M^{-1}$ $s^{-1}$, respectively (Amirbahman et al., 2013). In an $O_2$-saturated, non-bubbled solution, a photo-stationary state between $Hg^{II}$ and $Hg^0$ occurs, indicating that the reduction pathways (**Rxn 13 red.**) are gradually balanced by oxidation pathways (**Rxn 13 ox.**, Ababneh et al., 2006). In the absence of $Hg^0$ removal, solubility limitations are easily exceeded during experiments (Lin and Pehkonen, 1997; Ababneh et al., 2006), resulting in the precipitation of colloidal $Hg^0$. The removal of dissolved $Hg^0$ by sorption on hydrous iron oxides, which is relevant here, is also documented (Richard et al., 2016). In the presence of competing anions, such as chloride, the rate of reduction decreases, in part owing to the formation of metastable, poorly soluble dimeric mercurous salts that compete with the disproportionation of $Hg^I$ to $Hg^0$ and $Hg^{II}$ (Pasakarnis et al., 2013).

## 7.5 Field observations of photoreduction in precipitation, clouds and fog

In precipitation and clouds, a strong correlation between Hg and total organic carbon was observed (Li et al., 2018; Åkerblom et al., 2015), suggesting that Hg-organics complexes are also important in aerosols. Authentic rain samples, where Hg-organics complexes dominate, present photoreduction rates ranging from 0.02–0.2 $h^{-1}$ (Yang et al., 2019; Saiz-Lopez et al., 2019; Fu et al., 2021). There have been a handful of measurements of Hg in cloud water (Li et al., 2018; Weiss-Penzias et al., 2018; Malcolm et al., 2003; Gerson et al., 2017; Huang et al., 2016a), but thus far, only a few studies on the photoreduction rate in this category of water exist (Li et al., 2018; Zhen et al., 2023; Gao et al., 2023). Photolysis rates in cloud water samples of 0.07–0.21 $h^{-1}$ measured in situ under actinic light and in the laboratory under UV (> 290 nm) light are consistent with those observed in precipitation. Whether the photoreduction rates observed in rain or cloud water are representative of atmospheric aerosols is questionable. $Hg^{II}$ in snowfall or freshly fallen snow has been reported to be labile for photoreduction (Steffen et al., 2008; Faïn et al., 2013). In temperate urban and pristine rural snow, within 24 h, approximately 50% (Lalonde et al., 2003; Lalonde et al., 2002) and, within 48 h, up to 90% (Poulain et al., 2004) of the newly deposited Hg can be effectively recycled back to the atmosphere. The reduction is reportedly strong even under cloudy conditions and is not limited by light (Faïn et al., 2013). In general, less than 5% of the Hg content of a snowpack is in the elemental form ($Hg^0$), which is concentrated stratigraphically in the first few centimeters. Nevertheless, if the rates are implemented as a mean value (~0.07 $h^{-1}$), determining the lifetime of atmospheric Hg against wet deposition, then the model-estimated wet deposition underestimates the observations by an average of 25% globally. Current global chemistry and transport models (GMOS-Chem) consider photoreduction on particles with the pool of $Hg^{II}$ complexed with organic ligands as the reactant (Shah et al., 2021).

## 8 Mercury isotope systematics and fractionation

Natural Hg contains seven stable isotopes with mass numbers of 196, 198, 199, 200, 201, 202 and 204. In the 1920s, significant separation of Hg into its isotopes was achieved through vaporization in a vacuum (Harkins and Mulliken, 1921; Mulliken, 1923; Brønsted and De Hevesy, 1921), a process that is conducted on a preparative scale through electromagnetic (Love, 1973) and photochemical (Vyazovetskii, 2012) methods. When the feed flow is in the form of DMHg, total gram quantities of highly enriched Hg isotopes can be obtained through cascade centrifugation (Babaev et al., 2010). The longest-lived radioisotope is [194]Hg at 444 y. Since it does not occur naturally, it cannot be used in the dating typical of [14]C. Two additional unstable isotopes ([197]Hg and [203]Hg, with half-lives of 64.1 h and 46.6 d, respectively) are valuable for instrumental neutron activation analysis and radiolabeled Hg compounds because of their decay by emission of readily detectable γrays. It was not until after the turn of the century, 80 years later, with the development of high-precision analytical instruments such as multi-collector inductively coupled plasma–mass spectrometry (MC–ICP–MS), that it became possible to measure differences in the naturally stable Hg isotopic compositions in the environment (Jackson, 2001; Lauretta et al., 2001). Natural processes, including redox reactions, complexation, sorption, precipitation, dissolution, evaporation, diffusion, and biological processes,

can alter the isotopic composition, i.e., cause stable isotope fractionation (cf. **Fig. 17**). Stable isotope analyses can, therefore, provide a previously untapped source of valuable information on the sources and biogeochemical cycling of natural and anthropogenic Hg. Isotopic fractionation refers to the division of a sample into two (or more) parts with different ratios of "heavy" and "light" isotopes than the original ratio. In isotopic jargon, if one part contains more heavy isotopes, it is said to be "enriched," while the other part is said to be "depleted". Hg has extremely large isotopic variation in nature, which, when normalized by the relative mass difference between isotopes, approaches that of traditional light element isotopes (Wiederhold, 2015). However, the overlapping signals from different fractionation processes can be a major challenge in deciphering natural isotopic signatures when tracing sources. It is important to determine the Hg stable isotope fractionation for *individual* key processes, which can be accomplished, inter alia, through controlled laboratory and field experiments. Stable isotope variations are reported as relative values compared with a reference standard (NIST SRM 3133 Hg solution, Blum and Bergquist, 2007):

$$\delta^{xxx}Hg = 1000 \cdot \left[ \left( {}^{xxx}Hg/{}^{198}Hg \right)_{sample} / \left( {}^{xxx}Hg/{}^{198}Hg \right)_{NIST3133} - 1 \right] \tag{13}$$

where ${}^{xxx}Hg/{}^{198}Hg$ ($R^{xxx/198}$) is the ratio of the isotopes with mass numbers xxx and 198. The prevailing practice of expressing isotope ratios relative to the lightest stable isotope for each element is not applicable to Hg because of the rarity of ${}^{196}Hg$ (0.15% occurrence). The standard unit for $\delta$ values is per mill (‰). $\delta^{202}Hg$ expresses the total mass-dependent fraction (TMDF, containing contributions from conventional mass-dependent fractionation; hereafter, MDF and nuclear field shift (NFS) are described in **Section 8. 1**), while the isotope anomalies caused by mass-independent fractionation, MIF are expressed by capital deltas, $\Delta$ is defined as the difference between the measured $\delta$ value and that predicted from the measured $\delta^{202}Hg$ value and the scale factor for the kinetic MDF ($\beta_{KIE-MDF}^{xxx}$; see **Section 8. 1**) and is approximated for $\delta$ values < 10‰ according to:

$$\Delta^{xxx}Hg = \delta^{xxx}Hg - \beta_{KIE-MDF}^{xxx} \cdot \delta^{202}Hg \tag{14}$$

which is expressed numerically for each relevant Hg isotope:

$\Delta^{196}Hg = \delta^{196}Hg + 0.508 \cdot \delta^{202}Hg$ , $\Delta^{199}Hg = \delta^{199}Hg - 0.252 \cdot \delta^{202}Hg$ , $\Delta^{200}Hg = \delta^{200}Hg - 0.502 \cdot \delta^{202}Hg$ , $\Delta^{201}Hg = \delta^{201}Hg - 0.752 \cdot \delta^{202}Hg$ and $\Delta^{204}Hg = \delta^{204}Hg - 1.493 \cdot \delta^{202}Hg$.

The fractionation between two compounds A and B (assuming that A is a product of a reaction and that B is the remaining reactant) is expressed with the fractionation factor, $\alpha$, which is defined as the ratio of the isotope ratios in the compounds:

$$\alpha_{A-B}^{xxx} = R_A^{xxx/198} / R_B^{xxx/198} = R_A^{xxx} / R_B^{xxx} = \frac{1000 + (\delta^{xxx}Hg)_A}{1000 + (\delta^{xxx}Hg)_B} \tag{15}$$

The last term is obtained by substituting **Eq. 13** into the first term of **Eq. 15**. Actual $\delta$ values are usually very close to unity. Therefore, it is usually more practical to use an enrichment factor:

$$\varepsilon_{A-B}^{xxx} = (\delta^{xxx}Hg)_A - (\delta^{xxx}Hg)_B = 1000 \cdot (\alpha_{A-B}^{xxx} - 1) \cong 1000 \cdot \ln\alpha_{A-B}^{xxx} \tag{16}$$

The last similarity is valid only for $\delta$ values less than 10‰. Substitute **Eq. 14** into **Eq. 16** and obtain:

$$\varepsilon_{A-B}^{xxx} \cong \{ (\Delta^{xxx}Hg)_A - (\Delta^{xxx}Hg)_B \} + \beta_{KIE-MDF}^{xxx} \cdot [(\delta^{xxx}Hg)_A - (\delta^{xxx}Hg)_B] \tag{17}$$

**Eq. 17** expresses total fractionation during the process A → B, with the first term representing the MIF enrichment factor and the second term representing the total mass-dependent enrichment factor. Thus, the enrichment factor for MIF is written as a capital epsilon:

$$E_{A-B}^{xxx} = \{ (\Delta^{xxx}Hg)_A - (\Delta^{xxx}Hg)_B \} = \varepsilon_{A-B}^{xxx} - \beta_{KIE-MDF}^{xxx} \cdot \varepsilon_{A-B}^{202} \tag{18}$$

Many kinetic processes can be described as Rayleigh fractionation, which is an irreversible process in an open system involving the progressive removal of a fraction of a trace substance from a larger reservoir. It is described by the following differential equation:

$$d \ln R^{xxx} = (\alpha^{xxx} - 1) \cdot d \ln f_R \tag{19}$$

If the fractionation factor is constant, the differential equation can be integrated directly into the expression:

$$R^{xxx}/(R^{xxx})_0 = f_R^{(\alpha^{xxx}-1)} \tag{20}$$

where $(R^{xxx})_0$ is the isotope ratio of the initial reservoir (when $f_R = 1$) and where $R^{xxx}$ is the isotope ratio of the reservoir at a given time when the fraction of initial material remaining in the reservoir is defined by $f_R$. The following expression is often used to evaluate the fractionation factor:

$$\ln\frac{1000 + \delta^{xxx}Hg}{1000 + (\delta^{xxx}Hg)_0} = (\alpha^{xxx}-1) \cdot \ln f_R \tag{21}$$

The process tends to enrich the heavier isotopes in the reservoir ($\alpha < 1$, normal kinetic isotope effect, KIE) rather than removing the heavier isotopes from the reservoir more rapidly ($\alpha > 1$, inverse KIE).

**8.1 Conventional mass-dependent and mass-independent fractionation**

The scaling factor $\beta$ describes the relationship between the fractionation factors as follows:

$$\alpha^{xxx} = (\alpha^{202})^\beta \tag{22}$$

where $\beta$ for mass-dependent equilibrium fractionation ($\beta_{EIE-MDF}$) and kinetic fractionation ($\beta_{KIE-MDF}$) are as follows (Young et al., 2002):

$$\beta_{EIE-MDF} = \frac{1/m_{198} - 1/m_{xxx}}{1/m_{198} - 1/m_{202}} \tag{23}$$

$$\beta_{KIE-MDF} = \frac{\ln(m_{198}/m_{xxx})}{\ln(m_{198}/m_{202})} \tag{24}$$

The equilibrium MDF resulting from the differences in zero-point vibrational energy (ZPE) distances and the kinetic MDF resulting from the differences in dissociation energies between the isotopologues and their respective effects can be expressed in two rules: Heavier isotopes are preferentially concentrated in compounds with the highest force constant, where the element is most rigidly bound and has greater potential energy. Conversely, compounds enriched in lighter isotopes have weaker bonds and require less energy to break, so they preferentially enter chemical reactions and are enriched in the product (Criss, 1999). Combining kinetic and equilibrium MDF makes it possible to achieve a limit of approximately 10‰ fractionation (Sun et al., 2022).

Properties of nuclei, such as nuclear size and shape or the presence of non-zero nuclear spins, may trigger isotope fractionation that does *not* follow the expected MDF relationships. The nuclear field shift (NFS, Rosenthal and Breit, 1932) is the interaction of the nuclear volume with electrons (NVE, Schauble, 2007). It is highly relevant for very heavy metals, including Hg, Tl, Pb, and U. NFS involves a shift in the ground electronic energy of an atom or molecule due to differences in nuclear size and shape between isotopes. The shift caused by an odd (neutron number) nucleus scales non-linearly between those of the even isotopes of the next highest and lowest atomic masses. The odd isotope electronic energy level is shifted toward the next lower even nucleus (odd-even staggering). Owing to its smaller size and greater surface charge density, the electronic energy of a light isotope is lower than that of a heavier isotope. The amount of shift is a product of two factors: the electron density at the nucleus and the charge, size, and shape of the nucleus and the change in the latter two between isotopes. Hg orbital electrons significantly overlap with the nucleus, whereas 5p, 5d, and 4f orbitals do not, although f electrons in inner shells have a smaller screening effect on 6s-valence electrons (Bigeleisen and Wolfsberg, 1957). The lowest energy of a system occurs when the heavier isotopes of Hg are enriched in chemical species with the fewest s-electrons in the bonding or valence orbital. The largest shifts, therefore, occur when the number of Hg 6s electrons is greatly reduced by the formation of an ionic bond (to an electronegative element), while a covalent bond has less influence. Examples of Hg species in the former category are chloro- or aqua-complexes with high coordination numbers (e.g., $[Hg(H_2O)_6]^{2+}$), while the latter includes soft ligands with typical linear bi-coordination (e.g., $Hg(SH)_2$ and $(CH_3)_2Hg$). The scale factor of nuclear volume fractionation ($\beta_{NFS}$) is defined as follows:

$$\beta_{NFS} = \frac{\langle r_{198}^2 \rangle - \langle r_{xxx}^2 \rangle}{\langle r_{198}^2 \rangle - \langle r_{202}^2 \rangle} \tag{25}$$

where $\langle r^2 \rangle$ describes the mean-square nuclear charge radii of different isotopes. Coincidentally, MDF and NFS with $^{198}Hg$, $^{200}Hg$ and

[202]Hg show almost identical β values, but [199]Hg, [201]Hg and [204]Hg and, to a lesser extent, [196]Hg show distinct non-mass dependent signatures due to NFS. Only a small proportion of the NFS is mass independent because it creates a deviation from MDF (Yang and Liu, 2015). The mass-dependent part of the two effects can be synergistic (increasing TMDF) or antagonistic (decreasing TMDE), with the former being dominant for Hg redox chemistry (Hintelmann and Zheng, 2011; Jiskra et al., 2012). The MDF scale is proportional to $1/T^2$, whereas the NFS scale is proportional to $1/T$ and is more prominent than MDF for the Hg red-ox reactions

studied (Schauble, 2007). Among the commonly measured isotopes 198-202, a minor to moderate level of MIF has been experimentally observed in the odd isotopes 199 ($\leq 0.6‰$) and 201 as a result of NFS. NFS has been described for equilibrium exchange reactions but has never been extended to kinetic processes. In contrast to the small magnitude observed in natural samples, the possibility has recently been suggested that nonequilibrium isotopic effects of NFS in photodissociation may give rise to a significant magnitude of MIF (Motta et al., 2020b).

The only effect that has been documented to lead to significant odd-mass number Hg MIF (odd-MIF) in present-day surface ecosystems is the magnetic isotope effect (MgIE). MgIE is a purely kinetic effect triggered by the formation of a long-lived radical pair after a primary process that causes homolysis of a Hg-ligand bond upon photolytic excitation. (**Fig. 11**). Among the stable isotopes of Hg, only [199]Hg and [201]Hg (odd mass numbers) have non-zero nuclear spin and momentum, with half-integer ($^1/_2$ and $^3/_2$, respectively) spins. MgIE arises when hyperfine coupling (HFC) acts on a spin-coherent solvent-separated radical pair after

dissociation by changing the rate of intersystem crossing from singlet to triplet (S↔T) or vice versa (T↔S) in odd Hg isotopes. Radical pairing and MgIE are suppressed in mercuric complexes with strong spin–orbit coupling (containing bromine and iodine ligands), favoring spin mixing and to the ground state, while S-, Cl- and C-bonded complexes with generally weak spin–orbit coupling favor strong MgIE (Motta et al., 2020a). If the radical pair is born in the triplet state (lower panel of **Fig. 11**), HFCs are induced, enriching odd isotopes in the resulting singlet state. The singlet radical pair can then recombine to the ground state, resulting in

odd isotope enrichment in the reactant, expressed as (+)MgIE. When the radical pair is in the singlet state (top panel of **Fig. 11**), the overall effect is to deplete odd isotopes in the reactant, as expressed by (–)MgIE, because mainly the odd isotopes with the majority in the triplet radical pair dissociate into free radicals. A computational study has explained why the photodissociation of monomethyl Hg species in nature is observed to yield only (+)MgIE, whereas the photolysis of inorganic mercuric complexes may yield positive or negative MgIE, depending on the reaction conditions and the degree of complex ligation (**Section 8.4**).

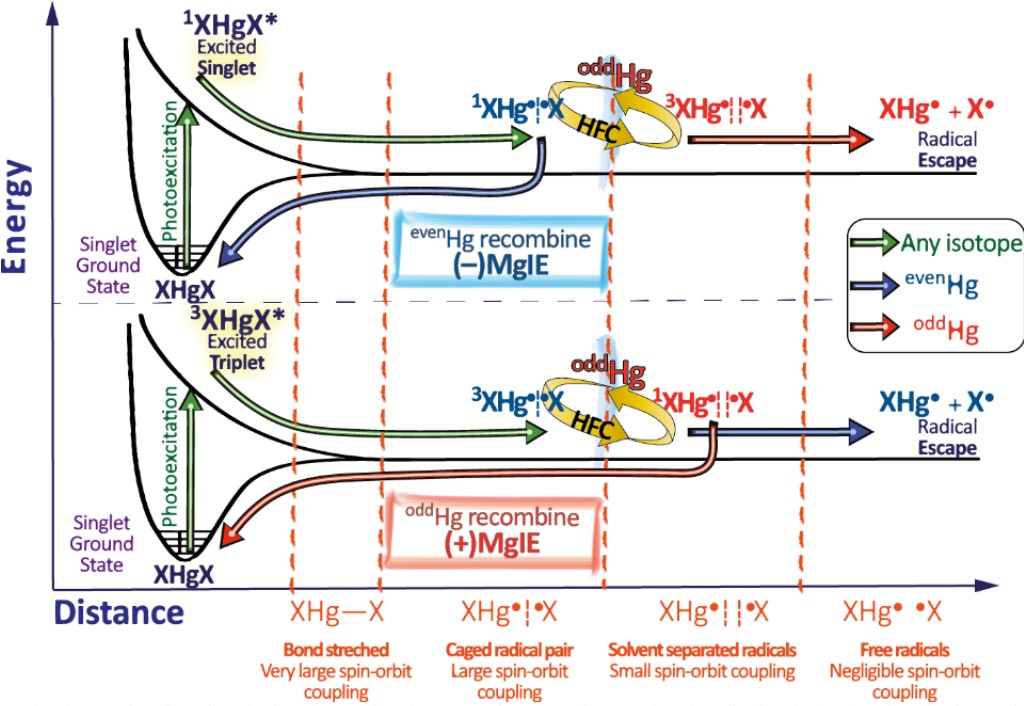

**Figure 11.** General scheme for the photolysis of a molecule to produce a spin-correlated radical pair (RP). The singlet and triplet RPs can be interconverted by intersystem crossing. Both the singlet and triplet RPs can escape from the solvent cage. Only the singlet RP can recombine. Adopted from Turro (1983) and Motta et al. (2020a).

Thus, odd-MIF results from both the MgIE and NFS mechanisms. MgIE is most effective in viscous solvents, where a "solvent cage" environment is possible (Turro, 1983). In addition, a seemingly enigmatic even mass number Hg isotope, MIF (even-MIF), has been observed in samples of atmospheric origin or deposition. However, analog atmospheric photochemical anomalous isotope fractionation is well known for the lighter (traditional) elements for which MIF (containing three or more stable isotopes), such as oxygen and sulfur, can be detected. However, understanding the underlying causes of multi-isotope anomalous fractionation is limited because investigations require detailed quantum mechanical calculations at the molecular isotope level, e.g., photodissociation. Recently, rock records have revealed significant even-MIF in the Archean atmosphere, which lacked an ozone ($O_3$) layer to filter UVC from actinic light, suggesting that contemporary UVC-induced atmospheric chemistry may be responsible for the coupled changes in even-MIF for both Hg and sulfur (Zerkle et al., 2020). Correlations between these entities have also been observed in marine aerosols in the Southern Hemisphere (Auyang et al., 2022).

### 8.1.1 MIF signatures as additional tracers

The isotopic measurement of Hg results in up to six useful isotopic signatures ($\delta^{202}Hg$, $\Delta^{196}Hg$, $\Delta^{199}Hg$, $\Delta^{200}Hg$, $\Delta^{201}Hg$ and $\Delta^{204}Hg$). In addition, pairs of these signatures have been utilized to distinguish between fractionation mechanisms. The relationship between these signatures is typically illustrated using a three-isotope plot. To interpret the experimental results satisfactorily, specific robust linear regression methods are recommended (Stephan and Trappitsch, 2023). On the basis of early results (Bergquist and Blum, 2007), it was assumed that photoreduction of $Hg^{II}$ to $Hg^0$ would result in a $\Delta^{199}Hg/\Delta^{201}Hg$ ratio of unity. Several investigated photoreactions exhibit just a ratio of 1 within the margin of error (refer to **Table 8**, **Section 8.4**). However, further data have shown that this is not always the case, as the slope depends on factors such as the complexing ligand and reaction conditions. Clearly, the odd-MIF signature for the photolysis of organomercurials is different from that stipulated for inorganic Hg. The photolytic degradation of $MMHg^+$ species results in a variation in $\Delta^{199}Hg/\Delta^{201}Hg$ ranging from 1.17 to 1.38 depending on the reaction conditions (Bergquist and Blum, 2007; Chandan et al., 2015; Rose et al., 2015; Malinovsky et al., 2010). Compared with MgIE, NFS generally results in a much weaker MIF, with greater anomalous fractionation of $^{199}Hg$ than of $^{201}Hg$, which approaches a ratio of ~1.6. However, the NFS should be confirmed via alternative methods when the experimentally measured NFS is too limited to determine a definitive odd-MIF ratio (Motta et al., 2020b). Another commonly used parameter is $\Delta^{199}Hg/\delta^{202}Hg$, which describes the degree of odd-MIF in relation to TMDF. The even-MIF signature of $\Delta^{200}Hg$ to $\Delta^{204}Hg$, which is negative in natural samples (air, rainfall, and fish), is discussed in **Section 8.2.4**.

### 8.2 Isotopic characteristics of atmospheric mercury

**Fig. 13** summarizes the magnitude of the isotopic observations reported in the literature on the main fractions of Hg in the atmosphere, namely, gaseous Hg dominated by $Hg^0$, particle-bound Hg and Hg associated with hydrometeors (rain, snow and water from clouds and fog). Here, $\delta^{202}Hg$, $\Delta^{199}Hg$, and $\Delta^{200}Hg$ are used to describe TMDF, odd-MIF, and even-MIF, respectively. The number of isotopically resolved samples has increased dramatically in recent years. Readers should consult the literature regularly to stay up to date. On the other hand, this development justifies revisiting the topic, even though it has been satisfactorily addressed in the recent past (Kwon et al., 2020). Notably, the spatial distribution of available data is heavily skewed toward North America, Europe, and East Asia, and observations from large parts of the world are missing (**Fig. 13**). However, as far as marine remote regions are concerned, recent oceanographic expeditions have contributed to an increasing amount of data. Further below, the in situ and laboratory experiments performed thus far to study the gas exchange of $Hg^0$ between air and water, soil and foliage in terms of isotope fractionation are discussed.

### 8.2.1 Gaseous Hg

The following part excludes an early series of measurements where the air was not filtered prior to sampling (Rolison et al., 2013). The referenced series of measurements should be considered TAM, not total gaseous mercury (TGM). TGM significantly affects $\delta^{202}Hg$ (–3.75 to 1.52‰) and $\Delta^{199}Hg$ (–0.62 to 1.32‰) but is more limited to $\Delta^{200}Hg$ (–0.22 to 0.11‰). Analogous to $Hg^{II}$ deposited in biomass and fossil fuels such as coal (Sun et al., 2014), $Hg^0$ in natural gas (Washburn et al., 2018) and in smoke from spontaneous combustion in coal fields (Sun et al., 2023) has strongly negative $\delta^{202}Hg$ values. This differs from the majority of $Hg^0$ in ambient air, which is isotopically heavy (often with positive $\delta^{202}Hg$ values). Terrestrial background air (rural, subpolar and forest in **Fig. 13a**) has higher $\delta^{202}Hg^0$ values because it is modified by vegetation, which preferentially incorporates lighter isotopes of $Hg^0$ into the foliage.

Foliar uptake of $Hg^0$ is discussed in more detail in **Section 8.6.2**. The estimate of atmospheric $Hg^0$ dry deposition to vegetation has recently been revised and constrained to approximately 2300 Mg $yr^{-1}$ (Feinberg et al., 2022), and together with the large negative $\varepsilon_{foliage/air}$ of the process, the global atmospheric $Hg^0$ pool is estimated to have a $\delta^{202}Hg$ mean of 0.5‰, in contrast to that of $Hg^0$ from anthropogenic sources (global bulk mean of –0.7‰, Sun et al., 2019). Studies examining the vertical distribution of mercury ($Hg^0$) concentrations from near the ground to above the canopy in different forest types reveal clear gradients averaging 10% (Wang et al., 2022) and 20% (Fu et al., 2016b) of ambient $Hg^0$. Under stable conditions, such as during summer nights, $Hg^0$ levels are strongly depleted below the canopy (Fu et al., 2016b; Mao et al., 2008; Poissant et al., 2008; Lan et al., 2012; Fu et al., 2019a). Thus, isotopic measurements of above-canopy air versus in-canopy air (Wang et al., 2022; Fu et al., 2016b) and daytime air versus nighttime air in forests (Kurz et al., 2020) show statistically significant differences ($p < 0.01$) in $\delta^{202}Hg^0$. For a deciduous forest in Northeast China, Fu et al. (2019a) reported that $\delta^{202}Hg^0$ in biweekly air samples during the growing season was 0.35 to 0.99‰ higher than that during the dormant season. In a subtropical, perennial forest in southwestern China, where there is little seasonal variation in the photosynthetic activity of vegetation, the existing seasonal variation in $\delta^{202}Hg^0$ (with an amplitude of 0.4 ‰) can be attributed to the influence of long-range anthropogenic emissions, which primarily occur during the warmer seasons. However, over the last five to seven years, the air concentrations of $Hg^0$ have decreased significantly in the two mentioned forest reserves due to reduced regional anthropogenic emissions, as evidenced by the median value of $\delta^{202}Hg^0$ shifting from 0.42 to 0.46‰ and from 0.17 to 0.57‰. The marine $\delta^{202}Hg^0$ data ($n = 112$) are significantly lower (Wilcoxon test, $p < 0.01$) than those from the forest ($n = 113$). Coastal measurements in the Gulf of Mexico show that the marine-influenced air isotopically represents background air modified by $Hg^0$ emitted from the sea after being formed in surface water by photoreduction (Demers et al., 2015). Measurements in the marine boundary layer of the offshore East China Sea indicated that airborne $Hg^0$ is essentially a binary mixture of anthropogenic outflow from mainland China and air masses from the sea (significantly correlated $\delta^{202}Hg^0$ and $\Delta^{199}Hg^0$ vs. $C_{Hg^0}^{-1}$), with an extrapolated $\Delta^{199}Hg^0$ of –0.26‰ for the marine component. The extrapolated $\Delta^{199}Hg^0$ value agrees well with observations made in Hawaii with passive samplers (Szponar et al., 2020) as well as with the signatures of a larger number of samples from Mauna Loa (3397 m a s l) in the free troposphere (Tate et al., 2023; Yamakawa et al., 2024).

The generally negative signature of $\Delta^{199}Hg^0$ in the background air indicates that $Hg^0$ has been added to the pool subsequent to $Hg^{II}$ photoreduction (of the variant that induces (+)MgIE in the reactant and complements it by depleting the product $Hg^0$ isotopically for odd isotopes) in oceans and aerosols. This is supported by atmospheric $Hg^0$ exhibiting $\Delta^{199}Hg/\Delta^{201}Hg$ slopes close to unity (Kwon et al., 2020), similar to aqueous photoreduction of inorganic $Hg^{II}$. However, not all photolytically controlled $Hg^0$ re-emissions from terrestrial ecosystems contribute to negative $\Delta^{199}Hg^0$ values in the atmosphere. An analysis of gas exchange in a subtropical beech forest revealed bidirectional fluxes of $Hg^0$, with uptake partially balanced by reemission of previously metabolized $Hg^{II}$. Photo reduction recirculates $Hg^0$, contradicting a retro-flux of deposited $Hg^0$ at the leaf surface (data in **Fig. 18a**). This re-emission is isotopically distinct in that it is enriched in odd isotopes compared with ambient air (Yuan et al., 2019b), indicating that leaf photoreduction induces (–)MgIE, as reported for $Hg^{II}$ bound to organic ligands containing sulfur or nitrogen in low oxidation states (Motta et al., 2020b; Zheng and Hintelmann, 2010b). A mass balance based on isotope measurements indicates that, compared with the uptake of $Hg^0$ from the air, re-emission from beech foliage gradually increases from emergence to senescence, accounting for an average of 30% (Yuan et al., 2019b). Observations from a temperate deciduous forest revealed 0.06–0.09 ‰ higher $\Delta^{199}Hg^0$ values during the growing season than in winter, suggesting that foliar $Hg^0$ efflux contributes to the atmospheric enrichment of odd Hg isotopes (Fu et al., 2019a).

The large spread of odd-MIF shown by $Hg^0$ in polar air (Araujo et al., 2022; Yu et al., 2021; Sherman et al., 2010, **Fig. 13a**) is due to the portion of the collected data that includes $Hg^0$ depletion events in the spring and $Hg^0$ enhancement during the summer, when reemissions of $Hg^0$ occur from the cryosphere. **Fig. 12** shows the isotopic compositions of airborne Hg fractionated into $Hg^0$ and $Hg^{II}$ (RM) during the Arctic spring (at three stations) compared with the corresponding data from a background station in the Pyrenees (Fu et al., 2021). With respect to $\Delta^{199}Hg$, a dichotomy between the polar and temperate data is striking for both $Hg^0$ and $Hg^{II}$ (RM), in that montane oxidized Hg is enriched in a limited range (0.14 to 0.77‰) whereas the polar $Hg^{II}$ is depleted in a greater range (–2.15 to –0.18‰), with a complementary relationship existing for $Hg^0$ (–0.31 to –0.16‰ versus –0.22 to 1.32‰). This relationship

could be caused by surface layer airborne Hg being strongly influenced by the oxidation of $Hg^0$ to $Hg^{II}$, which is controlled by halogen atoms during AMDEs, processes characterized by $E^{199}Hg$ values of –0.37‰ and –0.23‰ for $Cl^\bullet$-initiated and $Br^\bullet$-initiated oxidation, respectively **(Table 7**, Sun et al., 2016). In this way, the remaining reactant is driven to a higher $\Delta^{199}Hg^0$ and the molecular products assume negative $\Delta^{199}Hg^{II}$ values. However, this interpretation is not corroborated by the measured $\Delta^{199}Hg/\Delta^{201}Hg$ ratio of nearly unity in airborne Hg, which is more typically indicative of $Hg^{II}$ photo-reduction (-MgIE) occurring in snow. It has been proposed that this process also operates in aerosols of the boundary layer, with $Hg^0$ reemissions providing such a strong positive imprint that the entire boundary layer of the $Hg^0$ pool becomes enriched in odd isotopes (Araujo et al., 2022). Isotopic measurements of $Hg^{II}(g)$ separated from $Hg^{II}(p)$ using CEM (cf. **Section 3.1**) have commenced and are anticipated to elucidate the mechanisms underlying the pronounced fractionation of odd isotopes in airborne $Hg^0$ and $Hg^{II}$. Several such datasets are currently in preparation for publication. Furthermore, $Hg^0$ in the Arctic during the dark period of the year and from the Antarctic Peninsula throughout the year (Yu et al., 2021) shares a consistently slightly negative $\Delta^{199}Hg^0$ with other background air (represented by montane air in **Fig. 13a**). In the late Arctic summer, minimum $\Delta^{199}Hg^0$ values (approaching –0.5 ‰) are observed uniformly without much variation from coastal stations around the Arctic Ocean, which are thought to result from photoreduction of cryospheric $Hg^{II}$, a substrate that has been strongly depleted of odd isotopes during months of long sunshine (Araujo et al., 2022).

$\Delta^{200}Hg^0$ is generally negative for non-fossil/anthropogenic sources, while the remainder is significantly shifted to higher values (Wilcoxon T-test, e.g., natural gas vs. arid data, $p < 0.01$). As mentioned above, even-MIF is generated exclusively by atmospheric chemical processes, which may be mainly limited to molecular $Hg^{I,II}$ photolysis processes (Sun et al., 2022), of which $Hg^0$ is a product. The marine and polar $\Delta^{200}Hg^0$ data have the most negative values. For example, a recently published TGM record from Mauna Loa (not shown in **Fig. 13a**) in the Pacific Ocean has $\Delta^{200}Hg$ values as low as –0.20‰ (Yamakawa et al., 2024). The polar pool as a unit significantly shifted toward lower $\Delta^{200}Hg^0$ values than did the forest pool (Wilcoxon T test, $p < 0.05$). One can only speculate as to the reason, but it should be mentioned in the context of a halogen-rich environment that any presence of Cl-initiated $Hg^0$ oxidation in the gas phase will result in depletion of $^{200}Hg$ in the reactant pool ($E^{200}Hg \sim 0.06‰$, Sun et al., 2016). Owing to its relatively limited range, ambient $\Delta^{200}Hg^0$ and $\Delta^{204}Hg^0$ are considered conservative tracers of atmospheric $Hg^0$ deposition, and terrestrial surface and water $\Delta^{200}Hg$ and $\Delta^{204}Hg$ values can constrain the relative contribution of $Hg^0$ to $Hg^{II}$ deposition. Throughout, a median value of -0.05‰ (IQR –0.02 to –0.08‰) of $\Delta^{200}Hg^0$ was used to calculate this contribution to atmospheric transfer to soil ($\Delta^{200}Hg^{II}\sim0‰$, Enrico et al., 2016; Zhou et al., 2021; Zheng et al., 2016) and oceans ($\Delta^{200}Hg\sim0.04‰$, Jiskra et al., 2012). The quantitative AMDEs observed in Alaska are isotopically mass balanced in that the $\Delta^{200}Hg^{II}$ in snow (-0.06‰) corresponds, within the measurement uncertainty, to that in ambient $Hg^0$ (–0.05‰).

### 8.2.2 Aerosol-bound Hg

While $Hg^0$ has a relatively long lifetime and $Hg^{II}(g)$ has a short lifetime, the lifetime of particle-bound Hg (PBM, $Hg^{II}_p$) reflects that of particles, which varies from days to months due to their size and composition. Isotopic analyses have been performed on airborne $PM_{2.5}$, $PM_{10}$, and TSP, as well as on particles in precipitation. Studies of urban air, regionally polluted air, and air associated with anthropogenic emissions (CFPP, traffic and waste incineration, etc.) are well represented and strongly biased toward Asia. As reviewed and discussed in Kwon et al. (2020), attempts to decipher the cause of seasonal variations in urban and industrial air are challenging in environments with a plethora of local and regional emission sources. However, primary particles from fossil fuel and biomass combustion inherit the clearly negative but highly variable $\delta^{202}Hg^{II}(p)$ and the less negative $\Delta^{199}Hg^{II}(p)$ of the material. The large range in $\Delta^{199}Hg^{II}(p)$ (–0.93 to 1.5‰) around the origin depends on $Hg^{II}(p)$ photoreduction with (+)MgIE, halogen atom-initiated $Hg^0$ oxidation or, more speculatively, $Hg^{II}(p)$ photoreduction with (–)MgIE, driving the data to extremes. In a series of papers, including field measurements of particle-bound isotopic Hg in regionally polluted air (Huang et al., 2016b; Huang et al., 2019; Qiu et al., 2022; Zhang et al., 2022) and laboratory experiments (Huang et al., 2021; Huang et al., 2015), Chen and colleagues have focused on the effect of (+)MgIE photoreduction, which is accelerated in the presence of a particle surface liquid layer (wet haze) and water-soluble organic carbon as a reducing agent (Zhang et al., 2022). Several peripheral monitoring stations in China, primarily receptors of long-range particle transport, generally measure positive $\Delta^{199}Hg^{II}(p)$ values (Fu et al., 2019b). A strong anticorrelation between $\Delta^{199}Hg^{II}(p)$ (up to ~1.2‰, but initially at near zero) and the concentration of particle-bound Hg, rationalized as caused by photo-produced $Hg^0$ loss from aerosols, was observed in samples from these stations, with the major potential source area identified

as northeastern China and the regions along the lower reaches of the Yangtze River to its mouth (Fu et al., 2019b). The results indicate that the globally modeled tropospheric lifetime of $Hg^{II}$ against photoreduction in aerosols and clouds of nearly two weeks (Horowitz et al., 2017) is significantly shorter in East Asia, possibly because of a greater fraction of organic aerosols. As shown in **Fig. 13b**, there is a statistical anomaly in the PBM polar data for all reported isotopic signatures: positively shifted $\delta^{202}Hg^{II}(p)$, negatively shifted $\Delta^{199}Hg^{II}(p)$ and negatively shifted $\Delta^{200}Hg^{II}(p)$. It is represented in both Arctic (Araujo et al., 2022; Zheng et al., 2021) and Antarctic (Auyang et al., 2022; Li et al., 2020a) data in conjunction with AMDEs. In the high Arctic (~83°N), there is good isotopic agreement between ambient $Hg^0$ and PBM associated with nearly complete AMDEs, as would be expected. Moreover, for less quantitative oxidation, PBM is isotopically lighter than $Hg^0$, analogous to kinetic isotope fractionation during oxidation and subsequent uptake of $Hg^{II}(g)$ on particles (such as Arctic haze). As described above, halogen atom-driven gas-phase oxidation induces a negative $\Delta^{199}Hg^{II}$ (Sun et al., 2016; Auyang et al., 2022), which is consistent with the observed signature in PBM. The interpretation of Zheng et al. (2021) that gas-phase oxidation uniquely shapes isotopic fractionation has been challenged by Araujo et al. (2022), who instead consider (–)MgIE photoreduction in aerosols as the imprinting source. The Antarctic coast has shown uniquely high positive $\delta^{202}Hg^{II}(p)$ values (up to ~3‰ and anticorrelated with $\Delta^{199}Hg^{II}(p)$) in air masses transported by katabatic winds from the continental shelf, where oxidation of $Hg^0$ persists during summer (Li et al., 2020a). Under precipitation (**Section 8.2.3**), the high $\Delta^{200}Hg^{II}$ values measured in southern Canada are addressed, noting that this also applies to the particulate fraction in precipitation, which is included in the rural PBM category (**Fig. 13b**).

### 8.2.3 Hg in precipitation

Measurements of Hg isotopes in precipitation samples (including fog and cloud water) have been reported at sites in the Northern Hemisphere (map in **Fig. 13c**), mostly in North America. Compared with the $Hg^0$ and PBM samples, the precipitation samples presented the greatest scatter in both $\Delta^{199}Hg$ and $\Delta^{200}Hg$. Nevertheless, the isotopic distribution pattern in precipitation water is generally similar to that of PBM, which is scavenged in precipitation during rainout and washout processes. Precipitation in the vicinity of anthropogenic emission sources (such as CFPPs) tends to be isotopically distinct, with particularly negative $\delta^{202}Hg^{II}$ values (Sherman et al., 2012). Precipitation from more pristine areas has a $\delta^{202}Hg^{II}$ that is shifted in a positive direction (significant for marine, polar and rural categories, Wilcoxon t test, $p < 0.01$) compared with urban precipitation and precipitation near point sources. The general differences between $Hg^0$ and precipitation/PBM in terms of MIF signatures (negative $\Delta^{199}Hg^0$ & $\Delta^{201}Hg^0$ vs. positive $\Delta^{199}Hg^{II}$ & $\Delta^{201}Hg^{II}$ and negative $\Delta^{200}Hg^0$ & positive $\Delta^{204}Hg^0$ vs. positive $\Delta^{200}Hg^{II}$ & negative $\Delta^{204}Hg^{II}$, respectively) are explained by atmospheric redox processes (Auyang et al., 2022; Kwon et al., 2020). In the case of even-MIF, chlorine atom-initiated gas-phase oxidation is known to induce a limited positive $\Delta^{200}Hg^{II}$ in the product. However, its observed magnitude cannot explain the highest $\Delta^{200}Hg^{II}$ measured in precipitation (Kurz et al., 2021; Chen et al., 2012; Yuan et al., 2022) in North America. Cai and Chen (2015) reported a trend toward increasing $\Delta^{200}Hg^{II}$ in background precipitation as moving northward along the mid-latitudes of the Northern Hemisphere (~20–45°N), but only with data from a unique station anomalous with greater statistical significance. A one-year measurement north of Lake Ontario (Chen et al., 2012), separated by a full decade from measurements at the same site limited to the colder parts of the year (Yuan et al., 2022), has shown that precipitation in winter often contains high values of $\Delta^{200}Hg^{II}$ (and, at the same time, strongly negative $\Delta^{204}Hg^{II}$ values). During the full-year measurement in 2010, filtered precipitation samples presented a $\Delta^{200}Hg^{II}$ in the range of 0.21 to 1.24‰, whereas during the colder months around the turn of the year 2020-21, the same category of samples contained between 0.25 to 1.19‰ and between –1.97 to 0.37‰ for $\Delta^{200}Hg^{II}$ and $\Delta^{204}Hg^{II}$, respectively. During the last campaign, isotopic analysis was also performed on precipitation particles, which presented significantly lower positive $\Delta^{200}Hg^{II}$ values (up to 0.37‰) and less negative $\Delta^{204}Hg^{II}$ values (down to –0.84‰). Intermittently, the particle phase has the opposite sign to the solute phase in the same precipitation sample with respect to both odd- and even-MIF. This, together with a time series of unrelated odd-MIF and even-MIF trends during events with large fluctuations in these values, has been interpreted as the influence of the circumpolar vortex with varying contributions of tropospheric and stratospheric air, with the transport of the latter air masses explaining more extreme even-MIF values (Yuan et al., 2022). Compared with snow samples from the Canadian station north of Lake Ontario, rain samples from the Canadian station north of Lake Ontario generally have more moderately positive $\Delta^{200}Hg^{II}$ values, which is consistent with precipitation observations in the mid-latitudinal USA (Kurz et al., 2021; Demers et al., 2013; Gratz et al., 2010; Sherman et al., 2015), Europe (Fu et al., 2021; Enrico et al., 2016), the Tibetan Plateau (Yuan et al., 2015) and the Pacific

Ocean (Motta et al., 2019; Washburn et al., 2021). Although cloud water (Fu et al., 2021; Zhen et al., 2024) and fog water (Washburn et al., 2021) have been isotopically analyzed, there are no apparent differences between them or significant differences from rain samples. In cloud water, Hg speciation with increasing complexation with DOM has been shown to correlate with odd-MIF values (Zhen et al., 2024), which is consistent with the view that these mercuric complexes are photolabile. Polar precipitation samples (only those from AMDEs are reported in the literature, Araujo et al., 2022; Sherman et al., 2012; Zheng et al., 2021) consistently have slightly negative $\Delta^{200}Hg^{II}$ values, which differ from those of precipitation samples from all other provenances, which have positive median values. The reason for these observations is plausibly that oxidation is so advanced during these AMDEs that the $Hg^{II}$ scavenged by precipitation approaches the same isotopic values as the $Hg^0$ in the polar air before the AMDE.

### 8.2.4 Even-MIF ($\Delta^{200}Hg/\Delta^{204}Hg$) ratios in atmospheric samples

Early studies by Gratz et al. (2010) and Chen et al. (2012) revealed that MIF anomalies of even mass number isotope $^{200}Hg$ are regularly present in atmospheric precipitation. Later, measurements (Demers et al., 2013) were also made at $\Delta^{204}Hg$, which is more challenging due to the limitations of ion beam collector designs (Blum and Johnson, 2017). The anomaly of $\Delta^{204}Hg$ was generally larger and opposite to that of $\Delta^{200}Hg$. The $\Delta^{200}Hg/\Delta^{204}Hg$ ratio has been calculated based on spatial averages and exclusively on precipitation samples, which are usually above measurement uncertainty. For example, a slope of –0.5 was previously reported (Blum and Johnson, 2017) and later adjusted to –0.4 (Kwon et al., 2020) using this method as more data became available. However, when all individual precipitation data up to 2020 were combined, Kwon et al. (2020) obtained a significantly lower regression slope of – 0.24. **Fig. 14** shows the even-MIF data ($\Delta^{200}Hg$ vs. $\Delta^{204}Hg$) binned into geographical regions (categorized as $Hg^0$, rain/mist/cloud, PBM, RM, and snowfall samples). Linear regression of York-type $\Delta^{200}Hg$ against $\Delta^{204}Hg$ yields slopes between –0.07 and –0.53 for data grouped by site and category for data of statistical significance ($p \leq 0.05$, indicated by *). When the global data grouped by sample type are analyzed separately, significant ($p < 0.001$***) slopes of –0.51±0.02 (n = 45, Kurz et al., 2021; Yuan et al., 2022), – 0.41±0.03 (n = 108, Fu et al., 2021; Demers et al., 2015; Enrico et al., 2016; Sherman et al., 2012; Yuan et al., 2022; Demers et al., 2013; Donovan et al., 2013; Motta et al., 2019; Washburn et al., 2021), –0.29±0.06 (n = 58, Fu et al., 2019b) and –0.11±0.02 (n = 295, Fu et al., 2021; Kurz et al., 2020; Demers et al., 2015; Tate et al., 2023; Araujo et al., 2022; Enrico et al., 2016; Kurz et al., 2021; Demers et al., 2013; Yamakawa et al., 2017; Jiskra et al., 2019; Fu et al., 2016a; Wu et al., 2023a) are obtained for snowfall, rain and fog, particulate matter and $Hg^0$ respectively. The reaction mechanism triggering even-MIF could be photodissociation in the gas phase (Sun et al., 2022) or on surfaces (Fu et al., 2021). This should lead to varying degrees of fractionation depending on the species undergoing decomposition. As a result, the fractionation of atmospheric $Hg^I$ and $Hg^{II}$ species differs from one another, possibly explaining the divergent $\Delta^{200}Hg/\Delta^{204}Hg$ values for $Hg^{II}(aq)$, $Hg^{II}(p)$, and $Hg^0(g)$.

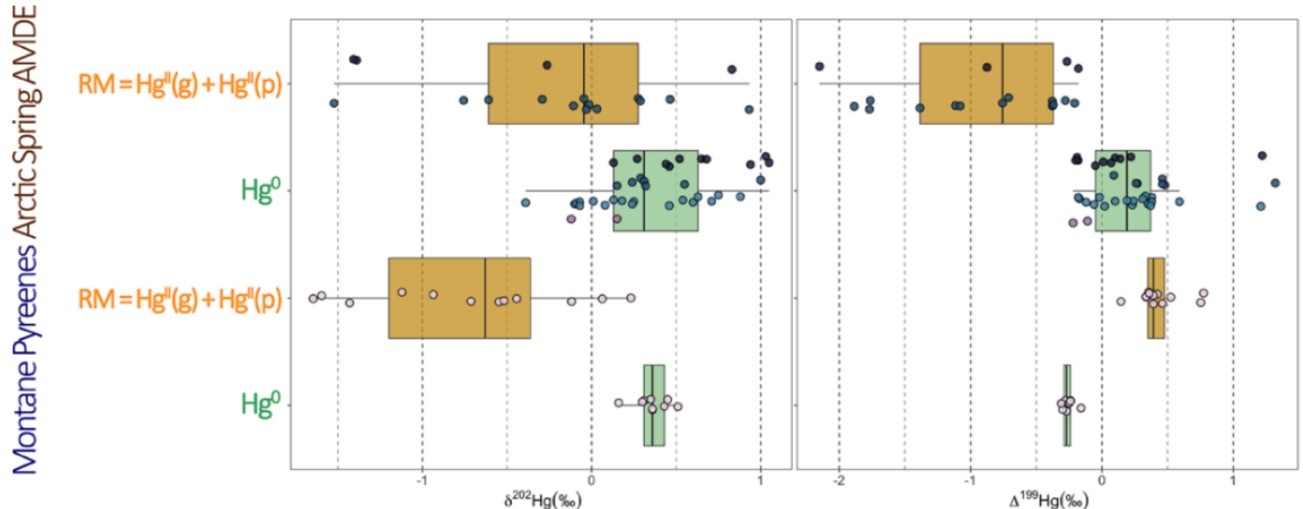

**Figure 12.** A comparison of the TMDF and odd-MIF signatures for atmospheric $Hg^0$ and RM, measured during Arctic AMDEs and in the Pyrenees during winter, reveals notable contrasts.

# Gaseous Hg (~Hg⁰)

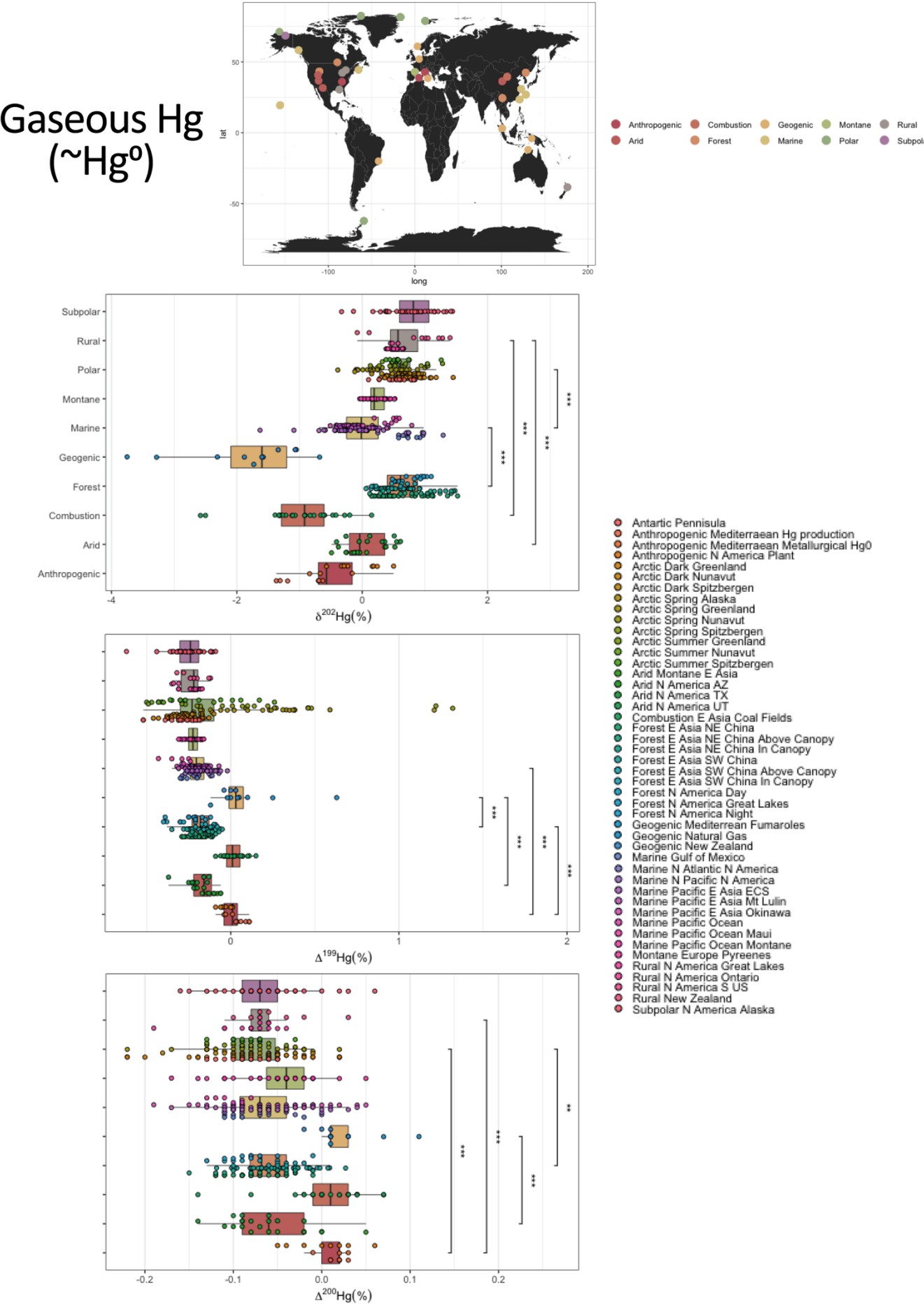

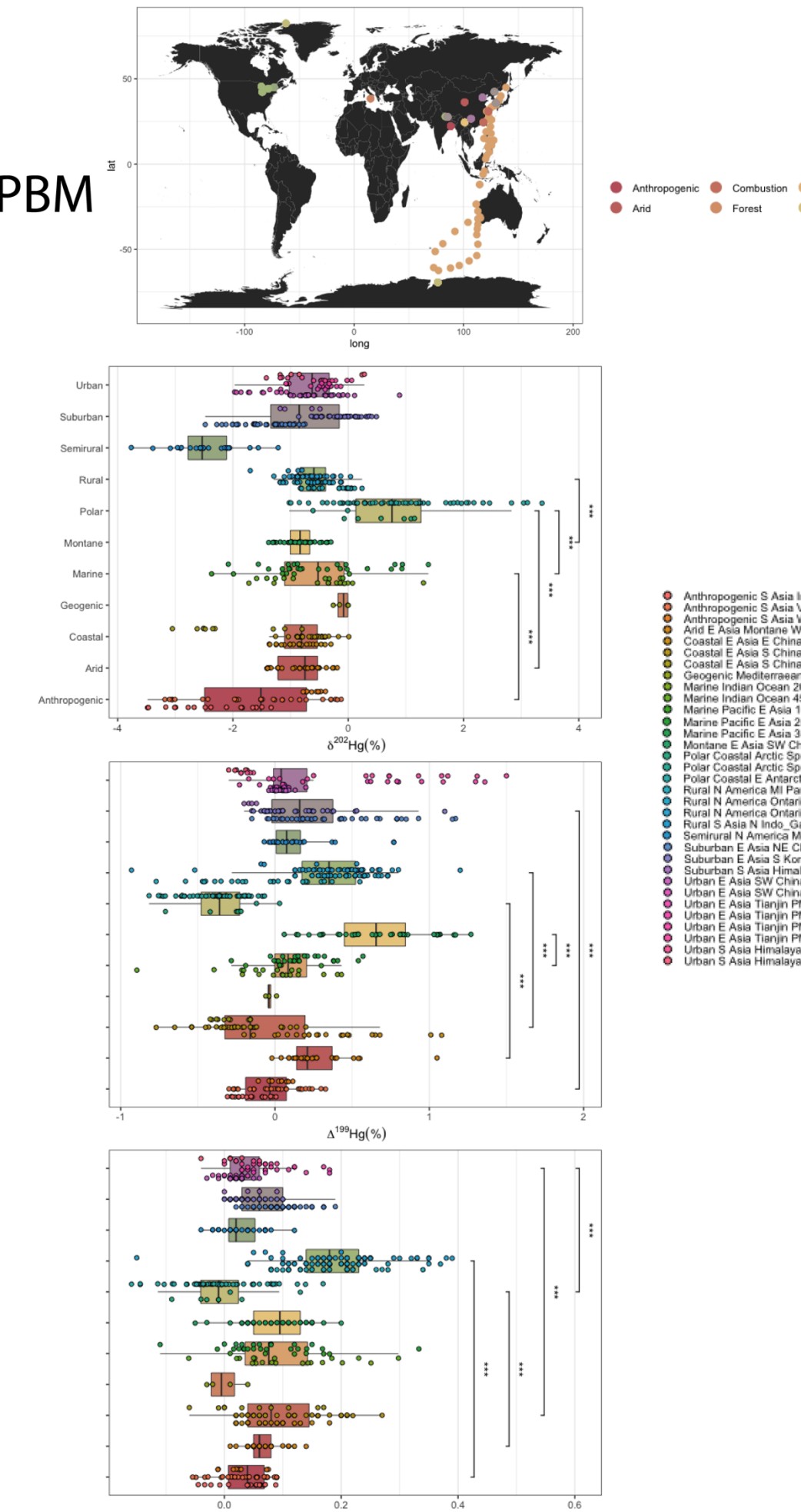

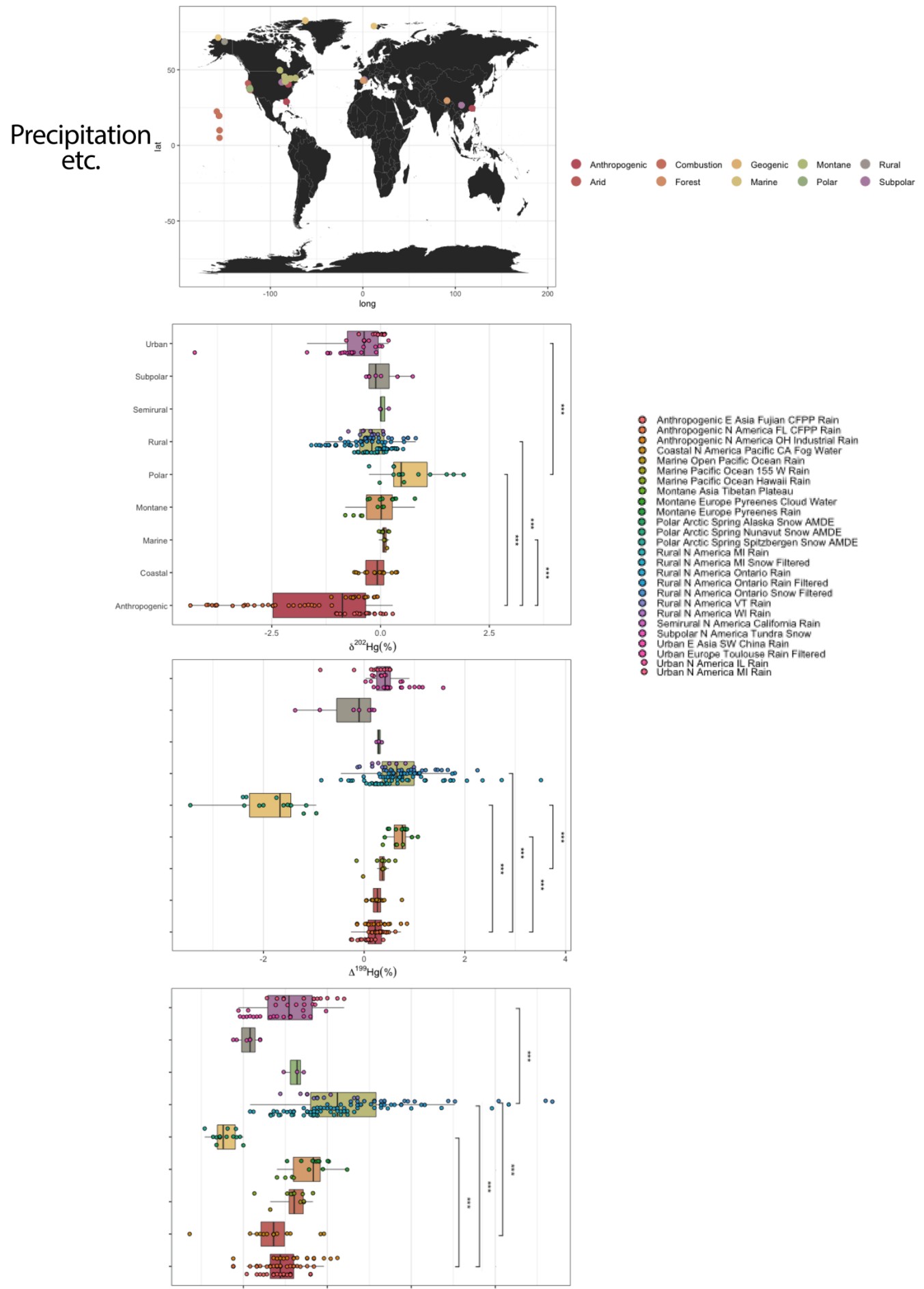

**Figure 13.** Global isotopic observations of gaseous Hg (~Hg⁰) (a), particulate Hg (PBM) (b) and Hg in precipitation (c) divided into $\delta^{202}$Hg (top), $\Delta^{199}$Hg (middle) and $\Delta^{200}$Hg (bottom).

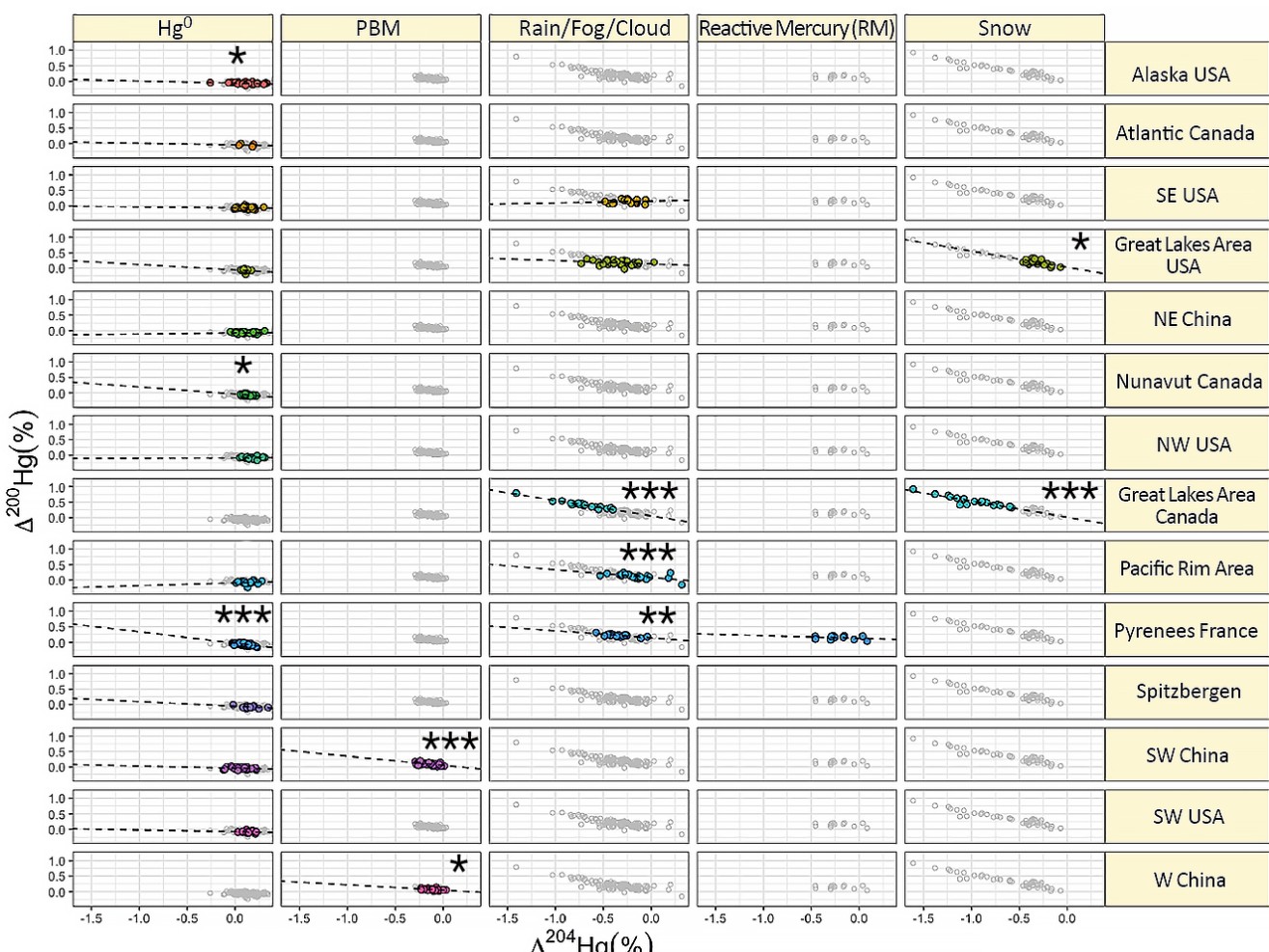

**Figure 14.** Global observations of even-MIF ($\Delta^{200}Hg$ vs. $\Delta^{204}Hg$) by, from left, $Hg^0$, PBM, rain/fog/cloud, reactive mercury (RM), and snow.

### 8.3 Isotope fractionation during gas-phase oxidation

Data on the stable isotopic fractionation of Hg during gas-phase chemical reactions are limited. However, in addition to the published studies on fractionation during the oxidation of $Hg^0$ initiated by Cl and Br atoms (Sun et al., 2016) and during the oxidation of electronically excited $Hg^0$ in the presence of synthetic air (Sun et al., 2022), the corresponding thesis provides additional data (Sun, 2018), which are highlighted here.

### 8.3.1 Ground-state $Hg^0$ oxidation in air

Isotope fractionation during the oxidation of $Hg^0$ vapor in the ground state has been studied for reactions initiated by $Cl^{\bullet}/Br^{\bullet}/^{\bullet}OH/O_3/BrO^{\bullet}$ in air at 750 Torr and 298 K, as listed in **Table 7**. **Fig. 15** shows that the $Br^{\bullet}$ and $^{\bullet}OH$ reactions produce a lighter isotope enrichment in the reactant $Hg^0$, unlike the other reactions that follow KIE. This deviation from KIE occurs because the $Hg^0$ to $Hg^I$ step (**Rxn G1--G3**, **Table 3**) in the overall $Hg^0$ to $Hg^{II}$ oxidation is reversible. EIE is especially notable for the $Br^{\bullet}$ and $^{\bullet}OH$ channels being affected by thermal and photolytic dissociation (**Rxn G14 & G53**), creating a cyclic replenishment of $Hg^0$ at higher temperatures, as discussed in **Section 5.1.2**. EIE predicts the enrichment of heavier isotopes in species with a stronger bonding environment (e.g., $HgBr_2$, $Hg(OH)_2$, Schauble, 2007). However, at temperatures in the upper atmosphere and during AMDEs in polar regions, the rate of **Rxn G14a & G53a** becomes much lower, and the oxidation mechanism moves toward irreversibility, potentially leading to the dominance of KIE at lower temperatures. The chlorine atom-initiated reaction already displays a KIE at 298 K, which is related to the relative thermal stability of the HgCl intermediate. All the atmospherically relevant reactions investigated ($Cl^{\bullet}$, $Br^{\bullet}$ and $^{\bullet}OH$) give rise to (+) odd-MIF, which is most pronounced for the Cl-initiated reaction ($E^{199}Hg = -0.37‰$) compared with the other reactions ($E^{199}Hg = -0.23‰$ and $-0.18‰$ for the Br and OH reactions, respectively). Analogous to $^{\bullet}OH + ^{\bullet}OH$ recombination, which yields $H_2O_2$ in the gas phase (Velivetskaya et al., 2016; Velivetskaya et al., 2018), odd-MIF plausibly occurs due to MgIE triggered by radical–radical ($^{\bullet}Hg^IX + Y^{\bullet}$) interactions that occur during reactions, leading to the

formation of XHg$^{II}$Y species. The diagnostic ratio of $\Delta^{199}Hg/\Delta^{201}Hg$ ~1.9, which is observed for the Hg$^0$ + Cl$^\bullet$ system, differs significantly from the ratios reported for the photoreduction of Hg$^{2+}$ complexes in water (**Section 8.4.1**).

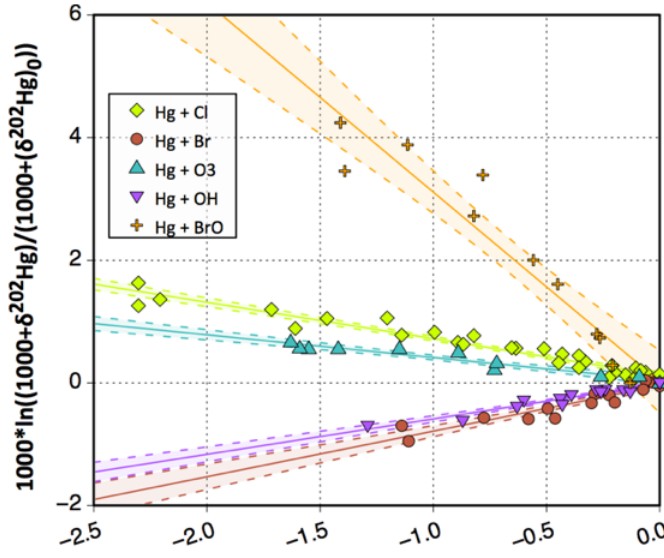

**Figure 15.** Linearized Rayleigh diagram for $\delta^{202}$Hg in Hg$^0$ during Cl, Br, OH, O$_3$ and BrO oxidation experiments at ~298 K showing normal and inverse KIEs. Each point represents a single experiment.

**Table 7.** Experimental fractionation factors determined in gas-phase oxidation studies.

| Oxidant | Precursor | Bath gas | $\varepsilon^{202}$Hg (‰) | E$^{199}$Hg (‰) | E$^{200}$Hg (‰) |
|---|---|---|---|---|---|
| Cl$^\bullet$ | CCl$_3$C(O)Cl + hv | air | –0.590 | –0.370 | 0.06 |
| Br$^\bullet$ | CHBr$_3$ + hv | air | 0.740 | –0.230 | |
| O$_3$ | n/a | air | –0.370 | –0.120 | |
| $^\bullet$OH | H$_2$O$_2$ + hv | air | 0.580 | –0.180 | |
| "BrO$^\bullet$" | CHBr$_3$ + O$_3$ + hv | air | –3.105 | 1.009 | |

### 8.3.2 Hg$^0$ oxidation initiated by photosensitized reactions

Ancient rock samples show a significant occurrence of even-MIF in the Archean atmosphere (~2.5 Ga, Zerkle et al., 2020), which lacked an O$_3$ layer to filter out deep UV light in the actinic zone. However, in the modern atmosphere, even MIF does not appear to occur significantly in Hg redox processes at the Earth's surface. The current atmospheric budget reveals notable imbalances between $\Delta^{200}$Hg in Hg emissions from and deposition to the Earth's surface (0.025 ± 0.032‰ vs. 0.073 ± 0.019‰, Fu et al., 2021). To maintain a steady state, even-MIF sources in the atmosphere are necessary. Studies have shown that UVC-induced Hg$^0$ vapor in the electronically excited state, Hg($^3$P$_1$), undergoes chemical transformation under both artificial (Mead et al., 2013) and modern (Sun et al., 2022) atmospheres, resulting in a large MIF of both odd and even Hg isotopes. There are claims (Blum and Johnson, 2017; Mead et al., 2013) that the $\Delta^{200}$Hg/$\Delta^{204}$Hg ratios found in nature are similar to those present in the glass housing of compact fluorescent lamps (CFLs). However, the $\Delta^{199}$Hg$^{II}$, $\Delta^{200}$Hg$^{II}$, and $\Delta^{204}$Hg$^{II}$ values in the CFL housing exhibit opposite signs to those observed in nature (cf. **Figs. 13 & 14**). Laboratory experiments have shown that the net oxidation of Hg$^0$ by the reaction between excited-state Hg$^0$ and atmospheric O$_2$, which is identical to the driving photosensitized reaction for the turnover of Hg$^0$ in the upper stratosphere (**Rxn G12b** counteracted by **Rxn G72**, **Table 3**), scrambles the systematics of all Hg isotopes in an entirely mass-independent manner. These laboratory experiments and atmospheric samples show similar observations for the $\Delta^{200}$Hg/$\Delta^{204}$Hg ratio, suggesting that photodissociation is a potential chemical mechanism for triggering even-MIF in the atmosphere (Sun et al., 2022). This review outlines new findings on atmospheric Hg chemistry, supporting the fundamental importance of photodissociation processes (**Sections 5.1.2 and 5.1.4**). In addition to the gas phase, surface-mediated photolysis of mercurous halide species has also been proposed as a mechanism for generating even-MIF (Fu et al., 2021). However, theoretical challenges still need to be solved at the quantum mechanics level to generically expand our understanding of anomalous isotope effects for traditional and non-traditional elements (Lin and Thiemens, 2024). Further field and laboratory research in this area should be encouraged.

### 8.4 Isotope fractionation during aqueous-phase red-ox transformation

Hg transformation in the aqueous phase has been reviewed extensively, including stable Hg isotope studies (Hintelmann and Zheng, 2011). The present study does not focus on biotic processes, such as microbial reduction, methylation and demethylation, or phototrophic microbial reduction. Kritee et al. (2013) and Tsui et al. (2020) provide overviews of this field. The focus is on abiotic processes, excluding those involving coordination with macromolecular heterogeneous ligands such as DOM or fractions, and instead on low-molecular-weight ligands, including those with N–, O–, S–, or (pseudo)halide donors. This includes inorganic and organic ligands and oxidizing and reducing processes (**Section 6**).

#### 8.4.1 Reduction

To recapitulate **Section 8.1**, in addition to MDF, isotopic effects in NFS and MgIE occur for Hg during chemical transformation in the aqueous phase. MDF and NFS are present in all reactions to varying magnitudes and in all mechanisms and have a thermodynamic nature. In contrast, MgIE is a kinetic effect and is indicative of spin-selective reactions involving a paramagnetic intermediate. Therefore, MgIE is the only isotope effect that detects the reaction mechanism. MgIE can be both thermally and photolytically induced and can be two-dimensional (+ or – depending on the reaction conditions, Zheng and Hintelmann, 2010b) or one-dimensional (exclusively +), depending on the identity of the $Hg^{II}$ complex (Motta et al., 2020a). In cases where the spin-selective reaction can be induced thermally, the radical pair is generated almost exclusively as a singlet (Buchachenko, 2018), which is spin-forbidden to react (dissociate) further into products. For a singlet spin forbidden reaction compared to a triplet spin allowed reaction, the magnitude of the MgIE-MIF is more limited. However, many $Hg^{II}$ complexes have a narrow energy separation of a variety of excited states, indicating that the intermediate radical pair can evolve into a triplet or singlet state. Studies of $Hg^{2+}$ photoreduction in the presence of organic ligands (which consistently follow a pseudo-first-order kinetic pathway) have shown that, depending on the degree of $Hg^{2+}$(aq) turnover, weak MIF is initially induced by NFS, and then, when most of the $Hg^{2+}$(aq) has been converted, there is a shift to strong MIF induced by MgIE, the onset of which coincides with strong suppression of MDF (Motta et al., 2020b; Zheng and Hintelmann, 2010b). One explanation for why MgIE first appears closer to complete $Hg^{II}$ reduction is, at least in part, that the termination radical–radical step when $Hg^0$ is split off (in a bimolecular reaction, such as $Hg^{+\bullet} + C_2O_4^{\bullet-}/CO_2^{\bullet-}$ in the photoreduction of $Hg(\eta^2\text{-}C_2O_4)$) is favored by a decreasing concentration ratio of oxidized Hg to bulk ligand (Zhao et al., 2021). As shown in **Fig. 11**, (–)MgIE is induced when the radical pair is generated in a singlet state, and (+)MgIE is induced when the excitation occurs in a triplet state. Ligand field strength, in combination with atomic orbital hybridization theory, has been used to illustrate MgIE in the (photo)reduction of $Hg^{II}$ complexes. This phenomenon has been suggested to vary as a function of, among other things, the arrangement of the ligands around $Hg^{2+}$, the coordination strength of the ligands, and the presence/absence of light along with its wavelength (Epov, 2011a; Epov, 2011b). As discussed in **Section 4.4**, reduced S- and reduced N-containing groups are soft (strong field) ligands, whereas O-donating groups are hard (weak field) ligands. Epov (2011a) rationalized mercuric complexes with strong field ligands such as cysteine ($Hg(cys)_2^{2-}$) and ethylenediamine ($Hg(en)^{2+}$) as bright singlets (i.e., in the presence of light) with sp-hybridization at the central Hg atom in two binding orbitals. To undergo singlet-triplet evolution by hyperfine coupling between magnetic nuclei ($^{199}Hg$ and $^{201}Hg$) and electrons to a paramagnetic state, the orbital hybridization of Hg must change from sp-linear to $sp^2d$-planar square so that the transfer of electrons from the soft ligand to Hg can be accomplished.

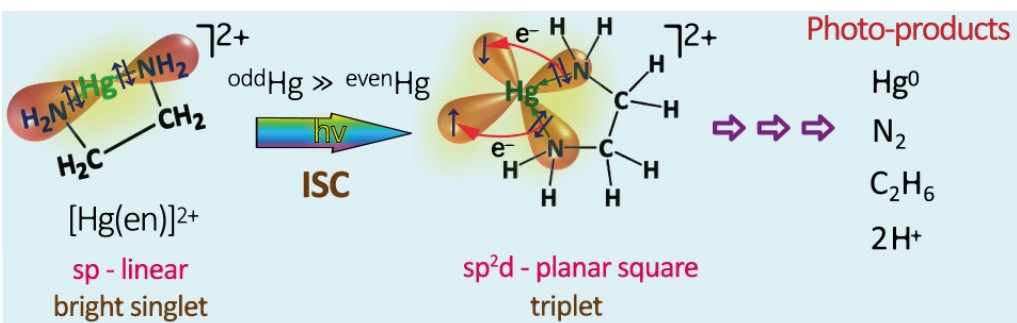

**Figure 16.** Mechanism proposed by Epov (2011b) for the photoreduction of $Hg(en)^{2+}$ via a bright singlet excited complex undergoing intersystem crossing preferentially for the odd isotope to the closest triplet state, which can dissociate following a complex reaction.

**Fig. 16** shows the schematic for Hg(en)$^{2+}$ + light, with (–)MgIE. It is postulated that mercuric complexes with O-binding ligands possess a bright triplet state that is more likely (spin allowed) to undergo Hg reduction via 1e-LMCT with an imprint of (+)MgIE.

For a change in the spin state to occur, spin–orbit coupling (SOC) must be induced, but if the SOC is elevated, spin relaxation or phosphorescence can be induced, which prevents the formation of a separating radical pair during dissociation, making MgIE less relevant (cf. **Fig. 11**). Coupling constants are known experimentally for only a few Hg-containing radicals (CH$_3$Hg$^•$, Karakyriakos and Mckinley, 2004; $^•$HgF, Knight Jr. et al., 1981; $^•$HgCN, Knight Jr. and Lin, 1972; $^•$HgH/$^•$HgD, Stowe and Knight Jr., 2002). Recently, published theoretical electronic structure simulations have been performed on environmentally interesting Hg halides (Cl, Br, I) and pseudohalides (methanethiol). The study (Motta et al., 2020a) reported that the coupling for reactions involving $^•$Hg$^I$Br and $^•$HgI is so high that radical pair formation is inhibited, whereas for $^•$Hg$^I$Cl and CH$_3$SHg$^{I•}$, coupling is sufficient in the caged pair as well as at a low level in the separated pair geometries, allowing MgIE to form. Depending on the identity of the Hg–ligand bond that undergoes homolysis to a radical pair, either quadruple (X = Cl, S) or double (Y = C) degeneracy can occur between the low-lying electronically excited levels and the ground state in the HgX$_2$ and HgXY compounds, respectively, allowing the photoreduction of HgX$_2$ to exhibit (+)MgIE or (–)MgIE while that of HgXY (i.e., MMHgX) exhibits only (+)MgIE. This is based on the premise that the photolysis of MMHgX is exclusively by cleavage of the weaker Hg–C bond rather than the stronger Hg–X bond. (+)MgIE is most evident for the photoreduction of MMHg$^+$ species, as its $^3\sigma\sigma^*$ 1e-LMCT state is energetically separated from other excited states in the paramagnetic intermediate, leading to the maximization of MgIE (Motta et al., 2020a). Stable Hg isotopes provide insight into the dynamics and metabolism of inorganic and methylated Hg in biota. Exposure to the former results in subtle odd-MIF with a $\Delta^{199}$Hg/$\Delta^{201}$Hg ratio close to unity at sampling, while for the latter, this ratio is greater (~1.3) with a large odd-MIF (up to ~5‰ in fish, Li et al., 2022b).

From **Table 8**, which summarizes the isotopic effects quantitatively observed in aqueous-phase laboratory studies, MgIE can have different signs for the same reactant depending on the reaction conditions, as exemplified by the Hg-cysteine-light system. Depending on the degree of photoconversion, the reduction of Hg$^{II}$ in the presence of water-soluble diesel soot (aromatic polyacids and humic-like structures) exhibits swings in the direction of MgIE (Huang et al., 2021). Another example of the impact of pH/complexation on the evolution of MgIE can be seen in the UVC photodegradation of MMHg$^+$ in acidic and alkaline (adjusted with NH$_3$) solutions. In the former, (+)MgIE is significant, but it is limited in the latter. For traditional elements with the same reaction mechanism, the strength development of MgIE depends on various factors, including viscosity, triplet sensitizer, and excited-state quenchers (Turro, 1983; Buchachenko, 2013). As seen in the laboratory experiments, both (+) and (–) net Hg MgIE were observed in samples related to the natural atmosphere, as previously reported in **Section 8.2**. The reaction conditions also affect the degree of turnover of the Hg reactant at which the onset of MgIE occurs, which incidentally does not correlate with a change in the overall reduction rate. To better interpret odd-MIF signatures and systematically elaborate the roles of reaction parameters (pH, presence of O$_2$, light wavelength, etc., Rose et al., 2015) in excited-state kinetic isotope effects, experimental research is needed. For example, dissolved O$_2$ is a well-known quencher of excited triplet states, but radical-O$_2$ reactions have also been described to induce significant MgIE (Pliss et al., 2019). For the photoreduction of Hg in the presence of multifunctional ligands (such as DOM), the stoichiometry (Hg:L ratio) has been shown to play an important role in the magnitude of MgIE induced. Zhang and Hintelmann (2009) observed an E$^{199}$Hg optimum ($\gtrsim$5%) in anoxic photo-experiments with the DOM fraction from Dorset Lake, Ontario. This optimum is associated with a ligation mode in which all S-bonding functional groups are saturated by Hg$^{2+}$ cations, increasing the proportion of Hg-O bonds and the ratio of bright triplets to bright singlets, thus making the MgIE increasingly positive. As the Hg:L ratio is further increased, the reduction rate (driven by Hg–O complexes) is significantly affected. The triplet-singlet spin evolution is limited to fewer HgL radical pairs, resulting in a lower –E$^{199}$Hg (Epov, 2011a). In contrast to freshwater DOM, photo experiments with Hg$^{II}$ in the presence of DOM extracted from marine phytoplankton yield (–)MgIE during reduction (Kritee et al., 2018).

**Table 8**. Experimental fractionation factors determined for a variety of Hg red-ox transformations.

| Initial Hg conc. (electrolyte) | | Reactant | Experimental conditions: L/Hg ratio | | Anoxic/ Oxic | Major Hg species | $\varepsilon^{202}$Hg (‰,±2σ) | $E^{199}$Hg (‰) | $\Delta^{199}$Hg/ $\Delta^{201}$Hg | Isotope effects | Reference |
|---|---|---|---|---|---|---|---|---|---|---|---|
| **Photoreduction of Hg$^{II}$** | | | | | | | | | | | |
| 0.5 μM NIST-3133 (Cl⁻) | | Cysteine (Hcys) | Quartz glass, ~2000:1 (M/M), Xe lamp (UVC-filter), pH 3.6 | | A | Hg(cys)₂ | −1.32 ± 0.07 | 1.02 | 1.46 ± 0.03 | MDF, NFS, (−)MgIE | Zheng & Hintelmann, 2010b |
| 0.17±0.04 μM | | | FEP Teflon, ≳2000:1 (M/M), natural sunlight, pH 3.2 | | | | | − | 1.34 ± 0.03 | | Motta et al., 2020b |
| | | | FEP Teflon, ≳2000:1 (M/M), natural sunlight, pH 7.2 | | | | −1.04 ± 0.09 | −0.25 | 0.99 ± 0.06 | MDF, (+)MgIE | |
| | | | FEP Teflon, ≳2000:1 (M/M), natural sunlight, pH 7.2 | | O | | | −1.15 | 1.11 ± 0.09 | | |
| 0.5 μM NIST-3133 (Cl⁻) | | Serine (Hser) | Quartz glass, 2000:1 (M/M), Xe lamp (UVC-filter), pH 3.8 | | A | Hg(ser)₂ | −1.71 ± 0.03 | 0.17 | 1.67 ± 0.28 | MDF, NFS, (+)MgIE[36] | Zheng & Hintelmann, 2010b |
| 0.17± 0.04 μM | | | FEP Teflon, ≳2000:1 (M/M), natural sunlight | | A/O | | −1.81 ± 0.04 | 0.06 ~−1.3 | ~1.6 1.08 ± 0.01 | MDF, NFS (+)MgIE[37] | Motta et al., 2020b |
| 0.17± 0.04 μM | | Ethylenediamine (en) | FEP Teflon, ≳20000:1 (M/M), natural sunlight, pH 7.4 | | O | Hg(en)₂²⁺ HgOH(en)⁺ | −0.9 ± 0.3 | 0.16 | 0.85 ± ±0.14 | MDF, (−)MgIE | Motta et al., 2020b |
| 1 μM (ClO₄⁻) | | Oxalate (ox²⁻) | Pyrex, 300:1 (M/M), UV-B light, pH 3.9 & 5.2 | | A/O | Hg-η-ox | −1.45 ± 0.06 | 0.15 | 1.39 ± 0.38 | MDF, NFS[38] | Zhao et al., 2021 |
| | | AQDS | Pyrex, 300:1 (M/M), UV-B light, pH 3.4 | | A | ? | −0.66 ± 0.10 | −0.86 | 1.00 ± 0.02 | MDF, (+)MgIE | |
| | | Salicylic acid (Hsal) | Pyrex, 300:1 (M/M), UV-B light, pH 4.3 | | A | Hg(sal)⁺? | −1.79 ± 0.30 | | | MDF, NFS | This work |
| | | 4-hydroxy-benzoic acid (HOBz) | Pyrex, 300:1 (M/M), UV-B light, pH 4.9 | | A | Hg(OBz)⁺? | −2.25 ± 0.10 | ~0.10 | 1.53 ± 0.02 | | |
| | | 4-aminobenzoic acid (HNBz) | Pyrex, 300:1 (M/M), UV-B light, pH 5.9 | | A | Hg(NBz)⁺? | −2.75 ± 0.40 | | | | |
| 0.3 − 0.5 μM | | Suwannee River fulvic acid | Quartz glass, ~10 − 17 (m/m), sunlight | | O | | −0.60 | −0.45 | 1.00 ± 0.02 | MDF, (+)MgIE | Bergquist & Blum, 2007 |
| ~10 μM | NIST-3133 (Cl⁻) | Dorset Lake bulk DOM pH 6.5 | Quartz glass, ~29000 (m/m), Xe lamp (UVC-filter) | | A | The proportion of Hg-O bonding increases as we move downward, and so does the reaction rate. | −0.77 ± 0.18 | −2.94 | 1.19 ± 0.02 | MDF, (+)MgIE | Zheng & Hintelmann, 2009 |
| | | | Quartz glass, ~6000 (m/m), Xe lamp (UVC-filter) | | | | −0.72 ± 0.10 | −4.12 | 1.22 ± 0.02 | | |
| ~50 μM | | | Quartz glass, ~1200 (m/m), Xe lamp (UVC-filter) | | | | −1.26 ± 0.07 | −6.29 | 1.24 ± 0.02 | | |
| | | | Quartz glass, ~1200 (m/m), sunlight | | | | −0.99 ± 0.02 | −5.57 | 1.26 ± 0.01 | | |
| ~500 μM | | | Quartz glass, ~120 (m/m), Xe lamp (UVC-filter) | | | | −1.06 ± 0.02 | −1.94 | 1.30 ± 0.02 | | |
| ~50 μM | | | | | | | −1.09 ± 0.04 | −1.99 | 1.31 ± 0.01 | | |
| 29 nM (NO₃⁻) | | Marine algal DOM (intracellular) | Teflon, 1.41 nmol chla⁻¹, UVB-light | | O | | −0.70 | 1.03 | 1.06 | MDF, (−)MgIE | Kritee et al., 2018 |
| 82 nM (Cl⁻) | | Water-soluble diesel soot extracts | Quartz glass, ~67:1 (M/M), Xe lamp, Instanteous removal of product (Hg⁰) | | A | | −1.30± ± 0.11 | −2.49 | 1.15 | MDF, (+)MgIE | Huang et al., 2021 |
| 10 nM (NO₃⁻) | | Dissolved black carbon (< 0.45 μm) | Glass, ~42000:1 (M/M), Xe lamp | | A | | | | 1.20 ± 0.10 | MDF, (−)MgIE | Li et al., 2020b |
| **Photoreduction of MMHg⁺** | | | | | | | | | | | |
| 50 μM | | CH₃HgCl | Hg-lamp (λ = 254 nm), pH 4.0 | | | CH₃HgCl | ~−0.25 | ~−0.5 | 1.26 ±0.06 | MDF, (+)MgIE | Malinovsky et al., 2010 |
| | | CH₃HgOH | Hg-lamp (λ = 254 nm), pH 8.6 | | | CH₃HgOH | ~−0.3 | < −0.06 | | MDF, supressed (+)MgIE | |
| 0.3 − 0.5 μM? | | | Suwannee River fulvic acid, sunlight | 10 mg C/L | | | −1.70± 0.30 | −7.9 | 1.36 ±0.02 | | Bergquist & Blum, 2007 |
| | | | | 1 mg C/L | | | −1.30 ± 0.20 | −3.3 | | | |
| 102 nM | | CH₃Hg⁺ | Suwannee River fulvic acid, Xe lamp (UVC-filter) | 2.13[39] | O | | −1.74 ±0.50 | −0.9 | 1.30 ±0.07 | MDF, (+)MgIE | Chandan et al., 2015 |
| 86 nM | | | | 0.72 | | | −4.64 ±1.64 | −5.0 | 1.28 ±0.09 | | |
| 101 nM | (Cl⁻) | | | 0.42 | | | −1.91 ±0.25 | −7.2 | 1.32 ±0.03 | | |
| 80 nM | | | | 0.17 | | | −1.77 ±0.81 | −7.2 | 1.30 ±0.03 | | |
| 97 nM | | | | 0.10 | | | −1.50 ±0.50 | −13.4 | 1.40 ±0.03 | | |
| 94 nM | | | | 0.07 | | | −2.24 ±0.44 | −16.2 | 1.37 ±0.03 | | |

---

[36] Appears at 4 h photoreduction and beyond with a $\Delta^{199}$Hg/$\Delta^{201}$Hg of 1.10–1.18

[37] Onset of (+)MgIE at $f_R$ = 0.40–0.76 depending on reaction conditions.

[38] A single experiment (anoxic, pH 6) on oxalate indicates (+)MgIE at $f_R$ = 0.11.

[39] MMHg/organic bound reduced sulfur (M/M)

| [Hg] | Reductant | Conditions | $f_R$ | O/A | Species | $\delta^{202}Hg$ | | | | Reference |
|---|---|---|---|---|---|---|---|---|---|---|
| 96 nM | | Pony Lake fulvic acid, Xe lamp (UVC-filter) | 0.01 | | | $-1.13 \pm 0.36$ | $-15.3$ | $1.36 \pm 0.01$ | | |
| 104 nM | | Nordic Lake DOM, Xe lamp (UVC-filter) | 0.41 | | | $-1.33 \pm 0.17$ | $-1.3$ | $1.17 \pm 0.04$ | | |
| 95 nM | | | 0.05 | | | $-2.23 \pm 0.68$ | $-14.6$ | $1.41 \pm 0.02$ | | |
| **Dark reduction of Hg$^{II}$** | | | | | | | | | | |
| 1 μM (ClO$_4^-$) | Benzoquinone C$_6$H$_4$(OH)$_2$, QH$_2$ | Pyrex, 300:1, dark, pH 4.6 | | O | Hg–QH$^+$? | $-1.25 \pm 0.19$ | 0.12 | $1.39 \pm 0.38$ | | Zhao et al., 2021 |
| 0.5 μM NIST-3133 (Cl$^-$) | SnCl$_2$ | Quartz, dark, low pH | | A | HgCl$_4^{2-}$ | $-1.56 \pm 0.11$ | 0.17 | $1.59 \pm 0.22$ | | Zheng & Hintelmann, 2010a |
| 1 μM (ClO$_4^-$) | Ascorbic acid | Pyrex, 300:1, dark, pH 5.1 | | O/A | ? | $-1.79 \pm 0.13$ | 0.08 | $1.48 \pm 0.35$ | | This work |
| NIST-3133 (Cl$^-$) | Dorset Lake bulk DOM | Quartz, dark, pH 6.5 | | A | | $-1.52 \pm 0.06$ | 0.19 | $1.54 \pm 0.34$ | MDF, NFS | Zheng & Hintelmann, 2009 |
| 1 μM NIST-3133 (NO$_3^-$) | FeCl$_2$ | Glass, 12.5:1, dark, pH 6.5, 0.5 mM Cl$^-$ | | A | Hg(OH)$_2$ | $-2.20 \pm 0.16$<br>$-2.44 \pm 0.17$[40] | 0.21<br>0.34 | $1.58 \pm 0.08$<br>$1.60 \pm 0.05$ | | Schwab et al., 2023 |
| | | Glass, 12.5:1, dark, pH 6.5, 10 mM Cl$^-$ | | | HgCl$_2$, Hg(OH)Cl | $-2.14 \pm 0.09$ | $0.24 \pm 0.01$ | | | |
| **Methylation of Hg$^{II}$** | | | | | | | | | | |
| 0.5 − 5 μM NIST-3133 (Cl$^-$) | Methylcobalamin | 1000:1, pH 5, (UV–A lamp) (λ~325 nm) | | O | | ~$-1.5$[41] | 0.21[42] | $1.20 \pm 0.17$ | MDF, (−)MgIE | Malinovsky & Vanhaecke, 2011 |
| | Acetate | | | | | | 0.22 | $1.20 \pm 0.26$ | | |
| **Oxidation of Hg$^0$** | | | | | | | | | | |
| 200 − 280 nM | Hydroxyl radicals | Pyrex, ≤ 600:1 (NaNO$_3$), pH 7, UV-lamp (λ > 300 nM) | | A | Hg(OH)$_2$ | $1.20 \pm 0.14$ | | | | Stathopoulos, 2014 |
| ~60 nM | Sulphanyl-acetic acid | Glass, 80:1 (M/M), dark, pH 7 | | A | | $1.25 \pm 0.11$ | $-0.14$ | $1.28 \pm 0.38$ | EIE-MDF, NFS | Zheng et al., 2019 |
| | 2-sulphanyl-propanoic acid | | | A | | $1.10 \pm 0.08$ | | | | |
| 115 nM | Reduced natural Elliott soil humic acid | Glass, ~0.6 − 1.2 S:Hg (M/M) | | A | | $1.54 \pm 0.10$ | $-0.18$ | | | |

As shown in **Table 8**, photoreduction of Hg$^{2+}$ often, but by no means always, is associated with high odd-MIF. For macromolecular entities such as DOM and fulvic acids and a selection of smaller organic ligands that use O−, N− and S-donor atoms to complex with Hg$^{2+}$, MgIE is initially induced in the photoreduction process, whereas for the amino acid serine, MgIE is triggered only after a significant turnover of Hg$^{2+}$, the onset of which varies significantly depending on the reaction conditions (Zheng and Hintelmann, 2010b; Motta et al., 2020b). The experimental $\Delta^{199}$Hg/$\delta^{202}$Hg data are described to follow the same trajectory, regardless of when MgIE kicks in during serine-assisted photo-reduction. When oxalic acid was screened with a single light experiment (anoxic, pH 6, $f_R = 0.11$), (+)MgIE was observed, anoxic time series experiments with UV-B irradiation at pH 3.9 and 5.2 revealed no evidence of MgIE in the range investigated down to $f_R = 0.01$. This is evidence that Hg oxalate complexes can be directly photodegraded by homolysis (Hg($\eta^2$–C$_2$O$_4$) $\xrightarrow{h\nu}$ Hg$^{\bullet+}$ + C$_2$O$_4^{\bullet-}$) as well as heterolysis (Hg($\eta^2$–C$_2$O$_4$) $\xrightarrow{h\nu}$ Hg + 2 CO$_2$). Heterolytic photoreduction does not induce MgIE but results in NFS with limited (−)odd-MIF, as is the case for ligation with the substituted aromatic carboxylic acids shown in the table. This also applies to thermal (dark) reduction by a uni− (e.g., Hg–QH$^+$ → Hg$^0$ + Q + H$^+$) or bimolecular (e.g., Hg$^{2+}$ + Sn$^{2+}$ → Hg$^0$ + Sn$^{IV}$) processes. Although NFS is a general isotopic effect, its magnitude depends on the shift in the 6s orbital electron density, which is greater for a red-ox reaction than for ligand exchange or evaporation. In turn, ionic Hg complexes have greater NFS than more covalent complexes upon reduction to Hg$^0$. NFS typically produces a characteristic $\Delta^{199}$Hg/$\Delta^{201}$Hg slope of ~1.54 to 1.66, as determined from experimental studies and theoretical calculations. However, the application of linear regression to NFS odd-MIF data ($\Delta^{199}$Hg vs. $\Delta^{201}$Hg) is limited in several cases because the observations are distributed over such a small range that they approach the scale of the corresponding analytical precision. **Table 8** gives two standard deviations of the slope of the linear fits using York's regression, and the uncertainty is so large that it does not allow a definitive $\Delta^{199}$Hg/$\Delta^{201}$Hg ratio to be determined. In these cases, it has been suggested that a better indicator of NFS is instead to confirm that the patterns of Hg isotope fractionation observed mimic the odd–even staggering pattern of nuclear charge radii (Motta et al., 2020b). The description of NFS is limited to equilibrium fractionation (**Eq. 25**) and predicts, similar to EIE-MDF (**Eq. 23**),

---

[40] Refers to a closed system

[41] Applies to dark conditions, under UVA irradiation demethylation gradually counteracts MMHg$^+$ formation.

[42] Potentially explained by photodegradation of MMHg$^+$

the enrichment of heavier isotopes in the oxidized fraction of the red–ox pair. Calculations performed for a series of $Hg^{II}$ complexes, both binary and heterogeneous, containing simple hard and soft ligands relative to $Hg^0$, show that NFS makes the most significant contribution to $\varepsilon^{202}Hg$ (ranging in total from 46 to 85% at 25 °C; Jiskra et al., 2012). The expected mass-independent enrichment $E^{199}Hg_{NFS}$ can be calculated based on the calculation of $\varepsilon^{202}Hg_{NFS}$, using the scale factors $\beta_{KIE–MDF}$ and $\beta_{NFS}$ (Jiskra et al., 2012):

$$E^{199}Hg_{NFS} = \varepsilon^{202}Hg_{NFS} \cdot (\beta_{NFS} - \beta_{KIE-MDF}) \approx -0.2 \cdot \varepsilon^{202}Hg_{NFS} \qquad (26)$$

Reduction by $Fe^{II}$ and p-substituted benzoic acids results in one of the highest magnitudes of experimentally observed kinetic MDF (**Table 8**). The former system has been studied anoxically both as an open and closed system (Schwab et al., 2023), where the fractionation is of the Rayleighian model (kinetic) and equilibrium type, respectively. The closed system permits overprinting with the signature of isotopic equilibrium fractionation between $Hg^0$ and hydrolyzed $Hg^{2+}$, which has been consistently determined in two independent studies to be –2.63 (Wang et al., 2021) and –2.44‰ (Schwab et al., 2023), respectively. As demonstrated below, the magnitude of the equilibrium isotope enrichment factor ($\varepsilon^{202}Hg$) between $Hg^0$ and thiol-bound $Hg^{II}$ is significantly lower (1.1–1.6‰), which is related to the lower vibrational energy of Hg-S bonds than that of Hg-O/Cl bonds.

### 8.4.2 Oxidation

To the extent that isotopic effects in aqueous-phase $Hg^0$ oxidation have been studied in the laboratory, it has been observed that oxidized Hg becomes isotopically heavier than the reactant. The observed fractionation does not conform to the Rayleigh model, but it is consistent with EIE in a closed system. Consequently, the isotope ratio of the product(s) linearly approaches that of the reactant at the beginning of the reaction. An example of atmospherically relevant oxidation is the rapid reaction with $^\bullet OH$ (**Rxn W2**, generated by photolysis of $NO_3^-$) with $\varepsilon^{202}Hg = 1.20 \pm 0.14$ ‰ (Stathopoulos, 2014). Experiments with thiol-substituted carboxylic acids in the dark produced similar fractionation results (**Table 8**, **Rxn W7**). Additionally, NFS produces a small odd-MIF signal that consistently acts in the opposite direction of mass-dependent fractionation (Zheng et al., 2019). The reason for observing EIE despite the continuous oxidation of $Hg^0$ without any indication of reversibility in the form of back reactions has been attributed to the rapid exchange of Hg isotopes between the remaining $Hg^0$ and the formed $Hg^{II}$ complexes (Wang et al., 2020). There is currently debate surrounding the mechanism by which this exchange occurs (Wang et al., 2020; Zheng et al., 2019; Wang et al., 2021). In the presence of humic acid, the oxidation of dissolved $Hg^0$ exhibits two kinetic regimes where the EIE is not fully established in the initial regime (Zheng et al., 2019). KIE-MDF during dark reduction in the presence of DOM and EIE-MDF during dark oxidation caused by humic acid results in fractionation in the same direction and magnitude, so unmasking the controlling redox process from isotopic measurements can be difficult.

### 8.5 Isotope fractionation during complexation, sorption and surface-catalyzed reduction

### 8.5.1 Processes interfacing the aqueous phase

Theoretical computations of EIE based on the MDF and NFS generally agree with experimentally determined fractionation factors for complexation. Competitive complexation of $Hg^{II}$ between one of the typical hard ligands $HO^-$ and $Cl^-$ and a soft ligand in the form of a thiol resin results in a lighter isotopic signature of the sulfur-bound $Hg^{II}$ pool ($\varepsilon^{202}Hg$ values of –0.62 and –0.53‰, respectively), which is related to increased covalent bonding and electron density in the 6s Hg orbital (Wiederhold et al., 2010). For the sorption of dissolved $Hg^{II}$ on $\alpha$-FeOOH, the observed isotopic fractionation ($\varepsilon^{202}Hg \sim -0.4‰$) is exclusively determined by the process in solution, where a vanishingly small pool (< 0.1%) of isotopically lighter cations is in equilibrium with a bulk of neutral $Hg^{II}$ molecules, with only the former being sorption active (Jiskra et al., 2012). Equilibration and kinetic fractionation have been reported to describe the precipitation process of $\beta$-HgS and HgO, respectively, from an initially acidic solution, with $\varepsilon^{202}Hg$ values between the precipitate and the supernatant being –0.63‰ and –0.32‰, respectively (Smith et al., 2015). Like adsorption on goethite, the observed fractionation during the precipitation of metacinnabar is interpreted as an effect of solution chemistry, in this case, a transition from O– to S–bonding for $Hg^{II}$. In addition to the homogeneous phase reduction of $Hg^{II}$ by $Fe^{II}$ in aqueous solutions (**Table 8**), the heterogeneous phase reduction of $Hg^{II}$ by surface-bound (adsorbed $Fe^{II}$ on goethite/boehmite) or structural $Fe^{II}$ (magnetite $Fe^{II}Fe_2^{III}O_4$, Schwab et al., 2023 and siderite/green rust $FeCO_3$, Wang et al., 2021) has been studied isotopically. As shown in **Table 9**, the isotopic fractionation in heterogeneous reduction is closely related in magnitude to that of homogeneous fractionation by $Fe^{II}$ (**Table 8**), except in the case of magnetite (whose iron structure is present in different oxidation states), which has a much more limited TMDF and MIF ($\varepsilon^{202}Hg = -1.38‰$ and $E^{199}Hg =$

0.13‰, respectively). All these processes, when determined with confidence, demonstrate $\Delta^{199}Hg/\Delta^{201}Hg$ ratios within the range of 1.56 to 1.62, which indicates that the observed MIF ($E^{199}Hg$ in the range of 0.13 to 0.34‰) is caused by NFS.

### 8.5.2 Processes interfacing the gas phase

**Section 8.4.1** and **Table 8** refer to a study of $Hg^{II}$ photoreduction of aqueous diesel soot, which includes experiments with a stationary soot phase mixed with $HgCl_2$ on a quartz plate over which a slow flow of Ar gas passes, as discussed below (Huang et al., 2021). In comparison, photoreduction in aqueous- and solid-phase diesel soot shows equivalent enrichment of heavier isotopes in the Hg reactant of 1.26–1.75‰. This value overlaps with the values typical of Hg redox chemistry (**Table 8**). In contrast to the aqueous phase, the photoreduction in the solid phase shows a continuous strong MIF (this time, positive MgIE induced in the $Hg^0$ product) throughout the reaction, whereas in the latter case, a large MIF of the opposite sign occurs after only ~60% of the reaction. Furthermore, the reduction rate increases with increasing carrier gas humidity. The photo-triggered MgIE is highest when the carrier gas is dehumidified, but decreases rapidly as the RH increases (**Table 9**).

**Table 9.** Experimental fractionation factors determined for $Hg^{II}$ complexation, sorption, surface-catalyzed reduction and processes interfacing the gas phase.

| Initial $Hg^{II}$ conc. (electrolyte) | Reactant | Experimental conditions: L/Hg ratio | | Anoxic (A)/Oxic (O) | $\varepsilon^{202}Hg$ (‰,±2σ) | $E^{199}Hg$ (‰) | $\Delta^{199}Hg/\Delta^{201}Hg$ | Isotope effects | Reference |
|---|---|---|---|---|---|---|---|---|---|
| **Complexation, sorption, precipitation of aqueous $Hg^{II}$** | | | | | | | | | |
| 196 µM (Cl⁻) | $HgCl_2$ | Complexation between $Hg^{II}$ and thiol resins | | O | −0.53± 0.15 | | | EIE-MDF, NFS | Wiederhold et al., 2010 |
| 207 µM (NO₃⁻) | $Hg(OH)_2$ | | | | −0.62± 0.17 | | | | |
| 5−25 µM | $HgOH^+$ $HgCl^+$ | $Hg^{II}$ sorption to α−FeOOH | | O | −0.37± 0.03 | −0.06 | | | Jiskra et al., 2012 |
| 100 µM | $Hg(OAc)_2$ | Sub-stoichiometric (10, 30, 50, 70%) amounts of $S^{2-}$ added at a start pH of 2.3–3.0 | | A | −0.63± 0.04 | | | KIE-MDF, NFS | Smith et al., 2015 |
| | $Hg^{2+}$ | Sub-stoichiometric (10, 30, 50, 70%) amounts of $OH^-$ added at a start pH of 1. | | A | −0.32 | | | | |
| **$Hg^{II}$ – $Hg^0$ equilibration** | | | | | | | | | |
| 300–328 nM $Hg^{II}$ | $Hg(OH)_2/Hg^0$ | Water | | A | 2.63± 0.37 | 0.28± 0.21 | 1.44 | EIE-MDF, NFS | Wang et al., 2021 |
| 150–173 nM $Hg^0$ | $HgCl_2/Hg^0$ | 10 mM NaCl | | | 2.77± 0.70 | | | | |
| **Heterogeneous $Hg^{II}$ reduction by surface-bound and structural $Fe^{II}$** | | | | | | | | | |
| 285 nM (NO₃⁻) | $Hg(OH)_2$ | $Hg^{II}$ reduction to $Hg^0$ by suspended $FeCO_3$ (s) | Siderite (0.1 g L⁻¹, pH 7.1) | A | 2.43 ± 0.38 | 0.09 | | EIE–MDF NFS | Wang et al., 2021 |
| | | | Green rust (0.01 g L⁻¹, pH 7.2) | | 2.28 ± 0.40 | | | | |
| 1 µM (NO₃⁻) | $Hg(OH)_2$ | $Hg^{II}$ reduction by suspended magnetite ($Fe^{II}Fe_2^{III}O_4$, surface area ~2 m² L⁻¹ | | A | −1.37 ± 0.07 | 0.13 ± 0.01 | 1.59 ± 0.09 | | Schwab et al., 2023 |
| **Photoreduction of $Hg^{II}$ doped on a diesel soot matrix** | | | | | | | | | |
| 12 µM (Cl⁻) | | Hg/C 7.8 × 10⁻⁵ (M/M) | Relative humidity 28% | A | −1.75 ±0.05 | 2.43 ±0.19 | 1.15 ± 0.01 | MDF, (−)MgIE | Huang et al., 2021 |
| | | | Relative humidity 68% | | −1.48 ±0.02 | 0.20 ±0.05 | | | |

### 8.6 Isotopic fractionation during air-surface $Hg^0$ gas exchange

The interaction between atmospheric Hg and the Earth's reservoirs has been discussed only briefly in **Section 3.2**, as this area has recently been covered by a literature review (Sommar et al., 2020). Importantly, the gas exchange of volatile Hg is bidirectional. Consequently, the net flux of Hg over an ecosystem may represent a delicate balance between opposing processes, including deposition/uptake versus re-emission. The end members of Hg exchange between the surface (biosphere, pedosphere, lithosphere, hydrosphere, and cryosphere) and atmosphere are all isotopically distinguishable (Liu et al., 2024). A combination of bulk measurements and analysis of stable Hg isotopic compositions enables separation of the contributions from atmospheric $Hg^{II}$ and $Hg^0$ deposition, as well as local partitioning between $Hg^0$ deposition and re-emission. The isotopic composition of atmospheric Hg is presented and discussed in **Section 8.2**. In addition to the data, an updated compilation of complementary isotopic Hg data for reservoirs that are in contact with the atmosphere and thus can undergo gas exchange has been produced during the preparation of this review (Liu et al., 2024). In the following, we express absolute deposition with negative values and vice versa for emission throughout.

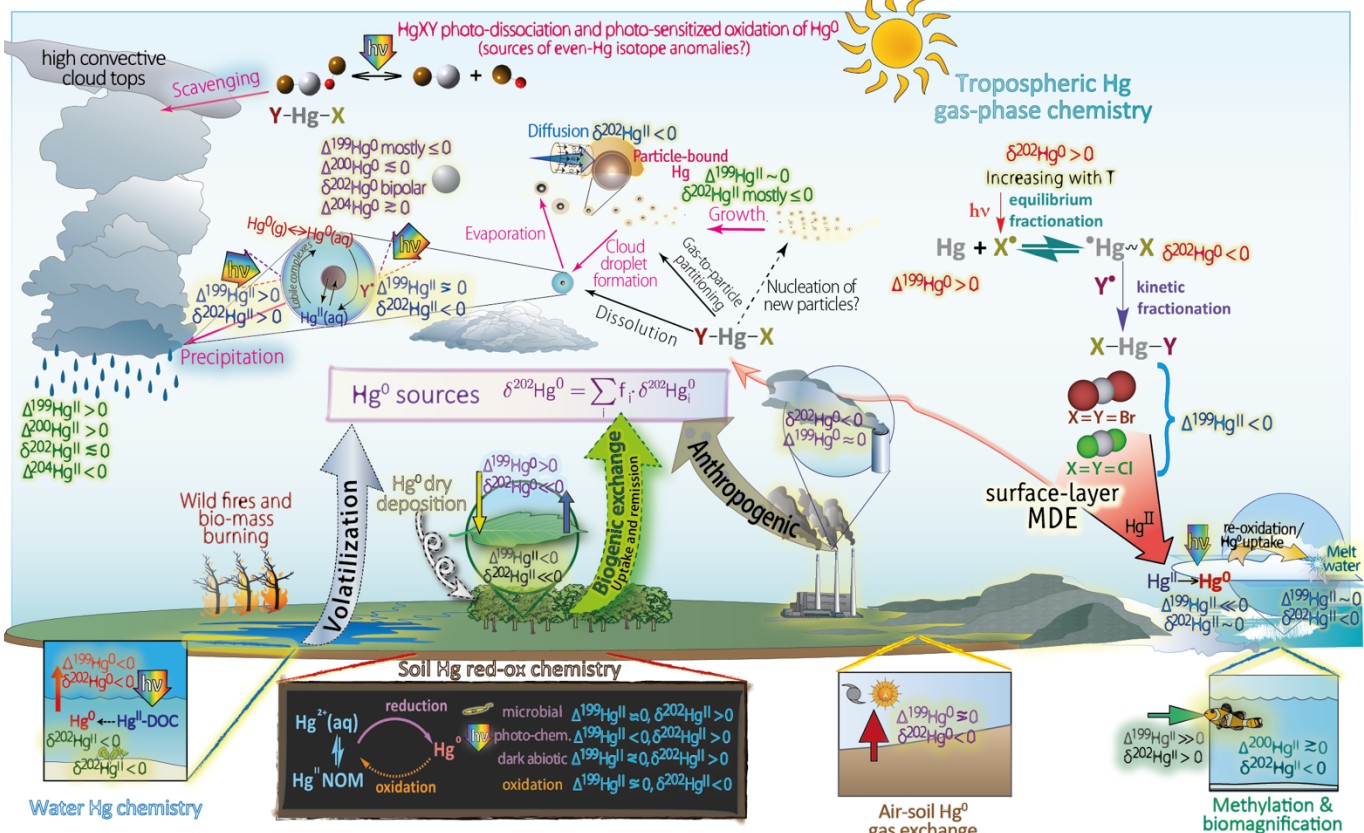

**Figure 17.** Schematic illustration showing key biogeochemical Hg processes in Earth's surface reservoirs and their associated Hg isotope fractionation along with corresponding isotopic composition observations focused on the atmosphere. **Section 8.2** addresses the isotopic characteristics of atmospheric mercury. The existing isotopic information on the gas-phase and water-phase redox transformations of mercury is presented in **Sections 8.3** and **8.4**, respectively. **Sections 8.5** and **8.6** describe isotopic fractionation in heterogeneous chemical processes and processes involving Hg$^0$ gas exchange between the atmosphere and the Earth's surface reservoirs, respectively.

### 8.6.1 Mixing and fractionation modeling of Hg$^0$ deposition and post-depositional processes

**Deposition**

Isotope-based modeling by binary (e.g., **Eq. 27**) and ternary mixing with MDF, odd-MIF, and even-MIF signatures of atmospheric Hg$^0$ and other Hg pools as end members has been applied to distinguish the fraction of Hg$^0$ deposition via vegetated surfaces (Wang et al., 2020b; Enrico et al., 2016; Obrist et al., 2017; Wang et al., 2019b; Li et al., 2022a; Li et al., 2023a; Li et al., 2023b), soil (Zheng et al., 2016; Obrist et al., 2017; Wang et al., 2019b; Wang et al., 2020a), water (Jiskra et al., 2021; Zhang et al., 2023a), throughfall (Wang et al., 2020b) and snow run-off (Douglas and Blum, 2019), estimated to be 60–90%, 32–105%, 50–85%, 34–82% and > 75% of total deposition, respectively. As a proxy for atmospheric Hg$^0$, foliage/litter Hg has been used as an end-member in mixing modeling of Hg$^0$ inputs to soil (Demers et al., 2013; Jiskra et al., 2015; Zhang et al., 2013), runoff (Jiskra et al., 2017), and stream water (Woerndle et al., 2018), which may introduce bias because a significant fraction of the gross air Hg$^0$ incorporated as Hg$^{II}$ in foliage is re-emitted after photoreduction (Yuan et al., 2019b). The contribution of Hg$^0$ deposition to vegetation Hg uptake is greatest in foliage, followed by branches, bark, stems and roots (Wang et al., 2020b; Liu et al., 2021a; Sun et al., 2017). The new Hg isotope evidence has demonstrated that Hg throughfall via the canopy and along stems, which was previously assumed to be derived mainly from wet and dry deposition of atmospheric RM (Wright et al., 2016), contains a larger proportion of Hg excreted from biomass, where it originated mainly from Hg$^0$ uptake followed by translocation. The isotope mixing formula is used to determine the proportions of different isotope sources in a mixture, the simplest form of which is as follows:

$$(\delta^{xxx}Hg)_{mix} = f_1 \cdot (\delta^{xxx}Hg)_1 + f_2 \cdot (\delta^{xxx}Hg)_2 \qquad (27)$$

$$f_1 + f_2 = 1$$

**Post-deposition**

Isotopic and concentration measurements of Hg$^0$ jointly in near-surface air and surface pore air/water, in addition to other isotopic

data, allow the inference of processes by mass balance or Rayleigh-type models at the air–soil interface and in the surface soil (Jiskra et al., 2019; Li et al., 2023a; Yuan et al., 2021; Chen et al., 2023). For poorly drained boreal organic soil horizons (histosols), in contrast to podzols, mixing modeling indicates significant reductive loss (24–33%) to the atmosphere by abiotic reduction (Jiskra et al., 2015). A further multi-process model is presented here, which is designed to elucidate the dynamic evolution of post-depositional Hg (>90% from litterfall) on the subtropical forest floor over a 500-year period (Yuan et al., 2020). The results indicate that photolytic and microbial reduction processes exert an influence during the initial few years but are subsequently superseded by dark redox processes (exhibiting NFS) in the compost, where $Hg^{II}$ finally becomes inert at depths of >10 cm in the horizon after approximately 420 years. Studies of forest soils in different climatic zones have shown that microbial reduction ($\varepsilon^{202}Hg = -0.4$‰, $E^{199}Hg \approx 0$, Kritee et al., 2007) plays a dominant role (Yuan et al., 2021; Chen et al., 2023), which, for rainforests, can explain up to 90% of the $Hg^{II}$ reduction in the upper soil horizon (Yuan et al., 2023b). In an open boreal peatland, photoreduction dominated the post-depositional process, accounting for the transformation of 30% of the annually deposited Hg (Li et al., 2023a).

## 8.6.2 Enclosure and related flux measurements

Experimental investigations employing dynamic flux chambers (DFCs) have been conducted in both ambient and controlled environments with the objective of elucidating the isotopic dynamics of $Hg^0$ exchange between the atmosphere and vegetation at the branch level (Yuan et al., 2019b; Chen et al., 2023), as well as between air and soil (Yuan et al., 2021; Chen et al., 2023; Zhu et al., 2022; Zhang et al., 2020), water (Zhang et al., 2023a), and snow (Sherman et al., 2010). For this application, in addition to traditional chambers (Demers et al., 2013; Chen et al., 2023; Zhu et al., 2024), a type was used that produces a uniform surface friction velocity over flat ground to couple with ambient shear conditions to scale to the ambient flux (Yuan et al., 2021; Yuan et al., 2023b; Lin et al., 2012). The surface-atmosphere $Hg^0$ flux is the result of complicated bidirectional processes, including $Hg^0$ efflux from the surface and direct atmospheric $Hg^0$ deposition.

**Deposition and sink processes**

When direct $Hg^0$ deposition is measured absolutely and isotopically with a DFC, enrichment factors for TMDF ($\varepsilon^{202}Hg_{air/surface}$) and odd-MIF ($E^{199}Hg_{air/surface}$) may be calculated via a linearized Rayleigh fractionation model (Zhu et al., 2022; Mariotti et al., 1981):

$$
\begin{aligned}
\delta^{202}Hg^0_{DFC} - \delta^{202}Hg^0_{air} &= \varepsilon^{202}Hg_{surface-air} \cdot \ln\left(c^{Hg^0}_{DFC}/c^{Hg^0}_{air}\right) \\
\Delta^{199}Hg^0_{DFC} - \Delta^{199}Hg^0_{air} &= E^{199}Hg_{surface-air} \cdot \ln\left(c^{Hg^0}_{DFC}/c^{Hg^0}_{air}\right)
\end{aligned}
\tag{28}
$$

where c represents the concentration and the indices air and DFC refer to the air entering and exiting the DFC, respectively. Alternatively, **Eq. 28** is applied to extract $\varepsilon^{202}Hg_{surface-air}$ using measurements of $c^{Hg^0}$ and $\delta^{202}Hg^0$ at two pristine sites with and without vegetation (Enrico et al., 2016) or using day- vs. night-time segregated ambient air data at the same site (Jiskra et al., 2019). When direct deposition is measured isotopically with a DFC, the residual $Hg^0$ in the chamber outlet shifts to be preferentially isotopically heavier, with a large but variable discrimination observed over soils ($\varepsilon^{202}Hg_{soil-air} = \sim 0$ to $-5.8$‰, Chen et al., 2023; Yuan et al., 2023b; Zhu et al., 2022) and over vegetation ($\varepsilon^{202}Hg_{foliage/air} = \sim -1$ to $-4.2$‰, Yuan et al., 2019b; Enrico et al., 2016; Demers et al., 2013; Jiskra et al., 2019; Chen et al., 2023). Deposition in contact with any surface does not result in a significant change in $\Delta^{199}Hg^0$, unlike the situation with $\delta^{202}Hg^0$.

Information on the sink processes of $Hg^0$ in the soil can be obtained by pursuing measurements of isotopic $Hg^0$ in the soil pore air under sub-ambient concentration regimes. In tundra (Jiskra et al., 2019) and peatlands (Li et al., 2023a), the isotopic differences between ambient $Hg^0$ and pore gas $Hg^0$, whose concentrations is sub-ambient ($\sim 0.4 - \sim 0.6$ and $\sim 0.2 - \sim 0.7$ ng m$^{-3}$) and therefore mediate $Hg^0$ net diffusion into the substrate via **Eq. 28,** have been linked to DOM-driven anaerobic oxidation in soil water exhibiting EIE (Zheng et al., 2019). Investigations of the $Hg^0$ level in the pore air of forest soils provide a mixed picture, ranging from sites with highly depleted air (Obrist et al., 2014) to sites with up to ten times enriched pore air (Yuan et al., 2019a) compared with the ambient concentrations above. In subtropical (Yuan et al., 2019a) and subalpine (Chen et al., 2023) forest soils, the concentration of $Hg^0$ in pore air is typically higher than that in near-surface ambient air and shows seasonal isotopic variations (TMDF and odd-MIF), suggesting complexity in $Hg^0$ gas exchange between air and soil. In tropical forest soils, pore air shifts from being nearly ambient during the rainy season to being markedly sub-ambient during the dry season (Yuan et al., 2023b). To resolve $Hg^0$ flux partitioning here, a combination of DFC measurements of net fluxes and forced unidirectional efflux, soil pore air, and Hg isotopic composition

in forest soil depth profiles are employed as inputs into isotope mass balance models based on odd-MIF (Yuan et al., 2021). Net fluxes measured by DFC are interpreted as a ternary mixing of deposition, $Hg^0$ losses from the surface soil via $Hg^{II}$ photoreduction, and a term generated by Hg redox processes (dark/microbial reduction vs. oxidation) in the organic soil horizon. Although associated with considerable uncertainties, the estimated gross deposition to the forest floor is between -7.8 and -1.8 ng m$^{-2}$ h$^{-1}$ for the subtropical site (Yuan et al., 2021) and between -6.7 and -4.4 ng m$^{-2}$ h$^{-1}$ for the tropical site (Yuan et al., 2023b), depending on the season, and between -4.9 and -2.0 ng m$^{-2}$ h$^{-1}$ for the subalpine site, depending on the type of forest floor (Chen et al., 2023).

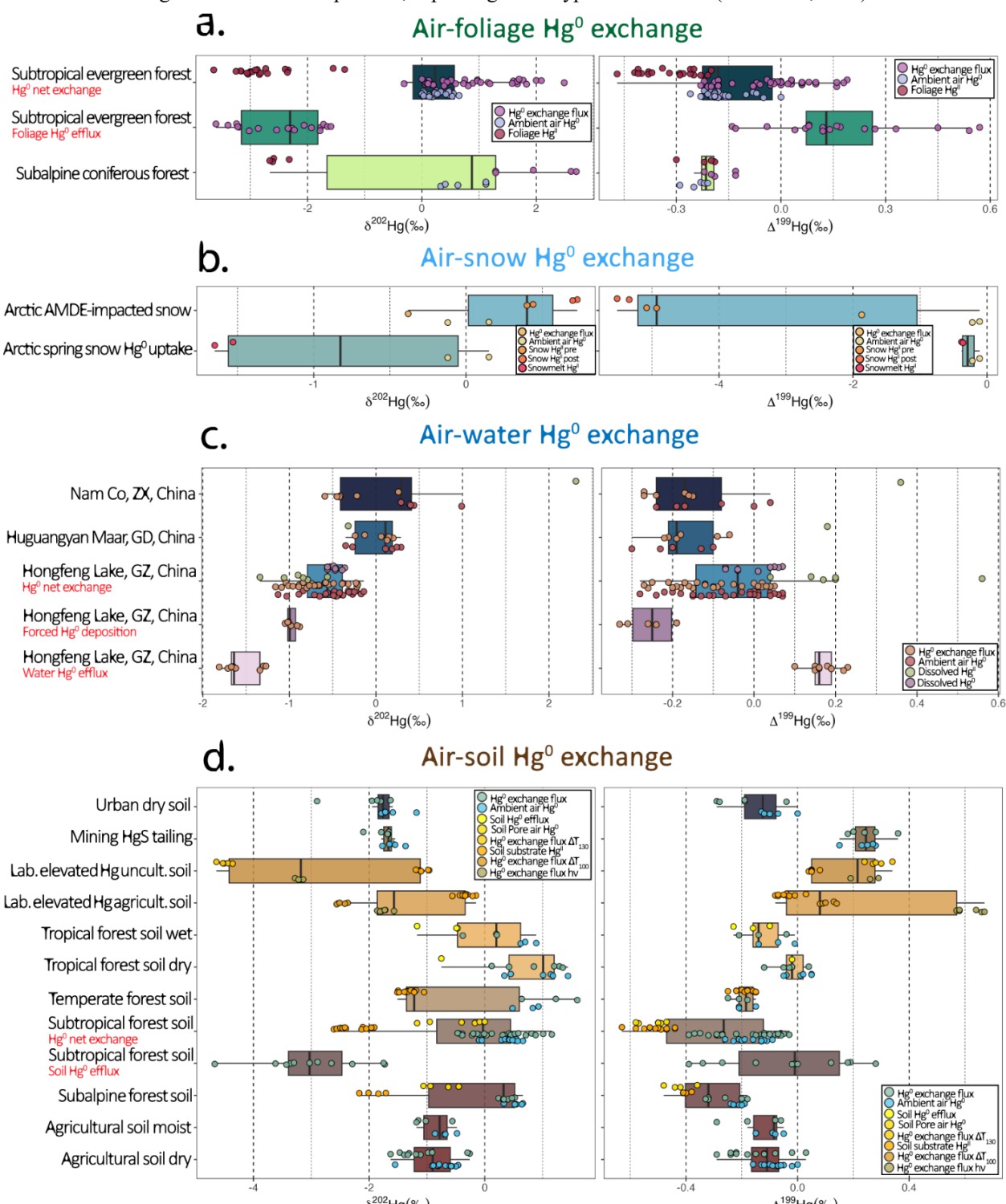

**Figure 18.** Statistical summary of observations from isotopic studies of $Hg^0$ exchange between the atmosphere and various groups of surface reservoirs on Earth. In the air-foliage group, data were taken from Yuan et al. (2019b) and Chen et al. (2023). For the air–snow group, data were taken from Sherman et al. (2010) and Douglas and Blum (2019). For the air–water group, data were taken from Zhang et al. (2023a; 2021a). For the air–soil group, data were taken from Zhang et al. (2020), Zhu et al. (2022), Yuan et al. (2021; 2023b) and Chen et al. (2023).

Foliar oxidation of $Hg^0$ drives its reactive uptake and is the most important step in the accumulation of initially $Hg^0$ uptake by plants (Liu et al., 2021b). Direct bio-oxidation from $Hg^0$ to $Hg^{II}$ has been traced to heme enzymes that catalyze the degradation of $H_2O_2$, specifically to a ferryl ($O=Fe^{IV}$) catalase radical cation complex (Ogata and Aikoh, 1984) that swiftly oxidizes $Hg^0$ ($1.4 \times 10^4$ $M^{-1}$ $s^{-1}$, Wigfield and Tse, 1986):

$$H_2O_2 + \langle Fe^{II}-E^{\bullet+} \leftrightarrow Fe^{III}-E \rangle \longrightarrow H_2O + \langle O=Fe^{IV}-E^{+\bullet} \leftrightarrow O=Fe^{III}-E \rangle$$
$$Hg^0 + \langle O=Fe^{IV}-E^{+\bullet} \leftrightarrow O=Fe^{III}-E \rangle + 2\,H^+ \longrightarrow Hg^{2+} + \langle Fe^{II}-E^{+\bullet} \leftrightarrow Fe^{III}-E \rangle + H_2O \qquad \text{(Rxn 14)}$$

here, E represents the heme group attached to the enzyme, which can provide an electron, reducing the formal oxidation number of iron from five to four. Divalent Hg readily binds to soft functional groups on the enzyme as soon as it is formed. MDF fractionation during oxidation of the absorbed isotopically light $Hg^0$ causes the product pool to be heavier than the reactant, which is consistent with observations that $Hg^{II}$ incorporated into leaf shoots is only slightly lighter than $Hg^0$ in ambient air. Notably, in contrast to the Hg pool in the leaf shoots, the Hg in the growing foliage of the current year shifted rapidly in the first months toward clearly negative

$\delta^{202}Hg^{II}$ signatures, the causes of which have been discussed elsewhere (Yuan et al., 2019b).

In contrast to the observations regarding the air modified by interactions with soil and foliage, the residual $Hg^0$ in the outgoing air is significantly lighter than that in the incoming air ($\delta^{202}Hg^0_{air} - \delta^{202}Hg^0_{DFC} = 0.38‰$), as observed by DFC, for deposition regimes over freshwater surfaces (Zhang et al., 2023a). This may be interpreted as the dissolved $Hg^0(aq)$ being consumed by oxidation, whereby the rapid exchange of Hg isotopes between the remaining $Hg^0$ and the formed $Hg^{II}$ (**Section 8.4.2**) causes the

2105 former, which is partially returned to the gas phase, to exhibit a more negative $\delta^{202}Hg^0$. In surface waters, photolytic re-reduction is also possible, which can be used for determining the isotopic composition of dissolved $Hg^0$ (Zhang et al., 2021a).

During colder seasons with limited solar radiation, there is a small but persistent net $Hg^0$ dry deposition over the snow-covered Arctic interior tundra (Obrist et al., 2017), whose interstitial snow air has sub-ambient concentrations (0.69 vs. 1.07 ng $m^{-3}$) with comparatively more positive $\delta^{202}Hg^0$ values (1.08 vs. 0.77‰, Jiskra et al., 2019). Using the exclusion method, this trajectory may

reflect $Hg^0$ uptake by ground lichens (Olson et al., 2019). Compared with the hinterland snowpack (~50 ng $m^{-2}$), the Arctic coastal snowpack has regionally much higher $Hg^{II}$ pools (>2000 ng $m^{-2}$), which are characteristically released as an ionic pulse in the runoff during snowmelt. High $Hg^{II}$ concentrations in the coastal marine cryosphere are partially explained by AMDEs (described in **Section 3.2**, Douglas et al., 2017). However, coastal AMDE deposition is mostly re-emitted as $Hg^0$ to the atmosphere before snow melts (see below). In contrast, the pulse in runoff appears to be related mainly to the reactive uptake of $Hg^0$ on marine snow,

which is rich in halogen compounds and other reactive species (see **Section 7.3**) (Douglas et al., 2017). In support of the significant reactive uptake of $Hg^0$ on salt-laden snow, analogous odd-MIF signatures between ambient air $Hg^0$ and snowmelt $Hg^{II}$ have been reported ($\Delta^{199}Hg$ values documented in **Fig. 18b**, Douglas and Blum, 2019).

**Net exchange, source processes and flux partitioning**

Owing to the length of time (typically a few days) required to accumulate sufficient Hg to perform isotopic analysis, samples from a

2120 DFC measurement are a composite of periods of net emission and net deposition, unless the $Hg^0$ concentration in the inlet is manipulated so that emission or deposition becomes persistent within the chamber. The TMDF and odd-MIF signatures from DFC measurements in ambient air ("net $Hg^0$ exchange") are calculated as follows (Zhu et al., 2022):

$$\delta^{202}Hg^0_{exchange} = \left( \delta^{202}Hg^0_{DFC} \cdot c^{Hg^0}_{DFC} - \delta^{202}Hg^0_{air} \cdot c^{Hg^0}_{air} \right) / \left( c^{Hg^0}_{DFC} - c^{Hg^0}_{air} \right)$$
$$\Delta^{199}Hg^0_{exchange} = \left( \Delta^{199}Hg^0_{DFC} \cdot c^{Hg^0}_{DFC} - \Delta^{199}Hg^0_{air} \cdot c^{Hg^0}_{air} \right) / \left( c^{Hg^0}_{DFC} - c^{Hg^0}_{air} \right) \qquad (29)$$

In the special case of using Hg-free air (zero air) to feed the DFC, $\delta^{202}Hg^0_{emission}$ and $\Delta^{199}Hg^0_{emission}$ can be determined. The enrichment factors during net $Hg^0$ exchange and emission are calculated using the following set of equations:

$$\varepsilon^{202}Hg_{exchange} = \delta^{202}Hg^0_{exchange} - \delta^{202}Hg_{surface} \quad , \quad E^{199}Hg_{exchange} = \Delta^{199}Hg^0_{exchange} - \Delta^{199}Hg_{surface}$$
$$\varepsilon^{202}Hg_{emission} = \delta^{202}Hg^0_{emission} - \delta^{202}Hg_{surface}, E^{199}Hg_{emission} = \Delta^{199}Hg^0_{emission} - \Delta^{199}Hg_{surface} \qquad (30)$$

In a series of light, temperature and substrate moisture controlled laboratory experiments with untilled (forest) and tilled (agricultural) soils, both with elevated Hg levels, enclosed in a DFC fed with Hg-free air, large $Hg^0$ fluxes (≥500 ng $m^{-2}$ $h^{-1}$) were unanimously

associated with the most negative $\delta^{202}Hg^0_{emisson}$ values (−2.9 to −2.2‰ and −4.4 to −4.2‰ for agricultural and forest soils, respectively) when substrates were exposed to elevated temperatures in the dark (100–130 °C vs. 40 °C), while treatments with light, moisture, or a combination of both at room temperature produced more moderately negative $\delta^{202}Hg^0_{emisson}$ values (−2.1 to −1.6‰ and −3.3 to −2.6‰ for agricultural and forest soils, respectively, Zhang et al., 2020). $E^{199}Hg_{emission}$ of agricultural and forest soils displays a value of approximately 0.2 ‰, and $\Delta^{199}Hg/\Delta^{201}Hg$ was ~ 1.55 for the temperature controls, suggesting that the treatment caused $Hg^0$ loss propelled by the thermally driven reduction in $Hg^{II}$ in the dark (**Section 8.4.1**). In the light and light–moisture exposure controls, the substrates differed in terms of the observed $E^{199}Hg_{emission}$, which for agricultural soils was 0.67 to 0.76‰ (mean) and for forest soils of a small magnitude, both positive and negative (−0.03 to 0.18‰, mean). The $E^{199}Hg_{emission}$ dichotomy may be interpreted as derived from a composite with $\Delta^{199}Hg$ contributions from both (−)MgIE and (+)MgIE-induced $Hg^{II}$ photoreduction pathways, almost completely dominated by (−)MgIE processes ($Hg^{II}$ bound to, e.g., N, S-containing ligands) for agricultural soils and for forest soils with a larger contribution from (+)MgIE processes ($Hg^{II}$ bound to, e.g., O-containing ligands), balancing odd-MIF fractionation from (−)MgIE processes. However, the agricultural soil placed under water (rice paddy) photoemits $Hg^0$ characterized by a negative $\Delta^{199}Hg^0$ ($\Delta^{199}Hg^0_{emission}$ = −0.38 ± 0.18‰, Zhang et al., 2024), which is indicative of all observed $Hg^{II}$ photoreduction in natural freshwaters studied in the laboratory as well as in situ.

A field study with DFC of cultivated or managed soils measured exchange fluxes (an MDF Rayleigh model yielded a 10-27 % contribution from deposition), which revealed net $Hg^0$ emissions (fraction) associated with average $\varepsilon^{202}Hg_{exchange}$ of −1.1 to −0.1‰ and −1.6 to −0.2‰ and average $E199Hg_{ex}$ values of −0.27 to −0.13‰ and 0.00 to 0.14‰ for rural and urban soils, respectively. The above enrichment factors and $E^{199}Hg_{exchange} \approx E^{201}Hg_{exchange}$ indicate that the emitted $Hg^0$ comes mainly from the pool produced by photoreduction. The air concentration positively influences the magnitude of deposition in soils so that at a critical concentration level (compensation point), the net flux tends to change direction. This is reflected in the apparent $\varepsilon^{202}Hg_{exchange}$, which varies with the ambient $Hg^0$ concentration (Zhu et al., 2022). Analogous to laboratory experiments, in situ experiments on the subtropical forest floor have revealed that soil emissions of $Hg^0$ are strongly negative $\delta^{202}Hg^0_{emission}$ (mean −3.0‰, Yuan et al., 2021), while the magnitude of $\delta^{202}Hg^0_{emission}$ for the tropical rainforest floor is much smaller but still negative (mean −0.7‰, Yuan et al., 2023b). $E^{199}Hg_{emission}$ for subtropical forest soils exhibit positive values for all seasons over a considerable range (mean 0.1–0.7‰), whereas for rainforests, $E^{199}Hg_{emission}$ is consistently positive, albeit to a lesser extent (mean 0.2–0.3‰). Limited negative $\delta^{202}Hg^0_{exchange}$ values (mean values of –0.26, –0.54, –0.07 and –0.09 ‰) and consistently positive $E^{199}Hg_{exchange}$ values (mean values of 0.42, 0.23, 0.39 and 0.30 ‰) are observed in net $Hg^0$ gas exchange experiments over subtropical (Yuan et al., 2021), tropical (Yuan et al., 2023b), subalpine (Chen et al., 2023) and temperate (Demers et al., 2013) forest soils, respectively. In conclusion, bare or cultivated soils result in a greater degree of MDF isotope fractionation associated with $Hg^0$ gas exchange with the atmosphere than do forest soils, where the effects of photic and thermal processes are limited by canopy shading. Temporally extensive chamber measurements conducted globally over the forest floor indicate net emissions (Yuan et al., 2019a). For the first three forest soil studies mentioned above, the DFC dataset also contains sufficient isotope data to enable the modeling of net flux partitioning into gross emission and gross deposition.

Re-emissions of $Hg^0$ from perennial foliage of three beech species show an average positive $\varepsilon^{202}Hg_{emission}$ and $E^{199}Hg_{emission}$ of 0.6 and 0.3‰, respectively. The studied net exchange of $Hg^0$ between foliage and air for montane evergreen deciduous (Yuan et al., 2019b) and spruce (Chen et al., 2023) forests is mostly on the uptake side, which indicates that $\delta^{202}Hg^0_{DFC}$ is generally more positive than that of ambient air (**Fig. 18a**, mean shift of 0.72‰ for the latter site). The presence of bidirectional fluxes is, however, reflected in the observation that the $E^{199}Hg_{exchange}$ for both sites is consistently positive (mean 0.08 and 0.13‰, respectively), albeit modestly, due to a contribution from $Hg^0$ emissions resulting from (−)MgIE-induced photoreduction.

Isotopic studies of air–snow $Hg^0$ interactions and post-depositional processes have typically been conducted in the Arctic (Araujo et al., 2022; Sherman et al., 2010; Zheng et al., 2021; Jiskra et al., 2019; Obrist et al., 2017; Douglas and Blum, 2019), with occasional studies at mid-latitudes (Kurz et al., 2021; Yuan et al., 2022). Hg in aging snowpacks exhibits by far the most extensive distribution of $\Delta^{199}Hg^{II}$ among Earth's surface reservoirs, with observations of $\Delta^{199}Hg$ progression reported in both positive (Kurz et al., 2021)

and negative (Sherman et al., 2010; Zheng et al., 2021; Douglas and Blum, 2019) directions relative to fresh snow. As discussed in **Section 8.2.1**, the larger $\Delta^{199}$Hg spread observed in polar airborne Hg (Hg$^0$ and RM) than in, for instance, high-altitude air from mid-latitudes can be attributed to the influence of AMDEs (during spring after sunrise and during summer) on a significant proportion of the collected polar data. Snow(fall) during the polar night is characterized by positive or near-zero $\Delta^{199}$Hg signatures, as is the case for most global precipitation data (**Fig. 13c**), while the $\Delta^{199}$Hg values of polar Hg$^0$ for the same period are all slightly negative, which is consistent with the global Hg$^0$ background pool (**Fig. 13a**). Only sporadic isotopic DFC measurements have been conducted over snow, yet ample measurements of polar air and snow as endmembers still offer an understanding of air-surface Hg$^0$ exchange following Hg$^{II}$ deposition associated with AMDE. A seminal set of isotope data (Sherman et al., 2010) demonstrating a substantial odd-MIF triggered because of Hg$^{II}$ photoreduction in snow was obtained from samples collected during a 9-day AMDE at the Alaskan Arctic coastline in conjunction with periods of minimal snowfall carrying high concentrations of scavenged Hg$^{II}$ (0.5 ± 0.4 µg L$^{-1}$; Johnson et al., 2008). Fresh snow, surface snow, and drifting snow presented, in order, rapidly increasing negative $\Delta^{199}$Hg$^{II}$ values of –0.95 to –1.20‰, –2.41 to –2.63‰, and –3.84 to –5.08‰, which, according to Rayleigh fractionation, can correspond to 5–30%, 35–50%, and 65–75% photoreduced Hg$^{II}$, respectively. A chamber measurement was conducted on AMDE-impacted drifting snow that had undergone substantial photoreduction ($\Delta^{199}$Hg ~–5.0‰) for 10.5 h of sunlight. The total DFC throughput, including the Hg$^0$ emissions corresponding to 6% of the total Hg$^{II}$ in the snow plot (whose $\Delta^{199}$Hg$^{II}$ dropped to ~–5.4‰), exhibited a $\Delta^{199}$Hg$^0$ of –1.87‰. Mid-latitude snow (MI, USA), derived from polar vortex-transported air masses originating in AMDE-affected subarctic regions, shows, when $\Delta^{199}$Hg is plotted against $\delta^{202}$Hg, a regression of -3.32 ± 1.19 (Kurz et al. 2021), which, given the uncertainty in the line fit, appears to agree well with the corresponding regression of data from the Alaska DFC snow experiment of -3.44 ± 0.70 (Sherman et al, 2010).

Perennial data from the Canadian High Arctic show that Hg$^{II}$ deposited on snow during the most frequent phase of AMDEs just after polar sunrise until early May, which is partly characterized by low temperatures and Arctic haze, has a significantly greater susceptibility to photoreduction and loss as Hg$^0$ (up to 60%) than that deposited later (<20%, Zheng et al., 2021). As previously stated in **Section 5.1.4**, airborne Hg$^{II}$ originating from high Arctic AMDEs undergoes rapid conversion to the particle phase between March and April, whereas unconverted GOM remains the dominant form between May and June. The cause of the reactivity of deposited Hg$^{II}$ is unclear (Sherman et al., 2010; Kurz et al., 2021). It has been speculated that components of Arctic haze, such as black carbon, that cause photoreactivity of particulate Hg$^{II}$ are the cause of the observed (–)MgIE signature (Zheng et al., 2021), which is supported by water-phase experiments with Hg$^{II}$ and dissolved black carbon (**Table 8**, Li et al., 2020b). Concurrently, the restricted Hg$^{II}$ reduction observed in Arctic snow toward the conclusion of spring is consistent with concurrent observations of substantial reactive uptake of Hg$^0$ (see above; Douglas and Blum, 2019), indicating that the snowpack then contains species with a predominant oxidative capacity. However, during snowmelt on the inland tundra, net Hg$^0$ deposition is disrupted by shifts in the isotopic signatures of snow interstitial air to those indicative of photoreduction, with $\Delta^{199}$Hg values decreasing to –1.37‰ in snow and –0.62‰ in snow interstitial air, which are consistently lower than those in ambient air (–0.23 ± 0.06‰). In contrast to Arctic snow, snow sampling in the U.S. Great Lakes area (with the exceptions noted above) generally results in increasing positive $\Delta^{199}$Hg$^{II}$ values (up to 3.51‰) in aging snow (Kurz et al., 2021). Indicative of (–) and (+) MgIE triggering photoreduction, respectively, the snow data from coastal Alaska (Sherman et al., 2010; Douglas and Blum, 2019) and the Great Lakes region (Kurz et al., 2021) show steeper $\Delta^{199}$Hg/$\delta^{202}$Hg trajectories than is the case for any of the well-studied Hg$^{II}$ complex photoreductions in the laboratory (**Tables 8 & 9**), leaving the question of which snow Hg$^{II}$ complexes are involved.

The mean MIF values ($\Delta^{199}$Hg$^{II}$ and $\Delta^{200}$Hg$^{II}$) in the pools of fresh and seawater are between the mean values of global atmospheric Hg$^0$ and wet precipitation. However, the variation is particularly pronounced for $\Delta^{199}$Hg$^{II}$ in coastal seawater, lakes, and river water (Liu et al., 2024). After three different categories of lakes with DFC were studied, a $\Delta^{200}$Hg isotope mass balance model was used to partition the overall net emission fluxes into gross emission and deposition fluxes, which ranged from 2.1 to 4.2 ng m$^{-2}$ h$^{-1}$ and from –2.3 to –1.2 ng m$^{-2}$ h$^{-1}$, depending on the lake (Zhang et al., 2023a). Hg$^0$ gross deposition exceeds the measured wet deposition across these lakes and accounts for 56–85% of the total deposition (Feng et al., 2022). The anomalous observation of preferential deposition of heavier Hg isotopes over water has already been discussed. The results of the volatilization experiments of dissolved Hg$^0$ in water

indicate an MDF enrichment factor ($\varepsilon^{202}Hg^0_{air-water}$) of −0.45‰ and a negligible $E^{199}Hg^0_{air-water}$ (Zheng et al., 2007). Emission-controlled experiments for one of the lakes yielded $E^{199}Hg_{emission}$ of –0.38‰ and $\varepsilon^{202}Hg_{emission}$ of –0.31‰, which are subject to large uncertainties, with a resulting $E^{199}Hg_{emission}/\varepsilon^{202}Hg_{emission}$ trajectory of 1.26 ± 0.72, which is within the margin of error for $Hg^{II}$ photoreduction mediated by fulvic acids (1.15 ± 0.07, Bergquist and Blum, 2007). The isotopic tracing of the formation of dissolved $Hg^0$ in peat-covered groundwater from $Hg^{II}$ in rainwater (1.24 ± 0.68) has also suggested that this process is the same type of photoreduction (Li et al., 2023a). The $E^{199}Hg_{exchange}$ was between –0.76 and –0.32‰, with the highest absolute value for a clear mountain lake fed mainly glacial water, indicating that (+)MgIE photoreduction plays an important role, as has been shown early in laboratory experiments on natural freshwater (Bergquist and Blum, 2007; Zheng and Hintelmann, 2009). The observed substantial positive $\Delta^{199}Hg^{II}$ shift of the sampled lake surface waters relative to $Hg^{II}$ in precipitation can be interpreted as an effect of partial photoreduction of $Hg^{II}$. However, other sources, including MMHg photodegradation, have been suggested (Chen et al., 2016). As discussed in **Section 4.2**, $Hg^0$ emissions from the ocean represent a primary source of Hg in the atmosphere. However, the isotopic signatures of this emission source remain largely unknown. In the absence of in-situ sampling, photoexperiments with $Hg^{II}$ in the presence of DOM extracted from marine phytoplankton produce (–)MgIE during reduction, in contrast to freshwater DOM (Kritee et al., 2018).

## 9. Future perspectives

### 9.1 Theoretical chemistry contributions & challenges

This examination of the advancements made in our comprehension of the mercury cycle in the troposphere and stratosphere reveals iterative interactions among three distinct branches of atmospheric chemistry (modeling, field measurements, and laboratory measurements). Advances in computational chemistry have made seminal contributions to our understanding of gas-phase $Hg^{(I,II)}$ molecules in terms of their geometries, energies, UV–VIS spectra, and reaction kinetics. The treatment of strong relativistic effects, which largely determine the chemistry of Hg-containing species, is crucial for accurate results. Ab initio thermochemical calculations for atmospheric Hg species are performed at a higher level of theory, which incorporates core–valence electron correlation and coupled-cluster methods. This approach yields a significantly improved accuracy of ≤ 4 kJ mol$^{-1}$, in accordance with high-quality experimental data. However, significant uncertainties in the estimates of the binding strength and thermal and photolytic stability of $^{\bullet}Hg^{I}I$ (**Section 5.1.2**) remain, limiting the ability to assess the occurrence and significance of iodine-induced $Hg^0$ oxidation in the troposphere and lower stratosphere, as has been suggested from atmospheric observations (Murphy et al., 2006; Lee et al., 2024).

Compared with ab initio thermodynamics, the calculation of ab initio kinetics is a much more challenging task, for which transition state theory (TST) and RRKM theory are often used for barrier and non-barrier bimolecular reactions, respectively. More flexible methods (e.g., variational TST) are now applied to optimize the position of the transition state (TS) by varying it along the reaction coordinate to minimize the free activation energy, which more accurately estimates the rate than traditional TST, which assumes a single, fixed TS that irreversibly leads to products. The calculation of TS energies is more challenging than the calculation of energies of relative minima (metastable species) because of the involvement of extended bonds where the electronic wave function is less dominated by a single electronic configuration. Obtaining a correct barrier energy is crucial for calculating reliable rate constants, as a bias of ~4 kJ/mol in the barrier height can lead to an error of nearly an order of magnitude in the resulting rate constant (Ariya and Peterson, 2005; Ariya et al., 2009).

For gas-phase reactions (**Section 5**), calculated rate constants have been presented and compared with those experimentally determined in the laboratory. The level of agreement varies from relatively good (≤ 30% as **Rxn G1 - G3**) to inconsistent (**Rxn G20a,b & G22**). Owing to the complex shape of their potential energy surfaces, the rates of assumed key reactions such as **Rxn G27**, **G45**, and **G63** are inherently difficult to constrain theoretically (**Section 5.1.4**) and thus require empirical verification, preferably using PLP-LIF or similar techniques. A direct reaction between water vapor and $YHg^{II}O^{\bullet}$ has recently been proposed for Y = OH (**Rxn G60**, Saiz-Lopez et al., 2022). If this reaction is realized with the given rate expression in models, it will result in the

conversion of essentially all HOHg$^{II}$O$^{\bullet}$ to the completely stable Hg(OH)$_2$ in the tropics. This type of reaction also requires empirical verification and should be given priority in laboratory experiments.

## 9.2 Laboratory measurement techniques & limitations

The absolute determination of rate constants experimentally with pulsed laser-assisted methods (reaction times typically < 0.1 ms), such as PLP-LIF, is more easily facilitated when secondary reactions are negligible and therefore does not contribute to the measured values. In general, absolute determination is conducted by obtaining pseudo-first-order conditions, whereby the more stable reactant is present in a density more than tenfold that of the other reactant. However, for Hg, this method is viable only for studies that are conducted at elevated temperatures (typically ≥ 100 °C). At atmospheric temperatures, the relatively low vapor pressure of Hg$^0$ (in comparison to, for instance, DMHg) precludes the possibility of such experiments. Despite the challenges for Hg$^0$, a flow PLP-LIF system has many advantages, including the ability to measure the rate coefficient over a wide range of temperatures and pressures and to test the effect of a change in the bath gas (third body). Nevertheless, to exploit these advantages, alternative methods have been used in which the Hg species is not in excess, but in which the excess is X = Cl and Br when the reaction Hg + X$^{\bullet}$ + M is studied, while it is instead Y = O$_3$, NO$_2$, NO, and O$_2$ when the interaction between $^{\bullet}$Hg$^I$Br and Y is studied. In the study of the former reaction type, X$^{\bullet}$ is present in excess, but its concentration decreases over time owing to the rapid three-body recombination of the species into X$_2$ and M. This results in additional Hg$^0$ exponential decay. To achieve a fit to the observed Hg$^0$ time profiles, rate coefficients must be obtained through numerical integration. This requires monitoring both the X$^{\bullet}$ and Hg$^0$ time profiles using LIF, with the absolute concentration of X atoms known with precision. The experimental measurements of the rate coefficients for the Hg + X$^{\bullet}$ + M reaction by Donohoue et al. (2005; 2006) are in accordance with the findings of theoretical computational studies.

The conversion of $^{\bullet}$Hg$^I$Br by bimolecular elimination reduction (**Rxn G18 & G20b**; Wu et al., 2020; Wu et al., 2022), addition (oxidation assisted by M [**Rxn G20a**]; Wu et al., 2020), or abstraction (**Rxn G22**, Gómez Martín et al., 2022) is constrained by the capacity to generate sufficiently high densities of $^{\bullet}$Hg$^I$Br through the gas-phase photolysis of HgBr$_2$ in deep UV. Because the vapor pressure of HgBr$_2$ is low (less than one-tenth that of Hg$^0$), it is necessary to keep the HgBr$_2$ source at least 30 °C and the flow tube reactor at least 10 °C higher to prevent vapor condensation. A higher temperature increases the thermal dissociation of $^{\bullet}$Hg$^I$Br; therefore, a large excess of Y is required for the $^{\bullet}$Hg$^I$Br + Y reaction to dominate the conversion of $^{\bullet}$Hg$^I$Br. In the context of laboratory experiments necessitating deep UV irradiation, it is essential to consider that oxygen atoms are formed through the partial photolysis of O$_3$ and NO$_2$, thereby enabling O($^3$P) to react with $^{\bullet}$Hg$^I$Br (O + HgBr$^{\bullet}$ → Hg + BrO$^{\bullet}$, **Rxn G23**). Experiments to study the reactions of $^{\bullet}$Hg$^I$Br with NO$_2$ and O$_3$ will inevitably result in the observation of a partially reversible oxidation process. This is due to the occurrence of secondary chemistry, including reactions **G14**, **G23**, **G24**, and **G29**, which take place concurrently with the title reactions **G20** and **G22**. Furthermore, to elucidate the influence of secondary chemistry on the observed $^{\bullet}$Hg$^I$Br disappearance, a comprehensive series of experiments must be conducted, with pressure, temperature, [$^{\bullet}$Hg$^I$Br], [Y], and [O] as variables. This necessitates numerical modeling to isolate the individual rate constants. While the laboratory study of $^{\bullet}$Hg$^I$Br + O$_3$ gives an experimental rate constant for the reaction **G22** that is in good agreement with computational predictions (Castro Pelaez et al., 2022), experimental kinetic data for $^{\bullet}$Hg$^I$Br + NO$_2$ (**Rxn G20**), which must be decoupled into termolecular oxidation (**Rxn G20a**) and reduction (**Rxn G20b**) reactions, respectively, indicate that computational methods overestimate the rate constants for both channels (Wu et al., 2020). Later, experimental investigations revealed that **Rxn G20** cannot fully account for observations but that significant losses of $^{\bullet}$Hg$^I$Br must occur via side reactions, probably involving **Rxn G23**, which was unexplored at the time. These intractable shortcomings present a challenge to validating a majority of the proposed reaction steps by computational quantum chemistry in the atmospheric Hg redox cycle, including YHg$^{II}$O$^{\bullet}$ chemistry, through experimental means. As requested by theoretical chemists (Edirappulige et al., 2023) and modelers (Shah et al., 2021), better rate constants are needed for YHgO$^{\bullet}$ + CH$_4$ and YHgO$^{\bullet}$ + CO reactions, especially for Y = Br and OH, to better assess the atmospheric fate of YHgO$^{\bullet}$, i.e., whether YHgO$^{\bullet}$ will be mainly reduced or form closed-shell Hg$^{II}$ compounds under different atmospheric conditions. As a workaround in the absence of experimentally

determined rate constants, Khiri et al. (2020) proposed efforts to perform molecular dynamics simulations via computationally more sophisticated variational TST with multidimensional tunneling.

    Many of the proposed key gas-phase Hg species lack experimental characterization (such as spectral proofs). The main method for studying such gas-phase molecules has been spectroscopy after preparation by matrix isolation, which has thus far been used to study the products of photochemical reactions of excited Hg atoms, e.g., $O_3$ (Butler et al., 1979), $O_2$ (Andrews et al., 2023), $H_2$ (Wang and

Andrews, 2005b), $H_2O$ (Wang and Andrews, 2005a), $F_2$ (Wang et al., 2007), and $OF_2$ (Andrews et al., 2012), in a matrix host of solid Ar and Ne at a cold (typically 4–7 K) surface. **Section 5.1.4** has already described some of the isolated molecules of interest, namely, $Hg(OH)_2$ (Wang and Andrews, 2005a) and the fluorine analog of $YHg^{II}O^{\bullet}$ (Andrews et al., 2012). Other studies involve mercury halide molecules (Loewenschuss et al., 1969) and their adducts (Tevault et al., 1977). The reaction mechanism for the formation of $Hg(OH)_2$ tentatively involves insertion as a first step: $Hg(^3P) + O_2 + H_2 \rightarrow (OHgO)^* + H_2 \rightarrow HOHgOH$, where OHgO ($^3\Sigma_g^-$) is

implicitly indicated as a reactive intermediate (c.f., **Rxn G12**, although, unlike the analogous complexes for the other Group 12 metals, OZnO and OCdO, it has yet to be identified by IR spectra, Chertihin and Andrews, 1997). Apart from MS experiments of the laser desorption ionization and time-of-flight type with solid HgO as the source and detection of $(HgO)_x$ clusters in the gas phase (Jayasekharan and Sahoo, 2014), there is one early (Butler et al., 1979) and one recent (Andrews et al., 2023) matrix study of the products of the $Hg(^3P) + O_2$ system, where both $^{16}O_2$ and $^{18}O_2$ were used as reagents. The former experiments required co-deposition

of Hg with 0.5 to 5% $O_3$ in excess of Ar under deep UV photolysis for oxidation to occur, while the latter experiments used laser-ablated Hg atoms energetic enough to form oxygen atoms when deposited in a cryogenic matrix doped with 0.3% $^{16}O_2$ or $^{18}O_2$, which reacts upon annealing to form $O_3$ and a series of $HgO_x$ species (x = 1 to 3). The observed fundamental harmonic vibrational frequencies in different cryogenic matrices for the simple oxide Hg–O are in the range of 500-600 $cm^{-1}$, as predicted by high-level calculations (Shepler and Peterson, 2003; Peterson et al., 2007), indicating the presence of a weakly ionic molecule. This is also true

for $HgO_2$ and $HgO_3$, which have superoxide ($Hg^{\bullet+}O_2^{\bullet-}$) and ozonide ($Hg^{\bullet+}O_3^{\bullet-}$) characteristics, respectively (Andrews et al., 2023). Notably, the study did not isolate linear mercury dioxide, OHgO, and evidence for this species remains weak. Nevertheless, this species is included as a metastable adduct in the **Rxn G12** scheme, the key reaction for $Hg^0$ turnover in the stratosphere, whose complex potential energy surface forms the basis of ab initio kinetic calculations. In these calculations (Saiz-Lopez et al., 2022), the energy of the Hg–O bond is assumed to be 27.3 kJ $mol^{-1}$, which is significantly higher than the most recently published high-level

calculation values (Peterson et al., 2007; Cremer et al., 2008). Increased activity and innovation in advanced experimental studies characterizing key species and reactions are needed to verify models of atmospheric Hg chemistry that currently appear overly reliant on computational chemistry. The innovation could be, for example, finding a laboratory method to capture the temporal behavior of HgO (perhaps generated by the spin-allowed $Hg(^1S) + O(^1D)$ reaction or by reacting DMHg with $O(^3P)$) in the presence of gas-phase coreactants (with reference to **Rxn G73 & G74**), performing a detailed study of **Rxn G8 & 12**, or finding a synthetic route to matrix-

isolate species such as $BrHgO^{\bullet}$ from laser-ablated Hg atoms.

### 9.3 Model validation & observational gaps

    Current limitations and challenges in accurately measuring speciated atmospheric mercury (Gustin et al., 2024; **Section 3.1**) mean that the basis for verifying models in detail is insufficient, despite reliable measurements of $Hg^0$ in air and $Hg^{II}$ in wet deposition. Nevertheless, some models have been developed by including KCl-denuder-based $Hg^{II}$ measurements in the reference material (Shah

et al., 2021; Fu et al., 2024), which are known to suffer from low bias, and others (Saiz-Lopez et al., 2020; Saiz-Lopez et al., 2025), which seem to stick strictly to RM data for validation or include KCl-denuder-based $Hg^{II}$ measurements corrected according to Marusczak et al. (2017); consequently, the evaluation is qualitative (Shah et al., 2021) and consistently fails to simulate the magnitude of recurring episodes of highly oxidized mercury originating in the free troposphere (underestimation by up to several hundred percent, Elgiar et al., 2025; Gustin et al., 2023). In **Section 3.2**, we highlighted the discrepancies that exist in terms of the atmospheric

budget and the fluxes into and out of it. Particularly, new model results concerning the importance of the stratosphere are inconsistent with existing empirical data and require further elaboration, as does the stratospheric chemistry discussed above. Recent observations

have shown that there are abundant anthropogenic emissions of reactive halogens (e.g., $Br_2$ and $BrCl$) over continental, densely populated areas of South and East Asia that also have high Hg emissions. This is now beginning to be modeled as a key component of the regional atmospheric Hg redox cycle (Fu et al., 2024), but more field measurements are needed for confirmation. Artisanal and small-scale gold mining sector is currently the largest source of anthropogenic $Hg^0$ emissions to the atmosphere globally. However, emission estimates using bottom-up approaches, such as emissions inventories, is highly unconstrained (Cheng et al., 2023). Independent techniques, such as top-down constraints in the form of inverse modeling of atmospheric observations, are plausibly necessary to ensure the accuracy of these estimates (Sommar et al., 2020).

The difficulties of accurately determining the speciation of $Hg^{II}$ in atmospheric water through equilibrium modeling and thus identifying the pool of reducible complexes have been described (**Section 4.3**). Additionally, the potential for simulating the gas–particle distribution of atmospherically oxidized Hg has been explored (**Section 7.1.1**). Stable isotope data have been analyzed to constrain Hg redox chemistry in the atmosphere (Song et al., 2024; Zhen et al., 2024), but there are profound knowledge gaps that require state-of-the-art theoretical and experimental investigations. **Section 8.2** describes the isotopic composition of atmospheric samples, a pool generally consisting of filtered air divided into PBM and gaseous mercury ($\sim Hg^0$), and precipitation samples, including those of cloud and fog water. The isotope measurements on RM, which are now also performed, are briefly presented here (Fu et al., 2021). With a measuring line consisting of a CEM ($Hg^{II}(g)$), a filter ($Hg^{II}(p)$) and a trap consisting of halogen-impregnated activated carbon ($Hg^0$) in series, the analysis shows that the three groups are clearly isotopically separated from each other and that the resulting samples can thus provide further insights into atmospheric processes (X. Fu, pers. comm). The discovery, made over a decade ago, that atmospheric samples contain a significant level of the even-mass-number isotope MIF with seasonal and geospatial variations has been a source of both benefit and puzzlement for scientists (**Sections 8.2.3 & -4**). As even Hg MIF variation is limited to samples from a few localities thus far (compare **Figs. 13 & 14**), $\Delta^{200}Hg$ and $\Delta^{204}Hg$ in the environment are considered conservative tracers because of their generally narrow range, and values of $\Delta^{200}Hg$ and $\Delta^{204}Hg$ on the land surface and in water confine the relative contribution of $Hg^0$ to the $Hg^{II}$ exchange process with the atmosphere. Nevertheless, the underlying chemical processes that give rise to anomalous MIF and the atmospheric conditions that facilitate its occurrence remain to be elucidated in greater detail. In addition to laboratory-based investigations, future field experiments that report vertical profiles of isotopic $Hg^0$ and $Hg^{II}$ in the atmosphere may prove invaluable in further constraining the sources of even-MIF.

### Author contributions

J. O. S. prepared the manuscript with contributions from all co-authors.

### Competing interests

The authors declare that they have no conflicts of interest.

### Financial support

This work was supported by the National Natural Science Foundation of China (grant nos. 42150710535 and 42373068).

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

## Appendix

## List of symbols and acronyms

| Symbol | Quantity | Unit etc. |
|---|---|---|
| $\alpha$ | mass accommodation coefficient | dimensionless |
| $\alpha^{xxx/198}$ | fractionation factor of the Hg isotope with mass number xxx (relative to 198) | |
| $\alpha_{A-B}$ | $\equiv \alpha_{A-B}^{xxx}$ isotope fractionation factor between any two parts (chemicals, phases etc.) of a system | |
| $\beta^{xxx}$ | $\equiv \beta^{xxx/198}$, scaling factor for an isotope effect acting on the Hg isotope with mass number xxx. | |
| $\beta_{NFS}$ | $\equiv \beta_{NFS}^{xxx/198}$, scaling factor for NFS acting on the Hg isotope with mass number xxx. | |
| $\beta_{EIE-MDF}$ | $\equiv \beta_{EIE-MDE}^{xxx/198}$, scaling factor for equilibrium MDF acting on the Hg isotope with mass number xxx. | |
| $\beta_{KIE-MDF}$ | $\equiv \beta_{KIE-MDE}^{xxx/198}$, scaling factor for kinetic MDF acting on the Hg isotope with mass number xxx. | |
| $\beta_{qr}$ | cumulative stability coefficient for a complex of the type $\{HgL_q(OH)_r\}^{(2-r)+}$ | miscellaneous |
| $\gamma$ | uptake coefficient (probability) | dimensionless |
| $\gamma_{net}^0$ | initial net uptake coefficient (probability) | |
| $\gamma_{net}^\infty$ | steady-state net uptake coefficient (probability) | |
| $\delta^{xxx}Hg$ | $\delta$-notation for the $^{xxx}Hg$ isotopic composition in a sample relative to that of the NIST3133 standard. | dimensionless (‰) |
| $\delta^{202}Hg$ | $\delta$-notation to describe total mass-dependent fractionation | |
| $\delta_A, \delta_B$ | $\delta$-notation for an isotope in reservoir A and B, respectively | |
| $\delta_0$ | $\equiv (\delta^{xxx}Hg)_0$, initial $\delta^{xxx}Hg$ of a process | |
| $\Delta^{xxx}Hg$ | $\Delta$-notation, deviation from mass dependence ($\delta^{202}Hg$) of the Hg isotope with mass number xxx. | |
| $\varepsilon^{xxx}Hg$ | enrichment factor for a Hg isotope with mass number xxx. | |
| $\varepsilon_{A-B}^{xxx}$ | $\equiv \varepsilon^{xxx}Hg_{A-B}$, enrichment factor for an isotope between two reservoirs A and B. | |
| $E^{xxx}Hg$ | enrichment factor for a Hg isotope of mass number xxx for the mass-independent part of a process. | |
| $E_{A-B}^{xxx}$ | $\equiv E^{xxx}Hg_{A-B}$, MIF enrichment factor for an isotope between two reservoirs A and B | |
| $\Gamma_g$ | conductance (= 1/resistance) of diffusion of a gas to the surface in a resistance model of gas-droplet interaction. | dimensionless |
| $\Gamma_{rxn}$ | conductance (= 1/resistance) of reaction in the liquid phase in a resistance model of gas-droplet interaction. | |
| $\Gamma_{sol}$ | conductance (= 1/resistance) of solubility and diffusion in the liquid phase in a resistance model of gas-droplet interaction. | |
| $\lambda$ | wavelength of light | nm |
| $\sigma$ | absorption cross section | $cm^2\ molecule^{-1}$ |
| $\tau$ | atmospheric lifetime | year |
| $\tau_{troposphere}$ | overall tropospheric lifetime | |
| $\tau_{rxn}$ | lifetime in the troposphere due to net oxidation | |
| $\tau_{ocean}$ | lifetime in the troposphere due to oceanic net uptake | |
| $\tau_{land}$ | lifetime in the troposphere due to terrestrial net uptake | |
| $\tau_{wash}$ | lifetime in the troposphere due to tropospheric wash-out | |
| $\tau_{stratosphere}$ | lifetime in the troposphere due to net transfer to the tropopause/stratosphere | |
| $\bar{\upsilon}_X$ | mean thermal velocity of a gas X | $m\ s^{-1}$ |
| $\phi$ | photolysis quantum yield | dimensionless |
| $a$ | parameter for T dependence of $K_{gp}$ | dimensionless |
| $b$ | parameter for T dependence of $K_{gp}$ | |
| $c$ | $\equiv c^{Hg^0}, c_{index}^{Hg^0}$, gas-phase mass concentration normalized to standard temperature (0 °C) and pressure (101.325 kPa). | $ng\ m^{-3}$ |
| $D_g$ | gas-phase diffusion coefficient | $m^2\ s^{-1}$ |
| $D_l$ | liquid-phase diffusion coefficient | $m^2\ s^{-1}$ |
| $E^0$ | standard electrode potential | V |
| $E$ | electrode potential | |
| $E_a$ | activation energy | $J\ mol^{-1}$ |
| $f$ | fraction (isotope mixing) | dimensionless |

| | | |
|---|---|---|
| $f_R$ | fraction of reactant remaining | |
| F | Faraday constant | 96485 C mol$^{-1}$ |
| $F_c$ | form factor describing the transition region of a gas-phase reaction, typically ~0.7 | dimensionless |
| $F(\lambda)$ | photon flux | photons cm$^2$ s$^{-1}$ |
| $\Delta G_R$ | Gibbs free energy of reaction | |
| $\Delta G^0$ | standard Gibbs free energy | |
| $\Delta H_f$ | enthalpy of formation | J mol$^{-1}$ |
| $\Delta H_R$ | enthalpy of reaction | |
| $\Delta H_{abs}^0$ | standard enthalpy of adsorption | |
| $J_X$ | net flux of the gas X into the condensed phase | mol m$^{-2}$ s$^{-1}$ |
| k, k(T) | rate coefficient | miscellaneous |
| $k_0$ | $\equiv k_0^T$, low-pressure limit gas-phase rate coefficient | cm$^6$ molecule$^{-2}$ s$^{-1}$ |
| $k_\infty$ | $\equiv k_\infty^T$, high-pressure limit gas-phase rate coefficient | cm$^3$ molecule$^{-1}$ s$^{-1}$ |
| $k_f$ | forward rate coefficient | miscellaneous |
| $k_r$ | reverse rate coefficient | |
| $k_H^{cc}$ | Henry's law coefficient | dimensionless |
| $k_H^{cp}$ | Henry's law coefficient | mol L$^{-1}$ atm$^{-1}$ |
| $k_{ads}$ | adsorption rate coefficient | miscellaneous |
| $k_{des}$ | desorption rate coefficient | |
| $k_{het}$ | heterogeneous rate coefficient | |
| $k_{obs}$ | effective first-order rate constant | s$^{-1}$ |
| $k_{gas}$ | bimolecular rate coefficient of the gas phase part of a partially heterogeneous reaction | cm$^3$ molecule$^{-1}$ s$^{-1}$ |
| $k_{surf}$ | surface bimolecular rate coefficient (normalized by reactor surface-volume ratio) | cm$^4$ molecule$^{-1}$ s$^{-1}$ |
| $K_a$ | acid constant (HA $\rightleftarrows$ H$^+$ + A$^-$) | mol L$^{-1}$ |
| $K_{gp}$ | coefficient for absorptive partitioning of GOM onto existing aerosol | m$^3$ µg$^{-1}$ |
| $K_q$ | stepwise stability coefficient for a HgL$_{q-1}$ + L $\rightleftarrows$ HgL$_q$ type equilibrium | L mol$^{-1}$ |
| [M] | third body concentration | molecule cm$^{-3}$ |
| m | empirically fitted exponent | dimensionless |
| $m_{xxx}$ | mass of the isotope $^{xxx}$Hg | amu |
| n | number of electrons transferred in a red-ox reaction | mol |
| $\Delta N / \Delta logr$ | particle number concentration in the size range $\Delta logr$ (log-normal distributed polydisperse aerosol) | m$^{-3}$ |
| pK | $-logK$ | dimensionless |
| $pK_a$ | $-log(K_a)$ | |
| PM | particulate matter | µg m$^{-3}$ |
| r | radius (droplet or tubular reactor) | m |
| $\langle r_{xxx}^2 \rangle$ | mean-square nuclear charge radius | fm$^2$ |
| R | gas constant | 8.314 J mol$^{-1}$ K$^{-1}$ |
| $R^{xxx/198}$ | $\equiv R^{xxx}$, ratio of isotope xxx to isotope 198 | dimensionless |
| $\Delta S_{abs}^0$ | enthropy of adsorption | J mol$^{-1}$ K$^{-1}$ |
| T | absolute temperature | K |
| V | volume | m$^3$ |
| $[X]_{g,\infty}$ | background (bulk) gas-phase concentration of species X | mol m$^{-3}$ |
| $[X]_{surf}$ | $[X]_g$ at the surface of a droplet | |
| $[X]_g$ | gas-phase concentration of species X | |
| $[X]_p$ | particle-phase concentration of species X | |

| Acronym | Plain text |
|---|---|
| AAS | atomic absorption spectroscopy |
| AMDE | atmospheric mercury depletion event |
| AOM | atmospheric organic matter |
| AQ | anthraquinone |
| CFPP | coal-fired power plant |
| CI-MS | chemical ionization mass spectrometry |
| CV-AFS | cold vapor atomic fluorescence spectroscopy |

| | |
|---|---|
| DFC | dynamic flux chamber |
| DMHg | dimethylmercury |
| DOM | dissolved organic matter |
| EIE | equilibrium isotope effect |
| EIE-MDF | equilibrium MDF |
| even-MIF | mass-independent fractionation of even Hg isotope ($^{200}$Hg, $^{204}$Hg) |
| FEP | polymeric fluorinated ethylene propylene |
| FF | fast flow reactor, chemical reactor designed for rapid mixing and reaction of gases or liquids |
| FF-ID-CI-MS | fast flow ion-drift chemical ionization mass spectrometry |
| FT-IR | Fourier Transform Infrared |
| GOM | $\equiv$ Hg$^{II}$(g), gaseous oxidized mercury |
| HMDE | hanging mercury drop electrode |
| HFC | hyperfine coupling |
| KIE | kinetic isotope effect |
| KIE-MDF | kinetic MDF |
| LIDAR | light detection and ranging |
| LMCT | ligand to metal charge transfer |
| LMWO | low molecular weight organics |
| LOD | limit of detection |
| MDF | mass-dependent fractionation |
| MgIE | magnetic isotope effect (acting on $^{199}$Hg, $^{201}$Hg) |
| MIF | mass-independent fractionation |
| MMHg$^+$ | species containing a methyl mercuric cation and an unspecified counteranion. |
| m/m | mass-to-mass ratio |
| M/M | mol-to-mol (stoichiometric) ratio |
| MRB | metal-reducing bacteria |
| MS | mass spectrometry |
| NFS | nuclear field shift, synonym for NVE |
| NVE | nuclear volume effect, synonym for NFS |
| odd-MIF | mass-independent fractionation of odd Hg isotope ($^{199}$Hg, $^{201}$Hg) |
| PBM | $\equiv$ Hg$^{II}$(p), particle-bound mercury |
| PLP-LIF | pulsed laser photolysis-laser induced fluorescence |
| PM | particulate matter, synonym for TSP |
| PM$_{2.5}$ | particulate matter $\leq 2.5$ μm |
| PM$_{10}$ | particulate matter $\leq 10$ μm |
| POA | primary organic aerosol |
| PTR-MS | proton transfer reaction mass spectrometry |
| RH | relative humidity (% of absolute humidity) |
| RKKM | Rice-Ramsperger-Kassel-Markus (theory) |
| RM | reactive mercury (GOM + PBM) |
| RR | determination of rate constant by a relative rate method, opposite to an absolute determination |
| Rxn | abbreviation for reaction, representing a chemical reaction |
| SOA | secondary organic aerosols |
| STP | standard temperature and pressure (one atmosphere, 101.325 kPa and 0 °C) |
| TAM | total atmospheric mercury (Hg$^0$ + GOM + PBM) |
| TGM | total gaseous mercury (Hg$^0$ + GOM), in practice not quantitative for GOM |
| TSP | total suspended particles, all particle sizes suspended in the air |
| TS | transition state |
| TST | transition state theory |
| UV-A | ultraviolet A, 315 – 400 nm |
| UV-B | ultraviolet B, 280 – 315 nm |
| UV-C | ultraviolet C, 100 – 280 nm, also called deep UV |
| UV–VIS | ultraviolet (A + B) + visible light (400 – 700 nm) |

3835