# Peer review of "Atmospheric Mercury: Recent advances in theoretical, computational, experimental, observational and isotopic understanding to decipher its redox transformations in the upper and lower atmosphere and interactions with Earth surface reservoirs"

_EGUsphere, 2024_

## Author Comment (AC1)

**Final response**

**Reply on RC1 (F. Slemr)**

First, we would like to express our sincere gratitude to Dr. Franz Slemr, one of the true doyens of the field, for taking the time to provide a thoughtful and constructive feedback on the manuscript.

**Responses to general comments**

1. "Because of the length of the review, a table of contents at the beginning would improve the orientation for readers looking for a specific topic".

   The only possible obstacle is getting the editor to consent to include a TOC.

2. "The ever-increasing complexity of mercury behaviour in the atmosphere seems at times to obscure the fact that it is a subject, as other trace gases, to general constraints imposed by atmospheric circulation...."

   To address this criticism, we have decided to rewrite, modify, and rearrange large portions of Section 3.2 (until L206 in the original manuscript) as follows:

[revised manuscript text omitted]
…". No reference is given for this statement, the references at the end of the sentence relate to exchange flux between stratosphere and troposphere. This statement cannot be true."

    We have not previously reflected further on the referenced figure of 20% derived from Shah et al. (2021). It is noteworthy that a comparable figure of 17 $\pm$ 4% is given in the study by Saiz-Lopes et al. (2025). We acknowledge the reviewer's skepticism and have made a substantial revision to address this concern, as outlined in **L183** through **L191** above.

4.  "In the paragraphs, L193 - 215 and 216 - 235, different lifetimes are presented and compared: the tropospheric, the chemical, the stratospheric, the lower stratospheric against surface deposition, the mid- and upper-stratospheric, and the mean atmospheric ones. This discussion needs a common denominator…"

    We agree with the reviewer on this comment and have made the following adjustments:

    The term atmospheric lifetime is presented (**L173 – L175**), the conditions to justify a global tropospheric lifetime (**L197 – L199**), the equation with explanatory notes (**L200 – L205**), the importance of the chemical lifetime (**L206-L210**), the final presentation of the Hg0 lifetime (**L230 – L235**), something about the effective lifetime for $Hg^I$ and $Hg^{II}$ (**L256 – L258**).

5.  "The definition inconsistencies in these paragraphs can be illustrated e.g. by the statement (lines 203-204) that "Based on correlations of $Hg^0$ with $N_2O$ in the stratosphere within 4 km above the thermal tropopause, Slemr et al. (2018) provided a lifetime estimate of 74 +/- 27 yr while Lyman and Jaffe (2012) inferred a relatively short lifetime for $Hg^0$ in intercepted descending air with stratospheric origin". Stratospheric lifetime estimated by Slemr et al. (2018) has been derived from hundreds of CARIBIC measurements and is consistent with the definition of Seinfeld and Pandis (1998). "A relatively short $Hg^0$ lifetime in the stratosphere of <1yr" mentioned by Lyman and Jaffe (2012) seems to be the local chemical one which cannot be compared to the lifetime according to the definition of Seinfeld and Pandis (1998)."

    We recognize that our review must be critical enough not to allow these references to stand next to each other without comment, and that the range of measurement data between the references is abysmal. We break out L200 - L206 and move parts of this paragraph to **L188 – L190** and to Section 5.1.5, which deals with the lower stratosphere. We now suggest reading from the middle of L713 to the end of the section:

713  …(**Fig. 8a,b**). The prediction of these model calculations, that $Hg^0$ converts to long-lived (photostable) oxidized

714      forms and thus leads to a higher RM/TAM ratio, is supported by hundreds of profile measurements made with an
715      Airbus 340-600 passenger aircraft in intercontinental traffic as an upper troposphere-lowermost stratosphere
716      observatory (Slemr et al., 2018). In addition to frequently observed higher RM/Hg$^0$ ratios, a steep decrease of the
717      Hg$^0$ mixing ratio occurs when crossing the tropopause. In the stratosphere, this ratio drops to 0.25 - 0.7 ng m$^{-3}$
718      (STP), measured up to an altitude of 4 km (Slemr et al., 2018). The results of both studies above show a more than
719      tenfold increase in Hg$^0$ lifetime in the lower stratosphere compared to the troposphere. The chemical lifetime of
720      Hg$^0$ increases and approaches 10 years the higher you go in the lower stratosphere bounded by the ozone layer.
721      (Saiz-Lopes et al. 2025).

**Responses to minor comments**

Line 58: Perhaps a reference Jiskra et al. (2018) should be added here.

This reference has been added.

Line 65: What do you mean with "reductionist"?

Try to give an explanation based on breaking the problem down into smaller and smaller units. We have chosen to delete the word from the sentence, as it is still clear that understanding is sought down to the molecular level.

Line 71: "compilation" is perhaps a better word than "tabulation".

Rephrasing applied.

Line 102: Why "filtered"? The AAS method will measure elemental mercury even with some aerosols as long as there is enough light coming through. LIDAR technique shows that aerosol poses no problem. In fact, back scattering on aerosol is the basis of the LIDAR techniques.

Correct. We delete the word.

Line 113: Awkward wording – "Since gold does not trap only Hg$^0$ but…" would be perhaps better.

We have adapted it.

Line 121: "…the risk of artifact formation of Hg$^{II}$ by co-sampling GOM with PBM…" – I think that no new Hg$^{II}$ is being formed but Hg$^{II}$ from GOM and PBM is being co-measured, later called RM.

We have deleted "artifact formation of Hg$^{II}$ by", so the sentence now reads:

"The automated KCl denuder method, with its variable efficiency, can thus lead to serious underestimation of GOM, while the refluxing mist chamber method, which is an alternative, carries the risk of co-sampling GOM with PBM (Gustin et al., 2021)".

Line 153-154: Awkward wording: "… is not recommended because it cannot be applied in multi-stage atmospheric pressure systems…" perhaps better.

We have adapted it.

Line 155: "…of ambient GOM species…" may be perhaps better.

We have adapted it.

Line 290: "Hinshelwood" instead of "Hinselwood".

We have adapted it.

Table 1, caption: "are" instead "is"

We have adapted it.

Table 3: In reactions G24 and G25 appears BrHg$^{II}$O* - where does it come from? Is the oxidation stage of Hg

consistently described? Please check the consistency of all chemical formulas, in the tables and in the text.

We have removed the oxidation numbers completely from Table 3 (i.e. for **G24 & G25**), but in our opinion they are still warranted occasionally in the text for accessibility to a less advanced reader.

Line 453: "…determined by Donohue…"

We have deleted "absolutely".

Line 466: Please define what RR studies are? If that means "reaction rate" it could be replaced "kinetics studies". Or omit it altogether – the type of studies is given by the context.

The acronym RR has been introduced in L452.

Line 538: decreases with decreasing temperature?

It should read "decreasing with increasing temperature".

Line 542: "and" instead "och"

Mistake corrected.

Line 574: "they" – Dibble et al or Castro Pelaez et al?

"they" has been substituted by "Castro Pelaez et al. (2022)".

Lines 714-715: But above the $O_3$ maximum in the stratosphere the reactions of excited Hg atom become important? See section 5.1.6. Perhaps the Section 5.1.5 should be merged with 5.1.6 to avoid misunderstanding.

We made an error in using "above" instead of "below" in reference to the ozone layer. Paragraphs L713 - L715 have been completely rewritten. The new wording of the paragraph can be found above under the heading **Responses to specific comments**. In addition, the titles of Sections 5.1.5 and 5.1.6 have been changed to "**Chemical transformation of Hg in the lower stratosphere**" and "**Chemical transformation of Hg in the upper stratosphere**", respectively.

Line 765: "Hg($^3$P)" – Hg($^3$P$_1$) or Hg($^3$P$_0$) or both?

In order not to exclude the one from the other, we have added "states" after Hg($^3$P).

Line 983: "bis-sulphite complex is thermally stable" perhaps better.

"is not thermally unstable" has been changed to "is thermally stable".

Sections 6 and 7: Interaction of $HgCl_2$ with $H_2SO_4$ droplets would be interesting because they constitute the major aerosol particles in the stratospheric Junge layer. Are any data available? Any comment on this? Here or in the Section 9?

In reference to the response to RC2, the chapter "Gas to nucleation" has been renamed "Can mercury species nucleate in the atmosphere?" and its introduction has undergone complete revision. On line 694, the following statement appears: "Interestingly, Hg$^{II}$ is empirically correlated with bromine and iodine in these particles of organic-sulfate type and has the highest relative concentrations in the stratosphere near the tropopause (Murphy et al., 2006)." Therefore, there is empirical evidence that Hg$^{II}$ is trapped in heavier halogen-doped organics-sulfuric acid/sulfate-type particles in the stratosphere, but it is "only rarely observed in the relatively pure sulfuric acid particles characteristic of the main stratospheric aerosol layer."

Line 1215: "Brunauer-Emmett-Teller" instead of "BET" perhaps better for non-specialists.

"Brunauer-Emmett-Teller" has been adopted.

Line 1239: "aerosolized"?

Changed to "aerosols"

Lines 2110 - 2111: Reliable measurement of $Hg^0$ in wet deposition? Perhaps "…despite generally reliable measurements of $Hg^0$ in air and $Hg^{II}$ in wet deposition." would be less ambiguous. These measurements may not be sufficient for verifying model studies in detail but they provide constraints with which the model results have to comply.

> The sentence now reads: "… mean that the basis for verifying *models in detail* is insufficient, despite reliable measurements of $Hg^0$ *in air* and $Hg^{II}$ *in wet deposition*".

References: The titles of the papers are sometimes written with capital letters, sometimes without. Please homogenize.

> References have been homogenized, meaning that words are not now capitalized.

**References**

[revised manuscript text omitted]

**Reply to RC2 (anonymous)**

**Responses to general comments**

"It will be beneficial if the authors shape this manuscript more as a critical review".

> The point was made by both reviewers to be more critical in selecting the references that can be considered well-founded and to provide a base line for comparison, so to speak, commented or uncommented. In the response to RC1 under point 3, we have removed one of the references. Here, we have deleted L461 - L464 and removed the corresponding reference (L2268 - L2270). To be critical, it is perhaps questionable to include relative rate experiments, especially those performed on dirty systems, but for the OH reaction (**Rxn G3**), which has only been studied in absolute terms, leading to an upper limit on the rate constant, it seems justified. The two RR experiments are well re-analyzed and converge to the rate coefficient reported in Table 3, which comes from high-level quantum chemical calculations. Otherwise, Table 3 contains only experimental data determined in an absolute way (exceptions are **Rxn G75 & G76**).

"Hence, it would be beneficial for the reader to know which work provides more accurate data by design".

> To remedy the problem, we have revised section 4.7 L392: "Table 3 summarizes the gas-phase reactions with the rate coefficients *considered most accurate* and the corresponding reaction enthalpies".

"There are a number of other places where the authors might have indicated the confidence degree associated with different presented data. I understand that this is not always an easy task."

> We have reviewed the kinetic data presented and have double-checked that the data presented in Tables 3 and 4 are accompanied by confidence intervals, where these are available in the reference literature. Altogether, about a dozen adjustments were made.

"Regarding the increased stability of HgO at lower temperatures in the stratosphere. I suggest adding "at lower temperatures and pressures", as this reaction is collision activated. I should note that according to my estimate, the lifetime is still about 1 ms at 250 K and 0.1 atm, which is very short, making this species irrelevant."

> We follow the reviewer's suggestion and make changes starting in line 640:

> "Taken together, this information suggests that gas phase HgO in the troposphere is highly unstable. Although the decay slows down at lower temperatures and pressures as the reaction is collisionally activated, the thermal lifetime is still only about 1 ms at 250 K and 0.1 atm".

"The Gas to nucleation" subtitle is awkward and the message of this subsection is unclear. Do you imply that mercury oxidation products have a sufficiently low vapor pressure and sufficiently high concentration to nucleate new particles or contribute to their growth in the atmosphere? I highly doubt this proposition."

> We thank the reviewer for this constructive remark. We agree and have made a throughout revision as follows:

> **Can mercury species nucleate in the atmosphere?**

> While $Hg^0$ vapor has been observed to nucleate homogeneously in laboratory experiments conducted under high pressures (Martens et al., 1987), neither $Hg^0$ atoms nor GOM species, which are molecular rather than ionic entities, have a vapor pressure that is sufficiently low and a concentration that is sufficiently high in the atmosphere to nucleate new particles by simple condensation (Murphy et al., 1998). However, the concerted action with a foreign gas phase precursor (perhaps amines, highly oxygenated organics, sulfuric, nitric, and iodic acid, etc., as candidates, (Lehtipalo et al., 2025; He et al., 2021)) or, classically, heterogeneous condensation on pre-existing nuclei of subcritical or critical size may result in the transfer of GOM species to aerosols (Ariya et al., 2015). Measurements of individual aerosol particles have shown that a significant portion of the aerosols

| 694 | present in the lowest kilometers of the stratosphere contain small yet measurable amounts of $Hg^{II}$. Interestingly, |
|---|---|
| 695 | $Hg^{II}$ is empirically correlated with bromine and iodine in these particles of organic-sulfate type and has the |
| 696 | highest relative concentrations in the stratosphere near the tropopause, but that $Hg^{II}$ is only rarely observed in the |
| 697 | relatively pure sulfuric acid particles characteristic of the main stratospheric aerosol layer (Junge Layer) (Murphy |
| 698 | et al., 2006). While bromine and iodine aerosols are also observed throughout the troposphere, no Hg can be |
| 699 | detected in these, indicating that the $Hg^{II}$ products can evaporate rapidly into GOM species (Murphy et al., 2014). |
| 700 | Both Br and I, with the oceans as the primary sources, are injected into the stratosphere, where they account for |
| 701 | most of the ozone depletion caused by halogens (Koenig et al., 2020). It is challenging to determine whether |
| 702 | there is a causal mechanistic relationship and, if so, what it is that can explain the observed correlation between |
| 703 | aerosol Hg, Br, and I. Nevertheless, there is a plethora of clues that can be utilized to assemble a coherent |
| 704 | narrative. Firstly, the combination of Br$^{•}$ (**Rxn G14a**) and $O_3$ (**Rxn G22**) constitutes a significant oxidation |
| 705 | pathway for $Hg^0$ to $Hg^{II}$. However, as mentioned above, there is no firm evidence that this reaction pathway is |
| 706 | relevant when I$^{•}$ is a substitute for Br$^{•}$. Secondly, the gas phase system I$^{•}$ + $O_3$ + $H_2O$ has been identified as a |
| 707 | substantial precursor of particle nucleation (as iodine oxoacids) and growth that possess a considerable |
| 708 | significance within marine (Sipilä et al., 2016) and stratospheric (Koenig et al., 2020) environments. Thirdly, |
| 709 | condensed phase Br$^{-}$ and I$^{-}$ act as robust complexing ligands (Table 1) for GOM partitioned to the aerosol, |
| 710 | thereby impeding its re-cycling back to the gas phase. Presumably, the fundamentals are similar for a particle |
| 711 | formation event observed in the context of the polar spring partial AMDE in the East Antarctic pack ice by |
| 712 | Humphries et al. (2015), where the formation of 3 nm particles lags behind the phase of gaseous $Hg^0$ loss in the |
| 713 | air mass. |

**L712** (old L690) → old L707 (end): uncut text

Chapter 8 on isotope effects makes this review stand apart from other atmospheric mercury reviews. It has a very large amount of important information, but I found this chapter difficult to read…

We have *prepared* an illustration of the biogeochemical cycle of isotopic mercury focusing on the atmosphere, which could possibly serve as a reference point for the reader and make the reading less monotonous.

[Figure]

At a minimum, I urge the authors to add a table that summarizes the different parameters (both deltas, beta, alpha, etc.) along with their definitions, meaning, and usage.

*We have prepared a list of the symbols and acronyms used in this paper along with their explanations (See Appendix below).*

Perhaps, it would be beneficial for one of the co-authors with an intermediate knowledge of isotope fractionation (who is not an expert), to read this chapter and edit it a bit to make it more digestible for the readers who are less familiar with the isotope field, e.g., by making sure that every time a value of a parameter is mentioned, the meaning of this value is fully interpreted.

*We revised the chapter to improve its readability. We corrected sloppy mistakes and ensured that terms were explained the first time they appeared in the text. We also included these terms in the separate list of symbols and acronyms (See Appendix below).*

*1007 & -8, 1011 & -12, 1056, 1099, 1765, and Caption **Table 5**: "NOM" replaced by "DOM"*

*1351: Rewritten: "Natural processes, including redox reactions, complexation, sorption, precipitation, dissolution, evaporation, diffusion, and biological processes, can alter the isotopic composition, i.e., causing stable isotope fractionation".*

*1361: New symbols in equation 12*

$$\delta^{xxx}\mathrm{Hg} = 1000 \cdot \left[ \left( {}^{xxx}\mathrm{Hg}/{}^{198}\mathrm{Hg} \right)_{sample} / \left( {}^{xxx}\mathrm{Hg}/{}^{198}\mathrm{Hg} \right)_{NIST3133} -1 \right]$$

*1362: Re-shuffled: "...per mill (‰). $\delta^{202}\mathrm{Hg}$ expresses the total mass dependent fraction (TMDF, containing contributions from conventional mass-dependent fractionation, hereafter, MDF and nuclear field shift, NFS see **Section 8. 1**), while the isotope anomalies caused by mass-independent fractionation, MIF are expressed by capital delta, $\Delta$ is defined as the difference between the measured $\delta$-value and that predicted from the measured $\delta^{202}\mathrm{Hg}$ value and the scale factor for kinetic MDF ($\beta^{xxx}_{KIE\text{-}MDF}$, see **Section 8. 1**) and is approximated for $\delta$-values < 10‰ according to:*

$$\Delta^{xxx}\mathrm{Hg} = \delta^{xxx}\mathrm{Hg} \ - \beta^{xxx}_{KIE\text{-}MDF} \cdot \delta^{202}\mathrm{Hg} \qquad (13)$$

*which is expressed numerically for each relevant Hg isotope:*

*Inserted L1442-L1443. The fractionation between two compounds A and B (assume that A is a product of a reaction and B is the remaining reactant.) is expressed with the fractionation factor, $\alpha$, which is defined as the ratio of the isotope ratios in the compounds:*

$$\alpha^{xxx}_{A\text{-}B} = R_A^{xxx/198}/R_B^{xxx/198} = \frac{1000 + (\delta^{xxx}\mathrm{Hg})_A}{1000 + (\delta^{xxx}\mathrm{Hg})_B} \qquad (14)$$

*The last term is obtained by substituting Eq. 12 into the first term of Eq. 14.*

*Inserted L1365 followed by old Eq. 14 (now Eq. 15) and L1366 (… than 10‰).*

*Substitute Eq. 13 into Eq. 15 and obtain:*

$$\varepsilon^{xxx}_{A\text{-}B} \cong \{(\Delta^{xxx}\mathrm{Hg})_A - (\Delta^{xxx}\mathrm{Hg})_B\} + \beta^{xxx}_{KIE\text{-}MDF} \cdot [(\delta^{xxx}\mathrm{Hg})_A - (\delta^{xxx}\mathrm{Hg})_B] \qquad (16)$$

*Equation 16 expresses total fractionation during the process A → B, with the first term representing the MIF enrichment factor and the second term representing the total mass-dependent enrichment factor. Thus, the enrichment factor for MIF is written as a capital epsilon:*

$$E^{xxx}_{A\text{-}B} = \{(\Delta^{xxx}\mathrm{Hg})_A - (\Delta^{xxx}\mathrm{Hg})_B\} = \varepsilon^{xxx}_{A\text{-}B} - \beta^{xxx}_{KIE\text{-}MDF} \cdot \varepsilon^{202}_{A\text{-}B} \qquad (17)$$

Inserted L1366 "Many kinetics..." Numbering on mathematical equations updated (15') → (18) etc.

1373: Inserted "... normal kinetic isotope effect, **KIE**)..."

[revised manuscript text omitted]

1610: Inserted: "... consistent with **precipitation** observations in the..."

1641 Caption Fig. 12: "MDF" changed to "TMDF"

Caption Table 7 Inserted: "... factors ($\varepsilon^{202}Hg$, $E^{199}Hg$) determined..."

**Table 7:** reads "detected" should read "0.06‰"

1687: Corrected: **Rxn G72**

1699: "dissolved organic matter" replaced by "DOM".

1703: Deleted: conventional

1732: "The figure" changed to "Fig. 16".

1738: Corrected: Stowe and Knight **Jr.**

1755: Revised: "Another example of the impact of pH/complexation on the evolution of MgIE can be seen in the UVC photodegradation of MMHg$^+$ in acidic and alkaline (adjusted with NH$_3$) solutions. In the former, (+)MgIE is significant, but in the latter, it is very limited."

1759: Inserted: "As **seen** in..."

1762: Rewritten: "To better interpret odd-MIF signatures and systematically elaborate the roles of reaction parameters (pH, presence of O$_2$, light wavelength, etc., Rose et al., 2015) in excited state kinetic isotope effects, experimental research is needed."

1766: Rewritten: "Zhang and Hintelmann (2009) observed an E199Hg optimum ($\gtrsim$5%) in anoxic photo-experiments with the DOM fraction from Dorset Lake, Ontario. This optimum is associated with a ligation mode in which all S-bonding functional groups are saturated by Hg$^{2+}$ cations, increasing the proportion of Hg-O bonds and the ratio of bright triplets to bright singlets, thus making the MgIE increasingly positive."

**Table 8:** All occurrences of "NVE" have been replaced by "NFS". The entry "0.24 ± 0.01" under the reference to Schwab et al. (2023) is in the wrong column and should be moved one column to the left.

Footnote 40: Mistake corrected

1807: New entry: "**8.4.2 Oxidation**

To the extent that isotopic effects in the aqueous phase Hg$^0$ oxidation have been studied in the laboratory, it has been observed that oxidized Hg becomes isotopically heavier than the reactant. The observed fractionation does not conform to the Rayleigh model, but it is consistent with EIE in a closed system. Consequently, the isotope ratio of the product(s) linearly approaches that of the reactant at the beginning of the reaction. An example of atmospherically relevant oxidation is the rapid reaction with $^\bullet$OH (**Rxn W2**, generated by photolysis of NO$_3^-$) with $\varepsilon^{202}$Hg = 1.20 ± 0.14 ‰ (Stathopoulos, 2014). Experiments with thiol-substituted carboxylic acids in the dark produced similar fractionation results (see Table 8, **Rxn W7**). Additionally, the NFS produced a small odd-MIF signal that consistently acts in the opposite direction of the mass-dependent fractionation (Zheng et al., 2019). The reason for observing EIE despite the continuous oxidation of Hg$^0$ without any indication of reversibility in the form of back reactions has been attributed to a rapid exchange of Hg isotopes between the remaining Hg$^0$ and the formed Hg$^{II}$ complexes (Wang et al., 2020). There is currently debate surrounding the mechanism by which this exchange occurs (Wang et al., 2020; Zheng et al., 2019; Wang et al., 2021). In the presence of humic acid, the oxidation of dissolved Hg$^0$ exhibits two kinetic regimes where the EIE is not fully established in the initial regime (Zheng et al., 2019). KIE-MDF during dark reduction in the presence of DOM and EIE-MDF during dark oxidation caused by humic acid give rise to fractionation in the same direction and magnitude, so it can be difficult to unmask the controlling redox process from isotopic measurements."

1810, 1827 & 1871: "NVE" replaced by "NFS".

1824: "MDF" changed to "TMDF"

1833: Revised into "... this time, positive..."

1839: Correction: "Section **3.2**"

1847: Clarifying addition: "In the following, we express absolute deposition with negative values and vice versa for emission throughout."

1850: Inserted: "Isotope-based modeling by binary (**example in Eq. 25**) and... ", "MDF" changed to "TMDF"

1862: Inserted: "The isotope mixing formula is used to determine the proportions of different isotope sources in a mixture, which in its simplest form is:

$$(\delta^{xxx}Hg)_{mix} = f_1 \cdot (\delta^{xxx}Hg)_1 + f_2 \cdot (\delta^{xxx}Hg)_2 \qquad (25)$$
$$f_1 + f_2 = 1$$

1885 & 1903: "MDF" changed to "TMDF"

1929: (aq) deleted.

1939: Correction: "Section **3.2**"

1947: "MDF" changed to "TMDF"

2016: "Regression of the Alaskan DFC experiment $\Delta^{199}Hg$ plotted against $\delta^{202}Hg$ data ($-3.44 \pm 0.70$)." deleted.

2037: "(DGM)" deleted.

2038: "MDF" changed to "TMDF"

2047: Correction: " ... photodegradation **have been suggested** (Chen et al., 2016)"

Chapter 9 on future perspectives is a hit-and-miss. I suggest that the authors rethink the structure and contents of this chapter. It must summarize the achievements, identify the remaining challenges and gaps, propose future work, and when possible, suggest ways of addressing the challenges. Having a concluding sentence or brief paragraph would be nice, too.

We've revised and expanded the chapter, and we hope that we've been able to address the criticisms that were raised.

[revised manuscript text omitted]

2062     including in the reference material KCl-denuder based $Hg^{II}$ measurements (Shah et al., 2021; Fu et al.,
2063     2024), which are known to suffer from low bias, and others (Saiz-Lopez et al., 2020; Saiz-Lopez et al.,
2064     2025) that seem to stick strictly to RM data for validation or include KCl-denuder based $Hg^{II}$ measurements
2065     corrected according to (Maruszczak et al., 2017), which has the consequence that the evaluation is
2066     qualitative (Shah et al., 2021), consistently fails to simulate the magnitude of recurring episodes of highly
2067     oxidized mercury originating in the free troposphere (underestimation by up to several hundred percent,
2068     (Elgiar et al., 2025; Gustin et al., 2023)). In **Section 3.2**, we have highlighted the discrepancies that exist in
2069     the view of the atmospheric budget and the fluxes into and out of it. In particular, new model results on the
2070     importance of the stratosphere are inconsistent with existing empirical data and require further elaboration,
2071     as does the stratospheric chemistry discussed above. Recent observations have shown that there are
2072     abundant anthropogenic emissions of reactive halogens (e.g., $Br_2$ and BrCl) over continental, densely
2073     populated areas of South and East Asia that also have high Hg emissions. This is now beginning to be
2074     modeled as a key component of the regional atmospheric Hg redox cycle (Fu et al., 2024), but more field
2075     measurements are needed for confirmation.

2076     The difficulties of accurately determining the speciation of $Hg^{II}$ in atmospheric water through equilibrium
2077     modelling, and thus identifying the pool of reducible complexes, have been elaborated (**Section 4.3**).
2078     Additionally, the potential for simulating the gas-particle distribution of atmospherically oxidized Hg has
2079     been explored (**Section 7.1.1**). Stable isotope data have been analyzed to constrain the Hg redox chemistry
2080     in the atmosphere (Song et al., 2024; Zhen et al., 2024), but there are profound knowledge gaps that require
2081     state-of-the-art theoretical and experimental investigations. **Section 8.2** describes the isotopic composition
2082     of atmospheric samples, a pool generally consisting of filtered air divided into PBM and gaseous mercury
2083     ($\sim Hg^0$), and precipitation samples, including those of cloud and fog water. The isotope measurements on
2084     RM, which are now also performed, are briefly presented here (Fu et al., 2021). With a measuring line
2085     consisting of a CEM ($Hg^{II}(g)$), a filter ($Hg^{II}(p)$) and a trap consisting of halogen-impregnated activated
2086     carbon ($Hg^0$) in series, the analysis shows that the three groups are clearly isotopically separated from each
2087     other and that the resulting samples can thus provide further insights into atmospheric processes (X. Fu,
2088     pers. comm). The discovery, made over a decade ago, that atmospheric samples contain a significant level
2089     of the even-mass-number isotope MIF with seasonal and geospatial variations has been a source of both
2090     benefit and puzzlement for scientists (**Section 8.2.3 & -4**). As the even Hg MIF variation is limited to
2091     samples from a few localities so far (compare **Figs. 13 & 14**), $\Delta^{200}Hg$ & $\Delta^{204}Hg$ in the environment is
2092     considered a conservative tracer due to its generally narrow range, and values of $\Delta^{200}Hg$ & $\Delta^{204}Hg$ on the
2093     land surface and in water confine the relative contribution of $Hg^0$ to $Hg^{II}$ exchange process with the
2094     atmosphere. Nevertheless, the underlying chemical processes that give rise to the anomalous MIF and the
2095     atmospheric conditions that facilitate its occurrence remain to be elucidated in greater detail. In addition to
2096     laboratory-based investigations, future field experiments that report on the vertical profiles of isotopic $Hg^0$
2097     and $Hg^{II}$ in the atmosphere may prove invaluable in further constraining the sources of even-MIF.

2098

**Responses to minor comments**

L24: Consider replacing "impact" with "threat"

>Replaced.

L47: Consider replacing "However, it" with "Although this route was discarded, ozone"

>Replaced.

L48: Replace "more unstable" with "less stable"

>Replaced.

L59-63: Sort out by year, and possibly cluster by major focus of each review.

>*General* (*biogeochemical cycle* (Lindqvist and Rodhe, 1985; Lindqvist et al., 1991; Schroeder and Munthe, 1998; Selin, 2009; Lyman et al., 2020), *observations* (Slemr et al., 2003; Sprovieri et al., 2010; Dommergue et al., 2010; Fu et al., 2015; Steffen et al., 2015; Mao et al., 2016; Zhang et al., 2019; Custódio et al., 2022; Bencardino et al., 2024), *isotopic observational data* (Kwon et al., 2020; Liu et al., 2024), *atmospheric measurement techniques* (Pandey et al., 2011; Huang et al., 2014; Gustin et al., 2015; Davis and Lu, 2024; Gustin et al., 2024), *emissions* (*anthropogenic*) (Carpi, 1997; Zhang et al., 2016; Cheng et al., 2023), *natural volcanism* (Edwards et al., 2021), *physical removal and air-surface exchange* (Zhang et al., 2009; Sommar et al., 2013; Zhu et al., 2016; Agnan et al., 2016; Cooke et al., 2020; Sommar et al., 2020; Zhou et al., 2021; Liu et al., 2024) with emphasis on *global change* (Obrist et al., 2018; Sonke et al., 2023), *polar atmospheric surface layer mercury depletion events* (Steffen et al., 2008), *chemical conversion in the atmosphere* (Schroeder et al., 1991; Lin and Pehkonen, 1999; Lin et al., 2011; Si and Ariya, 2018), *aqueous homogeneous and heterogeneous photoredox chemistry* (Zhang, 2006; Si et al., 2022), *multi-phase atmospheric chemistry* (Ariya et al., 2015), *assessment of critical atmospheric chemical processes using state-of-the-art experimental and computational chemistry methods* (Ariya and Peterson, 2005; Ariya et al., 2008; Hynes et al., 2009), *receptor-* (Cheng et al., 2015) *and global models* (Lin et al., 2006; Lin et al., 2007; Subir et al., 2011, 2012; Amos et al., 2015; Travnikov et al., 2017).

L87: "as the electron approaches" – the electron on which orbit?

>We have rewritten the sentence, which now follows as:

>The electronic configuration of the mercury atom has filled f- and d-orbitals with a high density of 6s-valence electrons near the nucleus ([Xe]4f$^{14}$5d$^{10}$6s$^2$), which is related to a relativistic radial contraction of ***all*** s- and p-shells as the inner electrons approach a significant fraction of the speed of light (which for an Hg 1s electron is 58%, implying a radial shrinkage of 23%).

L240: Replace "at which it occurs" with "at which the reaction occurs"

>Replaced.

Ibid: "The rate of a reaction is determined by 240 the interaction between kinetics, a rate process, and thermodynamics that describes the energetics of the process" – This sentence is very imprecise. "kinetics" of what? For instance, for a gas phase reaction the rate depends on the number of collisions between the reactants and thermodynamics of their interactions (the change in entropy and enthalpy upon passing through the transition state).

>We thank the reviewer for this valuable comment and have amended the paragraph to read:

>... Elementary processes involve a transition between two atomic or molecular states separated by a potential energy barrier representing the activation energy. The rate of a gas-phase reaction depends on the number of collisions between the reactants and the thermodynamics of their interactions (i.e., the change in entropy, $\Delta S$, and enthalpy, $\Delta H$, upon passing through the transition state), whereas for the rate of a reaction in aqueous solution there are a number of additional factors that can influence the rate, such as solvation, ionic strength, pH, and diffusion rates.

L290: I believe that instead of Pankow, 2007 the reference should be to Mao, et al 2021.

We thank the reviewer for pointing this out. Corrected.

L303: "These positive potentials indicate"

Corrected.

L356-357: Replace M with Hg in chemicals formulas, as you focus solely on one metal – mercury – in this review

Corrected.

L370: Replace "it is mainly in hexavalent form" with "the sulfur it is mainly present in hexavalent form"

Replaced.

L371-373: "Whose presence in AOM is not universal", "which is not relevant in this context",

Replaced.

"questionable to apply speciation by geochemical equilibrium modeling". The meaning is not clear. Why is it questionable? In which context? For what specific reason?

The term geochemical, which includes both the solid Earth and the atmosphere, causes confusion. We have revised the sentence to use the more specific term geospheric:

"Therefore, the application of speciation by equilibrium modeling based on a geospherical basis to assess the interaction between atmospheric DOM and Hg$^{II}$, as in some studies, is questionable".

L377: Replace "speciation is controlled" with "speciation is represented"

Replaced.

L380: What does "It" refer to?

We feel that the sentence structure and referencing here is substandard and have made the following change:

"In this regard, Yang et al. used complexation with fulvic acids under conditions of binding to predominantly O-donors (1:2 complexes with logß = 5.6, Haitzer et al., 2002) as a surrogate for Hg$^{II}$-AOM interaction, which, when applied, was found to dominate overall Hg$^{II}$ speciation of rural and urban rainwater samples in France (Yang et al., 2019)."

L385: What do you mean by "less constrained"?

We have again failed to formulate ourselves well and decided to rearrange the sentence.

"In conclusion, until the complexation of Hg$^{II}$ with AOM is well understood, there is considerable uncertainty regarding the partitioning of aquatic Hg$^{II}$ between stable and reduction labile complexes in the photic atmosphere."

Table 1: Elemental mercury is shown as a ligand in the first row. Does it indeed complex with Hg$^{2+}$? Is it just complexation or a redox reaction?

The line concerns the reverse of the equilibrium introduced on L349, in which Hg$^{2+}$ and Hg$^0$ comproportionate into Hg$_2^{2+}$.

L401: Clarify what you mean by "in the electronic ground state of atmospheric importance". Perhaps "in the electronic ground state" would be sufficient.

We agree and have adjusted accordingly.

Table 2: Are photolysis rates calculated using the global annual average photon flux? There is an extra $O_2$ among the products in Reactions G22b and G22c.

> The calculations are indeed based on this flux. This information has been included in the caption of Table 3. **Rxn G22** is actually without branching. The last two entries (**Rxn G22b & c**) are completely wrong and the products listed are derived from **Rxn G25a & b**. We have corrected this gross error.

L542: "och"?

> 'och' is Swedish for 'and'. This embarrassing error has been corrected.

L559: Replace "obsolete" with a more appropriate word

> We have replaced "obsolete" with "infeasible" in a sentence that now reads:
>
> "…, was considered infeasible by theoretical calculations due to steric hindrance."

L580: Clarify what you mean by "speciated"

> We have replaced "speciated Hg " with "GOM" in a sentence that now reads:
>
> "Field observations of GOM in urban air may suggest…"

L584-586: Simplify the sentence, e.g., "HgX can neither abstract hydrogen atoms from volatile organic compounds nor add to double bonds".

> "Finally, it should be noted that the reactivity of $^\bullet Hg^I X$ towards volatile hydrocarbons is rather low, as $^\bullet Hg^I X$ does not abstract a hydrogen atom from an alkane (e.g., from $CH_4$) nor does it significantly add to a double bond of an alkene (e.g., to $CH_2=CH_2$)."

L889: Rewrite, e.g., "Although atmospheric generally more stable than Hg(I) species, Hg(II) species are still labile and the atmospheric pool"

> Rewritten

L592: "indicated"

> Corrected.

L596: I suggest to clarify that BrHgY are molecules while XHgO are radicals, e.g., "Mixed compounds such as $BrHg^{II}Y$ molecules (Y= ONO, OOH, OH, OCl, OBr etc.) and $XHg^{II}O^\bullet$ radicals (X = Br, OH)"

> OK! Revised.

L607: "computer-assisted theoretical calculations" – be more specific

> We have changed from "computer-assisted theoretical calculations" to "computer-aided calculations based on 2D potential energy surfaces"

L609: Did you mean that "Photolysis of BrHgONO forms NO and BrHgO"?

> Yes, corrected!

L614: What do tilde means in O~O~H?

> Tilde indicates that the bond is unstable. We have adjusted (sic) "$^\bullet Hg^I Br + O{\sim}O{\sim}H$ (2%)" to "$^\bullet Hg^I Br + O{\sim}O–H$ ($\leqslant 3\%$)", thus including both of the $^\bullet Hg^I Br + O + {^\bullet}OH$ (2%) and $^\bullet Hg^I Br + {^\bullet}OOH$ ($\leqslant 1\%$) product channels.

L636: Consider revising "The enthalpy of thermal decay of HgO is weakly endothermic", as for diatomic molecules thermal dissociation is always endothermic. Did you mean "only weakly endothermic"?

We are grateful for this comment, and have revised the sentence to read ad notam.

L646: "marginal" – did you mean "scarce"?

Yes, corrected!

L647: Rephrase "computational calculations"

This sentence has been rephrased as well as expanded.

"Initially, the focus of computer simulations was on the bromine analog and its reactions. Later, the scope was expanded to include the thermochemistry for the hydroxyl and chlorine analogs, which will be recapitulated below."

Figure 6: add a citation to the source of this figure.

Added to the caption: Adopted from Sun et al. (2016).
Figure 7: It looks pretty but highly overloaded and messy, making it hard to read. Consider removing molecular structures, keeping only the formulas. Spectra look aesthetically nice, but also complicate the figure. Aim at striking a balance between prettiness and legibility. The same applies to several following figures. In this figure caption, I would not put $NO_2$, BrO and OH as abundant radicals, as their concentrations are vastly different.

[Figure]

[Figure]

Revised figures 7 and 9 are attached above. Part of the caption to the former figure has been reworded as follows:

"… but also by radicals (in descending order of abundance) such as $NO_2 > HO_2 > BrO \approx OH$ etc. It should be noted that NO cannot efficiently oxidize…"

L739: "elemental oxygen"

We agree with this and change from "molecular" to "elemental".

L760: Instead of asterisk provide the specific state of $O_2$

The state $O_2(^3\Sigma_u^+)$ has been inserted in Rxn 5 & 6.

L775: "less energetic than the reactants" – awkward, consider revising.

Revised to "lower in energy…"

L789: Is it the excited HgO?

HgO ($^3\Pi$) is the ground state.

L819: "automatically measured" is confusing. Is "automatically" important here?

No, we delete "automatically".

Figure 10: The pathway resulting in polymerization of HgO is inconsistent with its extremely short lifetime and extremely low concentration under typical atmospheric conditions, even in the stratosphere.

It was premature and incorrect to indicate that the clustering of HgO would have some atmospheric significance. For this reason, Figure 10 has been redrawn and is attached below.

[Figure]

L861: I highly doubt that the MMHg$^+$ cation can be volatilized. Did you mean a species with its counter-anion, e.g., MMHgCl?

"MMHg$^+$ species" refers to the series of chemical compounds containing MMHg$^+$ group. For clarification, we have added "(e.g., MMHgCl)" to the sentence.

Table 4: W17 appears to show a wrong reaction mechanism

As stated in footnote 32, the actual reductant is provisionally determined to be a hydroxylated reduced form that emerges as a consequence of solution irradiation, rather than AQDS. The reaction formula provided is of a schematic nature rather than being exhaustive.

L1025: Is this a dark reaction? What do you mean by "divergent"? Divergent in what sense?

No, the reaction is totally photolytic. We've changed "strongly divergent" to "much larger".

Rxn10: flip the second structure horizontally

L1130-1131: This sentence is bizarre. Clarify why that literature is irrelevant to the atmosphere.

We have clarified with the following revised text:

Despite a wealth of studies addressing the multiphase chemical or physical transformation of Hg under processes such as those under simulated post-combustion conditions, which undoubtedly pertain to the interaction with certain environmental surfaces, the findings offer limited insight into the surface and heterogeneous atmospheric Hg chemistry. The subsequent chapter will address the studies that have been identified as contributing meaningfully to the advancement of understanding in this domain.

L1181: Define "FF"

The term "FF" (fast flow) has been previously defined (line 281). Would you consider this to be sufficient?

L1229: "its freezing point" – water freezing point?

No, revised into "the freezing point of the metal".

L1240: As written, this contradict the information given in the previous paragraph (L1235) that adsorption reduces the energy required for photoreduction. Consider rephrasing.

The reviewer's comment eludes us here. We don't understand what is contradictory or wrong.

L1251: A second order rate constant is given for a photolytic process. Why? Is this reaction limited by the rate of the complex formation? Does it depend on light intensity and wavelength? Please clarify.

According to the reference, it is evident that all experiments were conducted under conditions of a total irradiance of 99 W m$^{-2}$, within the wavelength range of 300 to 420 nm. This indicates that the experiments were performed without any variation in either wavelength or intensity. The variables of interest included pH, chemical composition (with the exception of the 60 nM Hg$^{II}$ doping, which was constant), and temperature. The complexity of evaluating the pseudo-first order rate constant (k') in the disappearance of Hg$^{II}$ prompted the study to employ the mean values of k' scaling linearly with [org] and to present the result in the form of an apparent bimolecular rate constant. In light of the study's remarkably scattered data, our revision is as follows:

"… the release of Hg$^0$, which was most rapid in the presence of benzophenone at high pH."

L1273: Replace "means" with "tweezer"

Replaced.

L1363: "Parenthesized"?

"The prevailing practice of expressing isotope ratios relative to the lightest stable isotope for each element is not applicable to Hg due to the rarity of $^{196}$Hg (0.15% occurrence)".

L1204 and L1407: Consider sticking to a single term (NFS vs NVF).

We stick to NFS and made 19 changes accordingly.

L1581: "Antarctica are interpreted"

where it says "interpreted as originating" it should just say "originate"

[revised manuscript text omitted]